# Representation Learning for Equivariant Inference with Guarantees

**Daniel Ordoñez-Apraez** [1 2 *]  **Vladimir R. Kostić** [1 3 *]  **Alek Fröhlich** [1 2 *]  **Vivien Brandt** [4]
**Karim Lounici** [4 *]  **Massimiliano Pontil** [1 5 *]

## Abstract

In many real-world applications of regression, conditional probability estimation, and uncertainty quantification, exploiting symmetries rooted in physics or geometry can dramatically improve generalization and sample efficiency. While geometric deep learning has made empirical advances by incorporating symmetry and geometry priors, less attention has been given to statistical learning guarantees. In this paper, we introduce an equivariant representation learning framework that simultaneously addresses regression, conditional probability estimation, and uncertainty quantification while providing first-of-its-kind non-asymptotic statistical learning guarantees. Grounded in operator and group representation theory, our framework approximates the spectral decomposition of the conditional expectation operator, building representations that are both equivariant and disentangled along independent symmetry quotient groups. Empirical evaluations on synthetic datasets and real-world robotics applications confirm the potential of our approach, matching or outperforming existing equivariant baselines in regression while providing well-calibrated uncertainty estimates.

## 1. Introduction

A central problem in machine learning is modeling conditional probabilities—understanding how the distribution of a target variable **y** changes with an observed variable **x**. This underpins robust reasoning under uncertainty in critical applications such as medicine, finance, robotics, and physics (Izbicki, 2025; Smith, 2013; Wasserman, 2007). However,

estimating conditional distributions remains challenging in high-dimensional settings without strong inductive biases (Scott, 1991; Nagler & Czado, 2016; Izbicki & B. Lee, 2017).

Symmetry priors, in the form of principled assumptions about invariance or equivariance in the underlying data-generating process, offer a compelling way to reduce sample complexity and improve generalization (Kashinath et al., 2021; Wang, 2021; Bronstein et al., 2021; Elesedy, 2023). These priors naturally arise in inference tasks in chemistry and particle physics (Dresselhaus et al., 2007), set-&-graph structured data (Bronstein et al., 2021), computer graphics (Mitra et al., 2013; Kovar et al., 2002), and dynamical systems with group-invariant/equivariant laws of motion, which are ubiquitous in fields like physics (Dresselhaus et al., 2007), fluid dynamics (Olver, 1993), and robotics (Ordoñez-Apraez et al., 2025; Zhu et al., 2022).

Over the past few years, Geometric Deep Learning (GDL) has produced a rich ecosystem of architectures that encode symmetries, achieving strong empirical performance across various supervised (Bronstein et al., 2021; Weiler et al., 2023; van der Pol et al., 2020) and unsupervised tasks (Dangovski et al., 2022; Keurti et al., 2023; Wang et al., 2024a). However, the field remains focused on application specific designs and architectural innovation, with limited understanding of how symmetry priors can be leveraged to *learn representations with provable generalization guarantees*.

In this work, we take a different route: rather than proposing new architectures or solving specific inference tasks, we ask *how to systematically learn symmetry-aware representations that best capture conditional structure in the data*. Specifically, how should equivariant networks be trained so that their learned features reveal conditional distributions, and how does the quality of these representations affect performance in downstream tasks such as regression, conditional probability estimation, and uncertainty quantification?

To answer these questions, we connect two fields rarely studied together: spectral contrastive learning (Zou et al., 2013), a self-supervised approach for learning deep representations via operator-theoretic models of conditional expectations (Tsai et al., 2021; Kostic et al., 2024a), and GDL (Bronstein et al., 2021), which imposes symmetry and geometry priors

---

*Equal contribution  [1]CSML, Italian Institute of Technology, Genova, Italy [2]Università di Genova, Genova, Italy [3]University of Novi Sad, Novi Sad, Serbia [4]CMAP, École Polytechnique, Palaiseau, France [5]University College London, London, UK. Correspondence to: Daniel Ordoñez-Apraez <daniel.ordonez@iit.it>.

*Proceedings of the 43rd International Conference on Machine Learning*, Seoul, South Korea. PMLR 306, 2026. Copyright 2026 by the author(s).

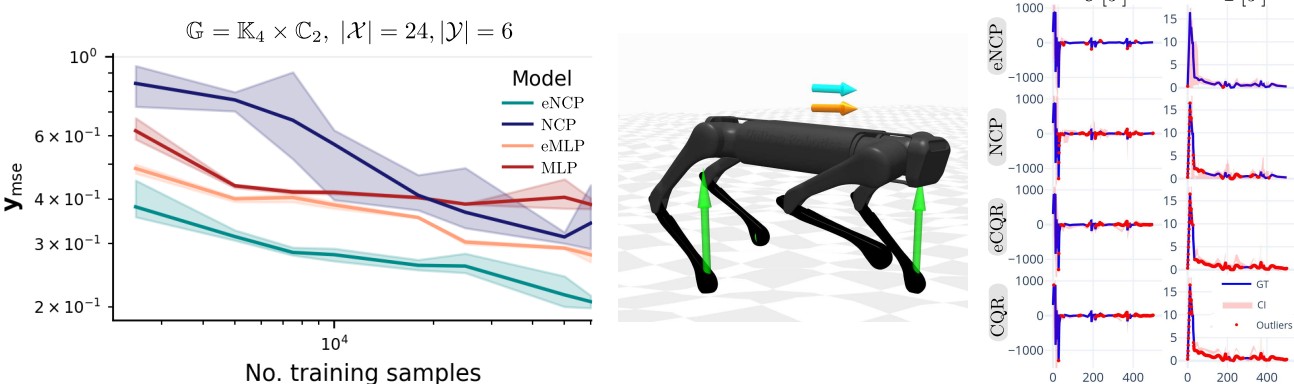

*Figure 1.* **Left**: Test set sample efficiency for $\mathbb{G}$-equivariant regression (MSE vs. training samples) when predicting the $\mathbb{G}$-equivariant linear and angular momentum of a quadruped robot's center of mass (CoM) from noisy joint positions and velocities. **Right**: Uncertainty quantification via $\mathbb{G}$-equivariant prediction of 90% Confidence Intervals (CIs) (light-red area) for the robot's instantaneous work $U_t$ and kinetic energy $T_t$ during locomotion over rough terrain for our method (eNCP) and competitors. The figure shows a trajectory with a strong initial disturbance, where blue markers denote samples within the predicted CI and red markers denote those outside. Note that only eNCP is able to predict well-calibrated CI intervals that cover both the disturbance and recovery phases.

through architectural constraints in Neural Networks (NNs). Our approach clarifies how symmetry constraints shape the representation space and improve generalization, opening new avenues for exchange between these fields. Concretely, we show that our method outperforms existing GDL techniques on regression tasks and provides reliable uncertainty quantification for robot locomotion (Fig. 1).

**Contributions** (1) Methodological framework: We introduce Equivariant Neural Conditional Probability (eNCP), the first framework to combine equivariant neural networks with operator-theoretic estimation of conditional distributions. (2) Task-agnostic representation learning: We show that any $\mathbb{G}$-equivariant architecture can be used to learn *disentangled, symmetry-respecting representations* that generalize across diverse downstream inference tasks. (3) Learning guarantees: By linking the representation quality directly to sample complexity, we provide the *first non-asymptotic statistical learning guarantees* for equivariant regression and symmetry-aware conditional probability estimation and uncertainty quantification. (4) Empirical results: [1] On both synthetic and real-world robotics tasks, eNCP consistently outperforms baselines, including contrastive methods Neural Conditional Probability (NCP) (Kostic et al., 2024a) and current equivariant models. In particular, eNCP achieves state-of-the-art performance in the challenging robotics task of contact force estimation.

**Paper Structure** Sec. 2 reviews modeling conditional probabilities with linear operators and NCP. Sec. 3 formally presents the symmetry priors we consider. Sec. 4 introduces our eNCP learning framework. Sec. 5 outlines our theoretical learning guarantees. Sec. 6 showcases experiments

---

[1] Code available at *github.com/Danfoa/symm_rep_learn*

on synthetic and real-world data. Furthermore, because the paper involves complex notation from probability, operator theory, and group theory, the appendices include a glossary of notation (App. A) as well as detailed expositions on representation theory (App. I), symmetric function spaces (App. J), and equivariant linear operators (App. K). Finally, App. C offers an in-depth discussion of related work, contrasting our work with the literature across these rich fields.

## 2. Background

We briefly review the operator-theoretic framework for modeling conditional probabilities, which underpins both NCP and our proposed eNCP method. We denote a random variable by $\mathbf{x}$, its realizations by $\boldsymbol{x} \in \mathcal{X}$, its probability distribution by $\mathbb{P}(\mathbf{x})$, and measure by $P_\mathbf{x}$. Expectations are written as $\mathbb{E}_\mathbf{x}[f(\mathbf{x})] = \int_\mathcal{X} f(\boldsymbol{x}) P_\mathbf{x}(d\boldsymbol{x})$. The same notation applies to other random variables such as $\mathbf{y}$.

**Modeling of Conditional Probabilities** Kostic et al. (2024a) proposed to model conditional probabilities by approximating the *conditional expectation operator* (Baker, 1973; Song et al., 2009; Ryu et al., 2024), $\mathsf{E}_{\mathbf{y}|\mathbf{x}} \colon \mathcal{L}_\mathbf{y}^2 \to \mathcal{L}_\mathbf{x}^2$, a linear integral operator acting on the Hilbert spaces $\mathcal{L}_\mathbf{x}^2 := \mathcal{L}_{P_\mathbf{x}}^2(\mathcal{X}, \mathbb{R})$ and $\mathcal{L}_\mathbf{y}^2 := \mathcal{L}_{P_\mathbf{y}}^2(\mathcal{Y}, \mathbb{R})$ of square-integrable functions of the random variables $\mathbf{x}$ and $\mathbf{y}$, respectively. The action of this operator on any function $h \in \mathcal{L}_\mathbf{y}^2$ returns the function's conditional expectation:

$$[\mathsf{E}_{\mathbf{y}|\mathbf{x}}h](\boldsymbol{x}) = \mathbb{E}[h(\mathbf{y})|\mathbf{x}=\boldsymbol{x}] := \int_\mathcal{Y} h(\boldsymbol{y}) P_{\mathbf{y}|\mathbf{x}}(d\boldsymbol{y}|\boldsymbol{x})$$
$$= \int_\mathcal{Y} h(\boldsymbol{y}) \frac{P_{\mathbf{yx}}(d\boldsymbol{y}, d\boldsymbol{x})}{P_\mathbf{x}(d\boldsymbol{x})} = \int_\mathcal{Y} h(\boldsymbol{y}) \kappa(\boldsymbol{x}, \boldsymbol{y}) P_\mathbf{y}(d\boldsymbol{y}),$$
$$\tag{1}$$

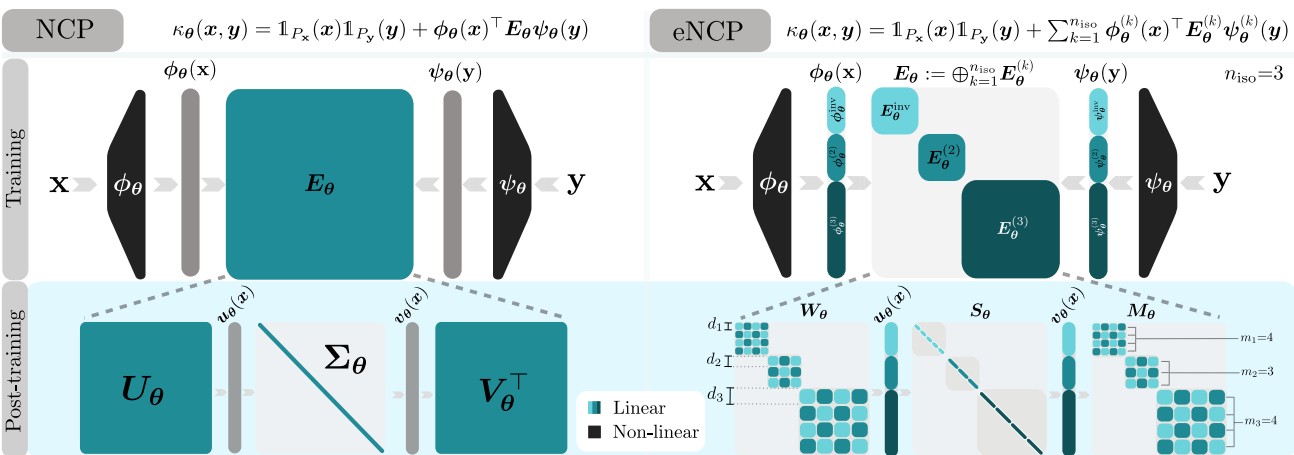

*Figure 2.* **Left:** NCP's bilinear NN architecture. **Right:** eNCP's $\mathbb{G}$-equivariant bilinear NN architecture, featuring $\phi_\theta$ and $\psi_\theta$ as $\mathbb{G}$-equivariant NNs and $E_\theta$ as a $\mathbb{G}$-equivariant block-diagonal matrix. Each block is equivariant to a quotient group $\mathbb{G}^{(k)} \subseteq \mathbb{G}$ and is constrained to have singular spaces of dimension at least $d_k$—the minimal dimension for a representation of the action of $\mathbb{G}^{(k)}$.

where $P_{\mathbf{y}|\mathbf{x}}$ is the conditional probability measure, and $\kappa(\boldsymbol{x}, \boldsymbol{y}) := \frac{P_{\mathbf{xy}}(d\boldsymbol{x}, d\boldsymbol{y})}{P_{\mathbf{x}}(d\boldsymbol{x})\, P_{\mathbf{y}}(d\boldsymbol{y})}$ is the Pointwise Mutual Dependency (PMD) (Sugiyama et al., 2012) kernel defining $\mathsf{E}_{\mathbf{y}|\mathbf{x}}$, obtained as the Radon-Nikodym derivative of the joint measure to the product of marginal measures (see Fig. 3 and App. H).

The conditional expectation operator is significant because it provides an infinite-dimensional linear model—in a nonlinear representation space—for computing conditional probabilities and expectations. To see this, note that for any $\boldsymbol{x} \in \mathcal{X}$ and any measurable set $\mathbb{B} \subset \mathcal{Y}$ we have that:

$$\mathbb{P}(\mathbf{y} \in \mathbb{B}|\mathbf{x}{=}\boldsymbol{x}) := \int_{\mathcal{Y}} \mathbb{1}_{\mathbb{B}}(\boldsymbol{y}) P_{\mathbf{y}|\mathbf{x}}(d\boldsymbol{y}|\boldsymbol{x}) = [\mathsf{E}_{\mathbf{y}|\mathbf{x}}\mathbb{1}_{\mathbb{B}}](\boldsymbol{x}), \text{ and}$$

$$\mathbb{E}[\mathbf{y}|\mathbf{x}{=}\boldsymbol{x}] := [\mathsf{E}_{\mathbf{y}|\mathbf{x}}\mathbf{y}](\boldsymbol{x}). \tag{2}$$

Therefore, to estimate conditional probabilities and expectations, NCP seeks the best finite-dimensional approximation of $\mathsf{E}_{\mathbf{y}|\mathbf{x}}$. As we explain next, this gives rise to a representation learning problem (Oord et al., 2018), in which the optimal representations of $\mathbf{x}$ and $\mathbf{y}$ are given by the top left and right singular functions of $\mathsf{E}_{\mathbf{y}|\mathbf{x}}$ (Kostic et al., 2024b; Ryu et al., 2024).

**Spectral Representation Learning** The problem of approximating the conditional expectation operator $\mathsf{E}_{\mathbf{y}|\mathbf{x}}$ as a rank-$r$ operator $\mathsf{E}_\theta$ with matrix representation $\boldsymbol{E}_\theta \in \mathbb{R}^{r \times r}$ is defined as:

$$\underset{\boldsymbol{\theta}}{\arg\min}\ \|\mathsf{E}_{\mathbf{y}|\mathbf{x}}{-}\mathsf{E}_\theta\|_{\mathrm{HS}}^2 = \mathbb{E}_{\mathbf{x}}\mathbb{E}_{\mathbf{y}}(\kappa(\mathbf{x}, \mathbf{y}) - \kappa_\theta(\mathbf{x}, \mathbf{y}))^2,$$

$$\text{s.t. } \mathbb{E}_{\mathbf{x}}\mathbb{E}_{\mathbf{y}}\kappa_\theta(\mathbf{x}, \mathbf{y}){=}1 \text{ and } \mathrm{rank}(\mathsf{E}_\theta) \leq r + 1. \tag{3}$$

The optimal solution, denoted $\mathsf{E}_\star$, is the $r$-truncated Singular Value Decomposition (SVD) of $\mathsf{E}_{\mathbf{y}|\mathbf{x}}$ (Eckart & Young, 1936;

Weidmann, 2012; Baker, 1973), namely

$$[\mathsf{E}_\star f](\boldsymbol{x}) = \sum_{i=0}^r \sigma_i \langle f, v_i \rangle_{P_{\mathbf{y}}} u_i(\boldsymbol{x}), \text{ with}$$

$$\sigma_i u_i(\boldsymbol{x}) = [\mathsf{E}_{\mathbf{y}|\mathbf{x}}v_i](\boldsymbol{x}), \ \forall i \in [r], \tag{4}$$

where $(\sigma_i, u_i, v_i)$ denotes the $i^{\text{th}}$ singular value and left/right singular functions of $\mathsf{E}_{\mathbf{y}|\mathbf{x}}$, with $(\sigma_0{=}1, u_0{=}\mathbb{1}_{P_{\mathbf{x}}}, v_0{=}\mathbb{1}_{P_{\mathbf{y}}})$ being the constant functions supported on $P_{\mathbf{x}}$ and $P_{\mathbf{y}}$.

Consequently, NCP parameterizes $\mathsf{E}_\theta$ by a bilinear model $\kappa_\theta(\boldsymbol{x}, \boldsymbol{y}) = \mathbb{1}_{P_{\mathbf{x}}}(\boldsymbol{x})\mathbb{1}_{P_{\mathbf{y}}}(\boldsymbol{y}){+}\phi_\theta(\boldsymbol{x})^\top \boldsymbol{E}_\theta \psi_\theta(\boldsymbol{y})$, composed of two encoder NNs $\phi_\theta \colon \mathcal{X} \to \mathbb{R}^r$ and $\psi_\theta \colon \mathcal{Y} \to \mathbb{R}^r$ that aim to approximate the *span* of the top $r$ (non-constant) left and right singular functions of $\mathsf{E}_{\mathbf{y}|\mathbf{x}}$. See Fig. 2-left.

As $\kappa$ is generally unavailable analytically, (3) is solved via the regularized contrastive low-rank (CLoRa) loss:

$$\mathcal{L}_\gamma(\boldsymbol{\theta}) = -2\mathbb{E}_{\mathbf{xy}}\kappa_\theta(\mathbf{x}, \mathbf{y}) + \mathbb{E}_{\mathbf{x}}\mathbb{E}_{\mathbf{y}}\kappa_\theta(\mathbf{x}, \mathbf{y})^2$$

$$+ 2\gamma\big(\|\mathbb{E}_{\mathbf{x}}\phi_\theta(\mathbf{x})\|_F^2 + \|\mathbb{E}_{\mathbf{y}}\psi_\theta(\mathbf{y})\|_F^2$$

$$+ \|\mathrm{Cov}(\phi_\theta) - \boldsymbol{I}_r\|_F^2 + \|\mathrm{Cov}(\psi_\theta) - \boldsymbol{I}_r\|_F^2\big), \tag{5}$$

where the first two regularization terms center the learned representations, ensuring that $\mathbb{E}_{\mathbf{x}}\mathbb{E}_{\mathbf{y}}\kappa_\theta(\mathbf{x}, \mathbf{y}){\approx}1$ (Kostic et al., 2024a), while the last two enforce approximate orthonormality of the learned bases in $\mathcal{F}_{\mathbf{x}}^\theta := \mathrm{span}(\phi_\theta) \subset \mathcal{L}_{\mathbf{x}}^2$ and $\mathcal{F}_{\mathbf{y}}^\theta := \mathrm{span}(\psi_\theta) \subset \mathcal{L}_{\mathbf{y}}^2$ (Izbicki & B. Lee, 2017). A key property of NCP is that the learned representations enables reliable regression and conditional probability estimation—and thus uncertainty quantification—via (2).

## 3. Problem Formulation

This paper tackles the problem of estimating the conditional expectation $\mathbb{E}[\mathbf{y}|\mathbf{x}{=}\cdot]$, and, more generally, conditional

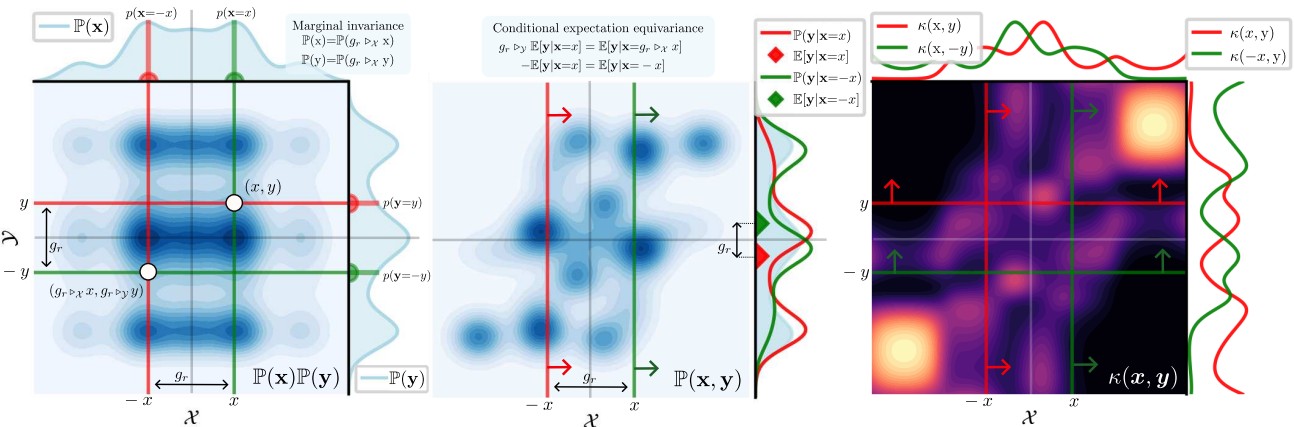

*Figure 3.* Example of symmetric random variables $(x, y) \sim \mathcal{X} \times \mathcal{Y} \subset \mathbb{R} \times \mathbb{R}$, whose marginals $\mathbb{P}(x)$ and $\mathbb{P}(y)$; joint $\mathbb{P}(x, y)$; and conditional $\mathbb{P}(y|x)$ distributions are invariant to reflections of the data: $g_r \triangleright_{\mathcal{X}} x = -x$ and $g_r \triangleright_{\mathcal{Y}} y = -y$, where $g_r$ denotes the reflection element of the reflection symmetry group $\mathbb{C}_2 := \{e, g_r | g_r^2 = e\}$. Consequently, the PMD $\kappa(x, y)$ is $\mathbb{C}_2$-invariant.

distribution $\mathbb{P}(\mathbf{y}|\mathbf{x})$, for random variables $\mathbf{x} \in \mathcal{X}$ and $\mathbf{y} \in \mathcal{Y}$, under the assumption that[2] $\mathbb{P}(\mathbf{y}|\mathbf{x})$ and $\mathbb{P}(\mathbf{x})$ are $\mathbb{G}$-invariant under *known* symmetry transformations of the data (as depicted in Fig. 3), i.e:

$$\mathbb{P}(\mathbf{y}|\mathbf{x}) = \mathbb{P}(g \triangleright_{\mathcal{Y}} \mathbf{y} | g \triangleright_{\mathcal{X}} \mathbf{x}), \quad \mathbb{P}(\mathbf{x}) = \mathbb{P}(g \triangleright_{\mathcal{X}} \mathbf{x}), \quad \forall g \in \mathbb{G}, \tag{6}$$

where $\mathbb{G}$ denotes a finite symmetry group (Def. I.1) acting on the data spaces $\mathcal{X}$ and $\mathcal{Y}$ via the group actions, $\triangleright_{\mathcal{X}} \colon \mathbb{G} \times \mathcal{X} \to \mathcal{X}$, and $\triangleright_{\mathcal{Y}} \colon \mathbb{G} \times \mathcal{Y} \to \mathcal{Y}$, with $g \triangleright_{\mathcal{X}} \boldsymbol{x} \in \mathcal{X}$ and $g \triangleright_{\mathcal{Y}} \boldsymbol{y} \in \mathcal{Y}$ denoting linear, invertible transformations of $\boldsymbol{x}$ and $\boldsymbol{y}$ defined by $g \in \mathbb{G}$ (see Fig. 3 and Def. I.2).

These priors imply the $\mathbb{G}$-invariance of the joint distribution $\mathbb{P}(\mathbf{x}, \mathbf{y})$ and of $\mathbf{y}$'s marginal distribution $\mathbb{P}(\mathbf{y})$, as well as the $\mathbb{G}$-equivariance of conditional expectations (see Fig. 3-middle and Prop. D.1):

$$g \triangleright_{\mathcal{Y}} \mathbb{E}[\mathbf{y}|\mathbf{x}{=}\boldsymbol{x}] = \mathbb{E}[\mathbf{y}|\mathbf{x}{=}g \triangleright_{\mathcal{X}} \boldsymbol{x}] \quad \forall g \in \mathbb{G}, \boldsymbol{x} \in \mathcal{X}. \tag{7}$$

Note that (7) implies the $\mathbb{G}$-equivariance of the regression function $\boldsymbol{x} \mapsto \mathbb{E}[\mathbf{y}|\mathbf{x}{=}\boldsymbol{x}]$ (see Fig. 3-middle). Therefore, the symmetry priors of (6) are satisfied whenever we approximate an equivariant/invariant function—that is, in virtually *all applications of GDL* (Bronstein et al., 2021).

The above symmetry priors represent a strong inductive bias for the conditional expectation operator (2), as they lead to the PMD kernel defining the operator to be $\mathbb{G}$-invariant (see Fig. 3-right):

$$\kappa(\boldsymbol{x}, \boldsymbol{y}) = \kappa(g \triangleright_{\mathcal{X}} \boldsymbol{x}, g \triangleright_{\mathcal{Y}} \boldsymbol{y}) \quad \forall g \in \mathbb{G}, \boldsymbol{x} \in \mathcal{X}, \boldsymbol{y} \in \mathcal{Y}. \tag{8}$$

In Sec. 4, we extend the NCP framework (Kostic et al., 2024a) by leveraging (8) to incorporate symmetry priors. This enables efficient estimation of $\mathbb{G}$-invariant conditional probabilities (6) and $\mathbb{G}$-equivariant regression (7) using GDL architectures, via (2), with strong learning guarantees.

---

[2]Throughout, with some abuse of notation we denote by $\mathbb{P}(\mathbf{x})$ and $\mathbb{P}(\mathbf{y}|\mathbf{x})$ both the probability and conditional probability, respectively, as well as the corresponding densities, when they exist.

## 4. ENCP Method for Equivariant Representation Learning

In this section, we show how to incorporate the symmetry priors (6) into NCP's representation learning framework. First, we analyze the symmetry constraints on the infinite-dimensional conditional expectation operator and prove that, for symmetric random variables $\mathbf{x}$ and $\mathbf{y}$, the optimal solution of (3) yields $\mathbb{G}$-equivariant representations $\phi_{\boldsymbol{\theta}}$ and $\psi_{\boldsymbol{\theta}}$ and approximates the operator with a $\mathbb{G}$-equivariant matrix $E_{\boldsymbol{\theta}}$. Then, we explain how to embed these structural constraints into the bilinear neural network architecture of NCP using *any* type of equivariant NNs.

**Symmetric Function Spaces** The assumption of $\mathbb{G}$-invariance of the marginal probabilities (Sec. 3) implies that the function spaces $\mathcal{L}_{\mathbf{x}}^2$ and $\mathcal{L}_{\mathbf{y}}^2$ are symmetric Hilbert spaces of $\mathbb{G}$-equivariant functions, as these inherit unitary group actions $\triangleright_{\mathcal{L}_{\mathbf{x}}^2} \colon \mathbb{G} \times \mathcal{L}_{\mathbf{x}}^2 \to \mathcal{L}_{\mathbf{x}}^2$ and $\triangleright_{\mathcal{L}_{\mathbf{y}}^2} \colon \mathbb{G} \times \mathcal{L}_{\mathbf{y}}^2 \to \mathcal{L}_{\mathbf{y}}^2$ defined via the push-forward of symmetry transformations of the data spaces (see details in App. J and in Fig. 21):

$$\begin{aligned} g \triangleright_{\mathcal{L}_{\mathbf{x}}^2} f(\cdot) &:= f(g^{-1} \triangleright_{\mathcal{X}} \cdot) \in \mathcal{L}_{\mathbf{x}}^2, \quad \text{and} \\ g \triangleright_{\mathcal{L}_{\mathbf{y}}^2} h(\cdot) &:= h(g^{-1} \triangleright_{\mathcal{Y}} \cdot) \in \mathcal{L}_{\mathbf{y}}^2, \quad \forall g \in \mathbb{G}. \end{aligned} \tag{9}$$

A fundamental property of $\mathbb{G}$-symmetric Hilbert spaces is their orthogonal decomposition into $n_{\text{iso}} \leq |\mathbb{G}|$ subspaces referred to as *isotypic subspaces*: $\mathcal{L}_{\mathbf{x}}^2 = \bigoplus_{k \in [1, n_{\text{iso}}]}^{\perp} \mathcal{L}_{\mathbf{x}}^{2(k)}$, and $\mathcal{L}_{\mathbf{y}}^2 = \bigoplus_{k \in [1, n_{\text{iso}}]}^{\perp} \mathcal{L}_{\mathbf{y}}^{2(k)}$ (see Thm. I.17). Where each $\mathcal{L}_{\mathbf{x}}^{2(k)}$ and $\mathcal{L}_{\mathbf{y}}^{2(k)}$ denote the spaces of $\mathbb{G}^{(k)}$-equivariant functions of $\mathbf{x}$ and $\mathbf{y}$, with $\mathbb{G}^{(k)} := \mathbb{G}/\mathbb{N}_k$ being a **quotient subgroup**, generated by a normal subgroup $\mathbb{N}_k$ defined by the kernel of the group's $k^{\text{th}}$ irreducible representation (see App. I.2).

This standard result from harmonic analysis (Mackey, 1980) enables us to express any $\mathbb{G}$-equivariant function as a sum of its projections onto the isotypic subspaces:

$$f(\cdot) = f^{\text{inv}}(\cdot) + \sum_{k=2}^{n_{\text{iso}}} f^{(k)}(\cdot), \quad h(\cdot) = h^{\text{inv}}(\cdot) + \sum_{k=2}^{n_{\text{iso}}} h^{(k)}(\cdot),$$
$$\text{with } f^{(k)} \in \mathcal{L}_{\mathbf{x}}^{2(k)}, h^{(k)} \in \mathcal{L}_{\mathbf{y}}^{2(k)}, \forall \, k \in [n_{\text{iso}}], \tag{10}$$

where $f^{(k)}$ and $h^{(k)}$ denote the $\mathbb{G}^{(k)}$-equivariant components of $f$ and $h$. Moreover, by convention, we associate the first subspace ($k = 1$) with the space of $\mathbb{G}$-invariant functions, i.e., $\mathbb{G}^{(1)} = \mathbb{G}^{\text{inv}} = \{e\}$ (see Example J.4 in the Appendix).

**Equivariant Conditional Expectation Operator** The $\mathbb{G}$-invariance of the PMD kernel (8), implies that $\mathsf{E}_{\mathbf{y}|\mathbf{x}}$ is a $\mathbb{G}$-equivariant linear operator (see Def. K.1). This means that $\mathsf{E}_{\mathbf{y}|\mathbf{x}}$ commutes with the group action on the function spaces, and consequently due to Shur's lemma (Lemma I.10), can be decomposed (disentangled) into a direct sum of operators acting on the corresponding isotypic subspaces (see details in App. K), that is:

$$g \triangleright_{\mathcal{L}_{\mathbf{x}}^2} [\mathsf{E}_{\mathbf{y}|\mathbf{x}} h](\cdot) = \mathsf{E}_{\mathbf{y}|\mathbf{x}}[g \triangleright_{\mathcal{L}_{\mathbf{y}}^2} h](\cdot) \quad \forall \, h \in \mathcal{L}_{\mathbf{y}}^2, g \in \mathbb{G}, \quad \text{gives}$$
$$[\mathsf{E}_{\mathbf{y}|\mathbf{x}} h](\cdot) = \sum_{k=1}^{n_{\text{iso}}} [\mathsf{E}_{\mathbf{y}|\mathbf{x}}^{(k)} h^{(k)}](\cdot) \tag{11}$$

where each $\mathsf{E}_{\mathbf{y}|\mathbf{x}}^{(k)} : \mathcal{L}_{\mathbf{y}}^{2(k)} \to \mathcal{L}_{\mathbf{x}}^{2(k)}$ models the conditional expectation for $\mathbb{G}^{(k)}$-equivariant functions.

**Equivariant Representation Learning** The $\mathbb{G}$-equivariant structure of $\mathsf{E}_{\mathbf{y}|\mathbf{x}}$ and its disentanglement (11) into isotypic components suggests that computing the conditional expectation of a $\mathbb{G}$-equivariant function is equivalent to summing the conditional expectations of its $\mathbb{G}^{(k)}$-equivariant components for all $k \in [n_{\text{iso}}]$. Therefore, the loss function of problem (3), where $\mathsf{E}_{\mathbf{y}|\mathbf{x}}$ is approximated in finite dimensional spaces $\mathcal{F}_{\mathbf{x}}^{\boldsymbol{\theta}}$ and $\mathcal{F}_{\mathbf{y}}^{\boldsymbol{\theta}}$, decouples into $n_{\text{iso}}$ independent (disentangled) components:

$$\arg\min_{\boldsymbol{\theta}} \, \|\mathsf{E}_{\mathbf{y}|\mathbf{x}} - \mathsf{E}_{\boldsymbol{\theta}}\|_{\text{HS}}^2 = \sum_{k=1}^{n_{\text{iso}}} \|\mathsf{E}_{\mathbf{y}|\mathbf{x}}^{(k)} - \mathsf{E}_{\boldsymbol{\theta}}^{(k)}\|_{\text{HS}}^2$$
$$= \mathbb{E}_{\mathbf{x}} \mathbb{E}_{\mathbf{y}} \sum_{k=1}^{n_{\text{iso}}} (\kappa^{(k)}(\mathbf{x}, \mathbf{y}) - \kappa_{\boldsymbol{\theta}}^{(k)}(\mathbf{x}, \mathbf{y}))^2,$$
$$\text{s.t. } \kappa_{\boldsymbol{\theta}}(g \triangleright_{\mathcal{X}} \boldsymbol{x}, g \triangleright_{\mathcal{Y}} \boldsymbol{y}) = \kappa_{\boldsymbol{\theta}}(\boldsymbol{x}, \boldsymbol{y}),$$
$$\mathbb{E}_{\mathbf{x}} \mathbb{E}_{\mathbf{y}} \kappa_{\boldsymbol{\theta}}(\mathbf{x}, \mathbf{y}) = 1, \quad \forall g \in \mathbb{G}, (\boldsymbol{x}, \boldsymbol{y}) \in \mathcal{X} \times \mathcal{Y}. \tag{12}$$

Importantly, to satisfy the $\mathbb{G}$-invariance constraint on $\kappa_{\boldsymbol{\theta}}$, the truncated operator $\mathsf{E}_{\boldsymbol{\theta}} : \mathcal{F}_{\mathbf{y}}^{\boldsymbol{\theta}} \to \mathcal{F}_{\mathbf{x}}^{\boldsymbol{\theta}}$ must act on symmetric finite-dimensional Hilbert spaces that are stable under $\mathbb{G}$, i.e., $g \triangleright_{\mathcal{L}_{\mathbf{x}}^2} f \in \mathcal{F}_{\mathbf{x}}^{\boldsymbol{\theta}}$ and $g \triangleright_{\mathcal{L}_{\mathbf{y}}^2} h \in \mathcal{F}_{\mathbf{y}}^{\boldsymbol{\theta}}$ for all $f \in \mathcal{F}_{\mathbf{x}}^{\boldsymbol{\theta}}$ and $h \in \mathcal{F}_{\mathbf{y}}^{\boldsymbol{\theta}}$, and have a $\mathbb{G}$-equivariant matrix representation $\boldsymbol{E}_{\boldsymbol{\theta}}$ (see Prop. K.2). Thus, as in the infinite-dimensional case, these finite-dimensional spaces decompose into $n_{\text{iso}}$ isotypic subspaces $\mathcal{F}_{\mathbf{x}}^{\boldsymbol{\theta}} = \bigoplus_{k=1}^{n_{\text{iso}}} \mathcal{F}_{\mathbf{x}}^{\boldsymbol{\theta}(k)}$ and $\mathcal{F}_{\mathbf{y}}^{\boldsymbol{\theta}} = \bigoplus_{k=1}^{n_{\text{iso}}} \mathcal{F}_{\mathbf{y}}^{\boldsymbol{\theta}(k)}$. Accordingly, the truncated operator matrix decomposes block-diagonally into $n_{\text{iso}}$ blocks, i.e., $\boldsymbol{E}_{\boldsymbol{\theta}} = \bigoplus_{k=1}^{n_{\text{iso}}} \boldsymbol{E}_{\boldsymbol{\theta}}^{(k)}$, where each $k$-th block needs to be a $\mathbb{G}^{(k)}$-equivariant matrix approximating the restriction of the conditional expectation

operator on its corresponding isotypic subspace (Salova et al., 2019; Ordoñez-Apraez et al., 2024), see Fig. 2-right.

Analogous to (4), the optimal truncation of $\mathsf{E}_{\mathbf{y}|\mathbf{x}}^{(k)}$ is given by its truncated SVD (Weidmann, 2012). However, the $\mathbb{G}^{(k)}$-equivariance of each disentangled component imposes additional structure on this SVD: any symmetry-transformed singular function remains a singular function in the same singular space (Ordoñez-Apraez et al., 2024). Consequently, each singular space must have dimension at least $d_k$—the smallest real vector-space dimension in which $\mathbb{G}^{(k)}$ can be faithfully represented, that is, the dimension of the irreducible representation of type $k$; see Fig. 2-right and Prop. K.3.

**Equivariant NN Parametrization** To solve $\mathbb{G}$-equivariant regression and estimate $\mathbb{G}$-invariant conditional probabilities via (12), we propose the disentangled bilinear NN setup in Fig. 2-right. This approach supports any GDL $\mathbb{G}$-equivariant backbone (e.g., MLP, CNN, GNN, Transformer), adapting to diverse data modalities and tasks. Our framework can handle *continuous* compact groups via group discretization, and *non-compact* groups by selecting appropriate backbones, as it is done with $\mathbb{G}$-steerable CNNs for image/audio processing with *non-compact translation* equivariance (see details in App. E.1 and Cesa et al. (2022)).

The representation functions $\boldsymbol{\phi}_{\boldsymbol{\theta}} : \mathcal{X} \to \mathbb{R}^r$ and $\boldsymbol{\psi}_{\boldsymbol{\theta}} : \mathcal{Y} \to \mathbb{R}^r$ can be parameterized by *any* $\mathbb{G}$-equivariant NN architecture, with an additional non-learnable linear output layer that applies a change of basis to the isotypic basis, i.e.,

$$\boldsymbol{\phi}_{\boldsymbol{\theta}}(\cdot) = \begin{bmatrix} \phi_{\boldsymbol{\theta}}^{\text{inv}}(\cdot) \\ \vdots \\ \phi_{\boldsymbol{\theta}}^{(n_{\text{iso}})}(\cdot) \end{bmatrix} = \boldsymbol{Q}_{\mathbf{x}}^{\top} (\boldsymbol{f}_{\boldsymbol{\theta}}(\cdot) - \widehat{\mathbb{E}}_{\mathbf{x}}[\boldsymbol{f}_{\boldsymbol{\theta}}(\mathbf{x})]) \quad \text{and}$$

$$\boldsymbol{\psi}_{\boldsymbol{\theta}}(\cdot) = \begin{bmatrix} \psi_{\boldsymbol{\theta}}^{\text{inv}}(\cdot) \\ \vdots \\ \psi_{\boldsymbol{\theta}}^{(n_{\text{iso}})}(\cdot) \end{bmatrix} = \boldsymbol{Q}_{\mathbf{y}}^{\top} (\boldsymbol{h}_{\boldsymbol{\theta}}(\cdot) - \widehat{\mathbb{E}}_{\mathbf{y}}[\boldsymbol{h}_{\boldsymbol{\theta}}(\mathbf{y})]).$$

Here, $\boldsymbol{f}_{\boldsymbol{\theta}} : \mathcal{X} \to \mathbb{R}^r$ and $\boldsymbol{h}_{\boldsymbol{\theta}} : \mathcal{Y} \to \mathbb{R}^r$ denote any $\mathbb{G}$-equivariant backbone architectures that output $r$-dimensional representations of $\mathbf{x}$ and $\mathbf{y}$. These representations are then centered to satisfy the centering constraint in (12). Moreover, $\boldsymbol{Q}_{\mathbf{x}}$ and $\boldsymbol{Q}_{\mathbf{y}}$ are orthogonal, non-learnable change-of-basis matrices that expose the isotypic subspaces of $\mathcal{F}_{\mathbf{x}}^{\boldsymbol{\theta}}$ and $\mathcal{F}_{\mathbf{y}}^{\boldsymbol{\theta}}$ (see details in Remark I.18).

Such an orthogonal decomposition of the learned equivariant representations is referred to in the representation learning literature as a *disentangled* representation (Higgins et al., 2018) (see Def. I.19). In our setting, disentanglement means that the representation decomposes into orthogonal components, each associated with a distinct group of symmetry transformations. Specifically, each component $\boldsymbol{\phi}_{\boldsymbol{\theta}}^{(k)} : \mathcal{X} \to \mathbb{R}^{r_k}$ and $\boldsymbol{\psi}_{\boldsymbol{\theta}}^{(k)} : \mathcal{Y} \to \mathbb{R}^{r_k}$ spans the approximated isotypic subspace of $\mathbb{G}^{(k)}$-equivariant functions, i.e.,

$$\mathcal{F}_{\mathbf{x}}^{\boldsymbol{\theta}} = \oplus_{k\in[1,n_{\mathrm{iso}}]}^{\perp}\mathcal{F}_{\mathbf{x}}^{\boldsymbol{\theta}(k)}, \quad \text{with } \mathcal{F}_{\mathbf{x}}^{\boldsymbol{\theta}(k)} := \mathrm{span}(\boldsymbol{\phi}_{\boldsymbol{\theta}}^{(k)}) \subset \mathcal{L}_{\mathbf{x}}^{2(k)}, \quad \text{and}$$
$$\mathcal{F}_{\mathbf{y}}^{\boldsymbol{\theta}} = \oplus_{k\in[1,n_{\mathrm{iso}}]}^{\perp}\mathcal{F}_{\mathbf{y}}^{\boldsymbol{\theta}(k)}, \quad \text{with } \mathcal{F}_{\mathbf{y}}^{\boldsymbol{\theta}(k)} := \mathrm{span}(\boldsymbol{\psi}_{\boldsymbol{\theta}}^{(k)}) \subset \mathcal{L}_{\mathbf{y}}^{2(k)}.$$

Furthermore, the truncated operator's matrix is parameterized in block-diagonal form $\boldsymbol{E}_{\boldsymbol{\theta}} = \oplus_{k=1}^{n_{\mathrm{iso}}}\boldsymbol{E}_{\boldsymbol{\theta}}^{(k)}$, with each block an $r_k \times r_k$ $\mathbb{G}^{(k)}$-equivariant matrix [3]. The corresponding approximated $\mathbb{G}$-invariant PMD kernel is given by:

$$\kappa_{\boldsymbol{\theta}}(\boldsymbol{x},\boldsymbol{y}) = \mathbb{1}_{P_{\mathbf{x}}}(\boldsymbol{x})\mathbb{1}_{P_{\mathbf{y}}}(\boldsymbol{y}) + \sum_{k=1}^{n_{\mathrm{iso}}}\kappa_{\boldsymbol{\theta}}^{(k)}(\boldsymbol{x},\boldsymbol{y}), \quad \text{with}$$
$$\kappa_{\boldsymbol{\theta}}^{(k)}(\boldsymbol{x},\boldsymbol{y}) := \boldsymbol{\phi}_{\boldsymbol{\theta}}^{(k)}(\boldsymbol{x})^{\top}\boldsymbol{E}_{\boldsymbol{\theta}}^{(k)}\boldsymbol{\psi}_{\boldsymbol{\theta}}^{(k)}(\boldsymbol{y}), \quad \forall k \in [1,n_{\mathrm{iso}}] \tag{13}$$

where $\mathbb{1}_{P_{\mathbf{x}}}(\boldsymbol{x})\mathbb{1}_{P_{\mathbf{y}}}(\boldsymbol{y})$ arises since the first singular functions of $\mathsf{E}_{\mathbf{y}|\mathbf{x}}$ are constant, similarly as in (4).

This parameterization inherently guarantees that the learned truncated operator $\boldsymbol{E}_{\boldsymbol{\theta}}$ satisfies the $\mathbb{G}$-equivariance constraint and that the learned representation spaces satisfy the symmetry constraints imposed on the singular spaces of each $\mathbb{G}^{(k)}$ isotypic subspace. These structural constraints enable sharper symmetry-aware statistical learning guarantees in theory and improved empirical performance in practice; see Secs. 5 and 6 and App. K.2.

**Disentangled Training Loss** Having introduced the equivariance constraints on the truncated operator matrix, we decompose the contrastive loss (5) to reflect the separability of the optimization arising from the operator's isotypic decomposition (11):

$$\mathcal{L}_{\gamma}(\boldsymbol{\theta}) = \sum_{k=1}^{n_{\mathrm{iso}}} -2\mathbb{E}_{\mathbf{x}\mathbf{y}}\kappa_{\boldsymbol{\theta}}^{(k)}(\mathbf{x},\mathbf{y}) + \mathbb{E}_{\mathbf{x}}\mathbb{E}_{\mathbf{y}}\kappa_{\boldsymbol{\theta}}^{(k)}(\mathbf{x},\mathbf{y})^2 +$$
$$\gamma\Omega^{(k)}(\boldsymbol{\theta}) + 2\gamma\big(\|\mathbb{E}_{\mathbf{x}}\boldsymbol{\phi}_{\boldsymbol{\theta}}^{\mathrm{inv}}(\mathbf{x})\|_F^2 + \|\mathbb{E}_{\mathbf{y}}\boldsymbol{\psi}_{\boldsymbol{\theta}}^{\mathrm{inv}}(\mathbf{y})\|_F^2\big). \tag{14}$$

This decomposes learning $\mathbb{G}$-equivariant representations of $\mathbf{x}$ and $\mathbf{y}$ into learning $n_{\mathrm{iso}}$ less constrained $\mathbb{G}^{(k)}$-equivariant representations for distinct quotient groups of $\mathbb{G}$.

Moreover, we improve the estimates of the regularization terms in (5) by leveraging our symmetry priors to: (i) tighten the centering regularization (14) given that functions in $\mathcal{F}_{\mathbf{x}}^{(k)}$ and $\mathcal{F}_{\mathbf{y}}^{(k)}$ are centered by construction for $k \neq \mathrm{inv}$ (see Cor. L.4)—and (ii) exploit the orthogonality between isotypic subspaces (10) to independently regularize orthonormality for each isotypic subspace (see example in Fig. 11), leading to better covariance estimates (Shah & Chandrasekaran, 2012):

$$\Omega^{(k)}(\boldsymbol{\theta}) := \sum_{k=1}^{n_{\mathrm{iso}}} \|\mathrm{Cov}(\boldsymbol{\phi}^{(k)}) - \boldsymbol{I}_{r_k}\|_F^2 + \|\mathrm{Cov}(\boldsymbol{\psi}^{(k)}) - \boldsymbol{I}_{r_k}\|_F^2. \tag{15}$$

Given a batch $\{(\boldsymbol{x}_n,\boldsymbol{y}_n)\}_{n=1}^N$ and their corresponding embeddings $\{(\boldsymbol{\phi}_{\boldsymbol{\theta}}(\boldsymbol{x}_n),\boldsymbol{\psi}_{\boldsymbol{\theta}}(\boldsymbol{y}_n))\}_{n=1}^N$, the empirical unregularized loss is estimated via U-statistics, yielding an unbiased estimate with an effective sample size of $N^2$ (Wang

---

[3]We chose square matrices for notational convenience. Dimensions for $\mathbf{y}$ and $\mathbf{x}$ spaces need not match

et al., 2022b; Tsai et al., 2020).

$$\widehat{\mathcal{L}}_0(\boldsymbol{\theta}) = \sum_{k\in[n_{\mathrm{iso}}]} \Bigg[ \frac{1}{N}\sum_{n\in[N]}\kappa_{\boldsymbol{\theta}}^{(k)}(\boldsymbol{x}_n,\boldsymbol{y}_n) +$$
$$\frac{1}{N(N-1)}\sum_{a\in[N]}\sum_{b\in[N]\setminus\{a\}}\kappa_{\boldsymbol{\theta}}^{(k)}(\boldsymbol{x}_a,\boldsymbol{y}_b)^2 \Bigg]. \tag{16}$$

Similarly, we use U-statistics to obtain unbiased estimates for orthonormal regularization in (15), achieving an effective sample size of $d_k N^2$ per isotypic subspace (see App. F.2). Consequently, standard NN optimization methods can be employed to learn equivariant representations via the approximate model of $\mathsf{E}_{\mathbf{y}|\mathbf{x}}$, enabling downstream inference tasks described in the next section.

## 5. Inference and Learning Guarantees

Once training is complete, the learned $\mathbb{G}$-invariant PMD from (13) can be used, via (2), for $\mathbb{G}$-equivariant regression and $\mathbb{G}$-invariant conditional probability estimation. These estimates are obtained using a NN architecture composed of $\boldsymbol{\phi}_{\boldsymbol{\theta}}$, $\boldsymbol{E}_{\boldsymbol{\theta}}$, and a final linear layer that delivers the basis expansion coefficients of the target variable in the $\mathbf{y}$ representation space $\mathcal{F}_{\mathbf{y}}^{\boldsymbol{\theta}} = \mathrm{span}(\boldsymbol{\psi}_{\boldsymbol{\theta}})$. For a summary of the estimates and their learning guarantees refer to Tab. 1.

Both estimates are derived from the general problem of vector-valued regression of a target function $\boldsymbol{z} \colon \mathcal{X} \to \mathcal{Z}$ defined by the conditional expectation of an observable $\boldsymbol{h} \in \mathcal{L}_{P_{\mathbf{y}}}^2(\mathcal{Y},\mathcal{Z})$, that is, $\boldsymbol{z}(\boldsymbol{x}) := \mathbb{E}_{\mathbf{y}}[\boldsymbol{h}(\mathbf{y})|\mathbf{x} = \boldsymbol{x}] = [\mathsf{E}_{\mathbf{y}|\mathbf{x}}\boldsymbol{h}](\boldsymbol{x})$, where $\mathcal{Z}$ is a symmetric vector space. Using the learned model, we estimate $\boldsymbol{z}(\boldsymbol{x}) := [\mathsf{E}_{\mathbf{y}|\mathbf{x}}\boldsymbol{h}](\boldsymbol{x})$ by:

$$\widehat{\boldsymbol{z}}_{\boldsymbol{\theta}}(\boldsymbol{x}) := \widehat{\mathbb{E}}_{\mathbf{y}}[\boldsymbol{h}(\mathbf{y})] + \boldsymbol{\phi}_{\boldsymbol{\theta}}(\boldsymbol{x})^{\top}\boldsymbol{E}_{\boldsymbol{\theta}}\widehat{\mathbb{E}}_{\mathbf{y}}[\boldsymbol{\psi}_{\boldsymbol{\theta}}(\mathbf{y})\otimes\boldsymbol{h}(\mathbf{y})], \tag{17}$$

where $\widehat{\mathbb{E}}_{\mathbf{y}}[\boldsymbol{\psi}_{\boldsymbol{\theta}}(\mathbf{y})\otimes\boldsymbol{h}(\mathbf{y})]$ represents the basis expansion coefficients of $\boldsymbol{h}$ in the learned basis of $\mathcal{F}_{\mathbf{y}}^{\boldsymbol{\theta}} \subset \mathcal{L}_{\mathbf{y}}^2$. Here, $\widehat{\mathbb{E}}_{\mathbf{x}} \colon \mathcal{L}_{\mathbf{x}}^2 \to \mathbb{R}$ and $\widehat{\mathbb{E}}_{\mathbf{y}} \colon \mathcal{L}_{\mathbf{y}}^2 \to \mathbb{R}$ are the $\mathbb{G}$-invariant empirical expectations defined by:

$$\widehat{\mathbb{E}}_{\mathbf{a}}[f(\mathbf{a})] = \frac{1}{|\mathbb{G}|N}\sum_{g\in\mathbb{G}}\sum_{n=1}^N f(g \triangleright_{\mathcal{A}} \boldsymbol{a}_n) \quad \text{for } \boldsymbol{a} \in \{\boldsymbol{x},\boldsymbol{y}\} \tag{18}$$

Hence, our method learns representations of $\mathbf{x}$ and $\mathbf{y}$ that transform *nonlinear regression of observables* into *linear regression* in the learned representation space. For example, assuming $\mathbf{y}$ has bounded variance and setting $\boldsymbol{h}(\boldsymbol{y}) = \boldsymbol{y}$, we recover the standard ($\mathbb{G}$-equivariant) regression solution (see Tab. 1-left). Equally important, by letting $\boldsymbol{h} = \mathbb{1}_{\mathbb{B}}$—the indicator of a measurable set $\mathbb{B} \subseteq \mathcal{Y}$—the model estimates conditional probabilities (see Tab. 1-right), thereby supporting both regression and uncertainty quantification (e.g., conditional quantiles, see Sec. 6 and Kostic et al. (2024b)). The following learning bounds cover this general setting.

| Task | $f(x) := \mathbb{E}_{\mathbf{y}}[\mathbf{y}|\mathbf{x}{=}x] \approx \hat{f}_{\boldsymbol{\theta}}(x)$ | $\mathbb{P}[\mathbf{y}{\in}\mathbb{B}|\mathbf{x} \in \mathbb{A}] \approx \widehat{\mathbb{P}}_{\boldsymbol{\theta}}[\mathbf{y}{\in}\mathbb{B}|\mathbf{x}{\in}\mathbb{A}]$ |
|---|---|---|
| **Estimate** | $\widehat{\mathbb{E}}_{\mathbf{y}}[\mathbf{y}]{+}\boldsymbol{\phi}_{\boldsymbol{\theta}}(x)^{\top}\boldsymbol{E}_{\boldsymbol{\theta}}\widehat{\mathbb{E}}_{\mathbf{y}}[\boldsymbol{\psi}_{\boldsymbol{\theta}}(\mathbf{y}) \otimes \mathbf{y}]$ | $\widehat{\mathbb{E}}_{\mathbf{y}}[\mathbb{1}_{\mathbb{B}}]{+}\dfrac{\widehat{\mathbb{E}}_{\mathbf{x}}[\mathbb{1}_{\mathbb{A}}(\mathbf{x}){\otimes}\boldsymbol{\phi}_{\boldsymbol{\theta}}(\mathbf{x})]^{\top}\boldsymbol{E}_{\boldsymbol{\theta}}\widehat{\mathbb{E}}_{\mathbf{y}}[\mathbb{1}_{\mathbb{B}}(\mathbf{y}){\otimes}\boldsymbol{\psi}_{\boldsymbol{\theta}}(\mathbf{y})]}{\widehat{\mathbb{E}}_{\mathbf{x}}[\mathbb{1}_{\mathbb{A}}(\mathbf{x})]}$ |
| **Guarantees** | $\|f{-}\hat{f}_{\boldsymbol{\theta}}\|_{\mathcal{L}_{\mathbf{x}}^2} \lesssim \sqrt{\mathrm{Var}[\|\mathbf{y}\|]}\left(\mathcal{E}_{\boldsymbol{\theta}}^r+\dfrac{\ln(d_{\mathrm{iso}}/\delta)}{(d_{\mathrm{iso}}N)^{\frac{\alpha}{1+2\alpha}}}\right)$ | $|\mathbb{P}{-}\widehat{\mathbb{P}}_{\boldsymbol{\theta}}| \lesssim \sqrt{\dfrac{\mathbb{P}[\mathbf{y}\in\mathbb{B}]}{\mathbb{P}[\mathbf{x}\in\mathbb{G}_{\triangleright_{\mathcal{X}}}\mathbb{A}]}}\left(\mathcal{E}_{\boldsymbol{\theta}}^r+\dfrac{\ln(d_{\mathrm{iso}}/\delta)}{(d_{\mathrm{iso}}N)^{\frac{\alpha}{1+2\alpha}}}\right)$ |

*Table 1.* Statistical guarantees for eNCP. The error bounds are governed by three factors: (i) the estimation error, impacted by the structure of the symmetry group $\mathbb{G}$—specifically, the sum of the minimum dimensionalities of the singular subspaces associated with each isotypic component, $d_{\mathrm{iso}} = \sum_{k=1}^{n_{\mathrm{iso}}} d_k$, equivalently the sum of the dimensions of the irreducible representations of the group (see Fig. 2), which increases the effective sample size; (ii) the representation learning error, measuring the quality of the learned representations, quantified by $\mathcal{E}_{\boldsymbol{\theta}}^r = \|\mathsf{E}_{\mathbf{y}|\mathbf{x}} - \mathsf{E}_{\boldsymbol{\theta}}\|_{\mathrm{op}} \leq \sqrt{\sigma_r/\sigma_r - \sigma_{r+1}}\sqrt{\mathcal{L}_{\gamma}(\boldsymbol{\theta}) - \mathcal{L}_{\gamma}(\star)}$; and (iii) the singular-value decay rate $\alpha > 0$ of the operator. Here, $\mathbb{G} \triangleright_{\mathcal{X}} \mathbb{A} := \cup_{g \in \mathbb{G}} g \triangleright_{\mathcal{X}} \mathbb{A}$ denotes the group orbit of $\mathbb{A}$ (Def. I.3).

**Theorem 5.1.** *Let the symmetry priors in (6) hold, $\mathsf{E}_{\mathbf{y}|\mathbf{x}}$ be a $1/\alpha$-Schatten-class $\mathbb{G}$-equivariant conditional expectation operator, as in (11), and $\mathsf{E}_{\boldsymbol{\theta}}$ be a truncated approximation, defined in (13) and trained via (14), with rank $r$ and parameters $\boldsymbol{\theta} \in \Theta$. Then, given an appropriate truncation dimension $r \asymp (N/d_{\mathrm{iso}}^{\alpha})^{1/(1+2\alpha)}$, the following results hold for any $\mathbb{G}$-equivariant/invariant $\boldsymbol{h} \in \mathcal{L}_{P_y}^2(\mathcal{Y}, \mathcal{Z})$, measurable set $\mathbb{A} \subset \mathcal{X}$, and $\mathbb{G}' \leq \mathbb{G}$; with probability at least $1 - \delta$ w.r.t. an iid draw of $\mathbb{D}_N = \{(\mathbf{x}_n, \mathbf{y}_n) \sim P_{\mathbf{x}\mathbf{y}}\}_{n=1}^N$:*

$$\|\boldsymbol{z} - \hat{\boldsymbol{z}}_{\boldsymbol{\theta}}\|_{\mathcal{L}_{P_x}^2(\mathcal{X}, \mathcal{Z})} \lesssim \sqrt{\mathrm{Var}[\|\boldsymbol{h}(\mathbf{y})\|_{\mathcal{Z}}]} \cdot \xi_{\boldsymbol{\theta}} \quad and \quad (19a)$$

$$\|\boldsymbol{z}(\mathbb{A}) - \hat{\boldsymbol{z}}_{\boldsymbol{\theta}}(\mathbb{A})\|_{\mathcal{Z}} \lesssim \frac{\sqrt{1+(|\mathbb{G}'| - 1)\gamma_{\mathbb{G}'}(\mathbb{A})}\sqrt{\mathrm{Var}[\|\boldsymbol{h}(\mathbf{y})\|_{\mathcal{Z}}]}}{\sqrt{|\mathbb{G}'|\mathbb{P}(\mathbf{x}\in\mathbb{A})}} \cdot \xi_{\boldsymbol{\theta}} \quad (19b)$$

*where the modeling/approximation and finite-sample-estimation error are given by:*

$$\xi_{\boldsymbol{\theta}} := \mathcal{E}_{\boldsymbol{\theta}}^r + \frac{\ln(d_{\mathrm{iso}}/\delta)}{(d_{\mathrm{iso}}N)^{\frac{\alpha}{1+2\alpha}}}.$$

*Here, $\mathcal{E}_{\boldsymbol{\theta}}^r = \|\mathsf{E}_{\mathbf{y}|\mathbf{x}}{-}\mathsf{E}_{\boldsymbol{\theta}}\|_{op}$ denotes representation learning modeling error (12), $r = \sum_{k=1}^{n_{\mathrm{iso}}} d_k m_k$ defines the representation space dimension, with $(d_k, m_k)$ denoting dimension and multiplicity of the group's irreducible representation of type $k$, and $d_{\mathrm{iso}} := \sum_{k=1}^{n_{\mathrm{iso}}} d_k \geq n_{\mathrm{iso}}$ (see Fig. 2).*

*Proof.* $\mathbb{G}$-invariance of $P_{\mathbf{x}}$ and $P_{\mathbf{y}}$ allows us to control both bias (Thm. M.2) and variance (Prop. M.3) of $\hat{\boldsymbol{z}}_{\boldsymbol{\theta}}$. A simple balancing of $r$ yields the final bound on the error. $\square$

We conclude by highlighting key theoretical and practical implications of Thm. 5.1. Since pointwise guarantees are essential in many applications, Eq. (19b) provides a learning bound conditioned on measurable sets $\mathbb{A} \subseteq \mathcal{X}$, yielding the estimate $\boldsymbol{z}(\mathbb{A}) := \mathbb{E}_{\mathbf{y}}[\boldsymbol{h}(\mathbf{y})|\mathbf{x} \in \mathbb{A}] \approx \widehat{\mathbb{E}}_{\mathbf{x}}[\hat{\boldsymbol{z}}_{\boldsymbol{\theta}}(x)]/\widehat{\mathbb{E}}_{\mathbf{x}}[\mathbb{1}_{\mathbb{A}}(x)]$. In this setting, exploiting symmetry helps mitigate the bottlenecks of rare-event estimation. To capture this effect, we introduce the *symmetry index* of $\mathbb{A}$ w.r.t. $P_{\mathbf{x}}$, which quantifies the degree of symmetry of $\mathbb{A}$ and appears in (19b):

$$\gamma_{\mathbb{G}}(\mathbb{A}) = \frac{1}{|\mathbb{G}| - 1} \sum_{g \in \mathbb{G}\setminus\{e\}} \frac{\mathbb{P}(\mathbf{x} \in \mathbb{A} \cap g \triangleright_{\mathcal{X}} \mathbb{A})}{\mathbb{P}(\mathbf{x} \in \mathbb{A})}, \quad (20)$$

Observe that $\gamma_{\mathbb{G}}(\mathbb{A}) \in [0, 1]$ for any $\mathbb{A} \subseteq \mathcal{X}$, with $\gamma_{\mathbb{G}}(\mathbb{A}) = 1$ if $\mathbb{A}$ is $\mathbb{G}$-invariant (e.g., the vertical and horizontal reflection planes in Fig. 3), while $\gamma_{\mathbb{G}}(\mathbb{A}) = 0$ if $\mathbb{A}$ is a $\mathbb{G}$-asymmetric set, that is, if $g \triangleright_{\mathcal{X}} \mathbb{A} \cap \mathbb{A} = \emptyset$ for all $g \in \mathbb{G}$ (e.g., any set disjoint from the reflection planes in Fig. 3). In particular, the effective rarity of $\mathbf{x} \in \mathbb{A}$ is captured by $\gamma_{\mathbb{G}'}(\mathbb{A})$, yielding a maximal gain of $|\mathbb{G}|\mathbb{P}[\mathbf{x} \in \mathbb{A}] \gg \mathbb{P}[\mathbf{x} \in \mathbb{A}]$ when $\mathbb{A}$ is asymmetric.

Equivariant disentangled representations increase the effective sample size according to $N \ll n_{\mathrm{iso}}N \leq d_{\mathrm{iso}}N \leq |\mathbb{G}|N$. Thus, they provide not only the expected $n_{\mathrm{iso}}$-fold gain from disentanglement, but also the stronger boost induced by $d_{\mathrm{iso}} = \sum_{k=1}^{n_{\mathrm{iso}}} d_k$ (see Fig. 2, right). This improvement in the estimation-error term of the bound is achieved by exploiting the structural constraints of the singular spaces associated with the irreducible representations of the group that appear in the chosen representation space. Moreover, any non-trivial symmetry group sharpens the bound. In the absence of a symmetry prior, namely when $\mathbb{G} = \{e\}$ and $|\mathbb{G}| = n_{\mathrm{iso}} = d_{\mathrm{iso}} = 1$, our framework recovers the symmetry-agnostic baseline results of (Kostic et al., 2024b).

The remaining term in the bound is the representation-learning error $\mathcal{E}_{\boldsymbol{\theta}}^r$ inherited from NCP (Kostic et al., 2024b). We do not provide a full statistical characterization of this term, as doing so would require additional architecture- and regularity-dependent assumptions without changing the main qualitative conclusions. Instead, we complement the theory with synthetic experiments in which the true operator is known, allowing us to empirically assess the impact of exploiting symmetry on $\mathcal{E}_{\boldsymbol{\theta}}^r$ (see Figs. 6 and 14).

Finally, the parameter $\alpha$ quantifies problem regularity through the decay rate of the operator's singular values, $\sum_{i \in \mathbb{N}} \sigma_i^{1/\alpha} < \infty$, with special cases including finite-rank operators ($\alpha = \infty$), compact operators ($\alpha = 0$), trace class operators ($\alpha = 1$), and Hilbert-Schmidt operators ($\alpha = 1/2$, equivalent to $\kappa \in \mathcal{L}_{P_{\mathbf{x}} \times P_{\mathbf{y}}}^2(\mathcal{X} \times \mathcal{Y})$; see App. M). Thus, our results cover learning rates ranging from arbitrarily slow rates as $\alpha \to 0$ to fast rates $[d_{\mathrm{iso}}N]^{-1/2}$ as $\alpha \to \infty$.

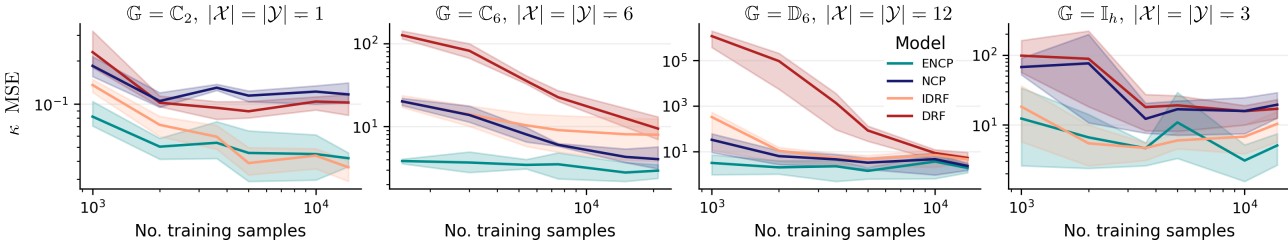

*Figure 4.* Sample-efficiency plots for the test-set PMD MSE $\kappa_{\mathrm{mse}} := \mathbb{E}_{\mathbf{x}}\mathbb{E}_{\mathbf{y}}(\kappa(\mathbf{x},\mathbf{y}) - \kappa_{\boldsymbol{\theta}}(\mathbf{x},\mathbf{y}))^2$ versus the number of training samples, on logarithmic scales. Each column corresponds to a symmetric cGMM with a different symmetry group and $(\mathbf{x},\mathbf{y})$ dimensionality. The tested groups are the cyclic groups $\mathbb{C}_2$ and $\mathbb{C}_6$, the dihedral group $\mathbb{D}_6$, and the icosahedral group $\mathbb{I}_h$ (order 60).

## 6. Experiments

We present three experiments evaluating our method in (i) approximating the conditional expectation operator and the use of the learned operator for (ii) $\mathbb{G}$-equivariant regression and (iii) symmetry-aware uncertainty quantification. For further details and additional experiments on synthetic regression, uncertainty quantification, symmetry misspecification and train/inference computational costs refer to App. G and the code repository[1]

**Conditional Expectation Operator Learning**   This experiment quantifies the Mean Squared Error (MSE) of approximating $\mathsf{E}_{\mathbf{y}|\mathbf{x}}$, i.e., $\kappa_{\mathrm{mse}} := \mathbb{E}_{\mathbf{x}}\mathbb{E}_{\mathbf{y}}(\kappa(\mathbf{x},\mathbf{y}) - \kappa_{\boldsymbol{\theta}}(\mathbf{x},\mathbf{y}))^2$. To achieve this, we extend the Conditional Gaussian Mixture Model (cGMM) of Gilardi et al. (2002) to parametrically construct symmetric vector-valued random variables $\mathbf{x} \in \mathcal{X}$ and $\mathbf{y} \in \mathcal{Y}$ that satisfy the symmetry priors in (6) for arbitrary finite symmetry groups (see example in Fig. 3). Each cGMM possess an analytical PMD, enabling direct $\kappa_{\mathrm{mse}}$ estimation, usually impossible for real-world datasets.

The results in Fig. 4 compare our eNCP against NCP (Kostic et al., 2024b), the baseline Density Ratio Fitting (DRF) (Tsai et al., 2020), and our Invariant Density Ratio Fitting (iDRF) adaptation. Note that the baselines approximate $\kappa$ as a single NN, $\kappa_{\boldsymbol{\theta}}^{\mathrm{drf}}: \mathcal{X} \times \mathcal{Y} \to \mathbb{R}^+$, trained via the contrastive loss in (5). Since they do not enforce the separable structure in (13), these methods cannot be used for regression or conditional probability estimation.

We evaluate performance across cGMMs with diverse symmetry groups and varying $(\mathbf{x},\mathbf{y})$ dimensions. Across all settings, eNCP achieves lower MSE and better sample efficiency than iDRF and the symmetry-agnostic models NCP and DRF (see Fig. 4). Moreover, Fig. 6 shows that the symmetry-agnostic models (NCP and DRF) struggle to recover the $\mathbb{G}$-invariance of the true PMD from data alone. In contrast, eNCP and iDRF explicitly enforce the invariance constraint in Eq. (12) through architectural NN constraints, preserving the desired $\mathbb{G}$-invariance throughout training up to numerical precision. These results show that eNCP accurately approximates the conditional expectation operator, consistently achieving lower empirical representation-

learning errors $\mathcal{E}_{\boldsymbol{\theta}}^r$ than all other models. This underscores the critical role of symmetry in improving both accuracy and sample efficiency.

$\mathbb{G}$-**Equivariant Regression**   To test our model's potential for performing $\mathbb{G}$-equivariant regression, we address the robot's Center of Mass (CoM) momenta regression task of (Ordoñez-Apraez et al., 2025). The goal is to predict a quadruped robot's CoM linear $\boldsymbol{l} \in \mathbb{R}^3$ and angular momenta $\boldsymbol{k} \in \mathbb{R}^3$ given the noisy observations of the robot's generalized positions $\boldsymbol{q} \in \mathbb{R}^{12}$ and velocity coordinates $\dot{\boldsymbol{q}} \in \mathbb{R}^{12}$, i.e., $[\boldsymbol{l}^\top, \boldsymbol{k}^\top]^\top = h_{\mathrm{CoM}}(\boldsymbol{q} + \epsilon_{\boldsymbol{q}}, \dot{\boldsymbol{q}} + \epsilon_{\dot{\boldsymbol{q}}})$ (see details in App. G.2 and Fig. 7 ). We compare eNCP against NCP and two baselines—a standard Multi-Layer Perceptron (MLP) and an Equivariant MLP (eMLP)—all with equivalent architectural footprint. The NCP and eNCP are trained using (5) and (14), respectively, while MLP and eMLP are trained using standard MSE.

The results in Fig. 1 demonstrate that our eNCP model outperforms all other baselines in both performance and sample complexity. Consistent with (Kostic et al., 2024b), the NCP model shows poorer sample complexity than MLP and eMLP due to its indirect approach to regression, via approximation of $\mathsf{E}_{\mathbf{y}|\mathbf{x}}$. However, by incorporating symmetry priors, eNCP appear to mitigate this limitation.

**Symmetry-Aware Uncertainty Quantification**   Finally, we demonstrate the practical impact of our approach on a core robotics problem: providing robust uncertainty quantification for unavailable yet crucial state observables for robot control and state estimation (Bledt et al., 2018; Maravgakis et al., 2023). Specifically, we use proprioceptive sensor readings to provide 90% CIs for the robot's Ground Reaction Forces (GRF) $\boldsymbol{\tau}_{\mathrm{grf}} \in \mathbb{R}^{12}$, the instantaneous work exerted or subtracted to the robot $U(\boldsymbol{q}, \dot{\boldsymbol{q}}, \boldsymbol{\tau}) \in \mathbb{R}$, and the kinetic energy $T(\boldsymbol{q}, \dot{\boldsymbol{q}}) \in \mathbb{R}$, while the robot traverses rough terrain (see App. G.6). Reliable probabilistic estimates of these quantities are of crucial relevance for optimal control (Bledt et al., 2018), contact detection (Maravgakis et al., 2023), state estimation (Nisticò et al., 2025), and system identification (Gautier, 1997).

This task tests our model's ability to learn conditional dis-

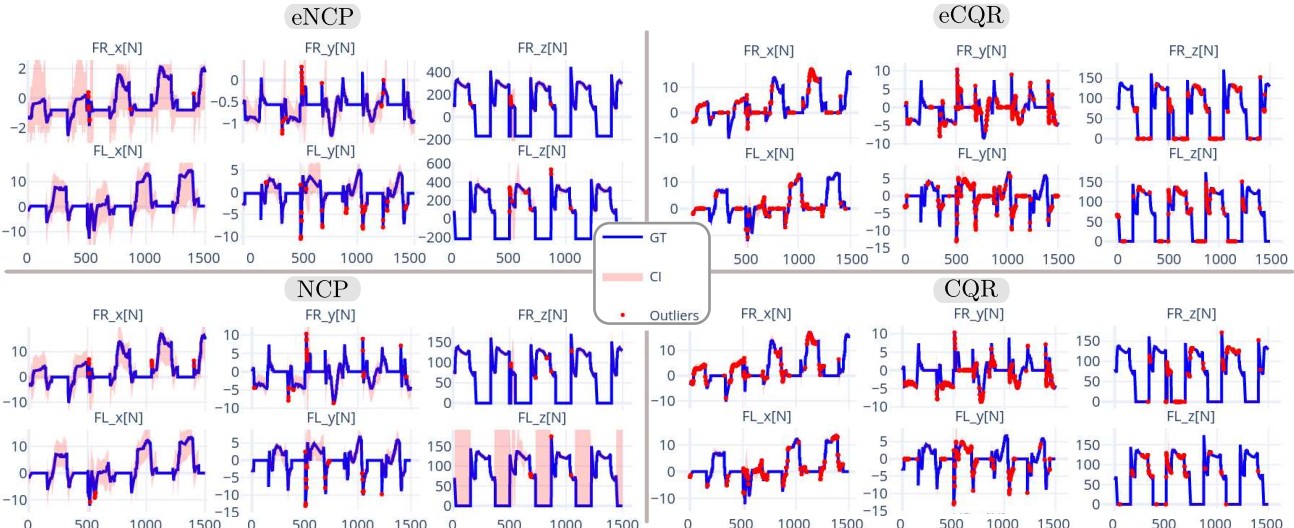

*Figure 5.* Prediction of 90% CIs (light red areas) for ground-reaction forces $\boldsymbol{\tau}_{\text{grf}} \in \mathbb{R}^{12}$ of a quadruped robot on rough terrain with varying friction. We compare eNCP, NCP, eCQR, and CQR models. CIs are computed along $x$, $y$, $z$ compoents of the forces on the front-right (FR) and front-left (FL) legs (see all legs predictions in Fig. 19). The trajectory of ground reaction forces in time is shown in blue lines, while sensed values outside of the predicted CIs are plotted in red markers.

tributions from high-dimensional data, considering that for the eNCP and NCP models, quantile estimation is done by regressing the conditional Cumulative Distribution Function (cCDF) for each dimension of $\mathbf{y} = [\mathbf{y}_1, \dots]$ and then applying a linear search to extract quantiles (see Figs. 10 and 14). This is achieved by discretizing the range of each $\mathbf{y}_i$ into $N_b$ bins and estimating $\mathbb{P}(\mathbf{y}_i \in \mathbb{A}_{i,n} | \mathbf{x} = \cdot) := [\mathsf{E}_{\mathbf{y}|\mathbf{x}} \mathbb{1}_{\mathbb{A}_{i,n}}](\cdot)$ for all $n \in [N_b]$ (see Sec. 5), where $\mathbb{A}_{i,n}$ consists of the first $n$ bins. In practice, this means regressing $|\mathcal{Y}| \times N_b$ conditional probabilities corresponding to sets of varying sizes in a *single forward pass* (see details in App. G.4). By contrast, the baseline Conditional Quantile Regression (CQR) (Feldman et al., 2023) and its equivariant adaptation Equivariant CQR (eCQR) directly regress quantiles for a fixed coverage level (i.e., the probability that an event lies within the predicted confidence interval) and need retraining for different coverage values.

The results, shown in Tab. 2 and Fig. 1 for work $U$ and kinetic energy $T$, and in Fig. 5 for the ground reaction forces $\boldsymbol{\tau}_{\text{grf}}$, identify eNCP as the only model that provides robust uncertainty quantification under both transient disturbances and nominal locomotion conditions. It is also the only

| | r-Coverage ↑ | Coverage ↑ | Set-size ↓ |
|---|---|---|---|
| eNCP | 99.5±0.1% | 95.0±0.4% | 4.3±3.6×10⁹ |
| NCP | 99.5±0.0% | 56.9±0.3% | 2.6±1.4×10¹⁰ |
| eCQR | 84.2±0.7% | 6.7±1.2% | 1.7±1.7×10⁷ |
| CQR | 80.5±3.7% | 8.5±0.9% | 1.4±0.1×10⁸ |

*Table 2.* Relaxed coverage (29), coverage (28), and set-size (30) for predicted CIs on the test set of the quadruped locomotion uncertainty-estimation task with $\mathbf{y} = [\boldsymbol{\tau}_{\text{grf}}^\top, U, T]^\top$. The target coverage is 90%. See all metrics in Tab. 4.

model whose empirical test-set coverage remains close to the target value, rendering the other models unreliable in practice. These results further highlight eNCP's potential for conditional probability estimation.

## 7. Conclusions

We introduce a novel framework for equivariant contrastive representation learning that enables equivariant regression, symmetry-aware uncertainty quantification, and conditional probability estimation with non-asymptotic statistical learning guarantees. Building on a recent contrastive representation learning method for approximating the spectral decomposition of the conditional expectation operator, our approach incorporates symmetry priors to impose additional structural constraints on the operator approximation, yielding a disentangled representation that admits stronger symmetry-aware statistical guarantees. We demonstrate the framework's benefits through theoretical learning bounds and empirical evaluations on robotics applications. Notably, we provide the first learning guarantees for equivariant regression and uncertainty quantification using neural network features, bridging spectral representation learning and geometric deep learning.

**Limitations and Future Work** Our method relies on fully specified symmetry priors, a natural assumption in GDL; however, some applications may have only partial or misspecified symmetries. Future work could accommodate partial or uncertain symmetry information and statistically test for its presence in data.

## Impact Statement

This paper presents work whose goal is to advance the field of Machine Learning. There are many potential societal consequences of our work, none which we feel must be specifically highlighted here.

## Acknowledgements

The work of DOA, VRK, AF and MP was supported by the EU Project ELIAS (grant No. 101120237), and by the European Union – NextGenerationEU and the Italian National Recovery and Resilience Plan through the Ministry of University and Research (MUR), under Project PE0000013 CUP J53C22003010006. KL acknowledges support from ELIAS and the French National Research Agency for the DECATTLON project (ANR-24-CE40-3341).

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

# Appendix

## Table of Contents

The appendix is organized as follows.

- App. A summarizes the notations used, while App. A provides a glossary.

- App. C offers a detailed discussion of related work on contrastive learning, equivariant representations, and statistical learning theory with symmetry priors.

- App. D-App. G detail our methodological contributions. In particular, App. E and App. F explain how our method leverages equivariant neural networks, and App. G complements Sec. 6 with additional experimental details and studies.

- We provide self-contained sections with unified notation covering essential preliminaries: group theory (App. I), representation theory in function spaces (App. J), equivariant linear operators (App. K), and the symmetries of covariance operators central to our work (App. L).

- Finally, App. M presents our theoretical contributions, summarized in Theorem Thm. 5.1. We first establish approximation error bounds using operator theory and equivariant representations, then combine group theory with concentration inequalities to derive estimation error bounds that fully expose the benefits of symmetry priors.

# A. Symbols and Notation

## Numbers and Arrays

| | |
|---|---|
| $x$ | A scalar, or scalar function $x(\cdot)$ |
| $\boldsymbol{x}$ | A vector, or vector-valued function $\boldsymbol{x}(\cdot)$ |
| $\boldsymbol{x}_1 \oplus \boldsymbol{x}_2$ | Direct sum (stacking) of vectors, such that $\boldsymbol{x}_1 \oplus \boldsymbol{x}_2 := \left[\begin{smallmatrix}\boldsymbol{x}_1\\\boldsymbol{x}_2\end{smallmatrix}\right]$ |
| $\boldsymbol{K}$ | A matrix |
| $\boldsymbol{A} \oplus \boldsymbol{B}$ | Direct sum of matrices, such that $\boldsymbol{A} \oplus \boldsymbol{B} := \left[\begin{smallmatrix}\boldsymbol{A} & \boldsymbol{O}\\\boldsymbol{O} & \boldsymbol{B}\end{smallmatrix}\right]$ |
| $\mathsf{K}$ | A linear operator |
| $\boldsymbol{I}$ | Identity matrix |
| $\delta_{i,j}$ | The Kronecker function, equal to 1 when $i = j$, and 0 when $i \neq j$ |

## Sets, Vector Spaces, and Function Spaces

| | |
|---|---|
| $\mathcal{X}, \mathcal{Z}, \mathcal{H}, \mathcal{F}$ | A vector or Hilbert space |
| $\bar{\mathcal{X}}, \bar{\mathcal{Z}}, \bar{\mathcal{V}}$ | An irreducible $\mathbb{G}$-stable space (Def. I.7) |
| $\mathbb{R}, \mathbb{C}$ | The set of real and complex numbers |
| $\mathcal{X} \oplus \mathcal{Y}$ | Direct sum of vector spaces $\mathcal{X}$ and $\mathcal{Y}$, such that if $\boldsymbol{x} \in \mathcal{X}$ and $\boldsymbol{y} \in \mathcal{Y}$, then $\boldsymbol{x} \oplus \boldsymbol{y} \in \mathcal{X} \oplus \mathcal{Y}$ |
| $\mathcal{L}_\mathbf{x}^2 := \mathcal{L}_{P_\mathbf{x}}^2(\mathcal{X}, \mathbb{R})$ | The space of square-integrable functions on $\mathcal{X}$. That is $\{f \mid \int_\mathcal{X} \lvert f(\boldsymbol{x})\rvert^2 P_\mathbf{x}(d\boldsymbol{x}) < \infty, \ f : \mathcal{X} \to \mathbb{R}\}$ |
| $\langle f, f'\rangle_{P_\mathbf{x}}$ | Inner product $\langle f, f'\rangle_{P_\mathbf{x}} := \int_\mathcal{X} f(\boldsymbol{x})f'(\boldsymbol{x})P_\mathbf{x}(d\boldsymbol{x})$ |

## Group and Representation theory

| | |
|---|---|
| $\mathbb{G}$ | A symmetry group |
| $g, g_1, g_a$ | A symmetry group element |
| $g \triangleright \boldsymbol{x}$ | The (left) group action of $g$ on $\boldsymbol{x}$ defined by $g \triangleright \boldsymbol{x} := \boldsymbol{\rho}_\mathcal{X}(g)\mathcal{X}$ |
| $\boldsymbol{\rho}_\mathcal{X}$ | A representation of the group $\mathbb{G}$ on the vector space $\mathcal{X}$, defined for a chosen basis of $\mathcal{X}$ |
| $\bar{\boldsymbol{\rho}}_k$ | An irreducible representation (Def. I.9) of the group $\mathbb{G}$ |
| $d_k := \lvert\bar{\boldsymbol{\rho}}_k\rvert$ | Dimensionality of the irreducible representation $\bar{\boldsymbol{\rho}}_k$ (see Fig. 2) |
| $m_k$ | Multiplicity of the irreducible representation $\bar{\boldsymbol{\rho}}_k$ in a given larger representation (see Fig. 2) |
| $n_{\mathrm{iso}}$ | Number of distinct irreps present in a given larger representation. |
| $\boldsymbol{\rho}_\mathcal{X}(g)$ | Representation of the group element $g$ on the vector space $\mathcal{X}$ |
| $\boldsymbol{\rho}_\mathcal{X} \oplus \boldsymbol{\rho}_\mathcal{Y}$ | Direct sum of group representations, such that $\boldsymbol{\rho}_\mathcal{X}(g) \oplus \boldsymbol{\rho}_\mathcal{Y}(g) := \left[\begin{smallmatrix}\boldsymbol{\rho}_\mathcal{X}(g) & \\ & \boldsymbol{\rho}_\mathcal{Y}(g)\end{smallmatrix}\right]$ |
| $\mathbb{G}\boldsymbol{x}$ | The group orbit of $\boldsymbol{x}$, defined as $\mathbb{G}\boldsymbol{x} := \{g \triangleright \boldsymbol{x} \mid g \in \mathbb{G}\}$ |
| $\gamma_{\mathbb{G}'}(A)$ | The symmetry index of a set $A \subseteq \mathcal{X}$ w.r.t. probability distribution on $\mathcal{X}$ and group elements $\mathbb{G}' \subseteq \mathbb{G}$ |
| $\mathbb{G}_a \times \mathbb{G}_b$ | Direct product of groups $\mathbb{G}_a$ and $\mathbb{G}_b$ |
| $\mathbb{U}(\mathcal{X})$ | Unitary group on the vector space $\mathcal{X}$ |
| $\mathbb{GL}(\mathcal{X})$ | General Linear group on the vector space $\mathcal{X}$, a.k.a the space of invertible matrices in $\mathbb{R}^{\lvert\mathcal{X}\rvert \times \lvert\mathcal{X}\rvert}$ |
| $\mathbb{C}_n$ | Cyclic group of order $n$ |
| $\mathbb{K}_4$ | Klein four-group |

## Probability Theory

| | |
|---|---|
| $\mathbf{x} \sim \mathbb{P}(\mathbf{x})$ | Random vector $\mathbf{x} \in \mathcal{X}$ has distribution $\mathbb{P}(\mathbf{x})$ |
| $P_\mathbf{x}$ | A probability measure on the space $\mathcal{X}$ |
| $\mathbb{E}_\mathbf{x}[f(\mathbf{x})]$ | Expectation of $f(\mathbf{x})$ with respect to $P_\mathbf{x}$ |
| $\mathrm{Cov}(f(\mathbf{x}))$ | Variance of $f(\mathbf{x})$ with respect to $P_\mathbf{x}$, define as $\mathbb{E}_\mathbf{x}(f(\mathbf{x}) - \mathbb{E}_\mathbf{x}f(\mathbf{x}))^2$ |
| $\mathrm{Cov}(f(\mathbf{x}), h(\mathbf{y}))$ | Covariance of $f(\mathbf{x})$ and $h(\mathbf{y})$ with respect to the joint distribution $P_{\mathbf{xy}}$, defined as $\mathbb{E}_{\mathbf{xy}}(f(\mathbf{x}) - \mathbb{E}_\mathbf{x}f(\mathbf{x}))(h(\mathbf{y}) - \mathbb{E}_\mathbf{y}h(\mathbf{y}))$ |
| $\mathcal{N}(\boldsymbol{x}; \boldsymbol{\mu}, \boldsymbol{\Sigma})$ | Gaussian distribution over $\boldsymbol{x}$ with mean $\boldsymbol{\mu}$ and covariance $\boldsymbol{\Sigma}$ |

# B. Acronyms

**cCDF** conditional Cumulative Distribution Function.

**cGMM** Conditional Gaussian Mixture Model (Gilardi et al., 2002): A parametric model for benchmark conditional density estimation datasets. Generates random variables $\mathbf{x}$ and $\mathbf{y}$ of arbitrary dimensions with varying mutual information. Enables analytical computation of the PMD density ratio (see Sec. 2), unavailable in real-world datasets, allowing direct quantification of approximation error for the conditional expectation operator $\mathsf{E}_{\mathbf{y}|\mathbf{x}}$ and its PMD density ratio .

**CI** Confidence Interval.

**CLoRa** Contrastive Low-Rank loss from (Kostic et al., 2024b; Ryu et al., 2024) for operator and representation learning. Used in density-ratio fitting (Sugiyama et al., 2012), representation learning (Wang et al., 2022b; HaoChen et al., 2021), and mutual information estimation (Tsai et al., 2020) .

**CNN** Convolutional Neural Network.

**CoM** Center of Mass.

**CQR** Conditional Quantile Regression (Feldman et al., 2023): A multivariate neural network approach that regresses upper and lower quantiles at a miscoverage level $\alpha$ using the pinball loss. Although CQR typically uses post-hoc conformal calibration, we omit it here for a fair comparison, as we evaluate the quality of the parametric estimates themselves, and conformal calibration is model-agnostic and applicable to eNCP as well .

**DoF** degree of freedom.

**DRF** Density Ratio Fitting (Tsai et al., 2020): A density ratio NN architecture that parameterizes the approximated PMD $\kappa_{\boldsymbol{\theta}} : \mathcal{X} \times \mathcal{Y} \to \mathbb{R}_+$ as a single NN. Consequently, this model cannot be used for downstream conditional probability estimation and regression—it is limited to estimating the mutual information between $\mathbf{x}$ and $\mathbf{y}$ (Tsai et al., 2020) .

**eCQR** Version of eCQR where the upper and lower parametric quantile functions are parameterized by $\mathbb{G}$-equivariant NNs .

**eMLP** Equivariant MLP.

**eNCP** Equivariant Neural Conditional Probability: Our proposed model integrating the symmetry priors Eq. (6) into the NCP deep representation learning algorithm .

**GDL** Geometric Deep Learning: A field of machine learning that incorporates geometric priors into deep learning models (Bronstein et al., 2021).

**GNN** Graph Neural Network.

**GRF** Ground Reaction Forces.

**iDRF** Invariant Density Ratio Fitting: This is a $\mathbb{G}$-invariant adaptation of the DRF model (Tsai et al., 2020) that parameterizes the approximated PMD $\kappa_{\boldsymbol{\theta}}$ as a $\mathbb{G}$-invariant NN .

**MLP** Multi-Layer Perceptron.

**MSE** Mean Squared Error.

**NCP** Neural Conditional Probability: A deep representation learning framework (Kostic et al., 2024b) for conditional probability estimation and regression with statistical guarantees via operator theory (Baker, 1973). This framework is symmetry-agnostic .

**NCPaug**  NCP with data augmentation.

**NN**  Neural Network.

**PMD**  Pointwise Mutual Dependency (Tsai et al., 2020): A pointwise dependency measure between random variables $\mathbf{x}$ and $\mathbf{y}$, defined as $\kappa(\boldsymbol{x}, \boldsymbol{y}) = \frac{dP_{\mathbf{xy}}(\boldsymbol{x}, \boldsymbol{y})}{d\left(P_{\mathbf{x}}(\boldsymbol{x}) \times P_{\mathbf{y}}(\boldsymbol{y})\right)} = \exp\left(\text{MI}(\boldsymbol{x}, \boldsymbol{y})\right)$ .

**SVD**  Singular Value Decomposition.

## C. Related Work

### C.1. Contrastive Representation Learning

Contrastive representation learning obtains high-dimensional representations from unlabeled data by contrasting positive and negative sample pairs via a noise contrastive loss (similar to Eq. (5)) (Le-Khac et al., 2020; Waida et al., 2023; Bao et al., 2022). Most works in this field aim to learn representations in a self-supervised fashion that transfer well to downstream classification tasks (Johnson et al., 2023; Cole et al., 2022; Tosh et al., 2021; Oord et al., 2018; Tsai et al., 2021; HaoChen et al., 2021). In contrast, our approach targets representations that effectively transfer to (equivariant) regression and uncertainty quantification, as in (Kostic et al., 2024a). Given a dataset $\mathbb{D} = \{(\boldsymbol{x}_n, \boldsymbol{y}_n)\}_{n=1}^N$ from a target (stochastic) function $\mathbf{y} = \boldsymbol{f}(\mathbf{x})$, we treat positive pairs as drawn from the joint distribution $(\boldsymbol{x}, \boldsymbol{y}) \sim \mathbb{P}(\mathbf{x}, \mathbf{y})$ and negative pairs as drawn from the product of the marginals $(\boldsymbol{x}, \boldsymbol{y}) \sim \mathbb{P}(\mathbf{x})\mathbb{P}(\mathbf{y})$. In this setting, our contrastive loss aims to learn representations that approximate the PMD ratio $\kappa(\boldsymbol{x}, \boldsymbol{y}) = \frac{\mathbb{P}(\boldsymbol{x}, \boldsymbol{y})}{\mathbb{P}(\boldsymbol{x})\mathbb{P}(\boldsymbol{y})}$, (Kostic et al., 2024a) or equivalently, the pointwise mutual information $\ln(\kappa(\boldsymbol{x}, \boldsymbol{y}))$ (Oord et al., 2018; Lin et al., 2024; Henaff, 2020; Ozair et al., 2019). Crucially, our work is the first study this problem when there is prior knowledge of the invariance of $\kappa$ under the action of a compact symmetry group, which occurs in most applications of GDL.

**Linear Transferability**  The goal of contrastive representation learning is to acquire representations that transfer to diverse downstream inference tasks (Bengio et al., 2013; Waida et al., 2023). While empirical studies demonstrate that contrastive learning can outperform supervised methods (Cole et al., 2022; Oord et al., 2018; Henaff, 2020), theoretical works aim to establish *linear separability/transferability* guarantees (HaoChen et al., 2022) [4]. That is, showing that linear functionals of the (frozen) learned representations suffice for regression/classification inference.

In the context of **classification**, (Waida et al., 2023; Bao et al., 2022; Johnson et al., 2023) show that contrastive learning losses serve as surrogates for standard supervised classification losses (e.g., the cross-entropy). Where the gap between the surrogate and supervised loss diminishes with the number of negative samples (Bao et al., 2022) ($N^2$ for the loss in Eq. (16)). To provide these transferability guarantees, these work assume $\mathcal{X} = \mathcal{Y}$, so that the PMD ratio $\kappa$ becomes a positive definite kernel. Consequently, kernel method guarantees can be transferred to the classification task, even when the representations are parameterized by NNs (Johnson et al., 2023; Bao et al., 2022; HaoChen et al., 2022).

Considerably fewer works have studied contrastive representation learning in the context of downstream **regression** tasks (Yerxa et al., 2024; Kostic et al., 2024a). Crucially, Kostic et al. (2024a) show that a contrastive learning loss serves as surrogate to the MSE regression loss (A summary of this method appears in Sec. 2 and in Tab. 3). While, to the best of our knowledge, (Yerxa et al., 2024) is the only work empirically studying contrastive learning for regression in the presence of symmetries.

### C.2. Equivariant Representation Learning

Equivariant contrastive representation learning (Dangovski et al., 2022; Wang et al., 2024b) aims to learn representations that are equivariant—instead of invariant—to data transformations. For example, Marchetti et al. (2023); Gupta et al. (2023); Lin et al. (2024) provide empirical evidence that representations of 3D scenes, images, and human body poses that are equivariant to translations, rotations, or reflections yield improved performance in *classification* tasks. Additionally, Yerxa et al. (2024) show that rotation- and reflection-aware image representations enhance the *regression* of neural responses in the macaque inferior temporal cortex, while also providing theoretical justification that such equivariant representations

---

[4]Also refeered to as linear evaluation protocol by Chen et al. (2020)

| Task | $\boldsymbol{f}(\boldsymbol{x}) := \mathbb{E}_{\mathbf{y}}[\mathbf{y}|\mathbf{x}{=}\boldsymbol{x}] \approx \hat{\boldsymbol{f}}_{\boldsymbol{\theta}}(\boldsymbol{x})$ | $\mathbb{P}[\mathbf{y}{\in}\mathbb{B}|\mathbf{x}{\in}\mathbb{A}] \approx \widehat{\mathbb{P}}_{\boldsymbol{\theta}}[\mathbf{y}{\in}\mathbb{B}|\mathbf{x}{\in}\mathbb{A}]$ |
|---|---|---|
| Estimate | $\widehat{\mathbb{E}}_{\mathbf{y}}[\mathbf{y}] + \boldsymbol{\phi}_{\boldsymbol{\theta}}(\boldsymbol{x})^\top \boldsymbol{E}_{\boldsymbol{\theta}}\widehat{\mathbb{E}}_{\mathbf{y}}[\boldsymbol{\psi}_{\boldsymbol{\theta}}(\mathbf{y})\otimes\mathbf{y}]$ | $\widehat{\mathbb{E}}_{\mathbf{y}}[\mathbb{1}_{\mathbb{B}}] + \dfrac{\widehat{\mathbb{E}}_{\mathbf{x}}[\mathbb{1}_{\mathbb{A}}(\mathbf{x})\otimes\boldsymbol{\phi}_{\boldsymbol{\theta}}(\mathbf{x})]^\top \boldsymbol{E}_{\boldsymbol{\theta}}\widehat{\mathbb{E}}_{\mathbf{y}}[\mathbb{1}_{\mathbb{B}}(\mathbf{y})\otimes\boldsymbol{\psi}_{\boldsymbol{\theta}}(\mathbf{y})]}{\widehat{\mathbb{E}}_{\mathbf{x}}[\mathbb{1}_{\mathbb{A}}(\mathbf{x})]}$ |
| Guarantees | $\|\boldsymbol{f}-\hat{\boldsymbol{f}}_{\boldsymbol{\theta}}\|_{\mathcal{L}^2_{\mathbf{x}}} \lesssim \sqrt{\mathrm{Var}[\|\mathbf{y}\|]}\left(\mathcal{E}^r_{\boldsymbol{\theta}} + \frac{\ln(1/\delta)}{N^{\frac{\alpha}{1+2\alpha}}}\right)$ | $|\mathbb{P}-\widehat{\mathbb{P}}_{\boldsymbol{\theta}}| \lesssim \sqrt{\frac{\mathbb{P}[\mathbf{y}\in\mathbb{B}]}{\mathbb{P}[\mathbf{x}\in\mathbb{A}]}}\left(\mathcal{E}^r_{\boldsymbol{\theta}} + \frac{\ln(1/\delta)}{N^{\frac{\alpha}{1+2\alpha}}}\right)$ |

*Table 3.* Statistical learning guarantees of NCP (Kostic et al., 2024a) for regression and conditional probability estimation. The bounds are shaped by the quality of the learned representations $\mathcal{E}^r_{\boldsymbol{\theta}} = \|\mathsf{E}_{\mathbf{y}|\mathbf{x}} - \mathsf{E}_{\boldsymbol{\theta}}\|_{\mathrm{op}} \leq \sqrt{\mathcal{L}_\gamma(\boldsymbol{\theta}) - \mathcal{L}_\gamma(\star)}$ (see (5)), the sample size $N$, and the decay rate of $\mathsf{E}_{\mathbf{y}|\mathbf{x}}$ singular-values $\alpha > 0$, which quantifies the difficulty of the problem.

mirror the known structure of animal visual perception. By introducing these transformations via data-augmentation of the training set, these methods inherently enforce symmetries in the data distributions, which are the fundamental priors assumed in Sec. 3.

**Disentangled Representations**   In equivariant representation learning, disentangled representations have been extensively studied (Wang et al., 2024a). Initially, (Bengio et al., 2013) defined disentanglement as decomposing representations into components that capture distinct, independently varying factors. Later, using group theory, Higgins et al. (2018) formalized that a representation is disentangled if its space decomposes into orthogonal subspaces reflecting a symmetry group decomposition, with each subspace influenced exclusively by one quotient group (see Def. I.19). As discussed in App. I, this aligns with the isotypic decomposition of a Hilbert space (Mackey, 1980): $\mathcal{H} = \oplus^{\perp}_{k=1} \mathcal{H}^{(k)}$—known in dynamical systems (Golubitsky et al., 2012)—when the symmetry group decomposes as $\mathbb{G} = \prod^{n_{\mathrm{iso}}}_{k=1} \mathbb{G}^{(k)}$. Orthogonality between subspaces follows from Schur's orthogonality relations via Cartan's and Peter-Weyl's theorems (Cartan, 1952). This symmetric structure is the cause of the achitectural constraints imposed in the eNCP architecture Fig. 2.

Several empirical works have explored disentanglement in representation learning. For instance, Keurti et al. (2023) proposed an autoencoder-based method to learn disentangled equivariant representations by using loss regularization to enforce latent space equivariance and sparsity for separating latent group actions. Unlike our approach, their method does not assume prior knowledge of the symmetry group and relies entirely on loss regularization rather than architectural constraints. Similarly, works such as Yang et al. (2023); Dangovski et al. (2022) have investigated various symmetry priors in latent space by examining the emergence of disentangled structures and enforcing algebraic constraints. Notably, in fields like molecular dynamics, physics, computer graphics, and robotics, symmetry priors are intrinsic to the task or system (Ordoñez-Apraez et al., 2025; Lin et al., 2024; Marchetti et al., 2023), making them natural assumptions. In a similar spirit to our work, Marchetti et al. (2023) leverage the known $\mathbb{SO}_3$ symmetries of the 3D world to learn $\mathbb{SO}_3$-disentangled equivariant representations using contrastive learning, thereby demonstrating the empirical advantages of symmetry-aware, disentangled representations for object classification.

### C.3. Symmetry-Aware Statistical Learning Theory

Existing literature on symmetry-aware learning focuses on group-invariant regression via kernel methods (Mroueh et al., 2015; Tahmasebi & Jegelka, 2023; Elesedy, 2021; Elesedy & Zaidi, 2021; Elesedy, 2023; Pal et al., 2017; Mei et al., 2021; Bietti et al., 2021; Donhauser et al., 2021). Most of these methods cannot be directly transferred to modern GDL architectures.

In contrast, in deep learning and GDL, while many works offer a group-theoretical analysis and empirical evidence of the benefits of incorporating symmetry priors (Kashinath et al., 2021; Wang et al., 2021; 2022a; Brandstetter et al., 2023), none, to our knowledge, provide statistical learning guarantees for equivariant **regression**. The only exception is (Behboodi et al., 2022), which derives generalization bounds for a MLP architecture in the context of $n$-class **classification** task using a margin loss. In contrast, our work provides statistical learning guarantees for equivariant **regression** and symmetry-aware **uncertainty quantification**, both as corollaries of Thm. 5.1.

## D. Symmetry Constraints on Conditional Expectations

Under the assumed symmetry priors in (6) the conditional expectation of $\mathbf{y}$ is a $\mathbb{G}$-equivariant function/map. This property is depicted in Fig. 3-center and proved in the following proposition.

**Proposition D.1** ($\mathbb{G}$-equivariant conditional expectations). *Let* $\mathbf{x} \in \mathcal{X}$ *and* $\mathbf{y} \in \mathcal{Y}$ *be two vector valued random variables*

*satisfying the symmetry priors of Eq. (6). Then, the conditional expectation of* $\mathbf{y}$ *given* $\mathbf{x}$ *is* $\mathbb{G}$*-equivariant, since, for every* $g \in \mathbb{G}, \boldsymbol{x} \in \mathcal{X}$,

$$
\begin{aligned}
\mathbb{E}[\mathbf{y}|\mathbf{x} = g \rhd_{\mathcal{X}} \boldsymbol{x}] &= g \rhd_{\mathcal{Y}} \mathbb{E}[\mathbf{y}|\mathbf{x} = \boldsymbol{x}] \\
&= \int_{\mathcal{Y}} g \rhd_{\mathcal{Y}} \boldsymbol{y} \; P_{\mathbf{y}|\mathbf{x}}(d\boldsymbol{y}|\boldsymbol{x}) = \int_{g^{-1} \rhd_{\mathcal{Y}} \mathcal{Y}} \boldsymbol{y} \; P_{\mathbf{y}|\mathbf{x}}\left(g^{-1} \rhd_{\mathcal{Y}} d\boldsymbol{y}|\boldsymbol{x}\right) \\
&= \int_{\mathcal{Y}} \boldsymbol{y} \; P_{\mathbf{y}|\mathbf{x}}(d\boldsymbol{y}|g \rhd_{\mathcal{X}} \boldsymbol{x}) \quad \textit{(by Eq. (6))} \\
&= \mathbb{E}[\mathbf{y}|\mathbf{x} = g \rhd_{\mathcal{X}} \boldsymbol{x}].
\end{aligned}
$$

## E. $\mathbb{G}$-Equivariant Bilinear NN Architecture

This section outlines how to construct a $\mathbb{G}$-equivariant disentangled representation for the random variables $\mathbf{x}$ and $\mathbf{y}$ using **any** type of $\mathbb{G}$-equivariant NN architecture backbone, such as MLP, CNNs, Transformers, and others.

Let $\boldsymbol{f_\theta} : \mathcal{X} \mapsto \mathbb{R}^r$ and $\boldsymbol{h_\theta} : \mathcal{Y} \mapsto \mathbb{R}^r$ be two $\mathbb{G}$-equivariant NNs, whose outputs will be interpreted as the basis functions of the truncated symmetric function spaces $\mathcal{F}_{\mathbf{x}} \subset \mathcal{L}_{\mathbf{x}}^2$ and $\mathcal{F}_{\mathbf{y}} \subset \mathcal{L}_{\mathbf{y}}^2$. Assume, the group representations on $\mathcal{F}_{\mathbf{x}}$ and $\mathcal{F}_{\mathbf{y}}$ are constructed from multiplicities of the group's regular representation, $\boldsymbol{\rho}_{\mathcal{F}_{\mathbf{x}}} = \bigoplus_{n=1}^{r/|\mathbb{G}|} \boldsymbol{\rho}_{\mathrm{reg}}$ and $\boldsymbol{\rho}_{\mathcal{F}_{\mathbf{y}}} = \bigoplus_{n=1}^{r/|\mathbb{G}|} \boldsymbol{\rho}_{\mathrm{reg}}$—as done usually in practice (Weiler et al., 2023; Cesa et al., 2022; Kondor et al., 2018). Since for (most) finite groups, the decomposition of $\boldsymbol{\rho}_{\mathrm{reg}}$ into *irreps* is known or can be computed (Babai & Rónyai, 1990; Unger, 2006), we can safely assume access to the analytical change of basis $\boldsymbol{Q}_{\mathbf{x}} : \mathcal{F}_{\mathbf{x}} \mapsto \mathcal{F}_{\mathbf{x}}$ and $\boldsymbol{Q}_{\mathbf{y}} : \mathcal{F}_{\mathbf{y}} \mapsto \mathcal{F}_{\mathbf{y}}$ to transition to the isotypic basis. Consequently, we can directly parameterize the representations of the random variables in disentangled form as:

$$
\boldsymbol{\phi_\theta}(\cdot) = \boldsymbol{Q}_{\mathbf{x}}^\top (\boldsymbol{f_\theta}(\cdot) - \mathbb{E}_{\mathbf{x}}[\boldsymbol{f_\theta}(\mathbf{x})]), \quad \boldsymbol{\psi_\theta}(\cdot) = \boldsymbol{Q}_{\mathbf{y}}^\top (\boldsymbol{h_\theta}(\cdot) - \mathbb{E}_{\mathbf{y}}[\boldsymbol{h_\theta}(\mathbf{y})]). \tag{21}
$$

Given that during training these representations are not orthogonal, the truncated operator is parameterized as the trainable $\mathbb{G}$-equivariant matrix $\boldsymbol{E_\theta} = \bigoplus_k^{n_{\mathrm{iso}}} \boldsymbol{E}_{\boldsymbol{\theta}}^{(k)} = \bigoplus_k^{n_{\mathrm{iso}}} \sum_{b \in \mathbb{B}} \boldsymbol{\Theta}_b^{(k)} \otimes \Psi(b)$, where $\mathbb{B}$ is the endomorphism's basis of the irreducible representation $\bar{\boldsymbol{\rho}}_k$, $\Psi : \mathbb{B} \to \mathbb{R}^{d_k \times d_k}$ is a mapping from the endomorphism's basis to the endomorphism space, and $\boldsymbol{\Theta}_b^{(k)} \in \mathbb{R}^{m_k^y \times m_k^x}$ represent the block's trainable parameters for each $b \in \mathbb{B}$, see details in Prop. I.14.

Note that after training, the SVD of the learned operator can be computed by exploiting the constraints imposed by the operator's $\mathbb{G}$-equivariance (see Thm. K.4 and Fig. 2).

### E.1. Continuous and Non-Compact Symmetry Groups

**Compatibility with Continuous Compact Symmetry Groups** Formally, Peter–Weyl/Cartan's theorem—the engine behind our isotypic decomposition—applies to every compact symmetry group, covering both compact Lie groups and finite groups. The challenge with compact continuous groups is that they possess infinitely many irreducible representations (i.e., $n_{\mathrm{iso}} \to \infty$), so a finite dimensional space cannot be partitioned into infinitely many isotypic subspaces.

Traditional GDL techniques overcome this by discretising the group. For $\mathbb{SO}_3$-equivariance, for instance, architectures are often restricted to a finite subgroup (e.g., icosahedral or octahedral); see (Cesa et al., 2022; Weiler et al., 2023) for details. Crucially, such discretization produces a finite symmetry subgroup, to which our formalism applies unchanged.

**Compatibility with Non-Compact Symmetry Groups** The proposed representation disentanglement applies to **compact** continuous/finite symmetry groups, given that compactness is a fundamental requirement of the isotypic decomposition. However, equivariance to non-compact symmetry groups can still be achieved by choosing an appropriate NN architecture for parameterizing the embedding functions $\boldsymbol{\phi_\theta} : \mathcal{X} \to \mathbb{R}^r$ and $\boldsymbol{\psi_\theta} : \mathcal{Y} \to \mathbb{R}^r$.

For instance, as is commonly done for image processing in GDL—where images are assumed to possess 2D rotational *and translational symmetries* $\mathbb{G} = \mathbb{SO}_2 \rtimes \mathbb{T}_2$—equivariance to the non-compact translation group $\mathbb{T}_2$ is naturally encoded via a Convolutional Neural Network (CNN) architecture, which is by construction $\mathbb{T}_2$-equivariant. Constraining the filters of the CNN then ensures the compact $\mathbb{SO}_2$-equivariance. These architectures are usually referred to as $\mathbb{G}$-steerable CNNs ; see Cesa et al. (2022); Weiler et al. (2023) for details.

Given that our method only assumes the embedding functions $\boldsymbol{\phi_\theta} : \mathcal{X} \to \mathbb{R}^r$ and $\boldsymbol{\psi_\theta} : \mathcal{Y} \to \mathbb{R}^r$ are equivariant to a compact symmetry group, we can use *any* NN backbone architecture which could encode equivariance to non-compact groups, like

$\mathbb{T}_2$ in image/video processing or $\mathbb{T}_1$ in time series data. That is, our method can also be used to train $\mathbb{G}$-steerable CNNs representations.

## F. Symmetry-Aware Orthonormalization of Disentangled Representations

This section covers how to compute unbiased empirical estimates of the orthonormalization and centering regularization terms in Eq. (14) in the presence of symmetries.

Let $\mathsf{E}_{\mathbf{y}|\mathbf{x}} : \mathcal{L}_{\mathbf{y}}^2 \mapsto \mathcal{L}_{\mathbf{x}}^2$ be the conditional expectation operator and $\mathsf{E}_{\boldsymbol{\theta}} : \mathcal{F}_{\mathbf{y}} \mapsto \mathcal{F}_{\mathbf{x}}$ be its $r$-rank approximation on the spaces $\mathcal{F}_{\mathbf{x}} = \text{span}(\{\phi_i\}_{i=1}^r)$ and $\mathcal{F}_{\mathbf{y}} = \text{span}(\{\psi_i\}_{i=1}^r)$. Denote by $\kappa(\boldsymbol{x}, \boldsymbol{y}) := \frac{P_{\mathbf{xy}}(\boldsymbol{x}, \boldsymbol{y})}{P_{\mathbf{x}}(\boldsymbol{x})P_{\mathbf{y}}(\boldsymbol{y})}$ and $\kappa_{\boldsymbol{\theta}}(\boldsymbol{x}, \boldsymbol{y}) := \sum_{i,j=1}^r [\boldsymbol{E}_{\boldsymbol{\theta}}]_{i,j}\phi_i(\boldsymbol{x})\psi_j(\boldsymbol{y}) = \phi(\boldsymbol{x})^\top \boldsymbol{E}_{\boldsymbol{\theta}}\psi(\boldsymbol{y})$ the kernel functions of the full and restricted operator, respectively. Then we have that:

$$\|\mathsf{E}_{\mathbf{y}|\mathbf{x}} - \mathsf{E}_{\boldsymbol{\theta}}\|_{\text{HS}}^2 \leq -2\langle \mathsf{E}_{\mathbf{y}|\mathbf{x}}, \mathsf{E}_{\boldsymbol{\theta}}\rangle_{\text{HS}} + \|\mathsf{E}_{\boldsymbol{\theta}}\|_{\text{HS}}^2, \tag{22a}$$

$$\leq -2\int_{\mathcal{X}\times\mathcal{Y}} \kappa(\boldsymbol{x}, \boldsymbol{y})\kappa_{\boldsymbol{\theta}}(\boldsymbol{x}, \boldsymbol{y})P_{\mathbf{x}}(d\boldsymbol{x})P_{\mathbf{y}}(d\boldsymbol{y}) + \int_{\mathcal{X}\times\mathcal{Y}} \kappa_{\boldsymbol{\theta}}(\boldsymbol{x}, \boldsymbol{y})^2 P_{\mathbf{x}}(d\boldsymbol{x})P_{\mathbf{y}}(d\boldsymbol{y})$$

$$\leq -2\int_{\mathcal{X}\times\mathcal{Y}} \kappa_{\boldsymbol{\theta}}(\boldsymbol{x}, \boldsymbol{y})P_{\mathbf{xy}}(d\boldsymbol{x}, d\boldsymbol{y}) + \int_{\mathcal{X}\times\mathcal{Y}} \kappa_{\boldsymbol{\theta}}(\boldsymbol{x}, \boldsymbol{y})^2 P_{\mathbf{x}}(d\boldsymbol{x})P_{\mathbf{y}}(d\boldsymbol{y})$$

$$\leq -2\mathbb{E}_{\mathbf{xy}}\kappa_{\boldsymbol{\theta}}(\mathbf{x}, \mathbf{y}) + \mathbb{E}_{\mathbf{x}}\mathbb{E}_{\mathbf{y}}\kappa_{\boldsymbol{\theta}}(\mathbf{x}, \mathbf{y})^2. \tag{22b}$$

For the purpose of our representation learning problem, we consider the scenario in which the chosen basis sets include the constant function, and all other basis functions are centered by construction. That is, $\mathbb{I}_{\mathcal{F}_{\mathbf{x}}} = \{\mathbb{1}_{P_{\mathbf{x}}}\} \cup \{\phi_i \mid \langle \phi_i, \mathbb{1}_{P_{\mathbf{x}}}\rangle_{\mathbf{x}} = 0\}_{i=1}^r$ and $\mathbb{I}_{\mathcal{F}_{\mathbf{y}}} = \{\mathbb{1}_{P_{\mathbf{y}}}\} \cup \{\psi_i \mid \langle \psi_i, \mathbb{1}_{P_{\mathbf{y}}}\rangle_{\mathbf{y}} = 0\}_{i=1}^r$. This results in the $(r+1)$-dimensional matrices:

$$\boldsymbol{V}_{\mathbf{x}} := \begin{bmatrix} 1 & \mathbf{0} \\ \mathbf{0} & \boldsymbol{C}_{\mathbf{x}} \end{bmatrix}, \quad \boldsymbol{V}_{\mathbf{y}} := \begin{bmatrix} 1 & \mathbf{0} \\ \mathbf{0} & \boldsymbol{C}_{\mathbf{y}} \end{bmatrix}, \tag{23}$$

where $\boldsymbol{C}_{\mathbf{x}} = \text{Cov}(\phi(\mathbf{x}), \phi(\mathbf{x})) \in \mathbb{R}^{r\times r}$, $\boldsymbol{C}_{\mathbf{y}} = \text{Cov}(\psi(\mathbf{y}), \psi(\mathbf{y})) \in \mathbb{R}^{r\times r}$ denote the matrix forms of the truncated covariance operators $\mathsf{C}_{\mathbf{x}} : \mathcal{F}_{\mathbf{x}} \mapsto \mathcal{F}_{\mathbf{x}}$ and $\mathsf{C}_{\mathbf{y}} : \mathcal{F}_{\mathbf{y}} \mapsto \mathcal{F}_{\mathbf{y}}$ (see Def. L.5), respectively. Then the orthonormality regularization of Eq. (5) becomes:

$$\|\boldsymbol{V}_{\mathbf{x}} - \boldsymbol{I}\|_{\text{F}}^2 = \|\boldsymbol{C}_{\mathbf{x}} - \boldsymbol{I}_r\|_{\text{F}}^2 + 2\|\mathbb{E}_{P_{\mathbf{x}}}\phi(\mathbf{x})\|^2 \quad \|\boldsymbol{V}_{\mathbf{y}} - \boldsymbol{I}\|_{\text{F}}^2 = \|\boldsymbol{C}_{\mathbf{y}} - \boldsymbol{I}_r\|_{\text{F}}^2 + 2\|\mathbb{E}_{P_{\mathbf{y}}}\psi(\mathbf{y})\|^2. \tag{24}$$

Since $\|\boldsymbol{C}_{\mathbf{x}}\|_{\text{F}}^2 = \text{tr}(\boldsymbol{C}_{\mathbf{xy}}^2)$ involves products of covariance matrices, we compute its empirical value using unbiased estimators. For generality, we present the unbiased estimator for the cross-covariance.

**Unbiased Estimation of Frobenious Norm of Cross-Covariance Operators** Since $\|\boldsymbol{C}_{\mathbf{xy}}\|_F^2 = \text{tr}(\boldsymbol{C}_{\mathbf{xy}}^2)$ involves products of covariance matrices, we obtain unbiased estimates from finite samples by computing the metric using two independent sampling sets from $P_{\mathbf{xy}}$, by:

$$\|\boldsymbol{C}_{\mathbf{xy}}\|_{\text{F}}^2 = \text{tr}(\boldsymbol{C}_{\mathbf{xy}}^2) = \sum_{i=1}^r [\boldsymbol{C}_{\mathbf{xy}}^2]_{i,i} = \sum_{i=1}^r \sum_{j=1}^r [\boldsymbol{C}_{\mathbf{xy}}]_{i,j}[\boldsymbol{C}_{\mathbf{xy}}]_{j,i}$$

$$= \sum_{i=1}^r \sum_{j=1}^r \mathbb{E}_{(\mathbf{x},\mathbf{y})\sim P_{\mathbf{xy}}}[\phi_{c,i}(\mathbf{x})\psi_{c,j}(\mathbf{y})]\mathbb{E}_{(\mathbf{x}',\mathbf{y}')\sim P_{\mathbf{xy}}}[\phi_{c,j}(\mathbf{x}')\psi_{c,i}(\mathbf{y}')]$$

$$= \mathbb{E}_{(\mathbf{x},\mathbf{y},\mathbf{x}',\mathbf{y}')\sim P_{\mathbf{xy}}}[\sum_{i=1}^r \phi_{c,i}(\mathbf{x})\psi_{c,i}(\mathbf{y}')\sum_{j=1}^r \phi_{c,j}(\mathbf{x}')\psi_{c,j}(\mathbf{y})] \tag{25}$$

$$= \mathbb{E}_{(\mathbf{x},\mathbf{y},\mathbf{x}',\mathbf{y}')\sim P_{\mathbf{xy}}}[(\phi_c(\mathbf{x})^\top \psi_c(\mathbf{y}'))(\phi_c(\mathbf{x}')^\top \psi_c(\mathbf{y}))]$$

$$\approx \frac{1}{N^2}\sum_{n=1}^N \sum_{m=1}^N (\phi_c(\boldsymbol{x}_n)^\top \psi_c(\boldsymbol{y}_m'))(\phi_c(\boldsymbol{x}_m')^\top \psi_c(\boldsymbol{y}_n)),$$

where $\phi_c(\mathbf{x}) = \phi(\mathbf{x}) - \mathbb{E}_{P_{\mathbf{x}}}\phi(\mathbf{x})$ denotes the centered basis functions, and $((\boldsymbol{x}, \boldsymbol{y}), (\boldsymbol{x}', \boldsymbol{y}')) \sim P_{\mathbf{xy}}$ indicates two independent sampling sets from $P_{\mathbf{xy}}$ used for the unbiased estimation of $\|\boldsymbol{C}_{\mathbf{x}}\|_F^2$. The final equation then provides the

unbiased empirical estimator computed on a dataset $\mathbb{D} = \{(\boldsymbol{x}_n, \boldsymbol{y}_n) \sim P_{\mathbf{xy}}\}_{n=1}^N$ and any random permutation of it, denoted as $\mathbb{D}' = \{(\boldsymbol{x}'_n, \boldsymbol{y}'_n) \sim P_{\mathbf{xy}}\}_{n=1}^N$.

### F.1. Unbiased Estimation of Orthonormal Regularization

The regularization term for optimizing the loss (5) involves encouraging the basis sets to be orthonormal. The metric quantifying the orthogonality of the basis sets is defined by:

$$
\begin{aligned}
\|\boldsymbol{V}_{\mathbf{x}} - \boldsymbol{I}\|_{\mathrm{F}}^2 &= \|\boldsymbol{C}_{\mathbf{x}} - \boldsymbol{I}_r\|_{\mathrm{F}}^2 + 2\|\mathbb{E}_{P_{\mathbf{x}}}\boldsymbol{\phi}(\mathbf{x})\|^2 = \mathrm{tr}(\boldsymbol{C}_{\mathbf{x}}^2) - 2\mathrm{tr}(\boldsymbol{C}_{\mathbf{x}}) + r + 2\|\mathbb{E}_{P_{\mathbf{x}}}\boldsymbol{\phi}(\mathbf{x})\|^2, \\
\|\boldsymbol{V}_{\mathbf{y}} - \boldsymbol{I}\|_{\mathrm{F}}^2 &= \|\boldsymbol{C}_{\mathbf{y}} - \boldsymbol{I}_r\|_{\mathrm{F}}^2 + 2\|\mathbb{E}_{P_{\mathbf{y}}}\boldsymbol{\psi}(\mathbf{y})\|^2 = \mathrm{tr}(\boldsymbol{C}_{\mathbf{y}}^2) - 2\mathrm{tr}(\boldsymbol{C}_{\mathbf{y}}) + r + 2\|\mathbb{E}_{P_{\mathbf{y}}}\boldsymbol{\psi}(\mathbf{y})\|^2.
\end{aligned}
\tag{26}
$$

Hence given a dataset of samples $\mathbb{D} = \{(\boldsymbol{x}_n, \boldsymbol{y}_n) \sim P_{\mathbf{xy}}\}_{n=1}^N$, and any random permutation of the dataset order $\mathbb{D}' = \{(\boldsymbol{x}'_n, \boldsymbol{y}'_n) \sim P_{\mathbf{xy}}\}_{n=1}^N$ we can derive unbiased empirical estimates of (26) as:

$$
\begin{aligned}
\|\boldsymbol{V}_{\mathbf{x}} - \boldsymbol{I}\|_{\mathrm{F}}^2 &\approx \widehat{\mathbb{E}}_{(\mathbf{x},\mathbf{x}')\sim P_{\mathbf{x}}}[(\boldsymbol{\phi}_c(\mathbf{x})^\top \boldsymbol{\phi}_c(\mathbf{x}'))^2] - 2\widehat{\mathbb{E}}_{P_{\mathbf{x}}}[\boldsymbol{\phi}_c(\mathbf{x})^\top \boldsymbol{\phi}_c(\mathbf{x})] + r + 2\|\widehat{E}_{P_{\mathbf{x}}}\boldsymbol{\phi}(\mathbf{x})\|^2 \\
&\approx \frac{1}{N^2}\sum_{n=1}^N \sum_{m=1}^N (\boldsymbol{\phi}_c(\boldsymbol{x}_n)^\top \boldsymbol{\phi}_c(\boldsymbol{x}'_m))^2 - 2\frac{1}{N}\sum_{n=1}^N \boldsymbol{\phi}_c(\boldsymbol{x}_n)^\top \boldsymbol{\phi}_c(\boldsymbol{x}_n) + r + 2\|\frac{1}{N}\sum_{n=1}^N \boldsymbol{\phi}(\boldsymbol{x}_n)\|^2, \\
\|\boldsymbol{V}_{\mathbf{y}} - \boldsymbol{I}\|_{\mathrm{F}}^2 &\approx \widehat{\mathbb{E}}_{(\mathbf{y},\mathbf{y}')\sim P_{\mathbf{y}}}[(\boldsymbol{\psi}_c(\mathbf{y})^\top \boldsymbol{\psi}_c(\mathbf{y}'))^2] - 2\widehat{\mathbb{E}}_{P_{\mathbf{y}}}[\boldsymbol{\psi}_c(\mathbf{y})^\top \boldsymbol{\psi}_c(\mathbf{y})] + r + 2\|\widehat{E}_{P_{\mathbf{y}}}\boldsymbol{\phi}(\mathbf{y})\|^2 \\
&\approx \frac{1}{N^2}\sum_{n=1}^N \sum_{m=1}^N (\boldsymbol{\psi}_c(\boldsymbol{y}_n)^\top \boldsymbol{\psi}_c(\boldsymbol{y}'_m))^2 - 2\frac{1}{N}\sum_{n=1}^N \boldsymbol{\psi}_c(\boldsymbol{y}_n)^\top \boldsymbol{\psi}_c(\boldsymbol{y}_n) + r + 2\|\frac{1}{N}\sum_{n=1}^N \boldsymbol{\psi}(\boldsymbol{y}_n)\|^2.
\end{aligned}
\tag{27}
$$

### F.2. Orthonormal Regularization of Symmetric Hilbert Spaces

Since the covariance operators $\mathsf{C}_{\mathbf{x}} : \mathcal{L}_{\mathbf{x}}^2 \mapsto \mathcal{L}_{\mathbf{x}}^2$ and $\mathsf{C}_{\mathbf{y}} : \mathcal{L}_{\mathbf{y}}^2 \mapsto \mathcal{L}_{\mathbf{y}}^2$ are $\mathbb{G}$-equivariant (see Prop. L.6), their matrix representations in the isotypic basis are constrained to be block-diagonal, with each block being composed of irrep endomorphisms (Prop. I.14). Hence (26) becomes:

$$
\begin{aligned}
\|\boldsymbol{V}_{\mathbf{x}} - \boldsymbol{I}\|_{\mathrm{F}}^2 &= \|\boldsymbol{C}_{\mathbf{x}} - \boldsymbol{I}_r\|_{\mathrm{F}}^2 + 2\|\mathbb{E}_{P_{\mathbf{x}}}\boldsymbol{\phi}(\mathbf{x})\|^2 \\
&= \|\oplus_{k=1}^{n_{\mathrm{iso}}} \boldsymbol{C}_{\mathbf{x}}^{(k)} - \boldsymbol{I}_r\|_{\mathrm{F}}^2 + 2\|\mathbb{E}_{P_{\mathbf{x}}}\boldsymbol{\phi}^{\mathrm{inv}}(\mathbf{x})\|^2, \\
&= \sum_{k=1}^{n_{\mathrm{iso}}} \|\boldsymbol{C}_{\mathbf{x}}^{(k)} - \boldsymbol{I}_r^{(k)}\|_{\mathrm{F}}^2 + 2\|\mathbb{E}_{P_{\mathbf{x}}}\boldsymbol{\phi}^{\mathrm{inv}}(\mathbf{x})\|^2 \\
&= \sum_{k=1}^{n_{\mathrm{iso}}} \left( \|\boldsymbol{C}_{\mathbf{x}}^{(k)}\|_F^2 - 2\mathrm{tr}(\boldsymbol{C}_{\mathbf{x}}^{(k)}) + r_k \right) + 2\|\mathbb{E}_{P_{\mathbf{x}}}\boldsymbol{\phi}(\mathbf{x})\|^2 \\
&= \sum_{k=1}^{n_{\mathrm{iso}}} \left( \|\sum_{b\in\mathbb{B}} \Theta_b^{(k)} \otimes \Psi_k(b)\|_F^2 - 2\mathrm{tr}(\sum_{b\in\mathbb{B}} \Theta_b^{(k)} \otimes \Psi_k(b)) + r_k \right) + 2\|\mathbb{E}_{P_{\mathbf{x}}}\boldsymbol{\phi}(\mathbf{x})\|^2, \quad \text{by (43)} \\
&= 2\|\mathbb{E}_{P_{\mathbf{x}}}\boldsymbol{\phi}(\mathbf{x})\|^2 + \sum_{k=1}^{n_{\mathrm{iso}}} \|\sum_{b\in\mathbb{B}} \Theta_b^{(k)} \otimes \Psi_k(b)\|_F^2 - 2\sum_{b\in\mathbb{B}} \mathrm{tr}(\Theta_b^{(k)})\mathrm{tr}(\Psi_k(b)) + r_k \\
&= 2\|\mathbb{E}_{P_{\mathbf{x}}}\boldsymbol{\phi}(\mathbf{x})\|^2 + \sum_{k=1}^{n_{\mathrm{iso}}} \|\sum_{b\in\mathbb{B}} \Theta_b^{(k)} \otimes \Psi_k(b)\|_F^2 - 2d_k\sum_{b\in\mathbb{B}} \mathrm{tr}(\Theta_b^{(k)}) + r_k \quad \text{by (40)} \\
&= 2\|\mathbb{E}_{P_{\mathbf{x}}}\boldsymbol{\phi}(\mathbf{x})\|^2 + \sum_{k=1}^{n_{\mathrm{iso}}} \sum_{b\in\mathbb{B}} \|\Theta_b^{(k)}\|_F^2 \|\Psi_k(b)\|_F^2 - 2d_k\sum_{b\in\mathbb{B}} \mathrm{tr}(\Theta_b^{(k)}) + r_k \quad \text{by (40)} \\
&= 2\|\mathbb{E}_{P_{\mathbf{x}}}\boldsymbol{\phi}(\mathbf{x})\|^2 + \sum_{k=1}^{n_{\mathrm{iso}}} d_k\sum_{b\in\mathbb{B}} \|\Theta_b^{(k)}\|_F^2 - 2d_k\sum_{b\in\mathbb{B}} \mathrm{tr}(\Theta_b^{(k)}) + r_k, \quad \text{s.t } \Psi_k(b)\Psi_k(b)^\top = \boldsymbol{I}_{d_k} \\
&= 2\|\mathbb{E}_{P_{\mathbf{x}}}\boldsymbol{\phi}(\mathbf{x})\|^2 + d_k\sum_{k=1}^{n_{\mathrm{iso}}} \sum_{b\in\mathbb{B}} \left( \|\Theta_b^{(k)}\|_F^2 - 2\mathrm{tr}(\Theta_b^{(k)}) \right) + r_k,
\end{aligned}
$$

Where the set of matrices $\{\Theta_b^{(k)} \in \mathbb{R}^{m_k \times m_k}\}_{b\in\mathbb{B}}$ define the free degrees of freedom of the covariance operator (see Prop. I.14), and $\{\Psi_k(b)\}_{b\in\mathbb{B}}$ denote the basis of endomorphisms of the irreducible representation $\bar{\rho}_k$ (see (40)). Crucially, $\|\Theta_b^{(k)}\|_F^2$ features an unbiased U-statistic estimator as in (25).

# G. Experimental Setup

In this section we provide details on the experimental setup. We first describe general design choices and hyperparameters and then provide details for each experiment.

## Experiments Overview

- App. G.1: conditional expectation operator approximation on symmetric cGMMs.

- App. G.2: $\mathbb{G}$-equivariant regression of robot CoM momenta from noisy proprioception.

- App. G.3: synthetic conditional quantile regression for symmetry-aware uncertainty quantification.

- App. G.4: one-dimensional synthetic regression where the conditional mean is insufficient for uncertainty quantification.

- App. G.5: robustness of eNCP under extrinsic and incorrect symmetry misspecification.

- App. G.6: uncertainty quantification for quadruped locomotion observables.

- App. G.7: training and inference costs of eNCP relative to NCP.

**Sample Efficiency Experiments** For both the conditional expectation operator approximation and the $\mathbb{G}$-equivariant regression experiments, we evaluate model performance by measuring sample efficiency/complexity. To do so, we partition the dataset $\mathbb{D} = \{(\boldsymbol{x}_n, \boldsymbol{y}_n)\}_{n=1}^N$ into training, validation, and testing splits in proportions of 70%, 15%, and 15%, respectively. With fixed validation and testing sets, we iteratively train the models on increasing portions of the training set and report the test performance for each size, for 3 to 5 different random seeds.

For each training set size, we select the model checkpoint with the best validation loss to compute the test performance. Thus, these experiments quantify the generalization error (or true risk) and its evolution as a function of the training set size.

**NNs Architectures and Hyperparameters** To compare our equivariant representation learning framework with other contrastive and supervised methods, all (inference) models share a similar fixed architectural footprint. For the baseline models, the only hyperparameter tuned is the learning rate, whereas for the NCP and eNCP models we additionally tune the regularization weight $\gamma$ in Eqs. (5) and (14). Further details for each experiment are provided in the corresponding sections below.

**Code Reproducibility** All experiments, plots and examples are provided in the open-access repository and python package *github.com/Danfoa/symm_rep_learn*.

## G.1. Conditional Expectation Operator Approximation

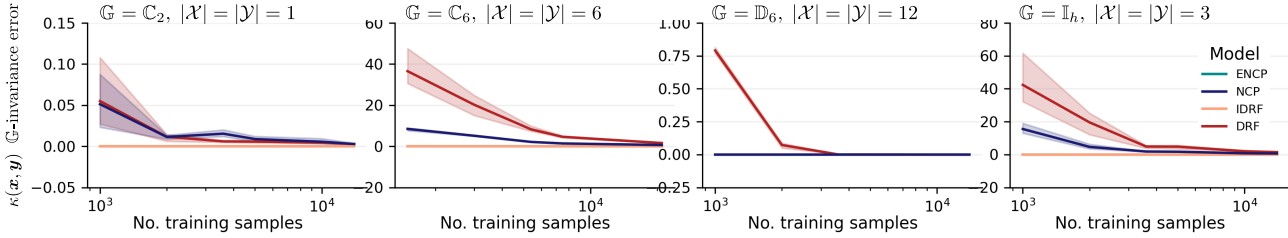

*Figure 6.* Sample-efficiency plots comparing the test-set PMD $\mathbb{G}$-invariance constraint (12) violation $\kappa_{\text{inv-err}} := \mathbb{E}_{\mathbf{x}}\mathbb{E}_{\mathbf{y}} \sum_{g \in \mathbb{G}} (\kappa_{\boldsymbol{\theta}}(\mathbf{x}, \mathbf{y}) - \kappa_{\boldsymbol{\theta}}(g \triangleright_{\mathcal{X}} \mathbf{x}, g \triangleright_{\mathcal{Y}} \mathbf{y}))^2$ versus the number of training samples, in log scales. Each column corresponds to a symmetric cGMM with distinct symmetry groups and $(\mathbf{x}, \mathbf{y})$ dimensionality. The tested groups are the cyclic groups $\mathbb{C}_2$ and $\mathbb{C}_6$, the Dihedral group $\mathbb{D}_6$ (order 12), and the Icosahedral group $\mathbb{I}_h$ (order 60).

In this experiment, we extend the conditional Gaussian Mixture Model (GMM) proposed by Gilardi et al. (2002) to parametrically construct symmetric random variables taking values in arbitrary data spaces $\mathcal{X}$ and $\mathcal{Y}$ and with arbitrary finite

symmetry groups $\mathbb{G}$. The GMM is defined by

$$\mathbf{z} := \mathbf{x} \oplus \mathbf{y} \sim \sum_{g \in \mathbb{G}} \sum_{c=1}^{n_g} \mathcal{N}(\boldsymbol{\rho}_{\mathcal{Z}}(g)\mu_{\mathbf{z},c} \, , \, \boldsymbol{\rho}_{\mathcal{Z}}(g)\Sigma_{\mathbf{z},c}\boldsymbol{\rho}_{\mathcal{Z}}(g)^{\top}),$$

where $\boldsymbol{\rho}_{\mathcal{Z}}(g) := \boldsymbol{\rho}_{\mathcal{X}}(g) \oplus \boldsymbol{\rho}_{\mathcal{Y}}(g)$ are arbitrary group representations of $\mathbb{G}$ and $n_g$ is the number of unique Gaussians with randomly sampled means $\mu_{\mathbf{z}} := \mu_{\mathbf{x}} \oplus \mu_{\mathbf{y}}$ and block-diagonal covariances $\Sigma_{\mathbf{z}} := \Sigma_{\mathbf{x}} \oplus \Sigma_{\mathbf{y}}$. Since every Gaussian appears in group orbits, this symmetric GMM has $\mathbb{G}$-invariant marginal distributions and an analytical expression for the conditional expectation operator kernel $\kappa(\boldsymbol{x}, \boldsymbol{y}) = {}^{p_{\mathbf{xy}}(\boldsymbol{x}, \boldsymbol{y})}/_{P_{\mathbf{x}}(\boldsymbol{x})P_{\mathbf{y}}(\boldsymbol{y})}$ (see 2D example in Fig. 3). Consequently, we can directly estimate the approximation of the conditional expectation operator (Eq. (5)) as the mean squared error between the true and learned density ratios, i.e., $\kappa_{\text{mse}} := \mathbb{E}_{\mathbf{x}}\mathbb{E}_{\mathbf{y}} \|\kappa(\mathbf{x}, \mathbf{y}) - \kappa_{\boldsymbol{\theta}}(\mathbf{x}, \mathbf{y})\|^2$.

To the best of our knowledge, this is the first synthetic experiment that directly estimates the truncation error of the conditional expectation operator in an inference task-agnostic setting, serving as a benchmark for future work.

Fig. 4 compares sample efficiency using $\kappa_{\text{mse}}$, while Fig. 6 shows the error in the $\mathbb{G}$-invariant of the learned $\kappa$ ratio versus sample size, highlighting that symmetry-aware methods encode this property as an architectural constraint, ensuring a strictly $\mathbb{G}$-invariant learned ratio.

### G.2. $\mathbb{G}$-Equivariant Regression of Robot's CoM Momenta

In this experiment, we evaluate the quality of the learned representations using the contrastive loss Eqs. (5) and (14) alongside supervised learning baselines trained with the standard MSE loss. The task is a $\mathbb{G}$-equivariant benchmark in robotics presented in (Ordoñez-Apraez et al., 2025), with the goal of predicting a quadruped robot's CoM linear $\boldsymbol{l} \in \mathbb{R}^3$ and angular momenta $\boldsymbol{k} \in \mathbb{R}^3$ from noisy observations of the robot's generalized positions $\boldsymbol{q} \in \mathbb{R}^{12}$ and velocity coordinates $\dot{\boldsymbol{q}} \in \mathbb{R}^{12}$. Consequently, the random variables are defined as $\boldsymbol{x} = \boldsymbol{q} + \epsilon_{\boldsymbol{q}} \oplus \dot{\boldsymbol{q}} + \epsilon_{\dot{\boldsymbol{q}}}$ and $\boldsymbol{y} = \boldsymbol{l} \oplus \boldsymbol{k}$, where $\epsilon_{\boldsymbol{q}} \in \mathbb{R}^{12}$ and $\epsilon_{\dot{\boldsymbol{q}}} \in \mathbb{R}^{12}$ are independent Gaussian noise terms that model sensor noise. The function computing the CoM momenta from these proprioceptive observations is highly non-linear and $\mathbb{G}$-equivariant whenever $\mathbb{G}$ is a morphological symmetry group of the robot (see Fig. 7 and Ordoñez-Apraez et al. (2025) for details).

The robot considered is the quadruped robot Solo (Fig. 7-right), which possesses a symmetry group of order 8: $\mathbb{G} = \mathbb{K}_4 \times \mathbb{C}_2$, as depicted in this animation showing 8 symmetric robot configurations along with their corresponding linear and angular momenta vectors.

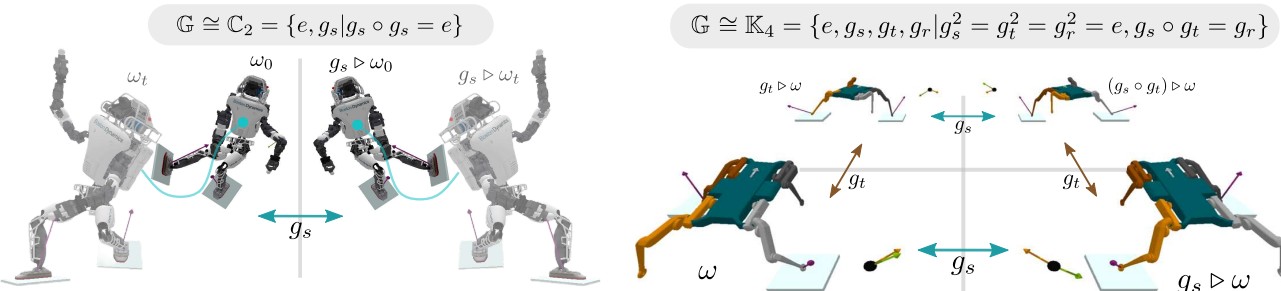

*Figure 7.* Example of morphological finite symmetry in robotics. **Left**: A humanoid robot with the reflectional symmetry group $\mathbb{G} \equiv \mathbb{C}_2$. **Right**: The quadruped robot Solo with the symmetry group $\mathbb{G} = \mathbb{K}_4 \times \mathbb{C}_2$ (only $\mathbb{K}_4$ is shown for clarity). The robot's center of mass linear $\boldsymbol{l} \in \mathbb{R}^3$ and angular $\boldsymbol{k} \in \mathbb{R}^3$ momentum are depicted as orange and green vectors, respectively, for each symmetric configuration. Images adapted from Ordoñez-Apraez et al. (2025) with author approval.

**NN Architectures** We configure all models under consideration (eNCP, NCP, eMLP, and MLP) to have an inference-time NN architecture with a similar footprint. In particular, the encoder network for $\mathbf{x}$ in NCP and eNCP is designed similarly to the NN used in MLP/eMLP. The idea is to test how a model with the same capacity performs on the downstream task of regression when trained using either the representation learning loss or a supervised learning loss. The backbone of all architectures is a standard multilayer perceptron consisting of three hidden layers, each with 512 units, followed by a final hidden layer containing 128 units. This final layer encodes the feature vector $r$ for the NCP and eNCP models. Crucially, since $\mathbb{G}$-equivariance enforces weight sharing in the NN architecture, the encoder NN for eNCP and eMLP comprises $\times 8$ fewer parameters than their symmetry-agnostic counterparts.

## G.3. Uncertainty Quantification via Conditional Quantile Regression

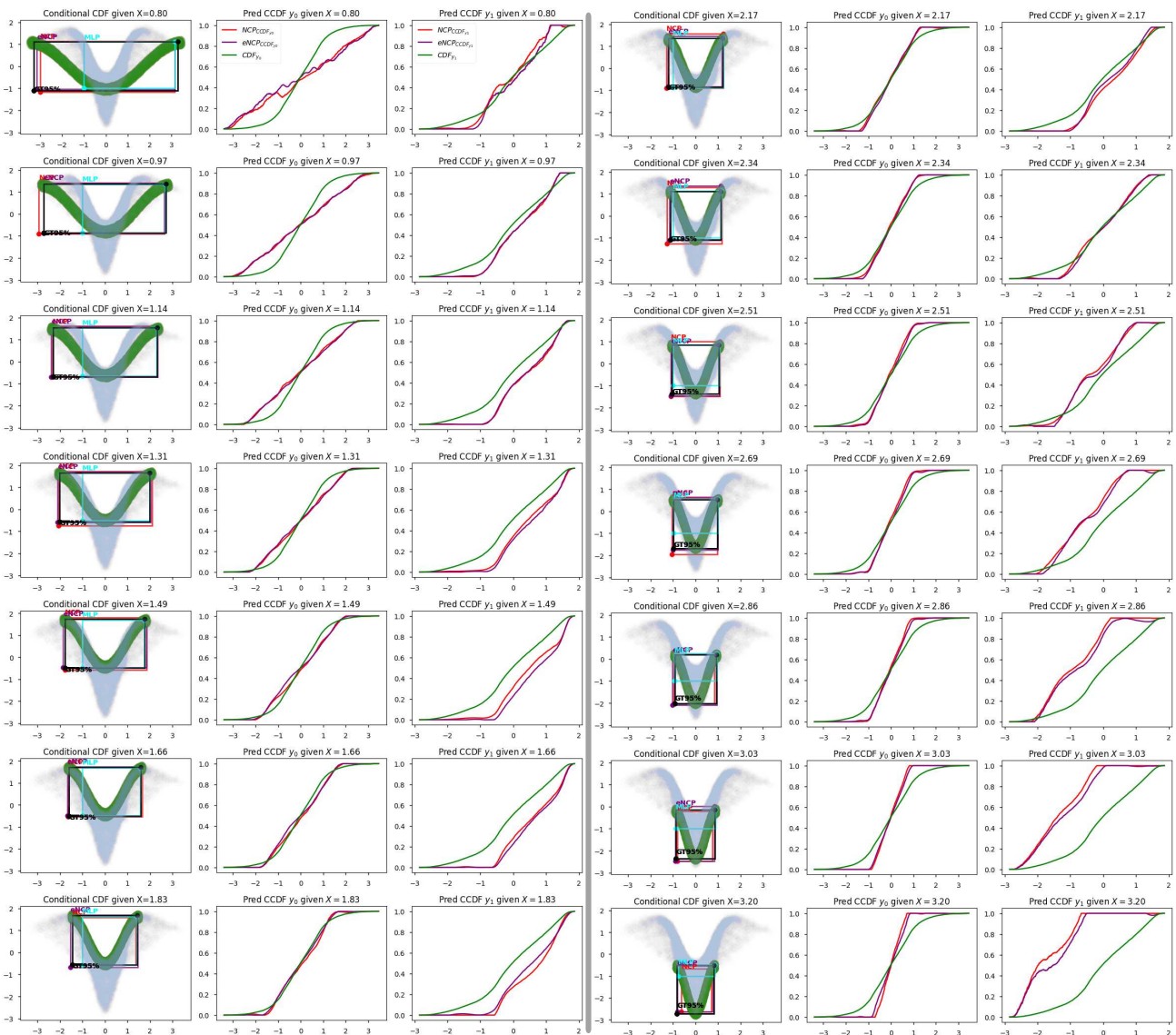

*Figure 8.* Results of a synthetic uncertainty quantification experiment comparing CQR, NCP, and eNCP models for predicting 95% CIs of $\mathbf{y} \in \mathbb{R}^2$ given $\mathbf{x} \in \mathbb{R}$. Left and fourth columns show conditional distributions $\mathbb{P}(\mathbf{y}|\mathbf{x} = \cdot)$ for different conditioning values. Second-third and fifth-sixth columns display cCDF predictions by eNCP and NCP models. While CQR directly regresses quantiles and requires retraining for different confidence levels, NCP and eNCP estimate the full cCDF, enabling adaptation to any confidence interval without retraining.

The goal of these experiments benchmark is to learn the family of conditional distributions $\mathbb{P}(\mathbf{y} \mid \mathbf{x} = \cdot)$ for a bivariate random variable $\mathbf{y} = [y_0, y_1] \in \mathbb{R}^2$ given a scalar covariate $\mathbf{x} \in \mathbb{R}$. Once $\mathbb{P}(\mathbf{y} \mid \mathbf{x})$ is recovered, the practitioner can estimate *conditional* $(1 - \alpha)$–confidence regions by regressing the lower and upper conditional quantiles $q_{\alpha/2}(\mathbf{x})$, $q_{1-\alpha/2}(\mathbf{x})$ for any desired miscoverage level $\alpha \in (0, 1)$. In particular, a 95% confidence region corresponds to $\alpha = 0.05$, so the two quantiles of interest are $q_{0.025}(\mathbf{x})$ and $q_{0.975}(\mathbf{x})$. See Fig. 9 for a visual representation of the problem.

**Conditional Quantile Regression Models** We compare the NCP and proposed eNCP models to a standard baseline for parametric NN conditional quantile regression, namely CQR Feldman et al. (2023), which uses two separate NNs to predict lower and upper quantiles, trained with pinball loss. Both models use MLP backbones with similar parameter counts, ensuring improvements are solely due to loss functions.

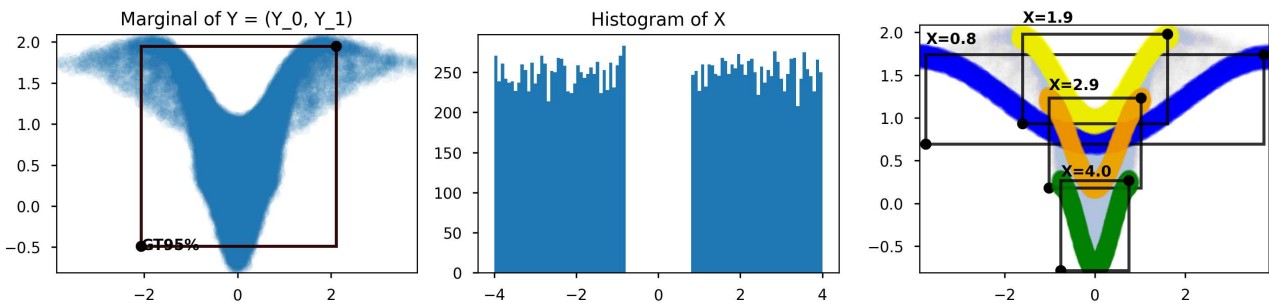

*Figure 9.* Uncertainty quantification experiment adapted from Feldman et al. (2023). Task: predict 95% CIs (black boxes) of $\mathbf{y} \in \mathbb{R}^2$ given $\mathbf{x} \in \mathbb{R}$. **Left:** Marginal $\mathbb{P}(\mathbf{y})$. **Middle:** Marginal $\mathbb{P}(\mathbf{x})$. **Right:** Conditional distributions $\mathbb{P}(\mathbf{y}|\mathbf{x} = \cdot)$ for different values.

Furthermore, CQR can only be trained for specific confidence intervals, requiring retraining for different quantiles. In contrast, the NCP and eNCP models, trained using the deep representation learning approach of Secs. 2 and 4, regress the cCDF of each dimension of $\mathbf{y}$ given $\mathbf{x}$. Thus, they can estimate conditional quantiles for any confidence interval via the quantile estimation algorithm from the cCDF described in Kostic et al. (2024a) without retraining. See details in Fig. 10.

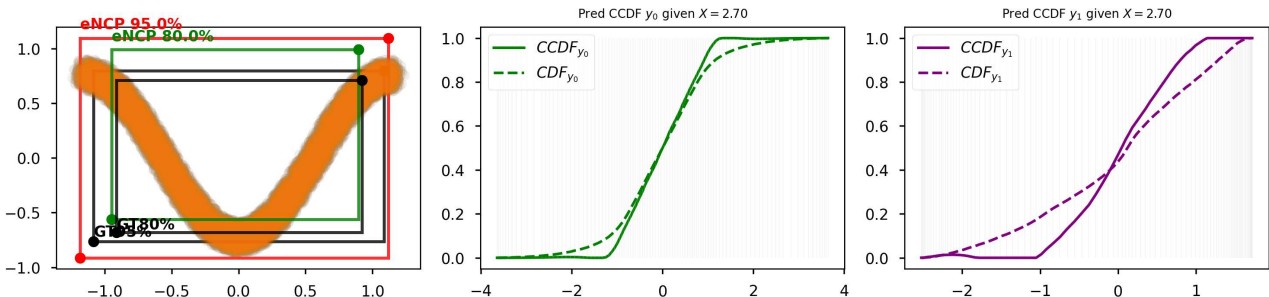

*Figure 10.* Prediction of the 80% and 95% CIs for the random variable $\mathbf{y}$ in experiment App. G.3 using the proposed eNCP model. The model estimates the cCDF by discretizing each dimension of $\mathbf{y} = [\mathrm{y}_1, \mathrm{y}_2]$ into 100 bins and computing the conditional probabilities $\mathbb{P}(\mathrm{y}_i \in \mathbb{A}_n | \mathbf{x} = \cdot) := [\mathsf{E}_{\mathbf{y}|\mathbf{x}} \mathbb{1}_{\mathbb{A}_n}](\cdot)$ for all $n \in [100]$ based on the learned conditional expectation operator $\kappa_{\boldsymbol{\theta}}(\mathbf{x}, \mathbf{y})$ (see Sec. 5). Here, $\mathbb{A}_n$ comprises the bins from the 0-th to the $n$-th. This yields the estimated cCDF for $\mathrm{y}_1$ (center) and $\mathrm{y}_2$ (right) at $\mathbf{x} = 2.7$. The cCDFs can then be used to estimate upper and lower quantiles for any confidence interval (Kostic et al., 2024a). In practice, the eNCP model regresses $2 \times 100$ variables in a single forward pass. Thus, the final layer of the conditional quantile regression model is a linear layer of size $r \times (2 \times 100)$, where $r$ is the number of features in the $\mathbf{y}$ representation (see Sec. 2). Note that the gray vertical lines in the cCDF plots indicate the discretization bins, estimated using a 'quantile_transformer', from sklearn (Pedregosa et al., 2011), fitted on the marginal distribution of each dimension of $\mathbf{y}$. This discretization procedure ensures high-resolution bins in high-density regions of $\mathbf{y}$ and coarser bins in low-density regions, while being able to cover the entire empirical support of $\mathbf{y}$ from the training set.

**Evaluation Metrics: Coverage and Set Size** Let $\mathbb{C}_{1-\alpha}(\mathbf{x}) \subseteq \mathbb{R}^d$ denote a *prediction set* of nominal level $(1 - \alpha)$ produced by a conditional quantile regression model for the response $\mathbf{y} \in \mathbb{R}^d$ given the covariate $\mathbf{x} \in \mathbb{R}^p$. In all experiments we assess two complementary metrics.

- **Coverage.** The conditional *coverage* of $\mathbb{C}_{1-\alpha}$ is the probability that the true response is captured by the predicted region,

$$c_{1-\alpha}(\mathbf{x}) := \mathbb{P}\big(\mathbf{y} \in \mathbb{C}_{1-\alpha}(\mathbf{x}) \mid \mathbf{x}\big), \qquad \text{with the target } c_{1-\alpha}(\mathbf{x}) \approx 1 - \alpha \; \forall \mathbf{x}. \tag{28}$$

In practice we report the *marginal* coverage $\widehat{\mathbb{E}}_{\mathbf{x}}[c_{1-\alpha}(\mathbf{x})]$, estimated on a large held-out sample; values above (resp. below) $1 - \alpha$ indicate over- (resp. under-) coverage.

- **Relaxed Coverage (r-Coverage).** The conditional *relaxed coverage* of $\mathbb{C}_{1-\alpha}$ is defined as the probability that each scalar component of the response lies within its corresponding predicted confidence interval. Formally, if

$\mathbf{y} = [\mathrm{y}_1, \ldots, \mathrm{y}_d]$ and $\mathbb{C}_{1-\alpha}(\mathbf{x})$ has corresponding marginal intervals $\mathbb{C}^{(i)}_{1-\alpha}(\mathbf{x})$ for $i \in \{1, \ldots, d\}$, then

$$\mathrm{rc}_{1-\alpha}(\mathbf{x}) := \prod_{i=1}^{d} \mathbb{P}\Big(\mathrm{y}_i \in \mathbb{C}^{(i)}_{1-\alpha}(\mathbf{x}) \,\Big|\, \mathbf{x}\Big), \tag{29}$$

with the target $\mathrm{rc}_{1-\alpha}(\mathbf{x}) \approx 1 - \alpha$ for all $\mathbf{x}$. As with coverage, we report the *marginal* relaxed coverage $\widehat{\mathbb{E}}_{\mathbf{x}}[\mathrm{rc}_{1-\alpha}(\mathbf{x})]$.

- **Set size.** To quantify how informative the region is, we measure its *size* (volume) under the Lebesgue measure $\lambda^d$:

$$\mathrm{Size}_{1-\alpha}(\mathbf{x}) := \mathrm{vol}\big(\mathbb{C}_{1-\alpha}(\mathbf{x})\big). \tag{30}$$

Smaller sets correspond to sharper uncertainty estimates, provided the required coverage is met. For multidimensional responses the volume is expressed in the natural units of $\mathbb{R}^d$; for $d = 1$ it reduces to the interval length. As with coverage, we report the marginal expectation $\widehat{\mathbb{E}}_{\mathbf{x}}[\mathrm{Size}_{1-\alpha}(\mathbf{x})]$ so that models can be compared fairly across the entire input distribution.

**Data Generation** The data is generated from a conditional distribution $\mathbb{P}(\mathbf{y} \mid \mathbf{x} = \cdot)$ for a bivariate random variable $\mathbf{y} = [\mathrm{y}_0, \mathrm{y}_1] \in \mathbb{R}^2$ given a scalar covariate $\mathbf{x} \in \mathbb{R}$. Adapted from Feldman et al. (2023), the covariate is sampled uniformly: $\mathbf{x} \sim \mathrm{Unif}\big([0.8, \, 4.0] \cup [-4.0, \, -0.8]\big)$, and the response variable $\mathbf{y}$ is produced by a non-linear transformation of auxiliary latent variables (see Fig. 9):

$$
\begin{aligned}
\mathrm{y}_0 &= \frac{\mathrm{z}}{\beta \, |\mathbf{x}|} \;+\; \mathrm{r}\cos\phi, & \mathrm{z} &\sim \mathrm{Unif}(-\pi, \pi), \\
& & \phi &\sim \mathrm{Unif}(0, 2\pi), \\
\mathrm{y}_1 &= \tfrac{1}{2}\big(-\cos \mathrm{z} + 1\big) \;+\; \mathrm{r}\sin\phi \;+\; \sin|\mathbf{x}|, & \mathrm{r} &\sim \mathrm{Unif}(-0.1, 0.1).
\end{aligned}
$$

Here, $\beta > 0$ is a scaling constant. Both marginal distributions (see Fig. 9) and the conditional probability distribution are invariant under the reflection symmetry group $\mathbb{G} = \mathbb{C}_2 = \{e, g_r \mid g_r \circ g_r = e\}$, acting on $\mathbf{y}$ as $g_r \triangleright_\mathcal{Y} \mathbf{y} = [-\mathrm{y}_0, \mathrm{y}_1]$ and on $\mathbf{x}$ as $g_r \triangleright_\mathcal{X} \mathbf{x} = -\mathbf{x}$.

**Results** The experimental results are shown in Fig. 8, where the NCP and eNCP models outperform the baseline CQR model in terms of both coverage and set size, for a diverse range of conditioning values.

Moreover, Fig. 11 illustrates the basis functions learned by the NCP and eNCP models for the random variable $\mathbf{y} = [\mathrm{y}_0, \mathrm{y}_1]$. In contrast to the standard NCP model, eNCP incorporates symmetry priors, enabling a clean separation of its latent representation into two orthogonal subspaces: one corresponding to $\mathbb{C}_2$-invariant functions and the other to functions that change sign under reflection. These correspond to the two isotypic subspaces of the representation space. Intuitively, by leveraging the orthogonality constraints between functions in these subspaces, eNCP performs independent representation learning within each subspace, leading to improved performance and satisfaction of the $\mathbb{G}$-equivariance of the learned conditional expectation operator, as shown in Fig. 6.

### G.4. $\mathbb{G}$-Equivariant Synthetic Regression with Uncertainty Quantification

This synthetic experiment demonstrates how the NCP and eNCP frameworks can train NNs for regression tasks under diverse noise conditions, particularly in applications where the conditional expectation, i.e., the regression function, may be uninformative due to skewed and asymmetric conditional distributions (see Fig. 12), and uncertainty quantification is therefore required. Crucially, in such settings, the representations learned by NCP and eNCP can be reused without retraining to predict conditional quantiles at arbitrary coverage levels (see Eq. (28)), which is not possible with standard supervised MSE regression training.

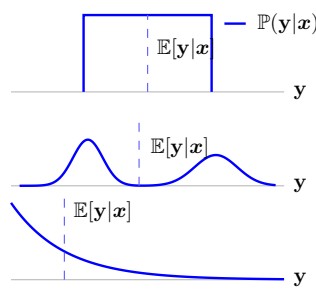

*Figure 12.* Example uninformative expected values.

The dataset is composed of a 1-dimensional samples $\mathbf{x} \in \mathbb{R}$ and $\mathbf{y} \in \mathbb{R}$ drawn from a piecewise conditional distribution with three distinct regimes (see Fig. 13):

- Skewed distribution ($|x| \leq 1$): This regime features a skewed conditional distribution with an exponential tail, with heteroscedastic noise.

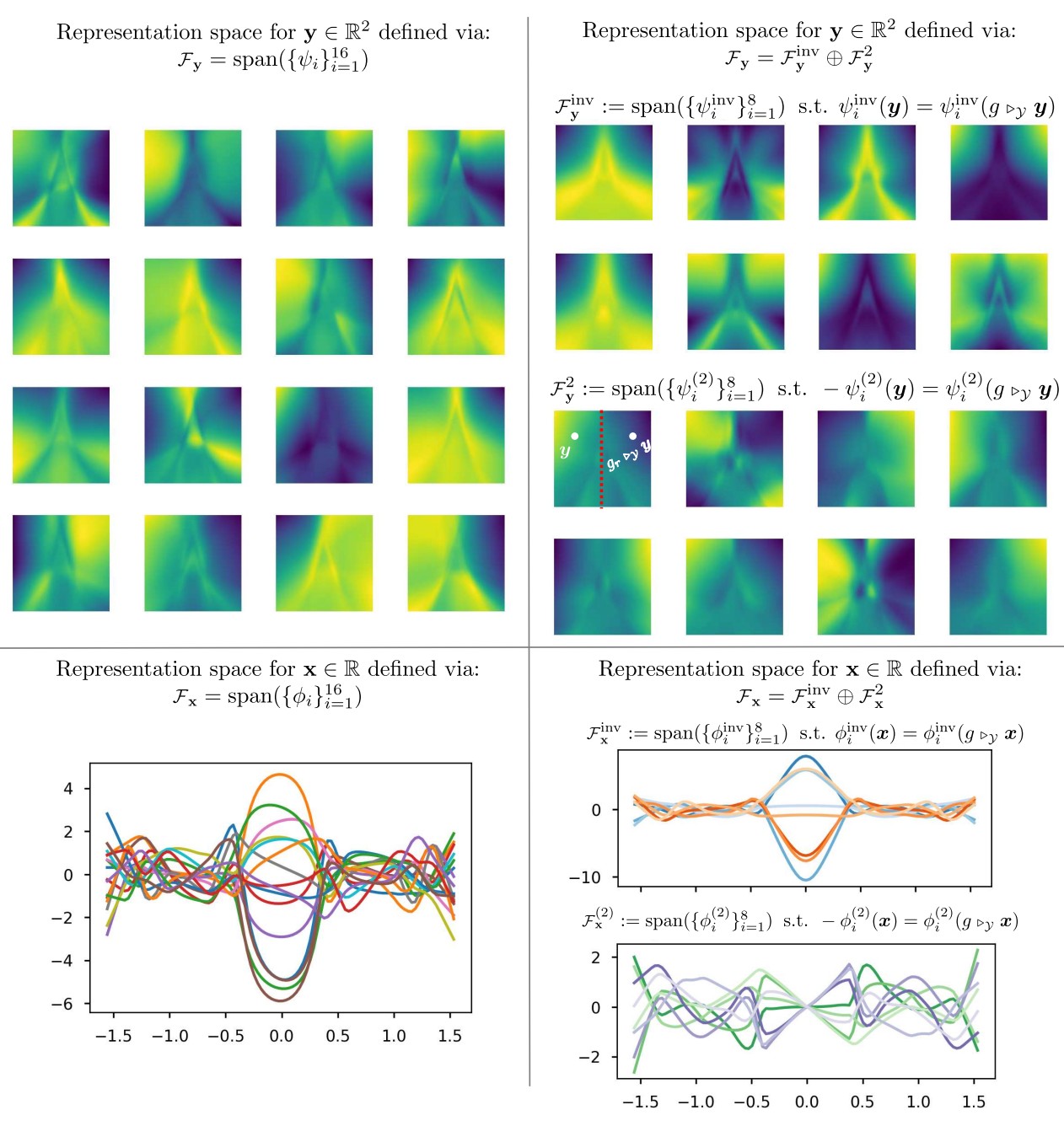

*Figure 11.* **Left:** Learned basis functions from the NCP model for $\mathbf{y} = [\mathrm{y}_0, \mathrm{y}_1]$ and $\mathbf{x}$. **Right:** Learned basis functions from the eNCP model for $\mathbf{y}$ and $\mathbf{x}$. The marginal distributions of $\mathbf{y}$ and $\mathbf{x}$ exhibit reflection symmetry (see Fig. 9) defined by the symmetry group $\mathbb{G} = \mathbb{C}_2\{e, g_r \mid g_r^2 = e\}$, with group actions $g_r \rhd_\mathcal{Y} \mathbf{y} = [-\mathrm{y}_0, \mathrm{y}_1]$ and $g_r \rhd_\mathcal{X} \mathbf{x} = -\mathbf{x}$. By incorporating this prior, the eNCP model learns nonlinear representation spaces for $\mathbf{x}$ and $\mathbf{y}$ that decompose into two isotypic subspaces, $\mathcal{F}_\mathbf{y} = \mathcal{F}_\mathbf{y}^{\mathrm{inv}} \oplus \mathcal{F}_\mathbf{y}^{(2)}$ and $\mathcal{F}_\mathbf{x} = \mathcal{F}_\mathbf{x}^{\mathrm{inv}} \oplus \mathcal{F}_\mathbf{x}^{(2)}$. The invariant subspaces (inv) consist of $\mathbb{C}_2$-invariant functions, while the complementary subspaces consist of functions that change sign under reflection.

- Symmetric distribution ($1 < |x| \leq 2$): This regime has a symmetric conditional distribution, with heteroscedastic noise.

- Bimodal distribution ($|x| > 2$): This regime exhibits a bimodal conditional distribution, with heteroscedastic noise.

Formally the piecewise conditional distribution is defined as follows:

$$\mathbb{P}(\mathbf{y} \mid \mathbf{x}) = \begin{cases} f_c(x) + \varepsilon_{\exp}, & |x| \leq 1, \\ f_c(x) + \sigma_h(|x|)Z, & 1 < |x| \leq 2, \\ S\,a(|x|) + \sigma_p(|x|)Z', & |x| > 2, \end{cases} \quad f_c(x) = \tfrac{1}{2}\cos\left(\tfrac{2\pi}{3}x\right) + \tfrac{1}{5}\cos\left(\tfrac{8\pi}{3}x\right) + \tfrac{1}{4},$$

Where $f_c$ is a deterministic function $\varepsilon_{\exp} \sim \mathrm{Exp}(\mathrm{scale} = s_{\exp}(|x|))$, $Z, Z' \sim \mathcal{N}(0,1)$, $S \in \{-1, +1\}$ is equiprobable. Furthemore $s_{\exp}$, $\sigma_p$ and $\sigma_h$ introduce heteroscedasticity (variance is conditioned on values of $\mathbf{x}$). The function is design such that the conditional distribution is $\mathbb{G}$-invariant under the relfection group $\mathbb{G} = \mathbb{C}_2$ acting as $g_r \rhd_{\mathcal{X}} = -x$ and $g_r \rhd_{\mathcal{Y}} = y$. Consequently, the target $\mathbb{G}$-invariant function defined by the conditional expectation is given as (see Fig. 13):

$$z(x) = \mathbb{E}[\mathbf{y} \mid \mathbf{x} = x] = \begin{cases} f_c(x) + \mathbb{E}[\varepsilon_{\exp}], & |x| \leq 1, \\ f_c(x), & 1 < |x| \leq 2, \\ 0, & |x| > 2, \end{cases}$$

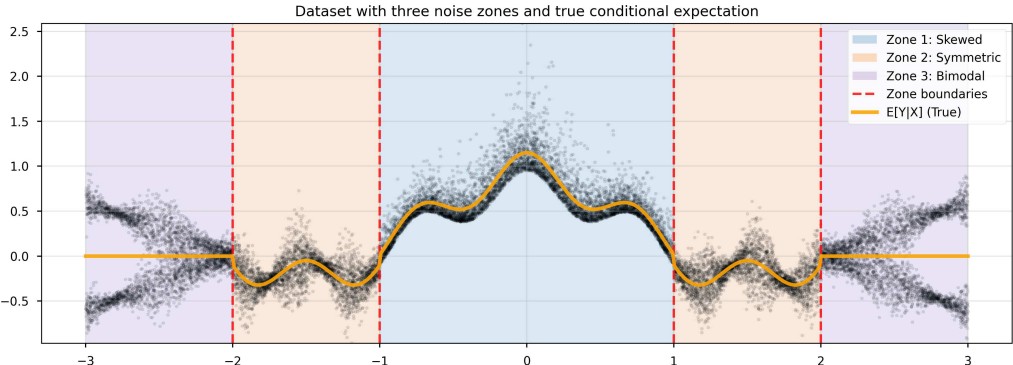

*Figure 13.* Synthetic regression experiment featuring three distinct zones with different families of conditional distributions for which the expected value is uninformative. Zone 1 ($|x| \leq 1$) exhibits a exponential skewed distribution, Zone 2 ($1 < |x| \leq 2$) has symmetric, and Zone 3 ($|x| > 2$) features a bimodal distributon. All zones are affected by heteroscedastic noise.

All models in the comparison share the same standard three-layer MLP backbone, with 64 hidden neurons per layer and ELU activations. The first baseline is the MLP architecture trained with the standard MSE regression loss. The NCP and NCP with data augmentation (NCPaug) models train the same architecture using the contrastive loss in Eq. (5), without and with data augmentation, respectively. Finally, the proposed eNCP model trains the same architecture using the disentangled contrastive loss in Eq. (14), which incorporates symmetry priors and orthogonality regularization to enforce the $\mathbb{G}$-equivariance of the learned conditional expectation operator.

All NCP-based models use orthogonality regularization with Lagrange multiplier $\lambda = 10^{-2}$ and momentum 0.995 to compute the orthonormality statistics through exponential moving-average estimates. All models are trained with early stopping on the validation objective, and the checkpoint with the best validation loss is used to compute test performance. The dataset is generated with $N = 20{,}000$ sample pairs.

After training, for both contrastive models, we fit two linear decoders from the learned representations to predict the conditonal expectation $\hat{z}_{\boldsymbol{\theta}}$, as described in (17), and to recover the cCDF with 200 support points for quantile extraction, as described in Fig. 10.

**Results** The results, shown in Fig. 13, illustrate that all models perform similarly in predicting the conditional expectation $\mathbb{E}[\mathbf{y} \mid \mathbf{x}]$. However, the key advantage of NCP and eNCP lies in their ability to model the conditional distribution, thereby

enabling uncertainty quantification. Here, this is shown through the prediction of conditional lower and upper quantiles with coverage levels of 90% and 70%. Recovering these quantiles is not possible when training the NN backbone using standard supervised MSE regression.

While both NCP and eNCP perform comparatively well and predict well-calibrated confidence intervals, eNCP achieves nearly perfect empirical coverage tracking for coverage levels ranging from 10% to 90%, as shown in Fig. 14 (middle). Furthermore, eNCP consistently predicts smaller CIs (bottom-middle), indicating accurate and non-conservative uncertainty estimates.

Finally, Fig. 15 shows the validation-loss dynamics during training for the NCP, NCPaug, and eNCP models. The results show that eNCP converges faster and to a better optimum than the NCP-based models, requiring approximately $2.5\times$ fewer epochs to reach the final loss values attained by NCP. Although this difference in sample efficiency is mildly mitigated by data augmentation, eNCP exhibits a clear optimality gap with respect to both NCP and NCPaug, achieving overall better validation performance.

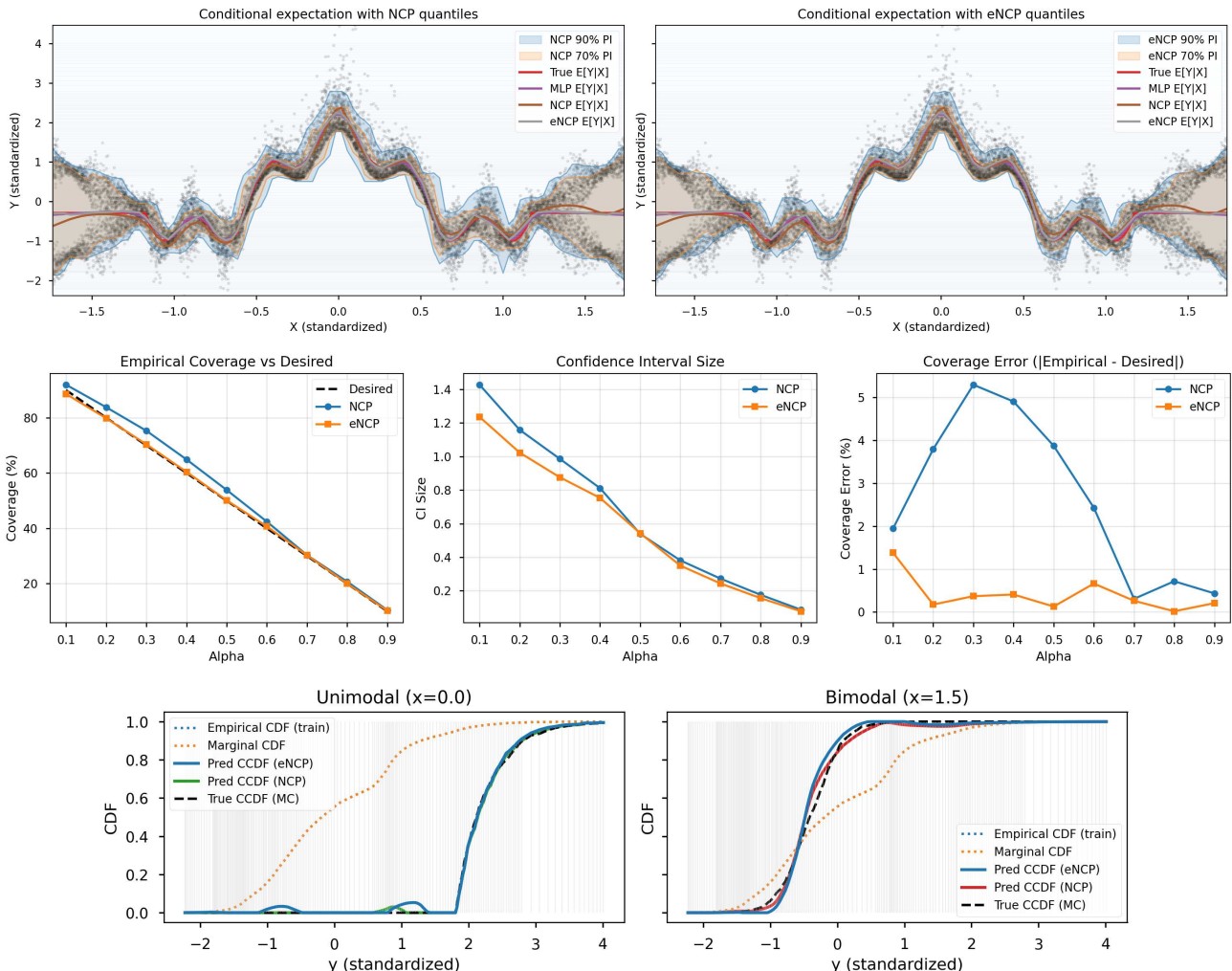

*Figure 14.* **Top:** Comparison of conditional expectation predictions $\mathbb{E}[\mathbf{y} \mid \mathbf{x}]$ and uncertainty quantification via 90% and 70% CIs for baseline MLP, NCP, and proposed eNCP models. **Middle:** Empirical coverage (28) tracking and CIs set size (30) for coverage levels from 10% to 90%. **Bottom:** Example cCDFs predicted by NCP and eNCP on the unimodal skewed distribution regime and the bimodal regime (see details in Fig. 10).

## G.5. Robustness to Symmetry Misspecification

The previous synthetic regression and uncertainty-quantification experiment assumes a correctly specified $\mathbb{C}_2$ symmetry prior: the marginal distribution $P_{\mathbf{x}}$ is invariant under $g_r \triangleright_{\mathcal{X}} x = -x$, and the conditional law $P_{\mathbf{y}|\mathbf{x}}$ is invariant under the

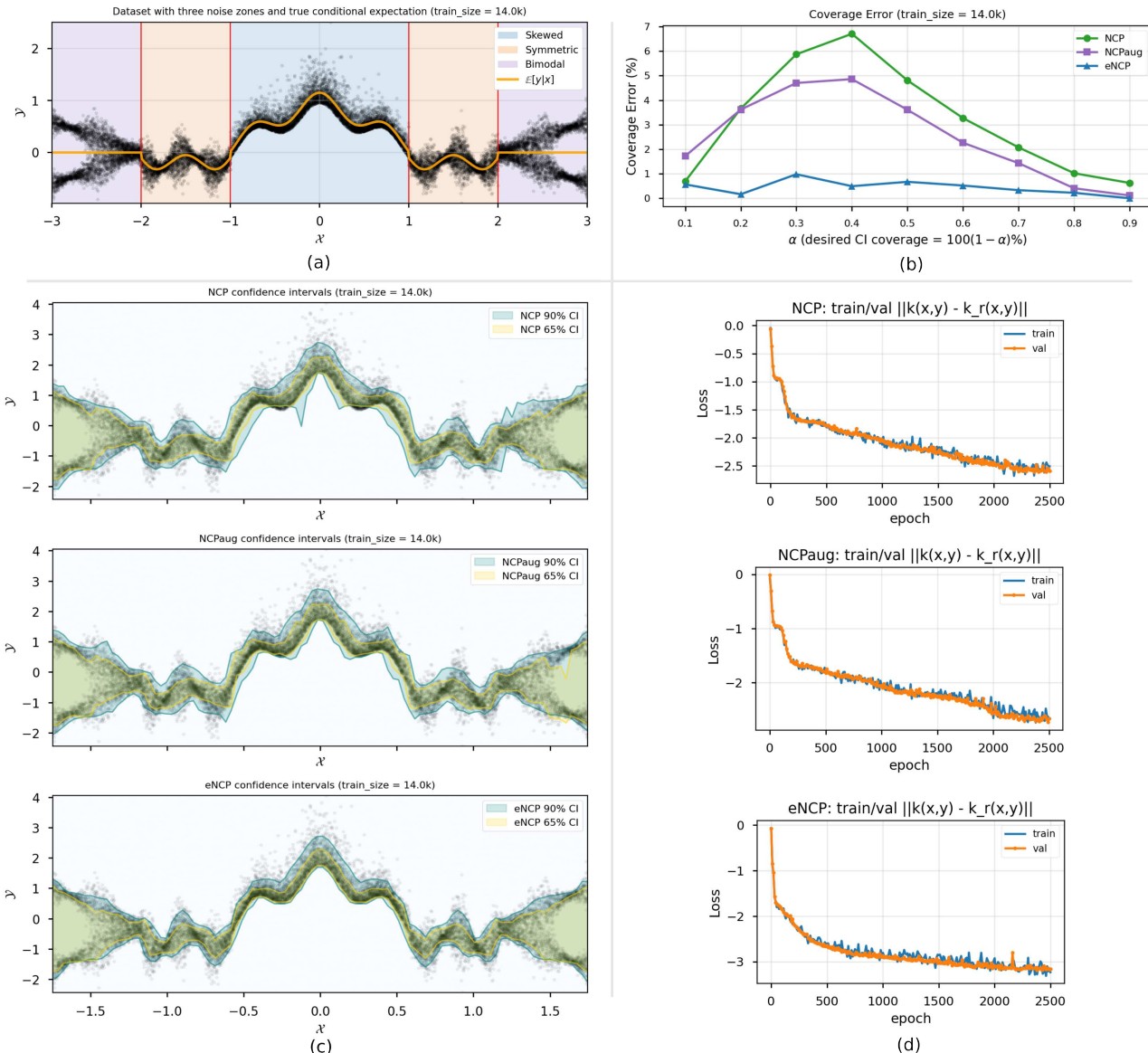

*Figure 15.* (a) Synthetic conditional distribution $P_{\mathbf{y}|\mathbf{x}}$ with three distinct zones whose conditional-distribution families share an uninformative conditional mean. Zone 1 ($|x| \leq 1$) is exponentially skewed, zone 2 ($1 < |x| \leq 2$) is symmetric, and zone 3 ($|x| > 2$) is bimodal. All zones include heteroscedastic noise. (b) Predicted confidence-interval coverage error for eNCP, NCP, and NCPaug across desired coverage levels, defined as $100\% \cdot (1 - \alpha)$. (c) Predicted confidence intervals at desired coverages of 90% and 65% for NCP, NCPaug, and eNCP. (d) Training curves of train and validation losses for NCP, NCPaug, and eNCP.

same action on $\mathcal{X}$ with the trivial action on $\mathcal{Y}$. Correctly specified symmetry priors is the core assumption of GDL, and the main motivation for incorporating them is to improve generalization by leveraging the inductive bias of symmetry.

However, symmetries can be used in practice also as *local* or *approximate* priors, used to regularize learning methods. In such cases, symmetry priors may hold only approximately or only on parts of the domain. Following the taxonomy of Wang et al. (2023), we therefore test eNCP under extrinsic and incorrect symmetry misspecification.

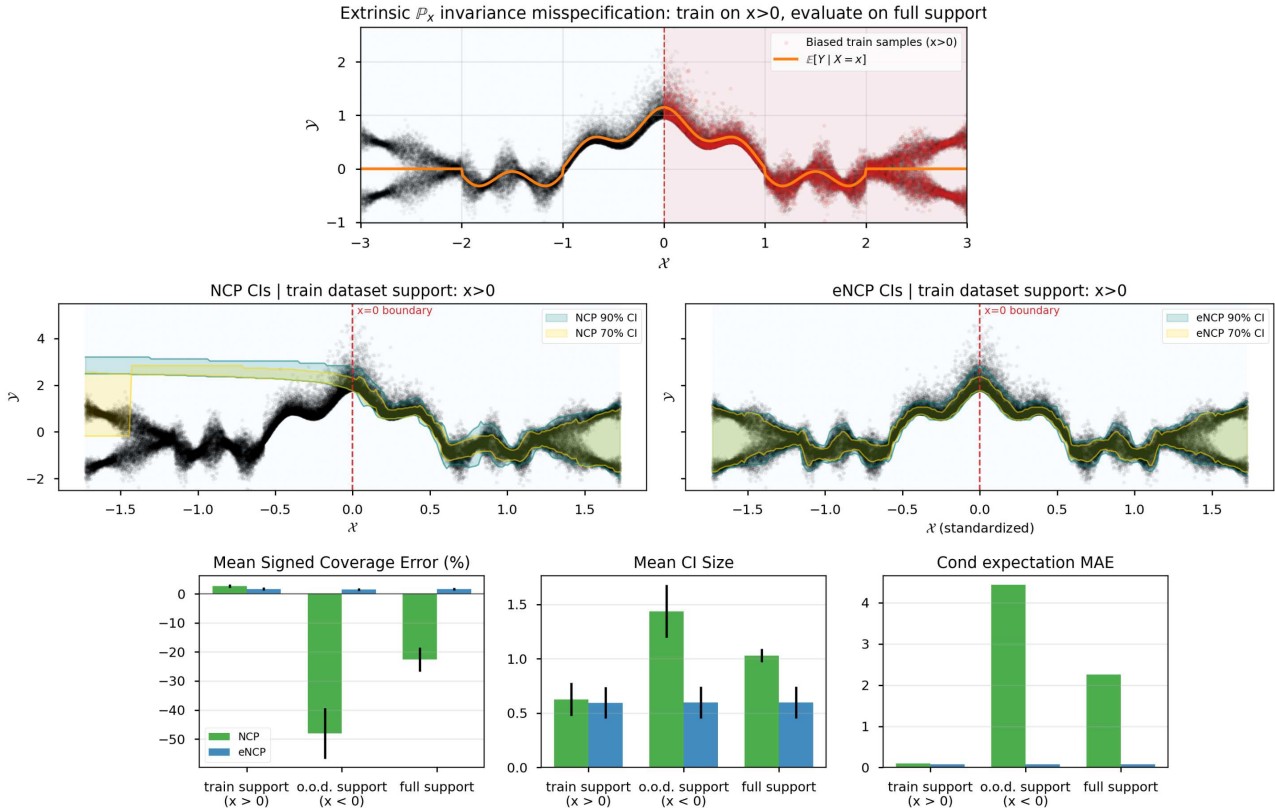

*Figure 16.* Top: Illustration of extrinsic symmetry misspecification, where the train/validation/test support is biased toward the coset subspace $\mathbf{x} > 0$, violating the assumed $\mathbb{C}_2$-invariance of $P_\mathbf{x}$. Middle: Predicted confidence intervals at 90% and 70% coverage for NCP and eNCP under extrinsic symmetry misspecification. Bottom: Mean coverage error, confidence-interval volume, and conditional-expectation Maximum Absolute Error (MAE) for NCP and eNCP, evaluated on the biased support $\mathbf{x} > 0$, the out-of-distribution (o.o.d.) support $\mathbf{x} < 0$, and the full support $\mathcal{X} \equiv \mathbb{R}$.

**Misspecification Types** We use the same one-dimensional regression and uncertainty-quantification setup as in App. G.4, and perturb only the symmetry assumptions. In the extrinsic case, the support of the train, validation, and test samples is biased toward the right coset subspace $\mathbf{x} > 0$. The conditional law remains symmetric, but the marginal $P_\mathbf{x}$ no longer satisfies the assumed $\mathbb{C}_2$-invariance. In the incorrect case, $P_\mathbf{x}$ remains symmetric, but $P_{\mathbf{y}|\mathbf{x}}$ violates the assumed symmetry locally on $|\mathbf{x}| > 1$. We consider two variants: an unbiased violation that increases the heteroscedastic noise scale by a factor $C \in [1.0, 6.0]$ only on $\mathbf{x} > 1$, and a biased violation that introduces a linear conditional-mean shift with slope $b \in [0.0, 0.75]$ only on $\mathbf{x} > 1$. The former preserves the equivariance of the conditional expectation $\mathbb{E}[\mathbf{y} \mid \mathbf{x} = \cdot]$, whereas the latter violates it.

**Metrics** We compare NCP and eNCP using the same learned representations and conditional-quantile extraction procedure described in App. G.4. For each model, we report predicted 90% and 70% CIs, mean signed coverage error, CI volume, and the maximum absolute error of the conditional-expectation estimate. For extrinsic misspecification, the metrics are reported on the biased support $\mathbf{x} > 0$, the out-of-distribution support $\mathbf{x} < 0$, and the full support $\mathcal{X} \equiv \mathbb{R}$. For incorrect misspecification, the metrics are reported on the perturbed support $\mathbf{x} > 1$, the symmetric counterpart $\mathbf{x} < -1$, and the full support.

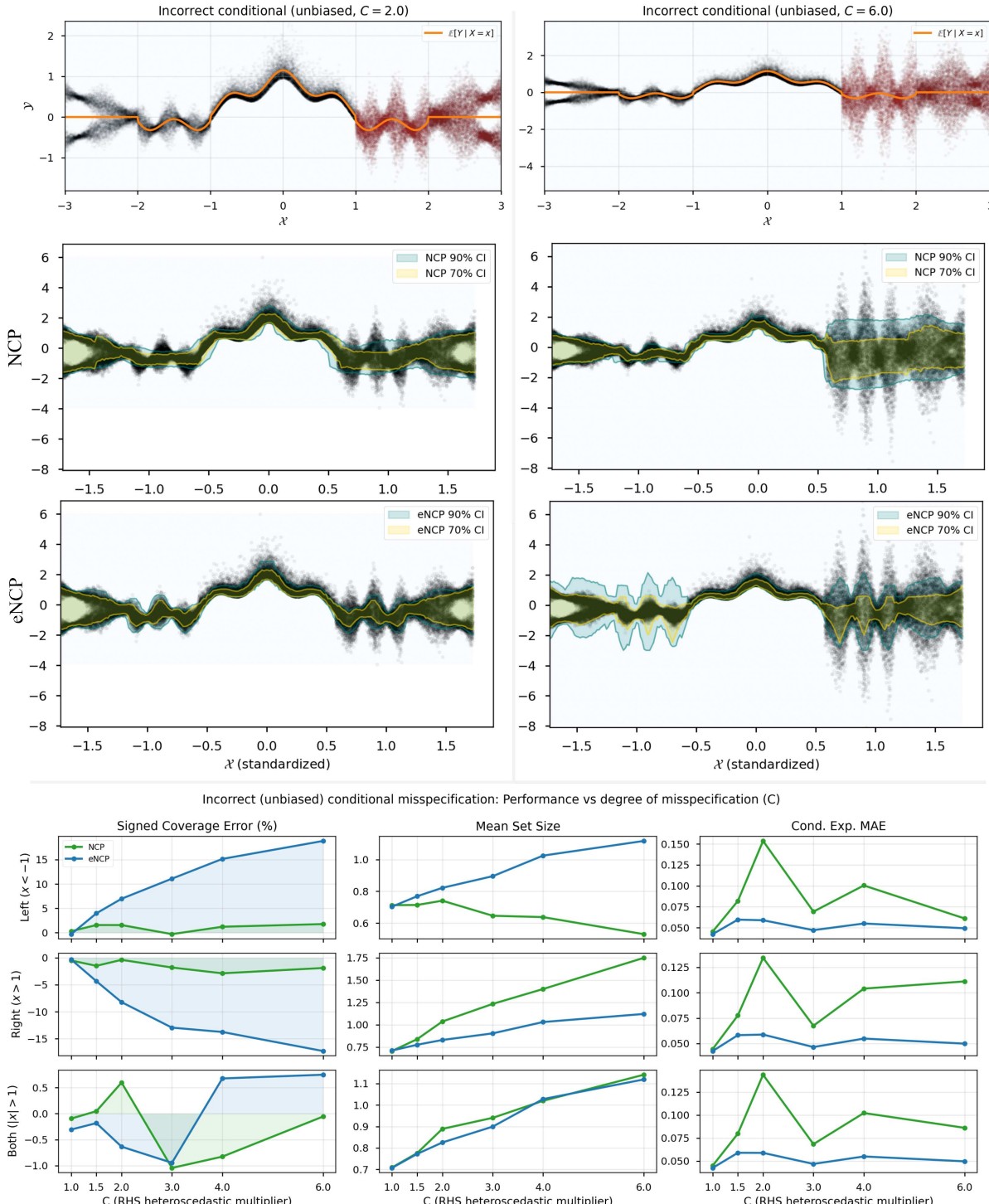

*Figure 17.* First row: Illustration of incorrect (unbiased) conditional misspecification. The $\mathbb{C}_2$-invariance of $P_{\mathbf{y}|\mathbf{x}}$ is violated on $|\mathbf{x}| > 1$ by increasing the heteroscedastic noise scale with $C \in [1.0, 6.0]$ only on the right-coset subspace $\mathbf{x} > 1$. This misspecification is termed unbiased because the conditional expectation $\mathbb{E}[\mathbf{y} \mid \mathbf{x} = \cdot]$ remains $\mathbb{C}_2$-equivariant. Second and third rows: Predicted confidence intervals at 90% and 70% coverage for NCP and eNCP with $C = 2.0$ and $C = 6.0$ under this incorrect unbiased conditional misspecification. Bottom: Mean (signed) coverage error, confidence-interval volume, and conditional-expectation Maximum Absolute Error (MAE) for NCP and eNCP, evaluated on the misspecified supports $\mathbf{x} > 1$ and $\mathbf{x} < -1$, and on the full support $\mathcal{X} \equiv \mathbb{R}$, across noise scales $C \in [1.0, 6.0]$.

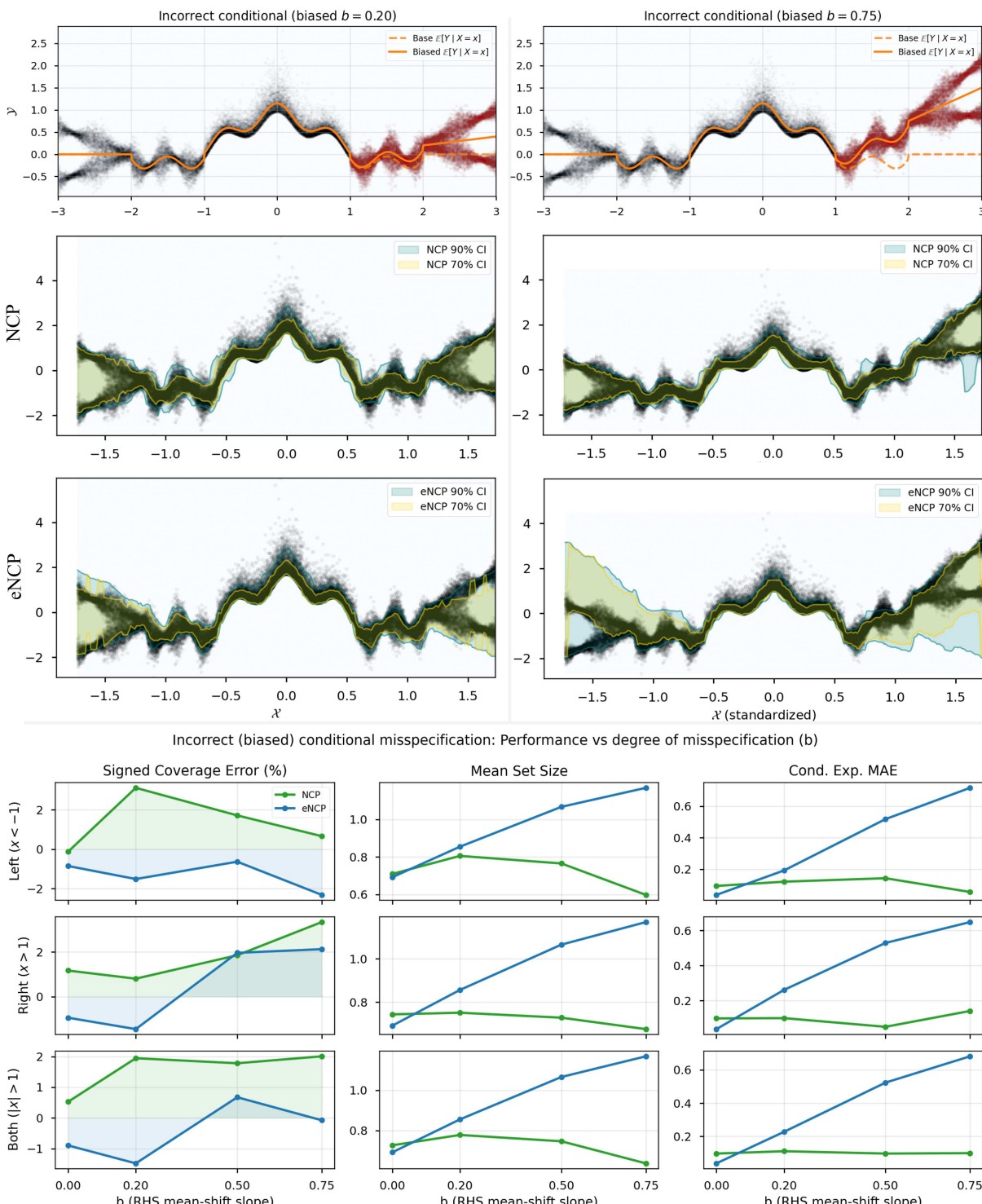

*Figure 18.* First row: Illustration of incorrect (biased) conditional misspecification. The $\mathbb{C}_2$-invariance of $P_{\mathbf{y}|\mathbf{x}}$ is violated on $|\mathbf{x}| > 1$ by introducing a linear bias with slope $b \in [0.0, 0.75]$ on the right-coset subspace $\mathbf{x} > 1$. This misspecification is termed biased because the conditional expectation $\mathbb{E}[\mathbf{y} \mid \mathbf{x} = \cdot]$ is no longer $\mathbb{C}_2$-equivariant. Second and third rows: Predicted confidence intervals at 90% and 70% coverage for NCP and eNCP with $b = 0.2$ and $b = 0.75$ under this incorrect biased conditional misspecification. Bottom: Mean (signed) coverage error, confidence-interval volume, and conditional-expectation Maximum Absolute Error (MAE) for NCP and eNCP, evaluated on the misspecified supports $\mathbf{x} > 1$ and $\mathbf{x} < -1$, and on the full support $\mathcal{X} \equiv \mathbb{R}$, across bias slopes $b \in [0.0, 0.75]$.

**Results** Fig. 16 shows that, under extrinsic misspecification, enforcing the $\mathbb{C}_2$ prior acts as a useful regularizer: eNCP improves out-of-distribution generalization on $\mathbf{x} < 0$ without degrading the in-support estimates on $\mathbf{x} > 0$. This is the benign regime of misspecification in which the conditional relation remains symmetry-consistent and the violation comes from sampling support.

Figs. 17 and 18 show the two incorrect conditional-misspecification regimes. As the violation strength increases, performance degrades continuously rather than catastrophically, and the degradation remains concentrated on the regions where the symmetry assumption is violated. In the unbiased heteroscedastic case, the conditional expectation remains $\mathbb{C}_2$-equivariant, so the main effect appears in coverage error and interval volume on the perturbed support. In the biased case, the conditional expectation itself is no longer equivariant, and the conditional-expectation error increases with the bias slope $b$. In both cases, the comparison between the misspecified and symmetric supports helps localize where the assumed symmetry is no longer compatible with the data.

### G.6. Uncertainty Quantification in Quadruped Legged Locomotion

We test how well conditional-quantile models can recover the conditional $95\%$ confidence regions of three physically meaningful observables produced by a simulated AlienGo quadruped walking over rough terrain (see Fig. 1) under varying friction coefficients. The dataset was collected using the Quadruped-PyMPC simulation framework and model predictive controller from (Turrisi et al., 2024).

The observables for which state-dependent uncertainty estimates are desired are $\boldsymbol{y}_t = [U_t, \ T_t, \ \boldsymbol{\tau}_t^{\mathrm{grf}}]^\top$, with each component defined as follows:

- $\mathbb{G}$-**invariant Kinetic Energy.** $T(\boldsymbol{q}, \dot{\boldsymbol{q}}) = \frac{1}{2} \dot{\boldsymbol{q}}^\top M(\boldsymbol{q}) \dot{\boldsymbol{q}} \in \mathbb{R}$, where $M(\boldsymbol{q})$ is the configuration-dependent inertia matrix. Noise is introduced through sensor measurement errors on the robot's degree of freedom (DoF) position $\boldsymbol{q} \in \mathbb{R}^{12}$ and velocity $\dot{\boldsymbol{q}} \in \mathbb{R}^{12}$.

- $\mathbb{G}$-**invariant Instantaneous Mechanical Work.** $U(\boldsymbol{q}, \dot{\boldsymbol{q}}, \boldsymbol{\tau}) \in \mathbb{R}$, representing the instantaneous mechanical work exerted or absorbed by the robot. This quantity depends on the actuator torques (typically measured with noisy, biased sensors) as well as the external forces (e.g. gravity, contact forces) that are not reliably measurable due to unobserved terrain parameters.

- $\mathbb{G}$-**equivariant Ground-Reaction Forces** $\boldsymbol{\tau}_{\mathrm{grf}} \in \mathbb{R}^{12}$, a fundamental quantity in quadruped control, whose reliable estimation and uncertainty quantification are critical for downstream tasks in robotics (Nisticò et al., 2025; Liu et al., 1994).

The observables of interest are predicted using a suit of onboard proprioceptive sensory signals available at time $t$:

$$\boldsymbol{x}_t = \left[\boldsymbol{q}_t, \ \dot{\boldsymbol{q}}_t, \ \boldsymbol{a}_t, \ \boldsymbol{v}_t, \ \boldsymbol{v}_{t,\mathrm{err}}, \ \boldsymbol{\omega}_t, \ \boldsymbol{\omega}_{t,\mathrm{err}}, \ \boldsymbol{g}_t, \ \dot{\boldsymbol{p}}_{t,\mathrm{feet}}, \ \boldsymbol{\tau}_t^{\mathrm{cmd}}\right]^\top.$$

Specifically, $\boldsymbol{q}_t \in \mathbb{R}^{n_q}$ and $\dot{\boldsymbol{q}}_t \in \mathbb{R}^{n_q}$ are the joint positions and velocities, respectively; $\boldsymbol{a}_t \in \mathbb{R}^3$ is the linear acceleration of the robot's base frame measured by the IMU; $\boldsymbol{v}_t \in \mathbb{R}^3$ is the base linear velocity, while $\boldsymbol{v}_{t,\mathrm{err}} \in \mathbb{R}^3$ the command error base linear velocity; $\boldsymbol{\omega}_t \in \mathbb{R}^3$ and $\boldsymbol{\omega}_{t,\mathrm{err}} \in \mathbb{R}^3$ are the base angular velocity and its command error; $\boldsymbol{g}_t \in \mathbb{R}^3$ is the gravity vector expressed in the base frame; $\dot{\boldsymbol{p}}_{t,\mathrm{feet}} \in \mathbb{R}^{12}$ stacks the linear velocities of the four feet (three components each); and $\boldsymbol{\tau}_t^{\mathrm{cmd}} \in \mathbb{R}^{n_q}$ contains the commanded joint torques.

Hence we design the experiments to compare models of similar footprint in number of parameters, while the loss used for training differs between the NCP and eNCP models w.r.t to the CQR and eCQR models.

**NN Architectures** We configure all models (eNCP, NCP, eCQR, and CQR) with similar inference-time NN architectures. The backbone consists of three hidden layers with 512 units each, followed by a final 128-unit layer that encodes the feature vector $r$ for NCP and eNCP models. Due to $\mathbb{G}$-equivariance weight sharing, eNCP and eCQR have $\times 2$ fewer trainbale parameters than their symmetry-agnostic counterparts.

**Results.** Given sensory input $\mathbf{x}$, the model predicts a set $\mathbb{C}_{0.95}(\mathbf{x}) \subseteq \mathbb{R}^{14}$ satisfying $\mathbb{P}(\mathbf{y} \in \mathbb{C}_{0.95}(\mathbf{x}) \mid \mathbf{x}) \approx 0.95$, while minimizing its volume $\hat{\mathbb{E}}_{\mathbf{x}}[\mathrm{vol}(\mathbb{C}_{0.95}(\mathbf{x}))]$. High coverage implies that the true $\mathbb{G}$-invariant kinetic energy, instantaneous

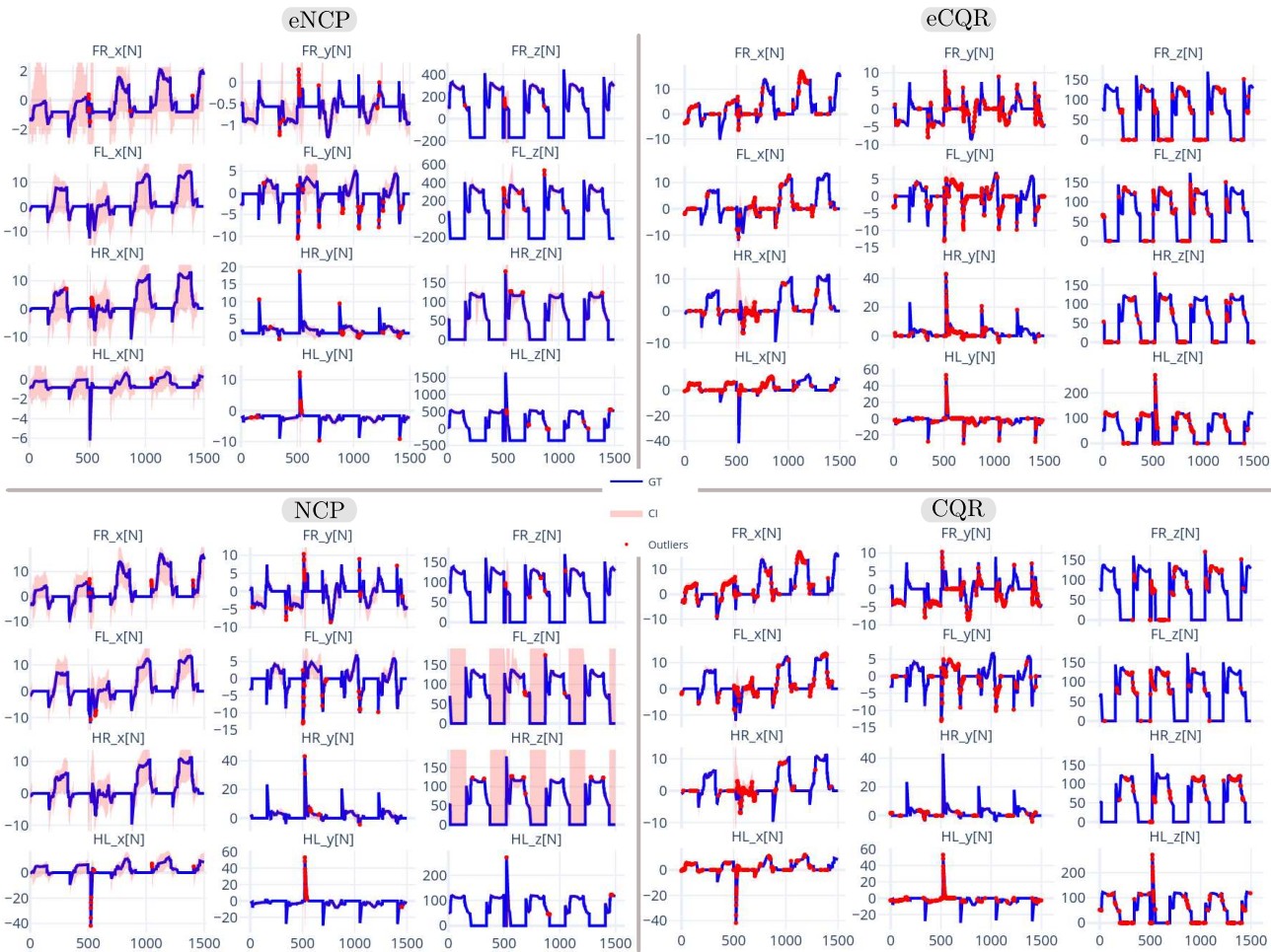

*Figure 19.* Prediction of 90% CIs for ground-reaction forces $\tau_{\mathrm{grf}} \in \mathbb{R}^{12}$ of a quadruped robot on rough terrain with varying friction. We compare eNCP, NCP, eCQR, and CQR models. CIs are computed for each leg—front-right (FR), front-left (FL), hind-right (HR), hind-left (HL)—along $x$, $y$, $z$ axes. Forces outside CIs are red, within CIs are blue. Terrain variations cause significant variability in $x/y$ components due to surface orientation and friction differences, while $z$ components are influenced by local height changes affecting contact timing and producing short-duration high-impact forces.

mechanical work, and $\mathbb{G}$-equivariant 12-dimensional ground-reaction forces lie within the predicted confidence set. Relaxed coverage (r-Coverage) quantifies the reliability of estimates on a per-dimension basis. Tab. 4 summarizes validation and test results for all models, and Fig. 19 illustrates a trajectory of GRF with respective $90\%$ confidence intervals. Both CQR and eCQR produce smaller CIs but fail to achieve desired coverage on the test set, implying unreliable CIs requiring further calibration through retraining or conformal calibration (Feldman et al., 2023). In contrast, the eNCP model achieves desired coverage on the test set while producing larger confidence intervals, hence yielding reliable uncertainty estimates.

Note that we omitted the use of conformal calibration on the CQR estimates to ensure a fair comparison between the parametric uncertainty estimates, considering that conformal calibration is model-agnostic and applicable to eNCP as well.

| | Validation | | | Test | | |
|---|---|---|---|---|---|---|
| | r-Coverage ↑ | Coverage ↑ | Set Size ↓ | r-Coverage ↑ | Coverage ↑ | Set Size ↓ |
| eNCP | 99.3±0.0% | 94.1±0.4% | 2.4±0.4×10^{10} | 99.5±0.1% | 95.0±0.4% | 4.3±3.6×10^9 |
| NCP | 96.4±0.0% | 56.9±0.1% | 3.9±4.5×10^{10} | 99.5±0.0% | 56.9±0.3% | 2.6±1.4×10^{10} |
| eCQR | 70.7±0.6% | 7.3±1.7% | 3.7±2.6×10^8 | 84.2±0.7% | 6.7±1.2% | 1.7±1.7×10^7 |
| CQR | 67.6±1.8% | 7.6±0.4% | 2.5±2.4×10^9 | 80.5±3.7% | 8.5±0.9% | 1.4±0.1×10^8 |

*Table 4.* Validation and test metrics for $95\%$ CIs on quadruped robot observables traversing rough terrain (see App. G.6). Metrics: **(i)** relaxed coverage (r-Coverage) (29), **(ii)** coverage (28), and **(iii)** set size (30). Best results in blue. While eCQR and CQR produce smaller confidence intervals, they fail to achieve expected $95\%$ coverage on validation and test sets. The eNCP model achieves best overall coverage for reliable uncertainty quantification. Importantly, eNCP and NCP models can provide CIs at any coverage level **without retraining**, whereas CQR and eCQR require retraining for each new level.

### G.7. Training and Inference Computational Costs

We compare the training and inference cost of eNCP and NCP across symmetry groups of increasing complexity. All timings are averaged over 200 passes after warmup, using a fixed batch size, fixed data dimensionality ($\dim \mathcal{X}$, $\dim \mathcal{Y}$), and encoder networks $\phi_\theta$ and $\psi_\theta$ parameterized by three-hidden-layer MLP/eMLP architectures.

| Group [Order] | Iso. subspaces | Model | No. trainable params (k) ↓ | Train forward (ms) ↓ | Train backward (ms) ↓ | Val. forward (ms) ↓ |
|---|---|---|---|---|---|---|
| $\mathbb{I}_h$ [60] | 5 | eNCP | 9.2 (60 × fewer) | 3.067 ± 0.105 (3.7×) | 2.237 ± 0.084 (2.5×) | 0.741 ± 0.091 (1.1×) |
| $\mathbb{I}_h$ [60] | 5 | NCP | 550.8 | 0.825 ± 0.032 | 0.896 ± 0.051 | 0.653 ± 0.080 |
| $\mathbb{D}_{10}$ [20] | 8 | eNCP | 16.7 (20 × fewer) | 3.761 ± 0.128 (5.4×) | 3.185 ± 0.109 (4.1×) | 0.487 ± 0.030 (1.2×) |
| $\mathbb{D}_{10}$ [20] | 8 | NCP | 333.2 | 0.692 ± 0.063 | 0.773 ± 0.046 | 0.415 ± 0.025 |
| $\mathbb{C}_{10}$ [10] | 6 | eNCP | 28.7 (10 × fewer) | 4.250 ± 0.492 (5.6×) | 2.707 ± 0.269 (3.4×) | 0.450 ± 0.033 (1.2×) |
| $\mathbb{C}_{10}$ [10] | 6 | NCP | 287.3 | 0.760 ± 0.298 | 0.797 ± 0.161 | 0.382 ± 0.028 |
| $\mathbb{C}_2$ [2] | 2 | eNCP | 139.3 (2 × fewer) | 2.315 ± 0.705 (3.2×) | 1.540 ± 0.552 (2.2×) | 0.353 ± 0.014 (1.2×) |
| $\mathbb{C}_2$ [2] | 2 | NCP | 278.5 | 0.721 ± 0.167 | 0.710 ± 0.114 | 0.301 ± 0.011 |

*Table 5.* Training and inference runtime comparison between eNCP and NCP across symmetry groups with different orders and isotypic structures. Values report the mean ± standard deviation over 200 passes after warmup. Parenthesized factors in the eNCP rows are measured relative to the corresponding NCP runtime or parameter count; for example, $(3.7x)$ indicates that the metric is 3.7 times larger for eNCP than for NCP.

**Training Time Overhead**   Tab. 5 shows that eNCP incurs a per-batch training overhead of roughly $3\times$–$6\times$ in the forward pass and $2.2\times$–$4.1\times$ in the backward pass. This overhead arises from the basis expansions in the equivariant layers (common to all equivariant NN implementations) and from the symmetry-aware estimation of means, covariances, and cross-covariances in the loss (14). Importantly, the computational overhead is governed by the number of isotypic subspaces rather than the order of the group, given that several algebraic operations are performed sequentially per each isotypic subspace.

**Inference Time Overhead**   At inference time, the equivariant NN can be run as a standard NN resulting in a minimal inference overhead (Ordonez Apraez, 2026), around $1.1\times$–$1.2\times$ across all groups. This mild increase is due to the additional change-of-basis layer used to expose the isotypic decomposition in the equivariant encoder (see Remark I.18). Despite the per-batch overhead, eNCP converges substantially faster: as shown in Fig. 15, it reaches a lower validation loss before epoch 500 than NCP achieves after 2500 epochs, corresponding to roughly $5\times$ fewer training batches to reach a better solution. Combined with the $2\times$–$60\times$ reduction in trainable parameters, this yields a favorable overall computational trade-off.

## H. Conditional Probability Modeling via the Conditional Expectation Operator

This section introduces the modelling of conditional probabilities for two random variables via the **conditional expectation operator**. Our goal is to understand conditional expectation from an operator-theoretic perspective. We begin by describing the marginal, joint, and conditional probabilities of the random variables within a measure-theoretic framework. This discussion extends the exposition of Kostic et al. (2024b).

Given two random variables $(\mathbf{x}, \mathbf{y})$ taking values in the measure spaces $(\mathcal{X}, \Sigma_{\mathcal{X}}, P_{\mathbf{x}})$ and $(\mathcal{Y}, \Sigma_{\mathcal{Y}}, P_{\mathbf{y}})$, we have that the marginal probability of any set $\mathbb{A} \in \Sigma_{\mathcal{X}}$ and $\mathbb{B} \in \Sigma_{\mathcal{Y}}$ are given by

$$\mathbb{P}(\mathbf{x} \in \mathbb{A}) = \int_{\mathcal{X}} \mathbb{1}_{\mathbb{A}}(\boldsymbol{x}) P_{\mathbf{x}}(d\boldsymbol{x}) = \int_{\mathbb{A}} P_{\mathbf{x}}(d\boldsymbol{x}) \quad \text{and} \quad \mathbb{P}(\mathbf{y} \in \mathbb{B}) = \int_{\mathcal{Y}} \mathbb{1}_{\mathbb{B}}(\boldsymbol{y}) P_{\mathbf{y}}(d\boldsymbol{y}) = \int_{\mathbb{B}} P_{\mathbf{y}}(d\boldsymbol{y}), \tag{31}$$

where $\mathbb{1}_{\mathbb{A}} \in \mathcal{L}_{\mathbf{x}}^2$ and $\mathbb{1}_{\mathbb{B}} \in \mathcal{L}_{\mathbf{y}}^2$ denote the characteristic functions of sets $\mathbb{A}$ and $\mathbb{B}$, respectively.

Furthermore, under the reasonable assumption that the joint probability measure is absolutely continuous w.r.t to the product of the marginals $P_{\mathbf{xy}} \ll P_{\mathbf{x}} \times P_{\mathbf{y}}$, we have that there exist a Radon-Nikodym derivative $\kappa : \mathcal{X} \times \mathcal{Y} \to \mathbb{R}_+$ such that $P_{\mathbf{xy}}(d\boldsymbol{x}, d\boldsymbol{y}) = \kappa(\boldsymbol{x}, \boldsymbol{y}) P_{\mathbf{x}}(d\boldsymbol{x}) P_{\mathbf{y}}(d\boldsymbol{y})$. Note that $\kappa$ is a kernel function that pointwise deforms the product of the marginals to produce the joint distribution (Sugiyama et al., 2012) (see Fig. 3). This kernel function enable us to express the joint probability by:

$$\mathbb{P}(\mathbf{x} \in \mathbb{A}, \mathbf{y} \in \mathbb{B}) = \int_{\mathcal{X} \times \mathcal{Y}} \mathbb{1}_{\mathbb{A}}(\boldsymbol{x}) \mathbb{1}_{\mathbb{B}}(\boldsymbol{y}) \underbrace{\kappa(\boldsymbol{x}, \boldsymbol{y}) P_{\mathbf{y}}(d\boldsymbol{y}) P_{\mathbf{x}}(d\boldsymbol{x})}_{P_{\mathbf{xy}}(d\boldsymbol{x}, d\boldsymbol{y})} = \int_{\mathbb{A} \times \mathbb{B}} \kappa(\boldsymbol{x}, \boldsymbol{y}) P_{\mathbf{x}}(d\boldsymbol{x}) P_{\mathbf{y}}(d\boldsymbol{y}). \tag{32}$$

Furthermore, given that $\mathbb{P}(\mathbf{y} \in \mathbb{B} | \mathbf{x} \in \mathbb{A}) = {}^{\mathbb{P}(\mathbf{x} \in \mathbb{A}, \mathbf{y} \in \mathbb{B})}/_{\mathbb{P}(\mathbf{x} \in \mathbb{A})}$, the conditional probability of any set $\mathbb{B} \in \Sigma_{\mathcal{Y}}$ given a value of the random variable $\mathbf{x} = \boldsymbol{x}$ is given by:

$$\mathbb{P}(\mathbf{y} \in \mathbb{B} | \mathbf{x} = \boldsymbol{x}) = \int_{\mathcal{Y}} \mathbb{1}_{\mathbb{B}}(\boldsymbol{y}) P_{\mathbf{y} | \mathbf{x}}(d\boldsymbol{y} | \boldsymbol{x}) = \int_{\mathcal{Y}} \mathbb{1}_{\mathbb{B}}(\boldsymbol{y}) \kappa(\boldsymbol{x}, \boldsymbol{y}) P_{\mathbf{y}}(d\boldsymbol{y}) = \int_{\mathbb{B}} \kappa(\boldsymbol{x}, \boldsymbol{y}) P_{\mathbf{y}}(d\boldsymbol{y}), \tag{33}$$

where $P_{\mathbf{y} | \mathbf{x}} : \Sigma_{\mathcal{Y}} \times \mathcal{X} \mapsto [0, 1]$ denotes the **conditional probability measure**. This is a well-defined probability measure considering that:

$$\mathbb{P}(\mathbf{x} \in \mathbb{A}) := \mathbb{P}(\mathbf{x} \in \mathbb{A}, \mathbf{y} \in \mathcal{Y}) = \int_{\mathbb{A}} \underbrace{\left( \int_{\mathcal{Y}} \kappa(\boldsymbol{x}, \boldsymbol{y}) P_{\mathbf{y}}(d\boldsymbol{y}) \right)}_{\mathbb{E} P_{\mathbf{y} | \mathbf{x}}(d\boldsymbol{y} | \mathbf{x} = \boldsymbol{x}) = 1 \ \forall \boldsymbol{x} \in \mathcal{X}} P_{\mathbf{x}}(d\boldsymbol{x}) = \int_{\mathbb{A}} P_{\mathbf{x}}(d\boldsymbol{x}).$$

**The Operator Perspective**   Every measurable function $h \in \mathcal{L}_{\mathbf{y}}^2$ can be approximated by simple functions—that is, as a combination of characteristic functions on measurable sets: $h(\cdot) \approx \sum_{i \in \mathbb{N}} \beta_i \mathbb{1}_{\mathbb{A}_i}(\cdot)$. Thus, Eq. (33) is a special case of the more general problem of approximating the conditional expectation of any function $h \in \mathcal{L}_{\mathbf{y}}^2$ given $\mathbf{x}$. This conditional expectation is captured by the action of a linear integral operator:

**Definition H.1** (Conditional expectation operator). *Let $(\mathbf{x}, \mathbf{y})$ be two random variables defined on the measure spaces $(\mathcal{X}, \Sigma_{\mathcal{X}}, P_{\mathbf{x}})$ and $(\mathcal{Y}, \Sigma_{\mathcal{Y}}, P_{\mathbf{y}})$, respectively, and let $\mathcal{L}_{\mathbf{x}}^2$ and $\mathcal{L}_{\mathbf{y}}^2$ denote the corresponding spaces of square-integrable functions. The conditional expectation operator $\mathsf{E}_{\mathbf{y} | \mathbf{x}} : \mathcal{L}_{\mathbf{y}}^2 \to \mathcal{L}_{\mathbf{x}}^2$ is the linear integral operator—defined via the PMD Radon–Nikodym derivative $\kappa(\boldsymbol{x}, \boldsymbol{y}) = {}^{P_{\mathbf{xy}}(d\boldsymbol{x}, d\boldsymbol{y})}/_{P_{\mathbf{x}}(d\boldsymbol{x}) P_{\mathbf{y}}(d\boldsymbol{y})}$ —which acts on any function $h \in \mathcal{L}_{\mathbf{y}}^2$ by computing its conditional expectation:*

$$[\mathsf{E}_{\mathbf{y} | \mathbf{x}} h](\boldsymbol{x}) = \mathbb{E}[h(\mathbf{y}) | \mathbf{x} = \boldsymbol{x}] := \int_{\mathcal{Y}} h(\boldsymbol{y}) P_{\mathbf{y} | \mathbf{x}}(d\boldsymbol{y} | \boldsymbol{x}) = \int_{\mathcal{Y}} h(\boldsymbol{y}) \frac{P_{\mathbf{xy}}(d\boldsymbol{y}, \boldsymbol{x})}{P_{\mathbf{x}}(d\boldsymbol{x})} = \int_{\mathcal{Y}} h(\boldsymbol{y}) \kappa(\boldsymbol{x}, \boldsymbol{y}) P_{\mathbf{y}}(d\boldsymbol{y}).$$

From a learning perspective, approximating the conditional expectation operator sufficiently well for a relevant set of functions in $\mathcal{L}_{\mathbf{y}}^2$ implies that we can approximate the conditional probability measure of any set $\mathbb{A} \in \Sigma_{\mathcal{Y}}$. This enables both regression *and* uncertainty quantification applications with a single model (see Eq. (2)).

## I. Background on Group and Representation Theory

**Group Actions and Representations**   This section provides a concise overview of the fundamental concepts in group and representation theory, which are used to define the symmetries of the random variables we consider in this work.

For a comprehensive background on these topics in finite-dimensional vector spaces, see Weiler et al. (2023); for the infinite-dimensional case, consult Knapp (1986). These concepts will be referenced as needed in the main text. To begin, we define a group as an abstract mathematical object.

**Definition I.1** (Group). *A group is a set $\mathbb{G}$, endowed with a binary composition operator defined as:*

$$
\begin{aligned}
(\circ): \quad \mathbb{G} \times \mathbb{G} &\longrightarrow \mathbb{G} \\
(g_1, g_2) &\longrightarrow g_1 \circ g_2,
\end{aligned}
\tag{34a}
$$

*such that the following axioms hold:*

$$
\begin{aligned}
\textit{Associativity:} \quad & (g_1 \circ g_2) \circ g_3 = g_1 \circ (g_2 \circ g_3), \quad \forall\, g_1, g_2, g_3 \in \mathbb{G}, && \text{(34b)} \\
\textit{Identity:} \quad & \exists\, e \in \mathbb{G} \text{ such that } e \circ g = g = g \circ e, \quad \forall\, g \in \mathbb{G}, && \text{(34c)} \\
\textit{Inverses:} \quad & \forall\, g \in \mathbb{G}, \exists\, g^{-1} \in \mathbb{G} \text{ such that } g \circ g^{-1} = e = g^{-1} \circ g. && \text{(34d)}
\end{aligned}
$$

We are primarily interested in symmetry groups, i.e., groups of transformations acting on a set $\mathcal{X}$. Each transformation is a bijection that leaves a fundamental property of the element of the set invariant. For example, if $\mathcal{X}$ represents states of a dynamical system, the invariant property is the state energy (see Fig. 7); if $\mathcal{X}$ is a data space, the preserved quantity is typically the probability density/distribution (see Fig. 3).

**Definition I.2** (Group action on a set (Weiler et al., 2023)). *Let $\mathcal{X}$ be a set endowed with symmetry group $\mathbb{G}$. The (left) group action of the group $\mathbb{G}$ on the set $\mathcal{X}$ is a map:*

$$
\begin{aligned}
(\triangleright): \quad \mathbb{G} \times \mathcal{X} &\longrightarrow \mathcal{X} \\
(g, \boldsymbol{x}) &\longrightarrow g \triangleright \boldsymbol{x}
\end{aligned}
\tag{35a}
$$

*that is compatible with the group composition and identity element $e \in \mathbb{G}$, in the sense that:*

$$
\begin{aligned}
\textit{Identity:} \quad & e \triangleright \boldsymbol{x} = \boldsymbol{x}, && \forall\, \boldsymbol{x} \in \mathcal{X} && \text{(35b)} \\
\textit{Associativity:} \quad & (g_1 \circ g_2) \triangleright \boldsymbol{x} = g_1 \triangleright (g_2 \triangleright \boldsymbol{x}), && \forall\, g_1, g_2 \in \mathbb{G}, \forall\, \boldsymbol{x} \in \mathcal{X}. && \text{(35c)}
\end{aligned}
$$

These sets of bijections describe structural properties of the set $\mathcal{X}$. To study these properties, we will frequently refer to the group of symmetry related elements of to a given element $\boldsymbol{x} \in \mathcal{X}$ as its *group orbit* of $\boldsymbol{x}$:

**Definition I.3** (Group orbit). *Let $\boldsymbol{x}$ be an element of the set $\mathcal{X}$ endowed with symmetry group $\mathbb{G}$. The group orbit of $\boldsymbol{x}$ is the the set of all symmetry related set elements, denoted by:*

$$
\mathbb{G}\boldsymbol{x} := \{g \triangleright_{\mathcal{X}} \boldsymbol{x} \mid g \in \mathbb{G}\} \equiv \{\boldsymbol{x} \triangleleft g \mid g \in \mathbb{G}\}.
$$

For our practical purposes, we focus on symmetry transformations acting on sets with a vector space structure. In most cases, the group action on a vector space is linear, which allows us to express symmetry transformations as (orthogonal) matrix-vector operations, once a basis for the space is chosen.

**Definition I.4** (Linear group representation). *Let $\mathcal{X}$ be a vector space endowed with symmetry group $\mathbb{G}$. A linear representation of $\mathbb{G}$ on $\mathcal{X}$ is a map, denoted by $\boldsymbol{\rho}_{\mathcal{X}}$, between symmetry transformation and invertible linear maps on $\mathcal{X}$ (i.e., elements of the general linear group $\mathbb{GL}(\mathcal{X})$):*

$$
\begin{aligned}
\boldsymbol{\rho}_{\mathcal{X}}: \quad \mathbb{G} &\longrightarrow \mathbb{GL}(\mathcal{X}) \\
g &\longrightarrow \boldsymbol{\rho}_{\mathcal{X}}(g),
\end{aligned}
\tag{36a}
$$

*such that the following properties hold:*

$$
\begin{aligned}
\textit{composition}: \quad & \boldsymbol{\rho}_{\mathcal{X}}(g_1 \circ g_2) = \boldsymbol{\rho}_{\mathcal{X}}(g_1)\boldsymbol{\rho}_{\mathcal{X}}(g_2), && \forall\, g_1, g_2 \in \mathbb{G}, && \text{(36b)} \\
\textit{inversion}: \quad & \boldsymbol{\rho}_{\mathcal{X}}(g^{-1}) = \boldsymbol{\rho}_{\mathcal{X}}(g)^{-1}, && \forall\, g \in \mathbb{G}. && \text{(36c)} \\
\textit{identity}: \quad & \boldsymbol{\rho}_{\mathcal{X}}(g \circ g^{-1}) = \boldsymbol{\rho}_{\mathcal{X}}(e) = \boldsymbol{I}, && && \text{(36d)}
\end{aligned}
$$

*Whenever the vector space is of finite dimension $n < \infty$, linear maps admit a matrix form $\boldsymbol{\rho}_{\mathcal{X}}(g) \in \mathbb{R}^{n \times n}$, once a basis set $\mathbb{I}_{\mathcal{X}}$ for the vector space $\mathcal{X}$ is chosen. In this case, Eqs. (36b) to (36d) show how the composition and inversion of symmetry*

*transformations translate to matrix multiplication and inversion, respectively. Moreover, $\boldsymbol{\rho}_\mathcal{X}$ allows to express a (linear) group action (Def. I.2) as a matrix-vector multiplication:*

$$
(\triangleright): \quad \begin{aligned} \mathbb{G} \times \mathcal{X} &\longrightarrow & \mathcal{X} \\ (g, \boldsymbol{x}) &\longrightarrow & g \triangleright \boldsymbol{x} := \boldsymbol{\rho}_\mathcal{X}(g)\boldsymbol{x}. \end{aligned}
\tag{36e}
$$

We will often study linear maps that preserve a vector space's symmetry structure by commuting with the group action. These maps are known as **endomorphisms** and include all change of basis and affine transformations that do not break the symmetry.

**Definition I.5** (Endomorphism). *Let $\mathcal{X}$ be a vector space endowed with symmetry group $\mathbb{G}$, with the group action $\triangleright_\mathcal{X}: \mathbb{G} \times \mathcal{X} \mapsto \mathcal{X}$. A linear map $\boldsymbol{A} : \mathcal{X} \mapsto \mathcal{X}$ is said to be an endomorphism if it commutes with the group action, such that:*

$$
\boldsymbol{\rho}_\mathcal{X}(g)\boldsymbol{A} = \boldsymbol{A}\boldsymbol{\rho}_\mathcal{X}(g), \quad \forall\, g \in \mathbb{G} \qquad \Longleftrightarrow \qquad
\begin{array}{ccc}
\mathcal{X} & \xrightarrow{\;\triangleright_\mathcal{X}\;} & \mathcal{X} \\
\downarrow{\scriptstyle \boldsymbol{A}} & & \downarrow{\scriptstyle \boldsymbol{A}} \\
\mathcal{X} & \xrightarrow{\;\triangleright_\mathcal{X}\;} & \mathcal{X}
\end{array}
$$

*We will denote the space of all endomorphisms of $\mathcal{X}$ as $\mathrm{End}_\mathbb{G}(\mathcal{X})$, such that any $\boldsymbol{A} \in \mathrm{End}_\mathbb{G}(\mathcal{X})$ satisfies the above commutation property.*

So far, we have studied symmetric vector spaces $\mathcal{X}$ endowed with a group action $\triangleright_\mathcal{X}$, which can be represented in matrix form via a group representation $\boldsymbol{\rho}_\mathcal{X}$ once a basis for $\mathcal{X}$ is chosen. However, while the choice of basis alters the group representation $\boldsymbol{\rho}_\mathcal{X}$, the underlying group action $\triangleright_\mathcal{X}$ remains invariant. This observation leads us to the concept of equivalent group representations.

**Definition I.6** (Equivalent group representations). *Let $\mathcal{X}$ be a vector space endowed with symmetry group $\mathbb{G}$, and let $\boldsymbol{\rho}'_\mathcal{X}$ and $\boldsymbol{\rho}_\mathcal{X}$ be two group representations of $\mathbb{G}$ on $\mathcal{X}$. They are said to be equivalent, denoted by $\boldsymbol{\rho}'_\mathcal{X} \sim \boldsymbol{\rho}_\mathcal{X}$, if there exists an invertible change of basis $\boldsymbol{Q} \in \mathrm{End}_\mathbb{G}(\mathcal{X})$ such that*

$$
\boldsymbol{\rho}'_\mathcal{X}(g) = \boldsymbol{Q}\boldsymbol{\rho}_\mathcal{X}(g)\boldsymbol{Q}^{-1}, \quad \forall\, g \in \mathbb{G}.
\tag{37}
$$

*Equivalent representations arise when the same group action $(\triangleright) : \mathbb{G} \times \mathcal{X} \to \mathcal{X}$ is expressed in different coordinate frames or bases. For instance, let $\mathbb{A}_\mathcal{X}$ and $\mathbb{B}_\mathcal{X}$ be two bases for $\mathcal{X} = \mathrm{span}(\mathbb{A}_\mathcal{X}) = \mathrm{span}(\mathbb{B}_\mathcal{X})$, and let $\boldsymbol{Q}^\mathbb{B}_\mathbb{A} : \mathcal{X} \to \mathcal{X}$ denote the change of basis from $\mathbb{A}_\mathcal{X}$ to $\mathbb{B}_\mathcal{X}$, so that $\boldsymbol{x}^\mathbb{B} = \boldsymbol{Q}^\mathbb{B}_\mathbb{A}\boldsymbol{x}^\mathbb{A}$ for all $\boldsymbol{x}^\mathbb{A} \in \mathcal{X}$. Then the group action admits equivalent representations, $\boldsymbol{\rho}^\mathbb{A}_\mathcal{X} \sim \boldsymbol{\rho}^\mathbb{B}_\mathcal{X}$, since*

$$
\begin{aligned}
g \triangleright \boldsymbol{x}^\mathbb{B} &:= \boldsymbol{Q}^\mathbb{B}_\mathbb{A}(g \triangleright \boldsymbol{x}^\mathbb{A}), \qquad \forall g \in \mathbb{G}, \\
\boldsymbol{\rho}^\mathbb{B}_\mathcal{X}(g)\boldsymbol{x}^\mathbb{B} &= \boldsymbol{Q}^\mathbb{B}_\mathbb{A}\big(\boldsymbol{\rho}^\mathbb{A}_\mathcal{X}(g)\boldsymbol{x}^\mathbb{A}\big) = \Big(\boldsymbol{Q}^\mathbb{B}_\mathbb{A}\boldsymbol{\rho}^\mathbb{A}_\mathcal{X}(g)\boldsymbol{Q}^{\mathbb{B}^{-1}}_\mathbb{A}\Big)\boldsymbol{x}^\mathbb{B}, \\
\boldsymbol{\rho}^\mathbb{B}_\mathcal{X}(g) &= \boldsymbol{Q}^\mathbb{B}_\mathbb{A}\boldsymbol{\rho}^\mathbb{A}_\mathcal{X}(g)\boldsymbol{Q}^{\mathbb{B}^{-1}}_\mathbb{A}.
\end{aligned}
\tag{38}
$$

To reveal the modular structure of symmetric vector spaces, we often change bases to decompose them into subspaces stable under the action of the group $\mathbb{G}$, termed $\mathbb{G}$-stable subspaces. This decomposition mirrors how a symmetry group can be broken down into products and direct products of smaller groups and is essential for analyzing and simplifying group representations. We introduce the following definition.

**Definition I.7** ($\mathbb{G}$-stable and irreducible subspaces). *Let $\mathcal{X}$ be a vector space endowed with a group action $(\triangleright)$ of the symmetry group $\mathbb{G}$. A subspace $\mathcal{X}' \subseteq \mathcal{X}$ is said to be $\mathbb{G}$-stable if the action of any group element on any vector in the subspace remains within the subspace, that is,*

$$
g \triangleright \boldsymbol{x} \in \mathcal{X}', \quad \forall\, \boldsymbol{x} \in \mathcal{X}' \subseteq \mathcal{X}, \forall\, g \in \mathbb{G}.
$$

*If the only $\mathbb{G}$-stable subspaces of $\mathcal{X}$ are $\{\boldsymbol{0}\}$ and $\mathcal{X}$ itself, then $\mathcal{X}$ is a irreducible $\mathbb{G}$-stable space. We will denote irreducible $\mathbb{G}$-stable spaces with an over bar, e.g., $\bar{\mathcal{V}}$.*

**Building Blocks of Symmetric Vector Spaces**    A recurrent theme in this work and in geometric deep learning is to study and process symmetric vector spaces by decomposing them in terms of irreducible $\mathbb{G}$-stable spaces. This process directly corresponds to the decomposition of the associated group representation $\rho_{\mathcal{X}}$ into smaller representations acting on these $\mathbb{G}$-stable subspaces.

**Definition I.8** (Decomposable representation). *Let $\mathcal{X}$ be a vector space with a group action ($\triangleright$) defined by the representation $\rho_{\mathcal{X}}$ in a chosen basis $\mathbb{A}_{\mathcal{X}}$. The representation is* decomposable *if it is equivalent to a direct sum of two lower-dimensional representations, $\rho_{\mathcal{X}} \sim \rho_{\mathcal{X}_1} \oplus \rho_{\mathcal{X}_2}$, where $\mathcal{X}_1$ and $\mathcal{X}_2$ are $\mathbb{G}$-stable subspaces of $\mathcal{X}$. Equivalently, there exists a change of basis $\boldsymbol{Q}_{\mathbb{A}}^{\mathbb{B}} : \mathcal{X} \to \mathcal{X}$ such that*

$$\boldsymbol{\rho}_{\mathcal{X}}^{\mathbb{B}} = \begin{bmatrix} \boldsymbol{\rho}_{\mathcal{X}_1} & \mathbf{0} \\ \mathbf{0} & \boldsymbol{\rho}_{\mathcal{X}_2} \end{bmatrix} = \boldsymbol{Q}_{\mathbb{A}}^{\mathbb{B}} \boldsymbol{\rho}_{\mathcal{X}} \boldsymbol{Q}_{\mathbb{A}}^{\mathbb{B}-1}, \; and \;\; g \triangleright \boldsymbol{x}^{\mathbb{B}} := \boldsymbol{\rho}_{\mathcal{X}}^{\mathbb{B}}(g) \boldsymbol{x}^{\mathbb{B}} = \begin{bmatrix} \boldsymbol{\rho}_{\mathcal{X}_1}(g) \boldsymbol{x}_1^{\mathbb{B}} \\ \boldsymbol{\rho}_{\mathcal{X}_2}(g) \boldsymbol{x}_2^{\mathbb{B}} \end{bmatrix}, \; where \; \boldsymbol{Q}_{\mathbb{A}}^{\mathbb{B}} \boldsymbol{x} = \begin{bmatrix} \boldsymbol{x}_1^{\mathbb{B}} \in \mathcal{X}_1 \\ \boldsymbol{x}_2^{\mathbb{B}} \in \mathcal{X}_2 \end{bmatrix}$$

*Hence, the representation's decomposition $\rho_{\mathcal{X}} \sim \rho_{\mathcal{X}_1} \oplus \rho_{\mathcal{X}_2}$ corresponds to* **decomposing the vector space** *into $\mathbb{G}$-stable subspaces, $\mathcal{X} = \mathcal{X}_1 \oplus \mathcal{X}_2$.*

When iteratively applying the decomposition process, we eventually reach representations that cannot be further decomposed. These are known as irreducible representations, or *irreps*, and they serve as the fundamental building blocks for all representations of a compact symmetry group $\mathbb{G}$. From a vector space perspective, the irreducible $\mathbb{G}$-stable subspaces (Def. I.7) associated with these irreps are the elementary subspaces that comprise any symmetric vector space, analogous to how one-dimensional subspaces are the fundamental components of standard vector spaces.

**Definition I.9** (Irreducible representation). *Let $\mathcal{X}$ be a vector space endowed with a group action ($\triangleright$) of a symmetry group $\mathbb{G}$. A representation $\rho_{\mathcal{X}}$ of $\mathbb{G}$ on $\mathcal{X}$ is said to be* irreducible *if it cannot be decomposed into smaller representations acting on proper $\mathbb{G}$-stable subspaces (Def. I.7). That is, the only $\mathbb{G}$-stable subspaces $\mathcal{X}' \subseteq \mathcal{X}$ are $\mathcal{X}' = \{\mathbf{0}\}$ and $\mathcal{X}' = \mathcal{X}$ itself.*

*To differentiate irreps from decomposable representations we will denote the formers and their associated irreducible $\mathbb{G}$-stable spaces with an over bar: $\bar{\rho}_{\bar{\mathcal{V}}} : \mathbb{G} \to \mathbb{GL}(\bar{\mathcal{V}})$.*

Crucially, any compact symmetry group $\mathbb{G}$ has a unique set of countably many irreps, denoted by $\{\bar{\rho}_k\}_{k \in [1, n_{\mathrm{iso}}]}$, where $k$ denotes the irrep type and $n_{\mathrm{iso}} \leq |\mathbb{G}|$ denotes the number of unique irreps of $\mathbb{G}$. A fundamental property of these irreps is that any two non-equivalent irreps act on vector spaces that are mutually **orthogonal**. This implies that whenever we decompose symmetric vector spaces into their irreducible subspaces we are inherently decomposing the space into orthogonal $\mathbb{G}$-stable subspaces, which will greatly simplify numerical and theoretical analyses. Formally, these orthogonality relations are a consequence of Schur's lemma, which we state below in its original form for the case of complex irreps and discuss its adaptation to real irreps.

**Lemma I.10** (Schur's Lemma for unitary (complex) representations (Knapp, 1986, Prop 1.5)). *Consider two complex Hilbert spaces, $\mathcal{H}$ and $\mathcal{H}'$, endowed with the (complex) irreducible unitary representations $\bar{\phi} : \mathbb{G} \mapsto \mathbb{U}(\mathcal{H})$ and $\bar{\phi}' : \mathbb{G} \mapsto \mathbb{U}(\mathcal{H}')$, respectively. Let $\boldsymbol{T} : \mathcal{H} \mapsto \mathcal{H}'$ be a linear map commuting with the group actions, such that $\boldsymbol{T} \in Homo_{\mathbb{G}}(\mathcal{H}, \mathcal{H}')$. Then, if the irreducible representations are not equivalent, i.e., $\bar{\phi} \nsim \bar{\phi}'$, then $\boldsymbol{T}$ is the trivial (or zero) map. Conversely, if $\bar{\phi} \sim \bar{\phi}'$, then $\boldsymbol{T}$ is a constant multiple of an isomorphism. Denoting $\boldsymbol{I}$ as the identity operator, this can be expressed as:*

$$\bar{\phi} \nsim \bar{\phi}' \iff \qquad\qquad \mathbf{0}_{\mathcal{H}'} = \boldsymbol{T}\boldsymbol{h} \mid \forall\, \boldsymbol{h} \in \mathcal{H} \qquad\qquad (39a)$$

$$\bar{\phi} \sim \bar{\phi}' \iff \qquad\qquad \boldsymbol{T} = \alpha \boldsymbol{U}, \alpha \in \mathbb{C}, \boldsymbol{U} \cdot \boldsymbol{U}^H = \boldsymbol{I} \qquad\qquad (39b)$$

$$\bar{\phi} = \bar{\phi}' \iff \qquad\qquad \boldsymbol{T} = \alpha \boldsymbol{I}, \; \alpha \in \mathbb{C} \qquad\qquad (39c)$$

The most common interpretation of Schur's lemma is that whenever the irreps are equivalent, $\bar{\phi} \sim \bar{\phi}'$, their associated spaces are isomorphic, $\mathcal{H} \sim \mathcal{H}'$ and $\boldsymbol{T}$ is an element of the endomorphism space $\mathrm{End}_{\mathbb{G}}^{\mathbb{C}}(\bar{\mathcal{H}})$, with $\bar{\mathcal{H}} \sim \mathcal{H} \sim \mathcal{H}'$ (see Def. I.5). Consequently, (39b) implies that the endomorphism space is one-dimensional, i.e., $\dim(\mathrm{End}_{\mathbb{G}}^{\mathbb{C}}(\bar{\mathcal{H}})) = 1$, with $\alpha \in \mathbb{C}$ denoting the only degree of freedom, and (39c) denotes the scenario in which the basis sets for the two spaces are identical.

However, this result holds only for *complex* irreducible representations and requires adaptation for the case of real irreducible representations. The main difference stems from the fact that if $\bar{\rho} : \mathbb{G} \to \mathbb{GL}(\bar{\mathcal{V}})$ is a real irrep, then the space of (real) endomorphisms, $\mathrm{End}_{\mathbb{G}}(\bar{\mathcal{V}})$, is no longer one-dimensional, but rather it can be $1$, $2$, or $4$ dimensional, depending on whether $\mathrm{End}_{\mathbb{G}}(\mathcal{V})$ is isomorphic to the real algebra ($\mathbb{R} = \mathrm{span}\{1\}$), complex algebra ($\mathbb{C} = \mathrm{span}\{1, i \mid i^2 = -1\}$), or quaternionic algebra ($\mathbb{H} = \mathrm{span}\{1, i, j, k \mid i^2 = j^2 = k^2 = -1\}$), respectively. Denoting by $\Psi : \mathbb{K} \to \mathrm{End}_{\mathbb{G}}(\bar{\mathcal{V}})$ the isomorphism of

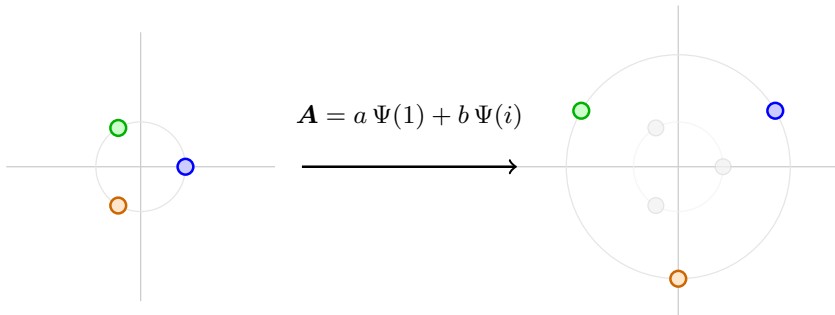

*Figure 20.* Example of an endomorphism acting on a $\mathbb{C}_3$-stable irreducible 2D space. The irreducible representation is of complex type, with endomorphism space $\text{End}_{\mathbb{C}_3}(\mathbb{R}^2) \sim \mathbb{C} = \text{span}\{1, i\}$, comprising all transformations that uniformly scale and rotate/reflect the plane.

basis elements of $\mathbb{K} \in \{\mathbb{R}, \mathbb{C}, \mathbb{H}\}$ with the basis elements of $\text{End}_{\mathbb{G}}(\bar{\mathcal{V}})$, we can summarize the basis sets of the three cases as follows (see Cesa et al. (2022, Appendix C) for details):

$$\mathbb{R} \sim \text{End}_{\mathbb{G}}(\mathcal{V}) = \text{span}\{\Psi(1) = \boldsymbol{I}_{\dim \bar{\phi}}\}$$

$$\mathbb{C} \sim \text{End}_{\mathbb{G}}(\mathcal{V}) = \text{span}\{\Psi(1) = \begin{bmatrix} \boldsymbol{I}_n & \boldsymbol{0} \\ \boldsymbol{0} & \boldsymbol{I}_n \end{bmatrix}, \Psi(i) = \begin{bmatrix} \boldsymbol{0} & -\boldsymbol{I}_n \\ \boldsymbol{I}_n & \boldsymbol{0} \end{bmatrix}\}$$

$$\mathbb{H} \sim \text{End}_{\mathbb{G}}(\mathcal{V}) = \text{span} \left\{ \begin{aligned} \Psi(1) = \begin{bmatrix} \boldsymbol{I}_n & \boldsymbol{0} & \boldsymbol{0} & \boldsymbol{0} \\ \boldsymbol{0} & \boldsymbol{I}_n & \boldsymbol{0} & \boldsymbol{0} \\ \boldsymbol{0} & \boldsymbol{0} & \boldsymbol{I}_n & \boldsymbol{0} \\ \boldsymbol{0} & \boldsymbol{0} & \boldsymbol{0} & \boldsymbol{I}_n \end{bmatrix}, \Psi(i) = \begin{bmatrix} \boldsymbol{0} & \boldsymbol{0} & -\boldsymbol{I}_n & \boldsymbol{0} \\ \boldsymbol{0} & \boldsymbol{0} & \boldsymbol{0} & -\boldsymbol{I}_n \\ \boldsymbol{I}_n & \boldsymbol{0} & \boldsymbol{0} & \boldsymbol{0} \\ \boldsymbol{0} & \boldsymbol{I}_n & \boldsymbol{0} & \boldsymbol{0} \end{bmatrix} \\ \Psi(j) = \begin{bmatrix} \boldsymbol{0} & -\boldsymbol{I}_n & \boldsymbol{0} & \boldsymbol{0} \\ \boldsymbol{I}_n & \boldsymbol{0} & \boldsymbol{0} & \boldsymbol{0} \\ \boldsymbol{0} & \boldsymbol{0} & \boldsymbol{0} & \boldsymbol{I}_n \\ \boldsymbol{0} & \boldsymbol{0} & -\boldsymbol{I}_n & \boldsymbol{0} \end{bmatrix}, \Psi(k) = \begin{bmatrix} \boldsymbol{0} & \boldsymbol{0} & \boldsymbol{0} & -\boldsymbol{I}_n \\ \boldsymbol{0} & \boldsymbol{0} & \boldsymbol{I}_n & \boldsymbol{0} \\ \boldsymbol{0} & -\boldsymbol{I}_n & \boldsymbol{0} & \boldsymbol{0} \\ \boldsymbol{I}_n & \boldsymbol{0} & \boldsymbol{0} & \boldsymbol{0} \end{bmatrix} \end{aligned} \right\} \tag{40}$$

While this result might appear complex, its interpretation is straightforward: given a $\mathbb{G}$-stable irreducible space $\bar{\mathcal{V}}$, the space of linear maps from the space to itself that preserve the symmetry structure consists of *linear transformations that scale all dimensions of $\bar{\mathcal{V}}$ uniformly, and possibly rotate or reflect the space*. Algebraically, this implies that any element of the algebra has a unique singular space, with a single singular value determined by the element's coefficients in the basis of (40). We summarize this result in the following proposition.

**Proposition I.11** (A real endomorphism has a single singular space). *Let $\mathbb{G}$ be a compact symmetry group, $(\bar{\rho}, \bar{\mathcal{V}})$ be an irreducible representation and its associated $\mathbb{G}$-stable space, and let $End_{\mathbb{G}}(\bar{\mathcal{V}})$ denote the space endomorphism algebra. Then every $\boldsymbol{A} \in End_{\mathbb{G}}(\bar{\mathcal{V}})$ admits an SVD*

$$\boldsymbol{A} = \boldsymbol{U} \gamma \boldsymbol{I}_d \boldsymbol{V}^\top,$$

*where $\gamma \in \mathbb{R}_{\geq 0}$ is the single singular value, repeated with multiplicity $d = |\bar{\rho}| = |\mathcal{V}_k|$. The right singular basis $\boldsymbol{V}$ can, without loss of generality, be taken as the canonical orthonormal basis of $\bar{\mathcal{V}}$, while the left singular basis is then $\boldsymbol{U} = \gamma^{-1} \boldsymbol{A} \boldsymbol{V}$, which is an orthogonal rotation/reflection of $\boldsymbol{V}$.*

($\mathbb{R}$) *Real case. Any $\boldsymbol{A} \in End_{\mathbb{G}}(\bar{\mathcal{V}})$ is of the form*

$$\boldsymbol{A} = a \Psi(1) = a \boldsymbol{I}_d, \qquad a \in \mathbb{R}.$$

*Hence*

$$\boldsymbol{A}^\top \boldsymbol{A} = a^2 \boldsymbol{I}_d, \qquad \sigma(\boldsymbol{A}) = \{|a|\}^{\times d}.$$

($\mathbb{C}$) *Complex case. Every element can be written as*

$$\boldsymbol{A} = a \Psi(1) + b \Psi(i) = \begin{bmatrix} a\boldsymbol{I}_n & -b\boldsymbol{I}_n \\ b\boldsymbol{I}_n & a\boldsymbol{I}_n \end{bmatrix}, \qquad a, b \in \mathbb{R}.$$

*Using $\Psi(i)^\top = -\Psi(i)$ and $\Psi(i)^\top \Psi(i) = \boldsymbol{I}$,*

$$\boldsymbol{A}^\top \boldsymbol{A} = (a^2 + b^2) \boldsymbol{I}_d, \qquad \sigma(\boldsymbol{A}) = \{\sqrt{a^2 + b^2}\}^{\times d}.$$

*(ℍ) Quaternionic case. Each element admits the expansion*

$$\boldsymbol{A} = a\,\Psi(1) + b\,\Psi(i) + c\,\Psi(j) + d\,\Psi(k), \qquad a, b, c, d \in \mathbb{R},$$

*where $\Psi(i), \Psi(j), \Psi(k)$ are the quaternionic structure matrices from (40), satisfying $\Psi(\alpha)^\top = -\Psi(\alpha)$, $\Psi(\alpha)^2 = -\boldsymbol{I}$, and $\Psi(\alpha)^\top \Psi(\alpha) = \boldsymbol{I}$, with the usual anti-commutation rules. Consequently,*

$$\boldsymbol{A}^\top \boldsymbol{A} = (a^2 + b^2 + c^2 + d^2)\,\boldsymbol{I}_d, \qquad \sigma(\boldsymbol{A}) = \big\{ \sqrt{a^2 + b^2 + c^2 + d^2} \big\}^{\times d}.$$

As an intuitive low-dimensional example, we can consider the case of a 2D rotational irrep of the cyclic group $\mathbb{C}_3$. The dimension of the irreducible $\mathbb{G}$-stable subspace is $\|barvsV\| = 2$, and the irreducible representation is of complex type, $\bar{\boldsymbol{\rho}} : \mathbb{C}_3 \to \mathbb{GL}(\bar{\mathcal{V}})$, with $\mathrm{End}_{\mathbb{G}}(\bar{\mathcal{V}}) \sim \mathbb{C} = \mathrm{span}\{1, i\}$ denoting the space of all rotations/reflections and uniform scaling of the plane, see Fig. 20.

## I.1. Maps between Symmetric Vector Spaces

We will frequently study and use linear and non-linear maps between symmetric vector spaces. Our focus is on maps that preserve entirely or partially the group structure of the vector spaces. These types of maps can be classified as $\mathbb{G}$-equivariant, $\mathbb{G}$-invariant maps:

**Definition I.12** ($\mathbb{G}$-equivariant and $\mathbb{G}$-invariant maps). *Let $\mathcal{X}$ and $\mathcal{Y}$ be two vector spaces endowed with the same symmetry group $\mathbb{G}$, with the respective group actions $\rhd_{\mathcal{X}}$ and $\rhd_{\mathcal{Y}}$. A map $f : \mathcal{X} \mapsto \mathcal{Y}$ is said to be $\mathbb{G}$-equivariant if it commutes with the group action, such that:*

$$g \rhd_{\mathcal{Y}} \boldsymbol{y} = g \rhd_{\mathcal{Y}} f(\boldsymbol{x}) = f(g \rhd_{\mathcal{X}} \boldsymbol{x}), \quad \forall \boldsymbol{x} \in \mathcal{X}, g \in \mathbb{G}. \tag{41a}$$
$$\boldsymbol{\rho}_{\mathcal{Y}}(g) f(\boldsymbol{x}) = f(\boldsymbol{\rho}_{\mathcal{X}}(g)\boldsymbol{x})$$

*A specific case of $\mathbb{G}$-equivariant maps are the $\mathbb{G}$-invariant ones, which are maps that commute with the group action and have trivial output group actions $\rhd_{\mathcal{Y}}$ such that $\boldsymbol{\rho}_{\mathcal{Y}}(g) = \boldsymbol{I}$ for all $g \in \mathbb{G}$. That is:*

$$\boldsymbol{y} = g \rhd_{\mathcal{Y}} f(\boldsymbol{x}) = f(g \rhd_{\mathcal{X}} \boldsymbol{x}), \quad \forall \boldsymbol{x} \in \mathcal{X}, g \in \mathbb{G}. \tag{41b}$$
$$\boldsymbol{y} = \boldsymbol{\rho}_{\mathcal{Y}}(g) f(\boldsymbol{x}) = f(\boldsymbol{\rho}_{\mathcal{X}}(g)\boldsymbol{x})$$

## Structure of $\mathbb{G}$-Equivariant Linear Maps

**Definition I.13** (Homomorphism and Isomorphism). *Let $\mathcal{X}$ and $\mathcal{Y}$ be two vector spaces endowed with the same symmetry group $\mathbb{G}$, with the respective group actions $\rhd_{\mathcal{X}}: \mathbb{G} \times \mathcal{X} \mapsto \mathcal{X}$ and $\rhd_{\mathcal{Y}}: \mathbb{G} \times \mathcal{Y} \mapsto \mathcal{Y}$. The spaces are said to be $\mathbb{G}$-homomorphic if there exists a linear map $\mathbb{A} : \mathcal{X} \mapsto \mathcal{Y}$ that commutes with the group action, such that $g \rhd_{\mathcal{Y}} (\boldsymbol{A}\boldsymbol{x}) = \boldsymbol{A}(g \rhd_{\mathcal{X}} \boldsymbol{x})$ for all $\boldsymbol{x} \in \mathcal{X}$. They are said to be $\mathbb{G}$-isomorphic if the linear map is invertible. Graphically, $\mathcal{X}$ and $\mathcal{Y}$ are $\mathbb{G}$-homomorphic or $\mathbb{G}$-isomorphic if the following diagrams commute:*

$$\boldsymbol{A} \in Homo_{\mathbb{G}}(\mathcal{X}, \mathcal{Y}) \qquad or \qquad \boldsymbol{A} \in Iso_{\mathbb{G}}(\mathcal{X}, \mathcal{Y}). \tag{42}$$

Homomorphism          Isomorphism

*Here, $Homo_{\mathbb{G}}(\mathcal{X}, \mathcal{Y})$ denotes the space of $\mathbb{G}$-equivariant linear maps between $\mathcal{X}$ and $\mathcal{Y}$, and $Iso_{\mathbb{G}}(\mathcal{X}, \mathcal{Y})$ denotes the space of $\mathbb{G}$-equivariant invertible linear maps between $\mathcal{X}$ and $\mathcal{Y}$.*

**Proposition I.14** (Structure of $\mathbb{G}$-homomorphisms / interwiners / $\mathbb{G}$-equivariant linear maps). *Let $\mathbb{G}$ be a compact group and $\boldsymbol{A} \in Homo_{\mathbb{G}}(\mathcal{X}, \mathcal{Y})$ be a $\mathbb{G}$-equivariant linear map between two (real) $\mathbb{G}$-symmetric vector spaces $\mathcal{X}$ and $\mathcal{Y}$, with isotypic decompostions:*

$$\mathcal{X} = \oplus_{k=1}^{n_{\mathrm{iso}}} \mathcal{X}^{(k)} = \oplus_{k=1}^{n_{\mathrm{iso}}} \oplus_{i=1}^{m_k^x} \mathcal{X}_i^{(k)} \quad and \quad \mathcal{Y} = \oplus_{k=1}^{n_{\mathrm{iso}}} \mathcal{Y}^{(k)} = \oplus_{k=1}^{n_{\mathrm{iso}}} \oplus_{j=1}^{m_k^y} \mathcal{Y}_j^{(k)},$$

*where $n_{\mathrm{iso}}$ denotes the number of isotypic subspaces, and $m_k^x$ and $m_k^y$ denote the multiplicities of the irreducible representation $\bar{\boldsymbol{\rho}}_k : \mathbb{G} \to \mathbb{GL}(\bar{\mathcal{V}}_k)$ in $\mathcal{X}$ and $\mathcal{Y}$, respectively. Each $\mathcal{X}_i^{(k)}$ and $\mathcal{Y}_j^{(k)}$ is isometrically isomorphic to $\bar{\mathcal{V}}_k$ (see Thm. I.17). Hence, in the isotypic bases, the map $\boldsymbol{A}$ decomposes block-diagonally into $n_{\mathrm{iso}}$ blocks corresponding to homomorphisms between isotypic subspaces of the same type, that is:*

$$\boldsymbol{A} = \oplus_{k=1}^{n_{\mathrm{iso}}} \boldsymbol{A}^{(k)} \qquad where \quad \boldsymbol{A}^{(k)} \in Homo_{\mathbb{G}}(\mathcal{X}^{(k)}, \mathcal{Y}^{(k)}).$$

*Furthermore, the map $\boldsymbol{A}^{(k)}$ decomposes into $m_k^x \times m_k^y$ blocks of endomorphisms of the irreducible subspace $\bar{\mathcal{V}}_k$. That is:*

$$\boldsymbol{A}^{(k)} = \begin{bmatrix} \boldsymbol{A}_{1,1}^{(k)} & \cdots & \boldsymbol{A}_{1,m_k^x}^{(k)} \\ \vdots & \ddots & \vdots \\ \boldsymbol{A}_{m_k^y,1}^{(k)} & \cdots & \boldsymbol{A}_{m_k^y,m_k^x}^{(k)} \end{bmatrix} \quad where \quad \boldsymbol{A}_{i,j}^{(k)} \in End_{\mathbb{G}}(\bar{\mathcal{V}}_k), \forall\ i \in [1, m_k^y], j \in [1, m_k^x].$$

*Consequently, depending of the type of irreducible representation $\mathbb{K} \in \{\mathbb{R}, \mathbb{C}, \mathbb{H}\}$, each sub-block is constrained to be in the span of the corresponding basis elements in Eq. (40). Consequently, if we denote by $\mathbb{B}$ the basis set of $\mathbb{K}$, we have that the map $\boldsymbol{A}^{(k)}$ can be expressed in tensor product form as:*

$$\boldsymbol{A}^{(k)} = \sum_{b \in \mathbb{B}} \boldsymbol{\Theta}_b^{(k)} \otimes \Psi_k(b), \quad where \quad \boldsymbol{\Theta}_b^{(k)} \in \mathbb{R}^{m_k^y \times m_k^x}, \Psi_k : \mathbb{B} \to End_{\mathbb{G}}(\bar{\mathcal{V}}_k) \tag{43}$$

*With $[\boldsymbol{\Theta}_b^{(k)}]_{i,j} = \langle \boldsymbol{A}_{i,j}^{(k)}, \Psi_k(b) \rangle$ denoting the basis expansion coefficient of the $i$-th, $j$-th endomorphism sub-block with the basis element $\Psi_k(b)$.*

## I.2. Isotypic Decomposition and Disentangled Representations

We now have all the necessary tools to decompose symmetric vector spaces into their fundamental building blocks: irreducible $\mathbb{G}$-stable subspaces (Def. I.7). This decomposition forms the theoretical foundation for disentangled representations, a concept introduced by Higgins et al. (2018) in the context of group theory and generalized here within the framework of representation theory.

By Maschke's theorem (Knapp, 1986), we have that irreducible representations are the fundamental building blocks of the representations of a compact symmetry group $\mathbb{G}$, given that any group representation $\boldsymbol{\rho}_{\mathcal{X}} : \mathbb{G} \to \mathbb{GL}(\mathcal{X})$ can be decomposed into a direct sum of irreducible representations, $\boldsymbol{\rho}_{\mathcal{X}} \sim \oplus_{i=1}^n \boldsymbol{\rho}_{\mathcal{X}_i}$, where each $\boldsymbol{\rho}_{\mathcal{X}_i}$ is isomorphic to one of the group's $n_{\mathrm{iso}} \leq |\mathbb{G}|$ irreducible representations (Defs. I.8 and I.9).

This decomposition will play a crucial role in facilitating numerical and theoretical analysis of operations on symmetric vector spaces. Therefore, we will frequenlt choose a convenient basis of the symmetric vector space which readily exposes this decomposition, termed isotypic basis. For the sake of generality we consider below the more general case of (finite and infinite dimensional) separable Hilbert spaces, which will enable us to extend this results to function spaces.

**Definition I.15** (Isotypic Basis). *Let $\boldsymbol{\rho}_{\mathcal{H}} : \mathbb{G} \to \mathbb{U}(\mathcal{H})$ be a unitary group representation of a compact group $\mathbb{G}$ on a separable Hilbert Space $\mathcal{H}$. The representation is said to be defined in an isotypic basis if it is defined by a direct sum of irreducible representations grouped by their type, that is, if:*

$$\boldsymbol{\rho}_{\mathcal{H}} = \oplus_{k=1}^{n_{\mathrm{iso}}} \oplus_{p=1}^{m_k} \bar{\boldsymbol{\rho}}_k, \tag{44}$$

*where $\{\bar{\boldsymbol{\rho}}_k : \mathbb{G} \to \mathbb{U}(\bar{\mathcal{H}}_k)\}_{k=1}^{n_{\mathrm{iso}}}$ are the $n_{\mathrm{iso}} \leq |\mathbb{G}|$ irreducible representations of $\mathbb{G}$, and $m_k \leq \infty$ is the multiplicity (i.e., number of copies) of the irrep type $k$ in the representation $\boldsymbol{\rho}_{\mathcal{H}}$.*

**Remark I.16.** Note that multiple isotypic bases exist for a given representation $\boldsymbol{\rho}_{\mathcal{H}}$, as both the irrep ordering and each irrep's multiplicity ordering can be arbitrarily permuted.

The utility of an isotypic basis stems from Schur's orthogonality relations (Lemma I.10), which ensure that any symmetric vector space decomposes into at most $n_{\text{iso}}$ orthogonal subspaces.

**Theorem I.17** (Isotypic decomposition of symmetric Hilbert spaces (Knapp, 1986)). *Let $\mathbb{G}$ be a compact group and $\mathcal{H}$ a separable Hilbert space with a unitary group representation $\boldsymbol{\rho}_{\mathcal{H}} : \mathbb{G} \to \mathbb{U}(\mathcal{H})$. Then we can identify $n_{\text{iso}} \leq |\mathbb{G}|$ irreducible representations $\bar{\boldsymbol{\rho}}_k : \mathbb{G} \to \mathbb{U}(\bar{\mathcal{H}}_k)$ that allow us to decompose $\mathcal{H}$ into a sum of orthogonal subspaces, denoted* isotypic *subspaces: $\mathcal{H} = \bigoplus^{\perp}_{1 \leq k \leq n_{\text{iso}}} \mathcal{H}^{(k)}$ where each $\mathcal{H}^{(k)} = \bigoplus^{m_k}_{p=1} \mathcal{H}^{(k)}_p$ is the sum of at most $m_k \leq \infty$ countably many subspaces isometrically isomorphic to $\bar{\mathcal{H}}_k$.*

**Remark I.18.** In practice, the isotypic decomposition of any finite-dimensional symmetric Hilbert space is obtained through a unitary or orthogonal change of basis. This change of basis can be computed numerically using Dixon's reduction method for finite-dimensional unitary (complex) representations (Dixon, 1970), with minor additional logic for the orthogonal (real) case. This method is implemented in `symm_learning` (Ordonez Apraez, 2026) through the `cplx_isotypic_decomposition` function.

For $\mathbb{G}$-equivariant NN backbone architecture, the torch module `symm_learning.nn.Change2DisentangledBasis` computes the required change of basis to an isotypic basis and applies it to the backbone output representations, thereby producing the disentangled representations used in this work and explain in detail next.

**Disentangled Representations**   The concept of isotypic decomposition is intricately linked to the idea of disentangled representations, introduced by Higgins et al. (2018) in the representation learning literature, restated below for completeness.

**Definition I.19** (Disentangled representation (Higgins et al. (2018))). *A vector representation is called a disentangled representation with respect to a particular decomposition of a symmetry group into subgroups, if it decomposes into independent subspaces, where each subspace is affected by the action of a single subgroup, and the actions of all other subgroups leave the subspace unaffected.*

Note that the independent subspaces of Def. I.19 refer to the orthogonal isotypic subspaces $\{\mathcal{H}^{(k)}\}^{n_{\text{iso}}}_{k=1}$, each of which is acted upon by a unique *quotient group*[5] defined by the kernel of the irrep acting on that isotypic subspace:

$$\mathbb{G}^{(k)} = \mathbb{G}/\mathbb{N}_k, \quad \text{where} \quad \mathbb{N}_k := \ker(\bar{\boldsymbol{\rho}}_k) = \{g \in \mathbb{G} \mid \bar{\boldsymbol{\rho}}_k(g) = \boldsymbol{I}_{d_k}\}. \tag{45}$$

Where each $\mathbb{G}^{(k)}$ is a well defined group of cosets generated by the normal subgroup $\mathbb{N}_k$. In practice, each $\mathbb{G}^{(k)}$ is isomorphic to the effective (matrix) group encoded by each irreducible representation $\bar{\boldsymbol{\rho}}_k : \mathbb{G} \mapsto \mathbb{U}(\bar{\mathcal{H}}_k)$.

## J. Representation Theory of Symmetric Function Spaces

In this section, we study symmetry group actions on infinite-dimensional function spaces and specify the conditions needed to approximate these spaces in finite dimensions. Specifically, given a set $\mathcal{X}$ with a compact symmetry group $\mathbb{G}$ acting via $(\triangleright)$ (Def. I.2), the space of scalar-valued functions on $\mathcal{X}$, $\mathcal{F} = \{f \mid f : \mathcal{X} \mapsto \mathbb{R}\}$, becomes a symmetric function space. The action of a symmetry transformation on a function is defined as:

**Definition J.1** (Group action on a function space). *Let $\mathcal{X}$ be a set endowed with the symmetry group $\mathbb{G}$, and let $\mathcal{F}$ be the space of scalar-valued functions on $\mathcal{X}$. The (left) action of $\mathbb{G}$ on a function $f \in \mathcal{F}$ is defined as the composition of $f$ with the inverse of the group element $g^{-1}$:*

$$\begin{aligned}(\triangleright_{\mathcal{F}}) : \quad &\mathbb{G} \times \mathcal{F} \quad \longrightarrow && \mathcal{F} \\ &(g, f) \quad \longrightarrow && [g \triangleright_{\mathcal{F}} f](\boldsymbol{x}) := [f \circ g^{-1}](\boldsymbol{x}) = f(g^{-1} \triangleright \boldsymbol{x}), \quad \forall \, \boldsymbol{x} \in \mathcal{X}.\end{aligned} \tag{46a}$$

*In other words, the point-wise evaluation of $f$ on a $g^{-1}$-transformed set $\mathcal{X}$ is equivalent to the evaluation of the transformed function $g \triangleright_{\mathcal{F}} f \in \mathcal{F}$ on the original set $\mathcal{X}$ (see simple examples in Fig. 21). Any function space that is stable under the group action Eq. (46a) is refereed to as a symmetric function space. Note that this action is compatible with the group composition and identity element $e \in \mathbb{G}$, such that the following properties hold:*

$$\text{Identity:} \quad e \triangleright_{\mathcal{F}} f(\cdot) = f(\cdot), \tag{46b}$$

$$\text{Associativity:} \quad [(g_2 \circ g_1) \triangleright_{\mathcal{F}} f](\cdot) = [g_2 \triangleright_{\mathcal{F}} [g_1 \triangleright_{\mathcal{F}} f]](\cdot), \qquad \forall \, g_1, g_2 \in \mathbb{G}. \tag{46c}$$

---

[5]The original definition of disentangled representations refers to subgroups, but in general the quotient grup $\mathbb{G}^{(k)}$ need not be a **subgroup** of $\mathbb{G}$.

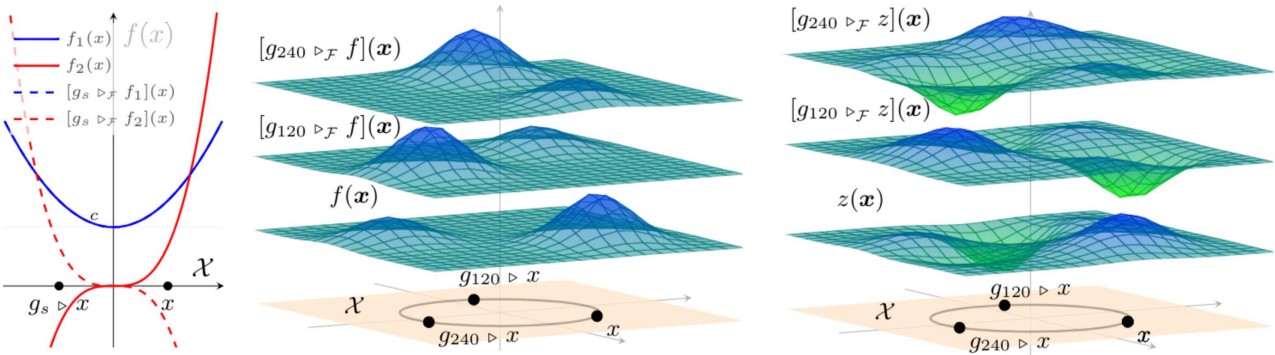

*Figure 21.* **Left:** Diagram of the group action $\rhd_\mathcal{F}$ on functions $f_1(x) = x^2 + c$ and $f_2(x) = x^3$ defined on the domain $\mathcal{X} := \mathbb{R}$ endowed with the reflectional symmetry group $\mathbb{G} := \mathbb{C}_2 = \{e, g_s\}$, with the reflection action acting on the domain by $g_s \rhd x = -x$ and on the function space $\mathcal{F} := \{f \mid f : \mathcal{X} \mapsto \mathbb{R}\}$ by $[g \rhd_\mathcal{F} f](\boldsymbol{x}) = f(g \rhd_\mathcal{X} \boldsymbol{x}) = f(-\boldsymbol{x})$. Hence we have that $f_1$ is a $\mathbb{G}$-invariant function, $g_s \rhd_\mathcal{F} f_1(x) = f_1(x)$ and $f_2$ a $\mathbb{G}$-equivariant function $g_s \rhd_\mathcal{F} f_2(x) = -x^3$. **Center:** Diagram representing the action $\rhd_\mathcal{F}$ on the (arbitrarily chosen) function $f(\boldsymbol{x}) = \mathcal{N}(\boldsymbol{x}; \boldsymbol{c}_1, 2) + \mathcal{N}(\boldsymbol{x}; \boldsymbol{c}_2, 1)$ defined over the symmetric domain $\mathcal{X} = \mathbb{R}^2$ with the cyclic symmetry group $\mathbb{G} = \mathbb{C}_3 = \{e, g_{120}, g_{240}\}$ and group action $g \rhd \boldsymbol{x} = \boldsymbol{\rho}_\mathcal{X}(g)\boldsymbol{x} = \boldsymbol{R}_g\boldsymbol{x}$, where $\boldsymbol{R}_g$ is a rotation matrix in 2D. Here, $g_{120} \rhd_\mathcal{F} f$ is equivalent to evaluating $f$ on a domain rotated by $-120°$. The same holds for $g_{240} \rhd_\mathcal{F} f$. Note that the $z$-offsets are added for visualization purposes. **Right:** Diagram representing the action $\rhd_\mathcal{F}$ on the function $z \in \widehat{\mathcal{F}}$, defined to be a member of the finite-dimensional symmetric function space $\widehat{\mathcal{F}} := \text{span}(\mathbb{I}_{\widehat{\mathcal{F}}})$, constructed from a basis set composed of the group orbit of the (arbitrarily chosen) function $f \in \mathcal{F}$, that is $\mathbb{I}_{\widehat{\mathcal{F}}} := \mathbb{G}f = \{f, g_{120} \rhd_\mathcal{F} f, g_{240} \rhd_\mathcal{F} f\}$. This function space is $\mathbb{G}$-stable by construction, since $\mathbb{G}\mathbb{I}_{\widehat{\mathcal{F}}} = \mathbb{I}_{\widehat{\mathcal{F}}}$. Note that the $z$-offsets are added for visualization purposes.

**Remark J.2.** From an algebraic perspective, the inversion $g^{-1}$ (*contragredient representation*) emerges to ensure that the associativity property of the group action (Eq. (46c)) holds:

$$[(g_2 \circ g_1) \rhd_\mathcal{F} f](\boldsymbol{x}) = [g_2 \rhd_\mathcal{F} [g_1 \rhd_\mathcal{F} f]](\boldsymbol{x}), \qquad \forall\, \boldsymbol{x} \in \mathcal{X}$$
$$f((g_2 \circ g_1)^{-1} \rhd \boldsymbol{x}) = [g_1 \rhd_\mathcal{F} f](g_2^{-1} \rhd \boldsymbol{x}) = f(g_1^{-1} \rhd (g_2^{-1} \rhd \boldsymbol{x}))$$
$$f((g_2 \circ g_1)^{-1} \rhd \boldsymbol{x}) = f((g_1 \circ g_2)^{-1} \rhd \boldsymbol{x}).$$

In the context of this work, we will study the scenario where the function space $\mathcal{F}$ is a separable Hilbert space and the group action of $\mathbb{G}$ on $\mathcal{F}$ is unitary, i.e., it preserves the inner product of the function space. This setup is crucial to enable us to approximate $\mathcal{F}$ and the group action on $\mathcal{F}$ in finite dimensions.

### J.1. Unitary Group Representation on Function Spaces

Assume our symmetric set $\mathcal{X}$ is endowed with a measure space structure $(\mathcal{X}, \Sigma_\mathcal{X}, P_\mathbf{x})$, where $P_\mathbf{x} : \Sigma_\mathcal{X} \mapsto \mathbb{R}$ is the space measure. Then, consider a function space with a separable Hilbert space structure $\mathcal{F} := \mathcal{L}^2_{P_\mathbf{x}}\mathcal{X}, \mathbb{R}$, and inner product $\langle f_1, f_2 \rangle_{P_\mathbf{x}} = \int_\mathcal{X} f_1(\boldsymbol{x})f_2(\boldsymbol{x})P_\mathbf{x}(d\boldsymbol{x})$ for all $f_1, f_2 \in \mathcal{F}$. Then, the action $\rhd_\mathcal{F}$ of the group $\mathbb{G}$ on the function space $\mathcal{F}$ is termed unitary if it preserves the inner product of the function space:

$$\langle f_1, f_2 \rangle_{P_\mathbf{x}} = \langle g \rhd_\mathcal{F} f_1, g \rhd_\mathcal{F} f_2 \rangle_{P_\mathbf{x}} \qquad \forall\, f_1, f_2 \in \mathcal{F}, g \in \mathbb{G}$$

$$\int_\mathcal{X} f_1(\boldsymbol{x})f_2(\boldsymbol{x})P_\mathbf{x}(d\boldsymbol{x}) = \int_\mathcal{X} (g \rhd_\mathcal{F} f_1)(\boldsymbol{x})(g \rhd_\mathcal{F} f_2)(\boldsymbol{x})P_\mathbf{x}(d\boldsymbol{x})$$

$$= \int_\mathcal{X} f_1(g^{-1} \rhd \boldsymbol{x})f_2(g^{-1} \rhd \boldsymbol{x})P_\mathbf{x}(d\boldsymbol{x}) \qquad (47)$$

$$= \int_{g \rhd \mathcal{X} = \mathcal{X}} f_1(\boldsymbol{x})f_2(\boldsymbol{x})P_\mathbf{x}(g \rhd d\boldsymbol{x}).$$

That is, the group action is unitary if $P_\mathbf{x}$ is a $\mathbb{G}$-invariant measure $P_\mathbf{x}(g \rhd d\boldsymbol{x}) = P_\mathbf{x}(d\boldsymbol{x})$, $\forall\, g \in \mathbb{G}, d\boldsymbol{x} \subseteq \mathcal{X}$. Note that an $\mathbb{G}$-invariant measure (and inner product) exists whenever $\mathbb{G}$ is finite, because for any measure $\eta : \Sigma_\Omega \mapsto \mathbb{R}$, we can use the group-average trick to obtain one, given by $P_\mathbf{x}(\mathbb{X}) = \Sigma_{g \in \mathbb{G}} \eta(g \rhd \mathbb{X})$.[6]

---

[6]Such a $\mathbb{G}$-invariant measure exists for any (finite or continuous) compact group. See discussion.

The importance of the Hilbert space structure is that it enables the definition of a unitary group representation. Unitary representations have a well-studied modular structure that allows their decomposition (Thm. I.17) into $\mathbb{G}$-stable subspaces (Def. I.7), which is crucial for approximating symmetric function spaces using a finite set of basis elements. Let $\mathbb{I}_{\mathcal{F}} = \{\phi_i \mid \phi_i \in \mathcal{L}_{\mathbf{x}}^2\}_{i \in \mathbb{N}}$ be an orthogonal basis for the function space $\mathcal{F} = \text{span}(\mathbb{I}_{\mathcal{F}})$, so that any function $f \in \mathcal{F}$ can be represented by its basis expansion coefficients $\boldsymbol{\alpha} = [\langle \phi_i \rangle_{P_{\mathbf{x}}} f]_{i \in \mathbb{N}}$, since $f_{\boldsymbol{\alpha}}(\boldsymbol{x}) = \sum_{i \in \mathbb{N}} \langle \phi_i, f \rangle_{P_{\mathbf{x}}} \phi_i(\boldsymbol{x})$. In this basis, the group action of $\mathbb{G}$ on $\mathcal{F}$ defines a unitary group representation mapping group elements to unitary linear integral operators on $\mathcal{F}$, which can be expressed in matrix form.

**Definition J.3** (Unitary group representation on a function space). *Let $\mathcal{F} = \mathcal{L}_{P_{\mathbf{x}}}^2 \mathcal{X}, \mathbb{R}$ be a separable Hilbert space of scalar-valued functions on a set $\mathcal{X}$ endowed with the symmetry group $\mathbb{G}$. Let $\mathbb{I}_{\mathcal{F}}$ be an orthogonal basis set spanning $\mathcal{F}$. Then, the group action of $\mathbb{G}$ on $\mathcal{F}$ (Def. J.1) defines a unitary group representation mapping group elements to unitary linear integral operators on $\mathcal{F}$:*

$$\boldsymbol{\rho}_{\mathcal{F}} : \begin{array}{ccc} \mathbb{G} & \longrightarrow & \mathbb{U}(\mathcal{F}) \\ g & \longrightarrow & \boldsymbol{\rho}_{\mathcal{F}}(g) \end{array}, \qquad s.t. \quad \boldsymbol{\rho}_{\mathcal{F}}(g)^* = \boldsymbol{\rho}_{\mathcal{F}}(g^{-1}). \tag{48}$$

Each unitary operator $\boldsymbol{\rho}_{\mathcal{F}}(g) : \mathcal{F} \mapsto \mathcal{F}$ admits an infinite-dimensional matrix representation with entries $[\boldsymbol{\rho}_{\mathcal{F}}(g)]_{i,j} := \langle \hat{f}_i, g \rhd_{\mathcal{F}} \hat{f}_j \rangle_{P_{\mathbf{x}}}$, which characterize how the group action transforms the chosen basis functions. Consequently, once the group representation for a chosen basis set is defined, the group action on a function $f_{\boldsymbol{\alpha}} \in \mathcal{F}$ can be expressed as an (infinite-dimensional) matrix transformation of its basis expansion coefficients $\boldsymbol{\alpha}$, given by:

$$[g \rhd_{\mathcal{F}} f_{\boldsymbol{\alpha}}](\cdot) := \sum_{i \in \mathbb{N}} \langle \hat{f}_i, g \rhd_{\mathcal{F}} f_{\boldsymbol{\alpha}} \rangle_{P_{\mathbf{x}}} \hat{f}_i(\cdot) = \sum_{i \in \mathbb{N}} \left( \sum_{j \in \mathbb{N}} \langle \hat{f}_i, g \rhd_{\mathcal{F}} \hat{f}_j \rangle_{P_{\mathbf{x}}} \underbrace{\langle \hat{f}_j, f \rangle_{P_{\mathbf{x}}}}_{\alpha_j} \right) \hat{f}_i(\cdot). \tag{49}$$

**Example J.4** (Isotypic decomposition of symmetric function space). Let $(\mathcal{X}, \Sigma_{\mathcal{X}}, P_{\mathbf{x}})$ be a symmetric 2D measure space with domain $\mathcal{X} \sim \mathbb{R}^2$ and cyclic symmetry group $\mathbb{G} := \mathbb{C}_3 = \{e, g_{120}, g_{240}\}$, acting on the 2D plane by $120°$ rotations (Fig. 22). Define the finite-dimensional function space $\mathcal{F}_{\mathbf{x}} \subset \mathcal{L}_{\mathbf{x}}^2$ with basis $\mathbb{I}_{\mathcal{F}_{\mathbf{x}}} = \{\phi, g_{120} \rhd \phi, g_{240} \rhd \phi\}$, where $\phi \in \mathcal{F}_{\mathbf{x}}$ is an arbitrary measurable function (Fig. 22-left). In this basis, the group action $\rhd_{\mathcal{F}_{\mathbf{x}}}$ for any function $z_{\boldsymbol{\alpha}} \in \mathcal{F}_{\mathbf{x}}$ is given by the regular representation $\boldsymbol{\rho}_{\mathcal{F}_{\mathbf{x}}} = \boldsymbol{\rho}_{\text{reg}}$ acting on the coefficient vector $\boldsymbol{\alpha} \in \mathbb{R}^3$ (Fig. 7-right).

$$[g \rhd_{\mathcal{F}_{\mathbf{x}}} z_{\boldsymbol{\alpha}}](\cdot) = \sum_{i=1}^3 \langle \phi_i, g \rhd_{\mathcal{F}_{\mathbf{x}}} z_{\boldsymbol{\alpha}} \rangle_{P_{\mathbf{x}}} \phi_i(\cdot) \equiv (\boldsymbol{\rho}_{\text{reg}}(g) \boldsymbol{\alpha})^\top \begin{bmatrix} \phi(\cdot) \\ g_{120} \rhd \phi(\cdot) \\ g_{240} \rhd \phi(\cdot) \end{bmatrix}, \quad \boldsymbol{\rho}_{\text{reg}}(g) = \begin{cases} \boldsymbol{I}_3, & \text{if } g = e \\ \begin{bmatrix} 0 & 1 & 0 \\ 0 & 0 & 1 \\ 1 & 0 & 0 \end{bmatrix}, & \text{if } g = g_{120} \\ \begin{bmatrix} 0 & 0 & 1 \\ 1 & 0 & 0 \\ 0 & 1 & 0 \end{bmatrix}, & \text{if } g = g_{240} \end{cases} \tag{50}$$

The group $\mathbb{C}_3$ possesses two types of (real-valued) irreducible representations, $n_{\text{iso}} = 2$: the trivial irreducible representation $\bar{\boldsymbol{\rho}}_{\text{inv}}$ and a 2D rotation representation $\bar{\boldsymbol{\rho}}_{2\pi/3}$, defined by:

$$\bar{\boldsymbol{\rho}}_{\text{inv}}(g) = \boldsymbol{I}_1, \forall g \in \mathbb{C}_3, \quad \text{and} \quad \bar{\boldsymbol{\rho}}_{2\pi/3}(g) = \begin{bmatrix} \cos(\theta) & -\sin(\theta) \\ \sin(\theta) & \cos(\theta) \end{bmatrix}, \quad s.t. \; \theta = \begin{cases} 0°, & \text{if } g = e \\ 120° & \text{if } g = g_{120} \\ 240° & \text{if } g = g_{240} \end{cases} \tag{51}$$

Applying the appropriate change of basis, we decompose the regular representation into a direct sum of the group's irreducible representations: $\boldsymbol{\rho}_{\text{reg}} = \boldsymbol{Q}(\bar{\boldsymbol{\rho}}_{\text{inv}} \oplus \bar{\boldsymbol{\rho}}_{2\pi/3})\boldsymbol{Q}^{-1}$, where $\boldsymbol{Q}$ transitions from the regular basis to the isotypic basis of $\mathcal{F}_{\mathbf{x}}$. Since $\mathbb{C}_3$ is abelian, $\boldsymbol{Q}$ corresponds to the linear map defining the Fourier transform.

By Thm. I.17, this results in the orthogonal decomposition of the finite-dimensional function space into two orthogonal subspaces; $\mathcal{F}_{\mathbf{x}} = \mathcal{F}_{\mathbf{x}}^{\text{inv}} \oplus^{\perp} \mathcal{F}_{\mathbf{x}}^{(2)}$, where $\mathcal{F}_{\mathbf{x}}^{\text{inv}}$ denotes the 1-dimensional subspace of $\mathbb{G}$-invariant functions ($\mathbb{G}^{(0)} = \{e\}$), and $\mathcal{F}_{\mathbf{x}}^{(2)}$ is the 2-dimensional subspace with group actions defined by the 2D irreducible representation $\bar{\boldsymbol{\rho}}_{2\pi/3}$ ($\mathbb{G}^{(1)} = \mathbb{C}_3$). We can construct the basis set in the isotypic basis given:

$$\mathbb{I}_{\mathcal{F}_{\mathbf{x}}}^{\text{iso}} = \boldsymbol{Q} \begin{bmatrix} \phi(\cdot) \\ g_{120} \rhd \phi(\cdot) \\ g_{240} \rhd \phi(\cdot) \end{bmatrix} = \begin{bmatrix} u^{\text{inv}}(\cdot) \\ u_1^{(2)}(\cdot) \\ u_2^{(2)}(\cdot) \end{bmatrix} \qquad s.t. \; \boldsymbol{Q} = \begin{bmatrix} 1/\sqrt{3} & 1/\sqrt{3} & 1/\sqrt{3} \\ 2/\sqrt{6} & -1/\sqrt{6} & -1/\sqrt{6} \\ 0 & 1/\sqrt{2} & -1/\sqrt{2} \end{bmatrix} \tag{52}$$

The new basis functions in the isotypic basis are depicted in Fig. 22-right, and elucidate that the symmetry constraints on this 3-dimensional function space, result in $m = 2$ unique functions, each associated with a unique irreducible representation.

Assuming $P_{\mathbf{x}}$ is a $\mathbb{G}$-invariant probability measure, we compute the expected value of each basis function. In the regular basis, functions related by a symmetry transformation share the same expected value, i.e., $\mathbb{E}_{\mathbf{x}} \phi = \mathbb{E}_{\mathbf{x}} g \rhd \phi$ for all $g \in \mathbb{C}_3$.

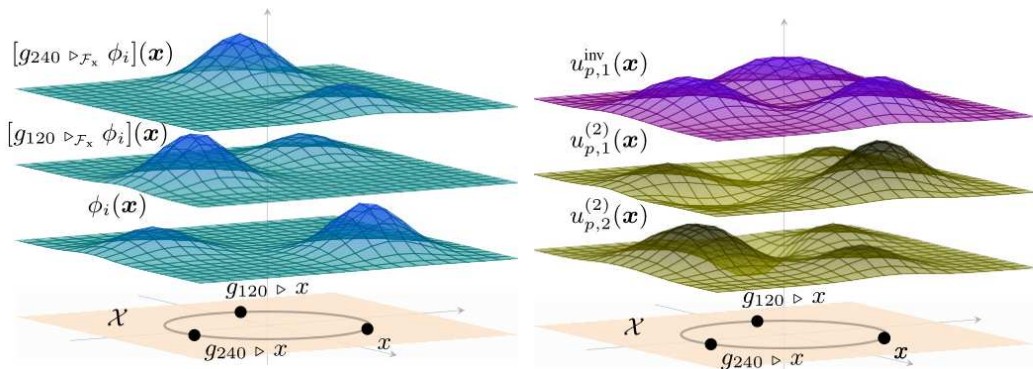

*Figure 22.* Visualization of the basis functions in the finite-dimensional symmetric function space $\mathcal{F}_\mathbf{x}$ from Example J.4. **Left**: Depiction of the basis functions in the regular basis $\mathbb{I}_{\mathcal{F}_\mathbf{x}} = \{\phi, g_{120} \triangleright \phi, g_{240} \triangleright \phi\}$, generated by the action of the cyclic group $\mathbb{C}_3$ on an arbitrary function $\phi \in \mathcal{F}_\mathbf{x}$. **Right**: Depiction of the basis functions in the isotypic basis $\mathbb{I}_{\mathcal{F}_\mathbf{x}}^{\text{iso}} = \{u^{\text{inv}}, u_1^{(2)}, u_2^{(2)}\}$, obtained via the change of basis matrix $\boldsymbol{Q}$. The first basis function $u^{\text{inv}}$ corresponds to the $\mathbb{G}$-invariant subspace $\mathcal{F}_\mathbf{x}^{\text{inv}}$ and is visually invariant under the action of $\mathbb{C}_3$ on $\mathcal{X}$. The other two basis functions $u_1^{(2)}, u_2^{(2)}$ are constrained to span a $\mathbb{G}$-stable subspace of $\mathcal{L}_\mathbf{x}^2$, denoted by $\mathcal{F}_\mathbf{x}^{(2)}$ that transform according to the irreducible representation $\bar{\boldsymbol{\rho}}_{2\pi/3}$. Meaning for any function $f \in \mathcal{F}_\mathbf{x}^{(2)}$, the group action $g \triangleright_{\mathcal{F}_\mathbf{x}} f$ can be computed by a linear transformation of its basis expansion coefficients.

In the isotypic basis, functions lacking a $\mathbb{G}$-invariant component (i.e., $u_1^{(2)}, u_2^{(2)}$) are centered: $\mathbb{E}_\mathbf{x} u_1^{(2)} = \mathbb{E}_\mathbf{x} u_2^{(2)} = 0$. In our example this constraint becomes clear from the nature of the change of basis $\boldsymbol{Q}$. Eq. (52).

## K. $\mathbb{G}$-Equivariant Linear Integral Operators

This section gives an overview of $\mathbb{G}$-equivariant linear integral operators between symmetric function spaces. We define these operators, discuss their properties, and specify conditions under which they commute with group actions. In App. K.1 we examine their infinite-dimensional matrix form and the resulting algebraic constraints from $\mathbb{G}$-equivariance. In App. K.2 we then show how to exploit these constraints in a finite-rank approximation.

Let $\mathbb{G}$ be a compact group acting on two measure spaces $(\mathcal{X}, \Sigma_\mathcal{X}, P_\mathbf{x})$ and $(\mathcal{Y}, \Sigma_\mathcal{Y}, P_\mathbf{y})$ via the group actions $\triangleright_\mathcal{X}$ and $\triangleright_\mathcal{Y}$ (see Def. I.2). Assume that the measures $P_\mathbf{x}$ and $P_\mathbf{y}$ are $\mathbb{G}$-invariant, i.e., $P_\mathbf{x}(g \triangleright_\mathcal{X} \mathbb{B}) = P_\mathbf{x}(\mathbb{B})$ and $P_\mathbf{y}(g \triangleright_\mathcal{Y} \mathbb{A}) = P_\mathbf{y}(\mathbb{A})$ for all $g \in \mathbb{G}, \mathbb{B} \in \Sigma_\mathcal{X}$, and $\mathbb{A} \in \Sigma_\mathcal{Y}$ (see Def. I.12).

Let $\mathcal{L}_\mathbf{x}^2 = \{f : \mathcal{X} \mapsto \mathbb{R} \mid \|f\|_{P_\mathbf{x}} < +\infty\}$ and $\mathcal{L}_\mathbf{y}^2 = \{h : \mathcal{Y} \mapsto \mathbb{R} \mid \|h\|_{P_\mathbf{y}} < +\infty\}$ be the Hilbert spaces of square-integrable functions with respect to $P_\mathbf{x}$ and $P_\mathbf{y}$, respectively. Since $\mathcal{X}$ and $\mathcal{Y}$ have a $\mathbb{G}$-action, the spaces $\mathcal{L}_\mathbf{x}^2$ and $\mathcal{L}_\mathbf{y}^2$ inherit group actions defined by $[g \triangleright_{\mathcal{L}_\mathbf{x}^2} f](\boldsymbol{x}) = f(g^{-1} \triangleright_\mathcal{X} \boldsymbol{x})$, $[g \triangleright_{\mathcal{L}_\mathbf{y}^2} h](\boldsymbol{y}) = h(g^{-1} \triangleright_\mathcal{Y} \boldsymbol{y})$, for all $f \in \mathcal{L}_\mathbf{x}^2$ and $h \in \mathcal{L}_\mathbf{y}^2$ (see Def. J.1).

We consider linear integral operators $\mathsf{T} : \mathcal{L}_\mathbf{x}^2 \mapsto \mathcal{L}_\mathbf{y}^2$ defined by

$$h(\boldsymbol{y}) = [\mathsf{T}f](\boldsymbol{y}) = \int_\mathcal{X} \kappa(\boldsymbol{x}, \boldsymbol{y}) f(\boldsymbol{x}) \, P_\mathbf{x}(d\boldsymbol{x}), \tag{53}$$

where $k : \mathcal{X} \times \mathcal{Y} \mapsto \mathbb{R}$ is the kernel function of $\mathsf{T}$. In this work we focus on those operators whose kernels are $\mathbb{G}$-invariant such operators are called $\mathbb{G}$-equivariant.

**Definition K.1** ($\mathbb{G}$-equivariant linear intergral operators). *Let $(\mathcal{X}, \Sigma_\mathcal{X}, P_\mathbf{x})$ and $(\mathcal{Y}, \Sigma_\mathcal{Y}, P_\mathbf{y})$ be two measure spaces endowed with group actions $\triangleright_\mathcal{X}$ and $\triangleright_\mathcal{Y}$ and $\mathbb{G}$-invariant measures $P_\mathbf{x}$ and $P_\mathbf{y}$ for a given compact symmetry group $\mathbb{G}$. Let $\mathsf{T} : \mathcal{L}_\mathbf{x}^2 \mapsto \mathcal{L}_\mathbf{y}^2$ be a linear integral operator between the spaces of square-integrable functions defined on the two measure spaces. The operator $\mathsf{T}$ is said to be $\mathbb{G}$-equivariant if it commutes with the group action, that is $\forall \ f \in \mathcal{L}_\mathbf{x}^2, g \in \mathbb{G}$ and*

$y \in \mathcal{Y}$:

$$[\mathsf{T}[g \rhd_{\mathcal{L}_x^2} f]](\boldsymbol{y}) = [g \rhd_{\mathcal{L}_y^2} [\mathsf{T}f]](\boldsymbol{y}) \tag{54a}$$

$$\int_{\mathcal{X}} \kappa(\boldsymbol{x}, \boldsymbol{y}) f(g^{-1} \rhd_{\mathcal{X}} \boldsymbol{x}) P_{\mathsf{x}}(d\boldsymbol{x}) = g \rhd_{\mathcal{L}_y^2} \left( \int_{\mathcal{X}} \kappa(\boldsymbol{x}, \boldsymbol{y}) f(\boldsymbol{x}) P_{\mathsf{x}}(d\boldsymbol{x}) \right)$$

$$\int_{\mathcal{X}} \kappa(g \rhd_{\mathcal{X}} \boldsymbol{x}, \boldsymbol{y}) f(\boldsymbol{x}) P_{\mathsf{x}}(g \rhd_{\mathcal{X}} d\boldsymbol{x}) = \int_{\mathcal{X}} \kappa(\boldsymbol{x}, g^{-1} \rhd_{\mathcal{Y}} \boldsymbol{y}) f(\boldsymbol{x}) P_{\mathsf{x}}(d\boldsymbol{x}) \qquad s.t. \ g \rhd_{\mathcal{X}} \mathcal{X} := \mathcal{X}$$

$$\int_{\mathcal{X}} \kappa(g \rhd_{\mathcal{X}} \boldsymbol{x}, \boldsymbol{y}) f(\boldsymbol{x}) P_{\mathsf{x}}(d\boldsymbol{x}) = \int_{\mathcal{X}} \kappa(\boldsymbol{x}, g^{-1} \rhd_{\mathcal{Y}} \boldsymbol{y}) f(\boldsymbol{x}) P_{\mathsf{x}}(d\boldsymbol{x}) \qquad s.t. \ P_{\mathsf{x}}(g \rhd_{\mathcal{X}} d\boldsymbol{x}) = P_{\mathsf{x}}(d\boldsymbol{x})$$

$$k(g \rhd_{\mathcal{X}} \boldsymbol{x}, \boldsymbol{y}) = \kappa(\boldsymbol{x}, g^{-1} \rhd_{\mathcal{Y}} \boldsymbol{y}) \iff k(g \rhd_{\mathcal{X}} \boldsymbol{x}, g \rhd_{\mathcal{Y}} \boldsymbol{y}) = \kappa(\boldsymbol{x}, \boldsymbol{y}). \tag{54b}$$

*Notice that the $\mathbb{G}$-equivariance of the operator $\mathsf{T}$ is linked to the $\mathbb{G}$-invariance of its kernel function, which is required to satisfy Eq. (54b).*

Multiple approaches exist to parameterize and approximate linear integral operators with finite resources (Kovachki et al., 2023, sec. 4). Here, we assume that both the input and output function spaces are separable Hilbert spaces, so that the operator can be represented as an infinite-dimensional matrix once appropriate basis sets are chosen. Its finite-dimensional (truncated or finite-rank) approximation is then obtained by selecting a finite number of basis functions in each space.

### K.1. Infinite-Dimensional Matrix Form of the Operator

Since $\mathcal{L}_{\mathsf{x}}^2$ and $\mathcal{L}_{\mathsf{y}}^2$ are Hilbert spaces with inner products $\langle \cdot, \cdot \rangle_{P_{\mathsf{x}}}$ and $\langle \cdot, \cdot \rangle_{P_{\mathsf{y}}}$ respectively, we can choose orthogonal bases for both spaces: $\mathbb{I}_{\mathcal{L}_{\mathsf{x}}^2} = \{\phi_i \mid \phi_i \in \mathcal{L}_{\mathsf{x}}^2\}_{i \in \mathbb{N}}$ and $\mathbb{I}_{\mathcal{L}_{\mathsf{y}}^2} = \{\psi_j \mid \psi_j \in \mathcal{L}_{\mathsf{y}}^2\}_{j \in \mathbb{N}}$. This choice allows any function $f \in \mathcal{L}_{\mathsf{x}}^2$ and $h \in \mathcal{L}_{\mathsf{y}}^2$ to be represented by their infinite-dimensional coefficient vectors $\boldsymbol{\alpha} = [\langle \phi_i, f \rangle_{P_{\mathsf{x}}}]_{i \in \mathbb{N}}$ and $\boldsymbol{\beta} = [\langle \psi_j, h \rangle_{P_{\mathsf{y}}}]_{j \in \mathbb{N}}$, so that:

$$f(\boldsymbol{x}) := f_{\boldsymbol{\alpha}}(\boldsymbol{x}) = \sum_{i=1}^{\infty} \langle \phi_i, f \rangle_{P_{\mathsf{x}}} \phi_i(\boldsymbol{x}) \equiv \boldsymbol{\alpha}^T \boldsymbol{\phi}(\boldsymbol{x}) \qquad h(\boldsymbol{y}) := h_{\boldsymbol{\beta}}(\boldsymbol{y}) = \sum_{j=1}^{\infty} \langle \psi_j, h \rangle_{P_{\mathsf{y}}} \psi_j(\boldsymbol{y}) \equiv \boldsymbol{\beta}^T \boldsymbol{\psi}(\boldsymbol{y}) \tag{55}$$

Here, $\boldsymbol{\alpha}^T \boldsymbol{\phi}(\boldsymbol{x})$ and $\boldsymbol{\beta}^T \boldsymbol{\psi}(\boldsymbol{y})$ represent the function as the dot product of its expansion coefficients with the basis evaluations $\boldsymbol{\phi}(\boldsymbol{x}) = [\phi_i(\boldsymbol{x})]_{i \in \mathbb{N}}$ and $\boldsymbol{\psi}(\boldsymbol{y}) = [\psi_j(\boldsymbol{y})]_{j \in \mathbb{N}}$. This notation is useful when we later select a finite number of basis functions to form a finite-dimensional approximation of $\mathsf{T}$.

With the chosen bases, the action of a linear integral operator $\mathsf{T} : \mathcal{L}_{\mathsf{y}}^2 \to \mathcal{L}_{\mathsf{x}}^2$ on any $f \in \mathcal{L}_{\mathsf{x}}^2$ is determined by its action on the basis functions:

$$[\mathsf{T}f_{\boldsymbol{\alpha}}](\boldsymbol{y}) = \int_{\mathcal{X}} \kappa(\boldsymbol{x}, \boldsymbol{y}) \left( \sum_{i \in \mathbb{N}} \alpha_i \phi_i(\boldsymbol{x}) \right) P_{\mathsf{x}}(d\boldsymbol{x}) = \sum_{i \in \mathbb{N}} \alpha_i \int_{\mathcal{X}} \kappa(\boldsymbol{x}, \boldsymbol{y}) \phi_i(\boldsymbol{x}) P_{\mathsf{x}}(d\boldsymbol{x}) = \sum_{i \in \mathbb{N}} \alpha_i [\mathsf{T}\phi_i](\boldsymbol{y}) \tag{56}$$

Since $[\mathsf{T}\phi_i] \in \mathcal{L}_{\mathsf{y}}^2$, each $[\mathsf{T}\phi_i](\boldsymbol{y})$ can be expanded using the output basis as $[\mathsf{T}\phi_i](\boldsymbol{y}) = \sum_{j \in \mathbb{N}} \langle \psi_j, \mathsf{T}\phi_i \rangle_{P_{\mathsf{y}}} \psi_j(\boldsymbol{y})$. Thus, the operator $\mathsf{T}$ can be represented by the infinite-dimensional matrix $\boldsymbol{T}$ with entries $\boldsymbol{T}_{ij} = \langle \psi_i, \mathsf{T}\phi_j \rangle_{P_{\mathsf{y}}}$. Therefore, the action of $\mathsf{T}$ on any $f_{\boldsymbol{\alpha}} \in \mathcal{L}_{\mathsf{x}}^2$ is given by the matrix–vector product $\boldsymbol{\beta} = \boldsymbol{T} \boldsymbol{\alpha}$, i.e.,

$$[\mathsf{T}f_{\boldsymbol{\alpha}}](\boldsymbol{y}) = \sum_{j \in \mathbb{N}} \alpha_j [\mathsf{T}\phi_j](\boldsymbol{y}) = \sum_{j \in \mathbb{N}} \alpha_j \sum_{i \in \mathbb{N}} \langle \psi_i, \mathsf{T}\phi_j \rangle_{P_{\mathsf{y}}} \psi_i(\boldsymbol{y})$$
$$= \sum_{i \in \mathbb{N}} \sum_{j \in \mathbb{N}} \boldsymbol{T}_{ij} \alpha_j \psi_i(\boldsymbol{y}) \equiv (\boldsymbol{T} \boldsymbol{\alpha})^T \boldsymbol{\psi}(\boldsymbol{y}) \tag{57}$$

Eq. (57) shows that knowing the action of $\mathsf{T}$ on the bases $\mathbb{I}_{\mathcal{L}_{\mathsf{x}}^2}$ and $\mathbb{I}_{\mathcal{L}_{\mathsf{y}}^2}$ determines its action on any function in $\mathcal{L}_{\mathsf{x}}^2$. In the sections that follow, we describe how symmetry constrains this action by requiring the bases to be $\mathbb{G}$-stable and by imposing $\mathbb{G}$-equivariance on $\boldsymbol{T}$, thereby introducing exploitable algebraic constraints for improved finite-rank approximations.

### K.1.1. $\mathbb{G}$-EQUIVARIANT MATRIX FORM OF THE OPERATOR

Whenever the function spaces carry a symmetry group $\mathbb{G}$, the group action on their bases $\mathbb{I}_{\mathcal{L}_{\mathsf{x}}^2}$ and $\mathbb{I}_{\mathcal{L}_{\mathsf{y}}^2}$ is defined by the unitary representations $\boldsymbol{\rho}_{\mathcal{L}_{\mathsf{x}}^2} : \mathbb{G} \to \mathbb{U}()\mathcal{L}_{\mathsf{x}}^2$ and $\boldsymbol{\rho}_{\mathcal{L}_{\mathsf{y}}^2} : \mathbb{G} \to \mathbb{U}()\mathcal{L}_{\mathsf{y}}^2$ (see Def. J.3). As in Eq. (57), these representations can be

expressed in (infinite-dimensional) matrix form so that the group action is given by a matrix-vector product:

$$
\begin{aligned}
{[g \rhd_{\mathcal{L}_{\mathbf{x}}^2} f_{\boldsymbol{\alpha}}](\cdot)} &\equiv (\boldsymbol{\rho}_{\mathcal{L}_{\mathbf{x}}^2}(g)\boldsymbol{\alpha})^T \boldsymbol{\phi}(\cdot), \quad \forall f_{\boldsymbol{\alpha}} \in \mathcal{L}_{\mathbf{x}}^2, g \in \mathbb{G} \\
{[g \rhd_{\mathcal{L}_{\mathbf{y}}^2} h_{\boldsymbol{\beta}}](\cdot)} &\equiv (\boldsymbol{\rho}_{\mathcal{L}_{\mathbf{y}}^2}(g)\boldsymbol{\beta})^T \boldsymbol{\psi}(\cdot), \quad \forall h_{\boldsymbol{\beta}} \in \mathcal{L}_{\mathbf{y}}^2, g \in \mathbb{G}
\end{aligned}
\tag{58}
$$

Since the operator $\mathsf{T}$ is $\mathbb{G}$-equivariant by construction (Eq. (54a)), the matrix form $\boldsymbol{T}$ of the operator must also be $\mathbb{G}$-equivariant with respect to the group representations $\boldsymbol{\rho}_{\mathcal{L}_{\mathbf{x}}^2}$ and $\boldsymbol{\rho}_{\mathcal{L}_{\mathbf{y}}^2}$:

$$
\begin{aligned}
{[\mathsf{T}[g \rhd_{\mathcal{L}_{\mathbf{x}}^2} f_{\boldsymbol{\alpha}}]](\boldsymbol{y})} &= [g \rhd_{\mathcal{L}_{\mathbf{y}}^2} [\mathsf{T}f_{\boldsymbol{\alpha}}]](\boldsymbol{y}) && \forall f_{\boldsymbol{\alpha}} \in \mathcal{L}_{\mathbf{x}}^2, g \in \mathbb{G}, \boldsymbol{y} \in \mathcal{Y} \\
(\boldsymbol{T}\boldsymbol{\rho}_{\mathcal{L}_{\mathbf{x}}^2}(g)\boldsymbol{\alpha})^\top \boldsymbol{\psi}(\boldsymbol{y}) &= (\boldsymbol{\rho}_{\mathcal{L}_{\mathbf{y}}^2}(g)\boldsymbol{T}\boldsymbol{\alpha})^\top \boldsymbol{\psi}(\boldsymbol{y}) && \text{s.t. Eqs. (57) and (58)} \\
\boldsymbol{T}\boldsymbol{\rho}_{\mathcal{L}_{\mathbf{x}}^2}(g) &= \boldsymbol{\rho}_{\mathcal{L}_{\mathbf{y}}^2}(g)\boldsymbol{T}
\end{aligned}
\tag{59}
$$

With bases $\mathbb{I}_{\mathcal{L}_{\mathbf{x}}^2}$ and $\mathbb{I}_{\mathcal{L}_{\mathbf{y}}^2}$ for $\mathcal{L}_{\mathbf{x}}^2$ and $\mathcal{L}_{\mathbf{y}}^2$, the kernel (Def. K.1) can be written as $\kappa(\boldsymbol{x},\boldsymbol{y}) = \sum_{i,j\in\mathbb{N}} \boldsymbol{T}_{i,j}\, \phi_j(\boldsymbol{x})\, \psi_i(\boldsymbol{y})$. Hence, the $\mathbb{G}$-invariance condition (Eq. (54b)) on the kernel directly implies that the matrix $\boldsymbol{T}$ is $\mathbb{G}$-equivariant, as stated in the following proposition:

**Proposition K.2** ($\mathbb{G}$-invariant kernel implies $\mathbb{G}$-equivariant matrix form). *Let $\mathsf{T} : \mathcal{L}_{\mathbf{x}}^2 \mapsto \mathcal{L}_{\mathbf{y}}^2$ be a $\mathbb{G}$-equivariant operator between symmetric function spaces endowed with the group actions $\rhd_{\mathcal{L}_{\mathbf{x}}^2}$ and $\rhd_{\mathcal{L}_{\mathbf{y}}^2}$ of a compact symmetry group $\mathbb{G}$. Let $\boldsymbol{\rho}_{\mathcal{L}_{\mathbf{x}}^2}$ and $\boldsymbol{\rho}_{\mathcal{L}_{\mathbf{y}}^2}$ be the group representation of the on the input/output function spaces on the chosen basis sets $\mathbb{I}_{\mathcal{L}_{\mathbf{x}}^2}$ and $\mathbb{I}_{\mathcal{L}_{\mathbf{y}}^2}$. Then the $\mathbb{G}$-invariance of the operator's kernel function (Eq. (54b)) implies that the matrix form of the operator, in the chosen basis sets, is $\mathbb{G}$-equivariant w.r.t the group representations $\boldsymbol{\rho}_{\mathcal{L}_{\mathbf{x}}^2}$ and $\boldsymbol{\rho}_{\mathcal{L}_{\mathbf{y}}^2}$ (Eq. (59)).*

*Proof.* The proof follows by choosing appropriate $\mathbb{G}$-stable basis sets $\{\phi_i\} \subset \mathcal{L}_{\mathbf{x}}^2$ and $\{\psi_j\} \subset \mathcal{L}_{\mathbf{y}}^2$, so that for all $g \in \mathbb{G}$ we have $g \rhd_{\mathcal{L}_{\mathbf{x}}^2} \phi_i = \phi_{g\rhd i}$ and $g \rhd_{\mathcal{L}_{\mathbf{y}}^2} \psi_j = \psi_{g\rhd j}$ with $g \rhd i, g \rhd j \in \mathbb{N}$ [7]. This basis sets the $\mathbb{G}$-invariance of the kernel translates into algebraic constraints on the matrix form $\boldsymbol{T}$.

$$
\begin{aligned}
k(\boldsymbol{x},\boldsymbol{y}) &= \kappa(g^{-1} \rhd_{\mathcal{X}} \boldsymbol{x}, g^{-1} \rhd_{\mathcal{Y}} \boldsymbol{y}) && \forall g \in \mathbb{G}, \boldsymbol{x} \in \mathcal{X}, \boldsymbol{y} \in \mathcal{Y} \\
\sum_{i\in\mathbb{N}} \sum_{j\in\mathbb{N}} \boldsymbol{T}_{i,j}\phi_i(\boldsymbol{x})\psi_j(\boldsymbol{y}) &= \sum_{i\in\mathbb{N}} \sum_{j\in\mathbb{N}} \boldsymbol{T}_{i,j}[g \rhd_{\mathcal{L}_{\mathbf{x}}^2} \phi_i](\boldsymbol{x})[g \rhd_{\mathcal{Y}} \psi_j](\boldsymbol{y}) = \sum_{i\in\mathbb{N}} \sum_{j\in\mathbb{N}} \boldsymbol{T}_{i,j}\phi_{g\rhd i}(\boldsymbol{x})\psi_{g\rhd j}(\boldsymbol{y})
\end{aligned}
\tag{60}
$$

That is, the kernel is $\mathbb{G}$-equivariant if the operator's matrix satisfies $\boldsymbol{T}_{i,j} = \boldsymbol{T}_{g\rhd i,\, g\rhd j}$ for all $g \in \mathbb{G}$, $i,j \in \mathbb{N}$. This condition exactly characterizes the $\mathbb{G}$-equivariance of the matrix form.

$$
\begin{aligned}
\boldsymbol{T}_{i,j} = \langle \psi_i, \mathsf{T}\phi_j \rangle_{P_{\mathbf{y}}} = \langle \psi_{g\rhd i}, \mathsf{T}\phi_{g\rhd j} \rangle_{P_{\mathbf{y}}} &= \boldsymbol{T}_{g\rhd i, g\rhd j} && \forall g \in \mathbb{G}, i,j \in \mathbb{N} \\
&= \langle g \rhd_{\mathcal{L}_{\mathbf{y}}^2} \psi_i, \mathsf{T}[g \rhd_{\mathcal{L}_{\mathbf{x}}^2} \phi_j] \rangle_{P_{\mathbf{y}}} \\
&= \langle g \rhd_{\mathcal{L}_{\mathbf{y}}^2} \psi_i, g \rhd_{\mathcal{L}_{\mathbf{y}}^2} [\mathsf{T}\phi_j] \rangle_{P_{\mathbf{y}}} && \text{s.t. Eq. (54a)} \\
&= \langle \psi_i, \mathsf{T}\phi_j \rangle_{P_{\mathbf{y}}} = \boldsymbol{T}_{i,j} && \text{s.t. Eq. (47)}
\end{aligned}
\tag{61}
$$

$\square$

### K.1.2. BLOCK-DIAGONAL STRUCTURE OF THE OPERATOR MATRIX FORM

According to Thm. I.17, a Hilbert space with a compact symmetry group $\mathbb{G}$ decomposes into $n_{\text{iso}}$ orthogonal subspaces—one for each irreducible representation type—yielding an orthogonal decomposition of the operator's input and output spaces:

$$
\mathcal{L}_{\mathbf{x}}^2 := \oplus_{1 \le k \le n_{\text{iso}}}^{\perp} \mathcal{L}_{\mathbf{x}}^{2(k)}, \qquad \text{and} \qquad \mathcal{L}_{\mathbf{y}}^2 := \oplus_{1 \le k \le n_{\text{iso}}}^{\perp} \mathcal{L}_{\mathbf{y}}^{2(k)},
\tag{62}
$$

where $\mathcal{L}_{\mathbf{x}}^{2(k)}$ and $\mathcal{L}_{\mathbf{y}}^{2(k)}$ denote the $k$-th isotypic subspaces of $\mathcal{L}_{\mathbf{x}}^2$ and $\mathcal{L}_{\mathbf{y}}^2$, respectively. Such that any function in these spaces can be decomposed into a sum of its projections onto the isotypic subspaces:

$$
f(\boldsymbol{x}) = \sum_{k=1}^{n_{\text{iso}}} f^{(k)}(\boldsymbol{x}), \quad h(\boldsymbol{y}) = \sum_{k=1}^{n_{\text{iso}}} h^{(k)}(\boldsymbol{y}) \quad \text{with} \quad f^{(k)} \in \mathcal{L}_{\mathbf{x}}^{2(k)}, h^{(k)} \in \mathcal{L}_{\mathbf{y}}^{2(k)}.
\tag{63}
$$

The orthogonal decomposition of the function spaces implies there exist unitary operators $\mathsf{A} : \mathcal{L}_{\mathbf{x}}^2 \to \mathcal{L}_{\mathbf{x}}^2$ and $\mathsf{B} : \mathcal{L}_{\mathbf{y}}^2 \to \mathcal{L}_{\mathbf{y}}^2$ (with matrix forms $\boldsymbol{A}$ and $\boldsymbol{B}$), that describe a change of basis from the canonical basis to an *isotypic basis*, $\mathbb{I}_{\mathcal{L}_{\mathbf{x}}^2}^{\text{iso}} = \cup_{k=1}^{n_{\text{iso}}} \mathbb{I}_{\mathcal{L}_{\mathbf{x}}^{2(k)}} =$

---

[7]By Cayley's theorem (Cayley, 1854), every finite symmetry group is isomorphic to a permutation group. From a representation-theoretic perspective, this guarantees the existence of a permutation (regular) basis for any finite group.

$A\mathbb{I}_{\mathcal{L}_{\mathbf{x}}^2}$ and $\mathbb{I}_{\mathcal{L}_{\mathbf{y}}^2}^{\text{iso}} = \cup_{k=1}^{n_{\text{iso}}} \mathbb{I}_{\mathcal{L}_{\mathbf{y}}^{2(k)}} = B\mathbb{I}_{\mathcal{L}_{\mathbf{x}}^2}$, where the group's representations decompose into a direct sum of representations per isotypic subspace (see Def. I.6):

$$\boldsymbol{\rho}_{\mathcal{L}_{\mathbf{x}}^2}^{\text{iso}}(\cdot) := \boldsymbol{A}\boldsymbol{\rho}_{\mathcal{L}_{\mathbf{x}}^2}(\cdot)\boldsymbol{A}^* = \oplus_{k=1}^{n_{\text{iso}}}\boldsymbol{\rho}_{\mathcal{L}_{\mathbf{x}}^{2(k)}}(\cdot) \quad \text{and} \quad \boldsymbol{\rho}_{\mathcal{L}_{\mathbf{y}}^2}^{\text{iso}}(\cdot) := \boldsymbol{B}\boldsymbol{\rho}_{\mathcal{L}_{\mathbf{y}}^2}(\cdot)\boldsymbol{B}^* = \oplus_{k=1}^{n_{\text{iso}}}\boldsymbol{\rho}_{\mathcal{L}_{\mathbf{y}}^{2(k)}}(\cdot). \tag{64}$$

Then, denoting the matrix form of $\mathsf{T}$ in the isotypic basis by $\boldsymbol{T}^{\text{iso}} = \boldsymbol{B}^*\boldsymbol{T}\boldsymbol{A}$, the $\mathbb{G}$-equivariance of $\mathsf{T}$ results in the matrix form of the operator in the isotypic basis being block-diagonal, with each block being $\mathbb{G}$-equivariant with respect to the group representations on the isotypic subspaces:

$$
\begin{aligned}
\boldsymbol{T}^{\text{iso}} &= \boldsymbol{\rho}_{\mathcal{L}_{\mathbf{y}}^2}^{\text{iso}}(g)\boldsymbol{T}^{\text{iso}}\boldsymbol{\rho}_{\mathcal{L}_{\mathbf{x}}^2}^{\text{iso}}(g^{-1}) \\
&= \oplus_{k=1}^{n_{\text{iso}}}\boldsymbol{\rho}_{\mathcal{L}_{\mathbf{y}}^{2(k)}}(g)\boldsymbol{T}^{\text{iso}} \oplus_{k=1}^{n_{\text{iso}}} \boldsymbol{\rho}_{\mathcal{L}_{\mathbf{x}}^{2(k)}}(g^{-1}) \qquad \text{s.t. Eqs. (59) and (64)}
\end{aligned}
\tag{65}
$$

$$\boldsymbol{T}^{(k)} = \boldsymbol{\rho}_{\mathcal{L}_{\mathbf{y}}^{2(k)}}(g)\boldsymbol{T}^{(k)}\boldsymbol{\rho}_{\mathcal{L}_{\mathbf{x}}^{2(k)}}(g^{-1}), \quad \forall\, k = 1, \ldots, n_{\text{iso}} \quad \boldsymbol{T}^{\text{iso}} = \oplus_{k=1}^{n_{\text{iso}}}\boldsymbol{T}^{(k)} = \begin{bmatrix} \boldsymbol{T}^{(1)} & & \\ & \ddots & \\ & & \boldsymbol{T}^{(n_{\text{iso}})} \end{bmatrix}.$$

Each $\boldsymbol{T}^{(k)}$ represents the matrix form of the operator $\mathsf{T}^{(k)} : \mathcal{L}_{\mathbf{x}}^{2(k)} \mapsto \mathcal{L}_{\mathbf{y}}^{2(k)}$ in the isotypic basis, acting on the isotypic subspaces of type $k$ in the input and output spaces. This shows that $\mathbb{G}$-equivariant operators preserve the structure of isotypic subspaces without mixing functions from different types.

This property is crucial for the finite-rank approximation of the operator $\mathsf{T}$, as it reduces the problem to approximating lower-rank operators $\mathsf{T}^{(k)} : \mathcal{L}_{\mathbf{x}}^{2(k)} \mapsto \mathcal{L}_{\mathbf{y}}^{2(k)}$, for $k \in [1, n_{\text{iso}}]$. Moreover, the block diagonal structure of $\boldsymbol{T}^{\text{iso}}$ allows us to rewrite Eq. (57) in the isotypic basis in terms of the action of each $\mathsf{T}^{(k)}$ on the projection $f^{(k)}$ of the function onto the $k^{\text{th}}$ isotypic subspace, see (63), such that:

$$[\mathsf{T}f_{\boldsymbol{\alpha}}](\boldsymbol{y}) = \sum_{k=1}^{n_{\text{iso}}}[\mathsf{T}^{(k)}f^{(k)}](\boldsymbol{y}) \equiv \sum_{k=1}^{n_{\text{iso}}}(\boldsymbol{T}^{(k)}\boldsymbol{\alpha}^{(k)})^{\top}\boldsymbol{\psi}^{(k)}(\boldsymbol{y}). \qquad \boldsymbol{\psi}^{(k)}(\cdot) = [\psi_j^{(k)}(\cdot)]_{j\in\mathbb{N}}, \forall\, \psi_j^{(k)} \in \mathbb{I}_{\mathcal{L}_{\mathbf{y}}^{2(k)}}. \tag{66}$$

In the isotypic basis $\mathbb{I}_{\mathcal{L}_{\mathbf{x}}^2}^{\text{iso}} = \cup_{k=1}^{n_{\text{iso}}}\mathbb{I}_{\mathcal{L}_{\mathbf{x}}^{2(k)}}$, the expansion coefficient vector $\boldsymbol{\alpha} = \oplus_{k=1}^{n_{\text{iso}}}\boldsymbol{\alpha}^{(k)}$ is formed from the projections of $f$ onto each isotypic subspace: $\boldsymbol{\alpha}^{(k)} = [\langle \phi_i^{(k)}, f \rangle_{P_{\mathbf{x}}}]_{i\in\mathbb{N}}$. The block-diagonal structure of $\boldsymbol{T}^{\text{iso}}$ is only one of the algebraic constraints imposed on the matrix form of $\mathsf{T}$ by the $\mathbb{G}$-equivariance condition. The next section describes the further structural constraints on each block.

### K.1.3. Structure of Operators between Isotypic Subspaces

In this section, we shift the focus from the input and output function spaces, $\mathcal{L}_{\mathbf{x}}^2$ and $\mathcal{L}_{\mathbf{y}}^2$; and the operator $\mathsf{T} : \mathcal{L}_{\mathbf{x}}^2 \mapsto \mathcal{L}_{\mathbf{y}}^2$, to their individual isotypic subspaces, $\mathcal{L}_{\mathbf{x}}^{2(k)}$ and $\mathcal{L}_{\mathbf{y}}^{2(k)}$ for $k \in [1, n_{\text{iso}}]$, and the operators $\mathsf{T}^{(k)} : \mathcal{L}_{\mathbf{x}}^{2(k)} \mapsto \mathcal{L}_{\mathbf{y}}^{2(k)}$ (Eq. (62)).

Recall from Thm. I.17, that each isotypic subspace possesses unitary group representations that decompose into direct sums of (infinitely many) multiplicities of the irreducible representation of type $k$; that is:

$$\boldsymbol{\rho}_{\mathcal{L}_{\mathbf{x}}^{2(k)}}(g) \sim \oplus_{p=1}^{\infty}\bar{\boldsymbol{\rho}}_k(g) \qquad \text{and} \qquad \boldsymbol{\rho}_{\mathcal{L}_{\mathbf{y}}^{2(k)}}(g) \sim \oplus_{p=1}^{\infty}\bar{\boldsymbol{\rho}}_k(g). \tag{67}$$

This implies that each isotypic subspace further decomposes into (infinitely many) finite-dimensional $\mathbb{G}$-stable subspaces: $\mathcal{L}_{\mathbf{x}}^{2(k)} := \oplus_{p=1}^{\infty}\mathcal{L}_{\mathbf{x}}^{2k,p}$ and $\mathcal{L}_{\mathbf{y}}^{2(k)} := \oplus_{p=1}^{\infty}\mathcal{L}_{\mathbf{y}}^{2k,p}$. Each subspace $\mathcal{L}_{\mathbf{x}}^{2k,p}$ (and similarly $\mathcal{L}_{\mathbf{y}}^{2k,p}$) has finite dimension $d_k \leq \infty$ and its elements transform according to the irreducible representation $\bar{\boldsymbol{\rho}}_k$ of the group $\mathbb{G}$.

The modular structure of the isotypic subspaces results in a similar structure of $\mathsf{T}^{(k)}$ as it decomposes into $\mathbb{G}$-equivariant maps between the multiplicities of finite-dimensional, irreducible, $\mathbb{G}$-stable subspaces $\mathsf{T}_{i,j}^{(k)} : \mathcal{L}_{\mathbf{x}}^{2k,i} \mapsto \mathcal{L}_{\mathbf{y}}^{2k,j}$ for $i, j \in \mathbb{N}$, such that, in such basis the infinite-dimensional matrix form of $\mathsf{T}^{(k)}$ can be expressed as:

$$\boldsymbol{T}^{(k)} = \begin{bmatrix} \boldsymbol{T}_{1,1}^{(k)} & \boldsymbol{T}_{1,2}^{(k)} & \cdots \\ \boldsymbol{T}_{2,1}^{(k)} & \ddots & \cdots \\ \vdots & \vdots & \ddots \end{bmatrix} \quad \text{s.t.} \quad \boldsymbol{T}_{i,j}^{(k)} \in \text{End}_{\mathbb{G}}(\bar{\mathcal{V}}_k), \forall\, i, j \in \mathbb{N}, \tag{68}$$

Where each $\boldsymbol{T}_{i,j}^{(k)}$ is constrained to be an (real) endomorphism of the vector space $\bar{\mathcal{V}}_k$ associated to the irreducible representation $\bar{\boldsymbol{\rho}}_k : \mathbb{G} \to \mathbb{U}(\bar{\mathcal{V}}_k)$.

In practical terms, this implies that each operator $\mathsf{T}^{(k)}$ is constructed from a combination of infinitely many elements of the endomorphism space $\mathrm{End}_{\mathbb{G}}(\bar{\mathcal{V}}_k)$ (Def. I.5). Crucially, for compact symmetry groups and real irreducible representations, the space of endomorphisms is $1$, $2$ or $4$ dimensional, and possess an analytical basis set, defined by the map $\Psi_k : \mathbb{K} \rightarrow \mathrm{End}_{\mathbb{G}}(\bar{\mathcal{V}}_k)$, where $\mathbb{K} \in \{\mathbb{R}, \mathbb{C}, \mathbb{H}\}$ denotes the basis algebra associated to the irreducible representation $\bar{\rho}_k$ (see details in Prop. I.14).

Such a constraint implies that each $\boldsymbol{T}_{i,j}^{(k)}$ can be expressed as a linear combination of endomorphism basis elements: $\boldsymbol{T}_{i,j}^{(k)} = \sum_{b \in \mathbb{K}} \alpha_{b,i,j} \Psi_k(b)$. Since every element of the endomorphism basis $\{\Psi_k(b)\}_{b \in \mathbb{K}}$ is a finite-dimensional orthogonal matrix, this structure can be interpreted as a constraint on the dimensionality of the singular spaces of the operator $\mathsf{T}^{(k)}$ to be of dimension larger than $d_k = |\bar{\rho}_k|$, as summarized in the following proposition:

**Proposition K.3** (Minimum dimensionality of singular space of $\mathbb{G}$-equivariant operators between isotypic subspaces)**.** *Let* $\mathsf{T}^{(k)} : \mathcal{L}_{\mathbf{x}}^{2(k)} \mapsto \mathcal{L}_{\mathbf{y}}^{2(k)}$ *be a* $\mathbb{G}$*-equivariant operator between isotypic subspaces* $\mathcal{L}_{\mathbf{x}}^{2(k)}$ *and* $\mathcal{L}_{\mathbf{y}}^{2(k)}$ *of type k. Then, the minimum dimension of a singular space of the operator is* $d_k$.

*Proof.* The proof follows naturally from Prop. I.11, which characterizes the singular spaces of each irrep endomorphism. $\square$

### K.2. Finite-Rank Approximation of $\mathbb{G}$-Equivariant Operators

In practical applications, infinite-dimensional operators are approximated by finite-dimensional ones to enable computation. For any linear integral operator $\mathsf{T} : \mathcal{L}_{\mathbf{x}}^2 \mapsto \mathcal{L}_{\mathbf{y}}^2$, the optimal rank-$r$ approximation in the Hilbert-Schmidt norm is obtained by truncating its SVD to the top $r$ singular values and associated left/right singular functions. Let $\{\sigma_i\}_{i=1}^{\infty}$ be the singular values of $\mathsf{T}$ in decreasing order and let $\{u_i\}_{i=1}^{\infty} \subset \mathcal{L}_{\mathbf{x}}^2$, $\{v_i\}_{i=1}^{\infty} \subset \mathcal{L}_{\mathbf{y}}^2$ be the corresponding singular functions satisfying $\langle v_i, \mathsf{T} u_i \rangle_{P_{\mathbf{y}}} = \sigma_i$ for each $i \in \mathbb{N}$ and $\langle v_i, \mathsf{T} u_j \rangle_{P_{\mathbf{y}}} = 0$ when $i \neq j$. The best rank-$r$ approximation of $\mathsf{T}$ is then given by (Eckart & Young, 1936):

$$\mathsf{T}_r f = \sum_{i=1}^{r} \sigma_i \langle u_i, f \rangle_{P_{\mathbf{x}}} v_i, \quad \forall f \in \mathcal{L}_{\mathbf{x}}^2, \qquad \Longleftrightarrow \qquad \kappa(\boldsymbol{x}, \boldsymbol{y}) \approx \sum_{i=1}^{r} \sigma_i u_i(\boldsymbol{x}) v_i(\boldsymbol{y}). \tag{69}$$

Since the left and right singular functions form orthonormal bases for $\mathcal{L}_{\mathbf{y}}^2$ and $\mathcal{L}_{\mathbf{x}}^2$, a rank-$r$ approximation reduces these infinite-dimensional spaces to the $r$-dimensional subspaces $\mathcal{F}_{\mathbf{x}} = \mathrm{span}(\{u_i\}_{i=1}^r)$ and $\mathcal{F}_{\mathbf{y}} = \mathrm{span}(\{v_i\}_{i=1}^r)$.

When $\mathcal{L}_{\mathbf{x}}^2$ and $\mathcal{L}_{\mathbf{y}}^2$ are symmetric function spaces with group actions $\triangleright_{\mathcal{L}_{\mathbf{x}}^2}$ and $\triangleright_{\mathcal{L}_{\mathbf{y}}^2}$ of a compact group $\mathbb{G}$, and $\mathsf{T}$ is $\mathbb{G}$-equivariant, the finite-rank approximation $\mathsf{T}_r : \mathcal{F}_{\mathbf{x}} \rightarrow \mathcal{F}_{\mathbf{y}}$ must satisfy that for all $f \in \mathcal{F}_{\mathbf{x}}$, $h \in \mathcal{F}_{\mathbf{y}}$, and $g \in \mathbb{G}$, both $g \triangleright_{\mathcal{L}_{\mathbf{x}}^2} f \in \mathcal{F}_{\mathbf{x}}$ and $g \triangleright_{\mathcal{L}_{\mathbf{y}}^2} h \in \mathcal{F}_{\mathbf{y}}$. This ensures that $g \triangleright_{\mathcal{L}_{\mathbf{y}}^2} [\mathsf{T}_r f] = \mathsf{T}_r [g \triangleright_{\mathcal{L}_{\mathbf{x}}^2} f]$ (see App. J).

Moreover, since $\mathcal{L}_{\mathbf{x}}^2$ and $\mathcal{L}_{\mathbf{y}}^2$ decompose orthogonally into isotypic subspaces, $\mathcal{L}_{\mathbf{x}}^2 = \oplus_{1 \leq k \leq n_{\mathrm{iso}}}^{\perp} \mathcal{L}_{\mathbf{x}}^{2(k)}$ and $\mathcal{L}_{\mathbf{y}}^2 = \oplus_{1 \leq k \leq n_{\mathrm{iso}}}^{\perp} \mathcal{L}_{\mathbf{y}}^{2(k)}$, the operator $\mathsf{T}$ is completely determined by the $n_{\mathrm{iso}}$ operators $\mathsf{T}^{(k)} : \mathcal{L}_{\mathbf{x}}^{2(k)} \rightarrow \mathcal{L}_{\mathbf{y}}^{2(k)}$ (see App. K.1.2). Thus, the $\mathbb{G}$-equivariance of $\mathsf{T}_r$ depends on that of each finite-rank operator $\mathsf{T}_{r_k}^{(k)} : \mathcal{F}_{\mathbf{x}}^{(k)} \rightarrow \mathcal{F}_{\mathbf{y}}^{(k)}$, which requires the approximated subspaces $\mathcal{F}_{\mathbf{x}}^{(k)}$ and $\mathcal{F}_{\mathbf{y}}^{(k)}$ to be $\mathbb{G}$-stable. For simplicity, we assume $|\mathcal{F}_{\mathbf{x}}^{(k)}| = |\mathcal{F}_{\mathbf{y}}^{(k)}| = r_k$, although this equality need not hold in general.

**Constraints on the Dimensionality of Truncation** Each approximation of an isotypic subspace $\mathcal{L}_{\mathbf{x}}^{2(k)}$ (and similarly $\mathcal{L}_{\mathbf{y}}^{2(k)}$) is $\mathbb{G}$-stable if the group representation is defined using a truncated multiplicity $m_k < \infty$ for the $k^{\mathrm{th}}$ irreducible representation, i.e. $\boldsymbol{\rho}_{\mathcal{F}_{\mathbf{x}}^{(k)}} \sim \oplus_{p=1}^{m_k} \bar{\rho}_k$ and $\boldsymbol{\rho}_{\mathcal{L}_{\mathbf{y}}^{2(k)}} \sim \oplus_{p=1}^{m_k} \bar{\rho}_k$. Consequently, the dimension of the approximated subspaces is multiple of the irreducible representation's dimension: $r_k = d_k m_k$ (see App. K.1.3).

**Spectral Decomposition** Given the block-diagonal structure of the operator $\mathsf{T}$ in the isotypic basis (Eq. (65)), the truncated SVD of $\mathsf{T}$ reduces to performing the truncated SVD of each per-isotypic operator $\mathsf{T}^{(k)}$. Let $\mathcal{F}_{\mathbf{x}} \subset \mathcal{L}_{\mathbf{x}}^2$ and $\mathcal{F}_{\mathbf{y}} \subset \mathcal{L}_{\mathbf{y}}^2$ be the $\mathbb{G}$-stable finite-dimensional approximations of the input/output spaces of $\mathsf{T}$, endowed with group representations $\boldsymbol{\rho}_{\mathcal{F}_{\mathbf{x}}} = \oplus_{k=1}^{n_{\mathrm{iso}}} \boldsymbol{\rho}_{\mathcal{F}_{\mathbf{x}}^{(k)}} = \oplus_{k=1}^{n_{\mathrm{iso}}} \oplus_{p=1}^{m_k} \bar{\rho}_k$ and $\boldsymbol{\rho}_{\mathcal{F}_{\mathbf{y}}} = \oplus_{k=1}^{n_{\mathrm{iso}}} \boldsymbol{\rho}_{\mathcal{F}_{\mathbf{y}}^{(k)}} = \oplus_{k=1}^{n_{\mathrm{iso}}} \oplus_{p=1}^{m_k} \bar{\rho}_k$. Here, $m_k \in \mathbb{N}$ denotes the multiplicity of the irreducible representation of type $k$, and $d_k := |\bar{\rho}_k|$ is its dimension. Then, the structural constraints on the SVD of the restriction of $\mathsf{T}$ to these spaces are summarized in the following theorem:

**Theorem K.4** (Isotypic-spectral basis)**.** *Let* $\mathsf{T}$ *be a* $\mathbb{G}$*-equivariant operator and let* $\mathsf{T}_{\star} : \mathcal{F}_{\mathbf{y}} \rightarrow \mathcal{F}_{\mathbf{x}}$ *be its* $\mathbb{G}$*-equivariant restriction in finite dimensions. Then, the singular value decomposition of the restricted operator matrix representation* $\boldsymbol{T}_{\star}$ *reduces to:*

$$\boldsymbol{T}_{\star} = \oplus_{k=1}^{n_{\mathrm{iso}}} \boldsymbol{T}_{\star}^{(k)} = \oplus_{k=1}^{n_{\mathrm{iso}}} \boldsymbol{W}_{\star}^{(k)} \boldsymbol{S}_{\star}^{(k)} \boldsymbol{M}_{\star}^{(k)\top} = \oplus_{k=1}^{n_{\mathrm{iso}}} \boldsymbol{U}_{\star}^{(k)} (\boldsymbol{\Sigma}_{\star}^{(k)} \otimes \boldsymbol{I}_{d_k}) \boldsymbol{V}_{\star}^{(k)\top}$$

Where $\boldsymbol{I}_{d_k}$ denotes the identity matrix in $d_k$-dimensions and $\boldsymbol{\Sigma}_\star^{(k)}$ denotes the infinite-dimensional diagonal matrix singular values per irreducible $\mathbb{G}$-stable subspace.

Thm. K.4 shows that symmetries force each isotypic subspace's singular space to have dimension at least $d_k$, which is the minimum required for a faithful representation of $\mathbb{G}^{(k)}$ (see Def. I.9). Because in practice our goal is to approximate the top $r$ singular spaces of $\mathsf{T}$, this result precisely characterizes the constraints imposed by $\mathbb{G}$-equivariance on the optimal rank-$r$ truncation's spectral basis and corresponding kernel function in Eq. (13), as summarized in the following corollary:

**Corollary K.5** (Symmetry constraints on the spectral basis). *Let* $\mathsf{T}$ *be a* $\mathbb{G}$-*equivariant operator and let* $\mathsf{T}_\star : \mathcal{F}_\mathbf{y} \to \mathcal{F}_\mathbf{x}$ *be its* $\mathbb{G}$-*equivariant restriction in* $r$-*dimensions. Then, the spectral basis of* $\mathsf{T}_\star$ *is given by:*

$$\kappa_\star(\boldsymbol{x}, \boldsymbol{y}) = \sum_{k=1}^{n_{\text{iso}}} \kappa_\star^{(k)}(\boldsymbol{x}, \boldsymbol{y}) = \sum_{k=1}^{n_{\text{iso}}} \sum_{s=1}^{r_k} \sigma_s^{(k)} \sum_{i=1}^{d_k} u_{s,i}^{(k)}(\boldsymbol{x}) v_{s,i}^{(k)}(\boldsymbol{y}), \tag{70}$$

*where* $\{u_{s,i}^{(k)}\}_{i \in [d_k]}$ *and* $\{v_{s,i}^{(k)}\}_{i \in [d_k]}$ *are the left and right singular basis sets of the* $s^{th}$ *singular space of* $\mathsf{T}^{(k)}$. *Note that the truncated dimension is restricted by the dimensionality and multiplicities of the individual irreducible representations* $r = \sum_{k=1}^{d_{\text{iso}}} r_k = \sum_{k=1}^{d_{\text{iso}}} d_k m_k$.

## L. Relevant $\mathbb{G}$-Equivariant Operators in Probability Theory

In this section we study the properties of expectations and covariances of functions of symmetric random variables in the presence our assumed symmetry priors Eq. (6). In a nutshell, we characterize how expectations of observables of symmetric random variables are invariant to the group action, and that the covariance and cross-covariance matrices in these spaces are $\mathbb{G}$-equivariant and hence inherit rich structural constraints that can aid in empirical estimation.

Let $(\mathbf{x}, \mathbf{y})$ be two vector-valued random variables over the probability spaces $(\mathcal{X}, \Sigma_\mathcal{X}, P_\mathbf{x})$ and $(\mathcal{Y}, \Sigma_\mathcal{Y}, P_\mathbf{y})$, with $\mathcal{L}_\mathbf{x}^2$ and $\mathcal{L}_\mathbf{y}^2$ being the corresponding square-integrable function spaces and $\mathbb{1}_{P_\mathbf{x}} \in \mathcal{L}_\mathbf{x}^2$, $\mathbb{1}_{P_\mathbf{y}} \in \mathcal{L}_\mathbf{y}^2$ the characteristic functions of sets with nonzero probability.

When $\mathcal{L}_\mathbf{x}^2$ and $\mathcal{L}_\mathbf{y}^2$ are symmetric function spaces (see App. J), denote their orthogonal isotypic decompositions by $\mathcal{L}_\mathbf{x}^2 := \bigoplus_{k=1}^{n_{\text{iso}}} \mathcal{L}_\mathbf{x}^{2(k)}$ and $\mathcal{L}_\mathbf{y}^2 := \bigoplus_{k=1}^{n_{\text{iso}}} \mathcal{L}_\mathbf{y}^{2(k)}$ (cf. Thm. I.17). Any function $f \in \mathcal{L}_\mathbf{x}^2$ or $h \in \mathcal{L}_\mathbf{y}^2$ decomposes as $f = \sum_{k=1}^{n_{\text{iso}}} f^{(k)}$ and $h = \sum_{k=1}^{n_{\text{iso}}} h^{(k)}$ (see Eq. (63)). By convention, the first isotypic subspace corresponds to the trivial group action. Thus, we write $\mathcal{L}_\mathbf{x}^{2\text{inv}} := \mathcal{L}_\mathbf{x}^{21} \subset \mathcal{L}_\mathbf{x}^2$ and denote the $\mathbb{G}$-invariant component of $f$ by $f^{\text{inv}} := f^{(1)}$ (and similarly for $\mathcal{L}_\mathbf{y}^2$).

### L.1. The Expectation Operator

The expected value of a function $f \in \mathcal{F} := \mathcal{L}_\mathbf{x}^2$ can be interpreted as the result of applying a linear integral operator that projects each $f \in \mathcal{F}$ to a constant function evaluating to the function's expected value $\mathbb{E}_{P_\mathbf{x}} f$.

**Definition L.1** (Expectation operator). *Let* $\mathcal{F} \subseteq \mathcal{L}_\mathbf{x}^2$ *be a function space. The expectation operator* $\mathsf{E}_\mathbf{x} : \mathcal{F} \mapsto \mathcal{F}$ *is a linear integral operator defined by a constant kernel function* $k_\mathbb{E}(\boldsymbol{x}, \boldsymbol{x}') = \mathbb{1}_{P_\mathbf{x}}(\boldsymbol{x}) \mathbb{1}_{P_\mathbf{x}}(\boldsymbol{x}')$ *for all* $\boldsymbol{x}, \boldsymbol{x}' \in \mathcal{X}$, *such that this operator maps any function* $f$ *to a constant function that evaluates to the function's expected value* $\mathbb{1}_{P_\mathbf{x}}(\cdot)\mathbb{E}_{P_\mathbf{x}} f$, *that is:*

$$[\mathsf{E}_\mathbf{x} f](\boldsymbol{x}') = \int_\mathcal{X} k_\mathbb{E}(\boldsymbol{x}, \boldsymbol{x}') f(\boldsymbol{x}) \mu(d\boldsymbol{x}) = \mathbb{1}_{P_\mathbf{x}}(\boldsymbol{x}') \int_\mathcal{X} f(\boldsymbol{x}) \mu(d\boldsymbol{x}) \equiv \mathbb{1}_{P_\mathbf{x}}(\boldsymbol{x}') \mathbb{E}_{P_\mathbf{x}} f. \tag{71}$$

Whenever $\mathcal{F}$ is a symmetric function space, the operator $\mathsf{E}_\mathbf{x}$ commutes with the group action and is $\mathbb{G}$-invariant (Def. I.12):

**Proposition L.2** ($\mathbb{G}$-invariant expectation operator). *Let* $\mathcal{F}$ *be a symmetric function space with the action* $\rhd_\mathcal{F}$ *of a compact symmetry group* $\mathbb{G}$. *Then, the expectation operator commutes with the group action and is a* $\mathbb{G}$-*invariant operator* $\mathsf{E}_\mathbf{x} : \mathcal{F} \mapsto \mathcal{F}^{inv} \subseteq \mathcal{F}$:

$$\mathsf{E}_\mathbf{x}[g \rhd_\mathcal{F} f] = g \rhd_\mathcal{F} [\mathsf{E}_\mathbf{x} f] \qquad \text{and} \qquad \mathsf{E}_\mathbf{x} f = \mathsf{E}_\mathbf{x}[g \rhd_\mathcal{F} f] \in \mathcal{F}^{inv}, \qquad \forall\, f \in \mathcal{F}, g \in \mathbb{G}. \tag{72}$$

*Proof.* The operator $\mathsf{E}_\mathbf{x}$ commutes with the group action as its kernel function $k_\mathbb{E}$ is constant and therefore $\mathbb{G}$-invariant (Def. K.1). Furthermore since the image of the expectation operator are constant functions, these functions belong to the subspace of $\mathbb{G}$-invariant functions, $\mathcal{F}^{\text{inv}}$. $\qquad\square$

As an operator that commutes with the group action, the expectation operator decomposes into $\mathsf{E}_{\mathbf{x}} := \oplus_{k=1}^{n_{\text{iso}}} \mathsf{E}_{\mathbf{x}}^{(k)}$, where $\mathsf{E}_{\mathbf{x}}^{(k)} : \mathcal{F}^{(k)} \mapsto \mathcal{F}^{(k)}$ denotes the restriction of $\mathsf{E}_{\mathbf{x}}$ to the isotypic subspace $\mathcal{F}^{(k)}$ (App. K.1.2). However, since the image of the operator lies in the subspace of $\mathbb{G}$-invariant functions, $\Im(\mathsf{E}_{\mathbf{x}}) \subset \mathcal{F}^{\text{inv}}$, it follows that $\mathsf{E}_{\mathbf{x}}^{(k)} = \mathbf{0}$ for every $k \neq \text{inv}$. Consequently, we obtain the following:

**Corollary L.3** (Expectation of a function depends only on its $\mathbb{G}$-invariant component). *For any function* $f \in \mathcal{F}$*, the expectation depends only on its $\mathbb{G}$-invariant component:*

$$[\mathsf{E}_{\mathbf{x}} f](\cdot) = \sum_{k=1}^{n_{\text{iso}}} [\mathsf{E}_{\mathbf{x}}^{(k)} f^{(k)}](\cdot) = [\mathsf{E}_{\mathbf{x}}^{inv} f^{inv}](\cdot) := \mathbb{1}_{\mu}(\cdot) \mathbb{E}_{\mu} f^{inv}. \tag{73}$$

**Corollary L.4** (Functions without a $\mathbb{G}$-invariant component are centered). *Any function* $f = \sum_{k=1}^{n_{\text{iso}}} f^{(k)} \in \mathcal{L}_{\mathbf{x}}^2$ *without a $\mathbb{G}$-invariant compoment, i.e.,* $f^{inv} = 0$*, is centered:*

$$[\mathsf{E}_{\mathbf{x}} f](\cdot) = \sum_{k=2}^{n_{\text{iso}}} [\mathsf{E}_{\mathbf{x}}^{(k)} f^{(k)}](\cdot) = \mathbb{1}_{\mu}(\cdot)0, \qquad \Longleftrightarrow \qquad \mathbb{E}_{\mu} f = 0, \quad \forall f \in \mathcal{L}_{\mathbf{x}}^{2inv\perp}. \tag{74}$$

To better comprehend these concepts we refer the reader to Example J.4.

## L.2. The Cross-Covariance Operator

Given two vector-valued random variables ($\mathbf{x} = [\mathsf{x}_1, \dots, \mathsf{x}_n], \mathbf{y} = [\mathsf{y}_1, \dots, \mathsf{y}_m]$) defined on the measure spaces $(\mathcal{X}, \Sigma_{\mathcal{X}}, P_{\mathbf{x}})$ and $(\mathcal{Y}, \Sigma_{\mathcal{Y}}, P_{\mathbf{y}})$, a key statistic assessing the linear relationship between scalar components is the covariance:

$$\text{Cov}(\mathsf{x}_i, \mathsf{y}_j) = \mathbb{E}_{P_{\mathbf{xy}}}[(\mathsf{x}_i - \mathbb{E}_{\mathbf{x}}[\mathsf{x}_i])(\mathsf{y}_j - \mathbb{E}_{\mathbf{y}}[\mathsf{y}_j])] = \mathbb{E}_{P_{\mathbf{xy}}}[\mathsf{x}_i \mathsf{y}_j] - \mathbb{E}_{\mathbf{x}}[\mathsf{x}_i] \mathbb{E}_{\mathbf{y}}[\mathsf{y}_j].$$

For vector-valued random variables, the cross-covariance matrix $\text{Cov}(\mathbf{x}, \mathbf{y}) \in \mathbb{R}^{n \times m}$ is defined entrywise by $\text{Cov}(\mathbf{x}, \mathbf{y})_{i,j} := \text{Cov}(\mathsf{x}_i, \mathsf{y}_j)$. The cross-covariance operator is the extension of this concept to the Hilbert spaces of functions $\mathcal{L}_{\mathbf{x}}^2$ and $\mathcal{L}_{\mathbf{y}}^2$.

**Definition L.5** (Cross-covariance operator (Fukumizu et al., 2004)). *Let* $\mathcal{F}_{\mathbf{x}} \subseteq \mathcal{L}_{\mathbf{x}}^2$ *and* $\mathcal{L}_{\mathbf{y}}^2 \subseteq \mathcal{L}_{\mathbf{y}}^2$ *be two Hilbert spaces of functions defined on the random variables* $\mathbf{x}$ *and* $\mathbf{y}$*, which take values in the measure spaces* $(\mathcal{X}, \Sigma_{\mathcal{X}}, P_{\mathbf{x}})$ *and* $(\mathcal{Y}, \Sigma_{\mathcal{Y}}, P_{\mathbf{y}})$*, respectively. The cross-covariance operator* $\mathsf{C}_{\mathbf{xy}} : \mathcal{L}_{\mathbf{y}}^2 \mapsto \mathcal{L}_{\mathbf{x}}^2$ *is a linear integral operator defined by*

$$\langle f, \mathsf{C}_{\mathbf{xy}} h \rangle_{P_{\mathbf{x}}} := \text{Cov}(f, h) = \mathbb{E}_{P_{\mathbf{xy}}}[f(\mathbf{x})h(\mathbf{y})] - \mathbb{E}_{\mathbf{x}}[f(\mathbf{x})] \mathbb{E}_{\mathbf{y}}[h(\mathbf{y})], \qquad \forall f \in \mathcal{L}_{\mathbf{x}}^2, \ h \in \mathcal{L}_{\mathbf{y}}^2. \tag{75}$$

*Choosing separable basis sets for the two spaces,* $\mathbb{I}_{\mathcal{L}_{\mathbf{x}}^2} = \{\phi_i\}_{i \in \mathbb{N}}$ *and* $\mathbb{I}_{\mathcal{L}_{\mathbf{y}}^2} = \{\psi_i\}_{i \in \mathbb{N}}$*, the matrix representation of the cross-covariance operator has entries* $[\boldsymbol{C}_{\mathbf{x},\mathbf{y}}]_{i,j} := \langle \phi_i, \mathsf{C}_{\mathbf{xy}} \psi_j \rangle_{P_{\mathbf{x}}} = \text{Cov}(\phi_i, \psi_j)$*, where the covariance is computed with respect to the joint measure* $P_{\mathbf{xy}}$ *and the marginals* $P_{\mathbf{x}}$ *and* $P_{\mathbf{y}}$*. Given a dataset of* $N$ *samples from the joint distribution* $(\boldsymbol{x}, \boldsymbol{y}) \sim P_{\mathbf{xy}}$*, the empirical estimate of the matrix form of the cross-covariance operator is*

$$\widehat{\boldsymbol{C}}_{\mathbf{xy}} = \frac{1}{N} \sum_{n=1}^{N} \boldsymbol{\phi}(\boldsymbol{x}_n)\boldsymbol{\psi}(\boldsymbol{y}_n)^\top - \widehat{\mathbb{E}}_{\mathbf{x}}[\boldsymbol{\phi}(\boldsymbol{x}_n)] \widehat{\mathbb{E}}_{\mathbf{y}}[\boldsymbol{\psi}(\boldsymbol{y}_n)]^\top, \quad \boldsymbol{\phi}(\cdot) = [\phi(\cdot)]_{i \in \mathbb{N}}, \ \boldsymbol{\psi}(\cdot) = [\psi(\cdot)]_{i \in \mathbb{N}}. \tag{76}$$

Note that the adjoint of the operator is defined by $\mathsf{C}_{\mathbf{xy}}^* = \mathsf{C}_{\mathbf{yx}} : \mathcal{L}_{\mathbf{x}}^2 \mapsto \mathcal{L}_{\mathbf{y}}^2$. In the case $\mathcal{L}_{\mathbf{x}}^2 = \mathcal{L}_{\mathbf{y}}^2$, the cross-covariance operator reduces to the covariance operator, and has an analog definition to Def. L.5.

**Covariance and Cross-Covariance Operators of Symmetric Hilbert Spaces of Functions**   Whenever $\mathcal{L}_{\mathbf{x}}^2$ and $\mathcal{L}_{\mathbf{y}}^2$ are symmetric function spaces, and the joint probability measure is $\mathbb{G}$-invaraint, the cross-covariance operator $\mathsf{C}_{\mathbf{xy}}$ commute with the group action and is $\mathbb{G}$-equivariant (App. I.1):

**Proposition L.6** ($\mathbb{G}$-equivariant cross-covariance operator). *Let* $\mathcal{L}_{\mathbf{x}}^2 \subseteq \mathcal{L}_{\mathbf{x}}^2$ *and* $\mathcal{L}_{\mathbf{y}}^2 \subseteq \mathcal{L}_{\mathbf{y}}^2$ *be symmetric Hilbert spaces of functions endowed with the group actions* $\rhd_{\mathcal{L}_{\mathbf{x}}^2}$ *and* $\rhd_{\mathcal{L}_{\mathbf{y}}^2}$ *of a compact symmetry group* $\mathbb{G}$*. Then, whenever the joint probability measure is* $\mathbb{G}$-invariant*, i.e.,* $P_{\mathbf{xy}}(\mathbb{B}, \mathbb{A}) = P_{\mathbf{xy}}(g \rhd_{\mathcal{X}} \mathbb{B}, g \rhd_{\mathcal{Y}} \mathbb{A})$ *for all* $g \in \mathbb{G}, \mathbb{B} \in \Sigma_{\mathcal{X}}, \mathbb{A} \in \Sigma_Y$*, the cross-covariance operator* $\mathsf{C}_{\mathbf{xy}} : \mathcal{L}_{\mathbf{y}}^2 \mapsto \mathcal{L}_{\mathbf{x}}^2$ *(Def. L.5) commutes with the group actions and is a* $\mathbb{G}$-equivariant operator *(Def. K.1):*

$$g \rhd_{\mathcal{L}_{\mathbf{x}}^2} [\mathsf{C}_{\mathbf{xy}} h] = \mathsf{C}_{\mathbf{xy}}[g \rhd_{\mathcal{L}_{\mathbf{y}}^2} h], \qquad \forall h \in \mathcal{L}_{\mathbf{y}}^2, g \in \mathbb{G}. \tag{77}$$

*Proof.* To proof that the operator is $\mathbb{G}$-equivariant we must show its kernel function is $\mathbb{G}$-invariant (see Def. K.1). The proof follows naturally in any regular basis of the input and output functions spaces $\mathbb{I}_{\mathcal{L}_{\mathbf{x}}^2} = \{\phi_i\}_{i \in \mathbb{N}}$ and $\mathbb{I}_{\mathcal{L}_{\mathbf{y}}^2} = \{\psi_i\}_{i \in \mathbb{N}}$, in which the group action on basis functions acts by permutations of basis functions, such that, $g \rhd_{\mathcal{L}_{\mathbf{x}}^2} \phi_i \equiv \phi_{g \rhd i} \in \mathbb{I}_{\mathcal{L}_{\mathbf{x}}^2}$ and $g \rhd_{\mathcal{L}_{\mathbf{y}}^2} \psi_j \equiv \psi_{g \rhd j} \in \mathbb{I}_{\mathcal{L}_{\mathbf{y}}^2}$, where $g \rhd i, g \rhd j \in \mathbb{N}$. Then we must show that that:

$$
\begin{aligned}
k(\boldsymbol{x}, \boldsymbol{y}) &= \kappa(g^{-1} \rhd_{\mathcal{X}} \boldsymbol{x}, g^{-1} \rhd_{\mathcal{Y}} \boldsymbol{y}) && \forall\, g \in \mathbb{G}, \boldsymbol{x} \in \mathcal{X}, \boldsymbol{y} \in \mathcal{Y} \\
\sum_{i \in \mathbb{N}} \sum_{j \in \mathbb{N}} [\boldsymbol{C}_{\mathbf{x},\mathbf{y}}]_{i,j} \phi_i(\boldsymbol{x}) \psi_j(\boldsymbol{y}) &= \sum_{i \in \mathbb{N}} \sum_{j \in \mathbb{N}} [\boldsymbol{C}_{\mathbf{x},\mathbf{y}}]_{i,j} [g \rhd_{\mathcal{L}_{\mathbf{x}}^2} \phi_i](\boldsymbol{x})[g \rhd_{\mathcal{Y}} \psi_j](\boldsymbol{y}) && \text{s.t. Defs. J.1 and L.5} \\
\sum_{i \in \mathbb{N}} \sum_{j \in \mathbb{N}} \mathrm{Cov}(\phi_i, \psi_j) \phi_i(\boldsymbol{x}) \psi_j(\boldsymbol{y}) &= \sum_{i \in \mathbb{N}} \sum_{j \in \mathbb{N}} \mathrm{Cov}(\phi_i, \psi_j) \phi_{g \rhd i}(\boldsymbol{x}) \psi_{g \rhd j}(\boldsymbol{y}).
\end{aligned}
\tag{78}
$$

Hence, the cross-covariance operator's kernel function is $\mathbb{G}$-invariant only if the covariance is $\mathbb{G}$-invariant:

$$
\begin{aligned}
\mathrm{Cov}(\phi_i, \psi_j) &= \mathrm{Cov}(g \rhd_{\mathcal{L}_{\mathbf{x}}^2} \phi_i, g \rhd_{\mathcal{Y}} \psi_j) && \forall\, g \in \mathbb{G}, i, j \in \mathbb{N} \\
\mathbb{E}_{P_{\mathbf{xy}}}[\phi_i(\mathbf{x}) \psi_j(\mathbf{y})] &= \mathbb{E}_{P_{\mathbf{xy}}}[\phi_i(g^{-1} \rhd_{\mathcal{X}} \mathbf{x}) \psi_j(g^{-1} \rhd_{\mathcal{Y}} \mathbf{y})] && \mathbb{E}_\mu f = \mathbb{E}_\mu g \rhd f \\
\int_{\mathcal{X} \times \mathcal{Y}} \phi_i(\mathbf{x}) \psi_j(\mathbf{y})\, P_{\mathbf{xy}}(d\mathbf{x}, d\mathbf{y}) &= \int_{\mathcal{X} \times \mathcal{Y}} \phi_i(g^{-1} \rhd_{\mathcal{X}} \mathbf{x}) \psi_j(g^{-1} \rhd_{\mathcal{Y}} \mathbf{y})\, P_{\mathbf{xy}}(d\mathbf{x}, d\mathbf{y}) \\
&= \int_{\mathcal{X} \times \mathcal{Y}} \phi_i(\mathbf{x}) \psi_j(\mathbf{y})\, P_{\mathbf{xy}}(g \rhd d\mathbf{x}, g \rhd d\mathbf{y}) \\
&= \mathrm{Cov}(\phi_i, \psi_j).
\end{aligned}
\tag{79}
$$

$\square$

An equivalent result follows for covariance operators of symmetric Hilbert spaces.

## M. Statistical Learning Theory

This section provides the development and proofs of the statistical learning guarantees in Thm. 5.1 for regression and conditional probability estimation using our proposed model.

Recall that regression and conditional probabilities can be expressed in terms of the conditional expectation operator $\mathsf{E}_{\mathbf{y}|\mathbf{x}} \colon \mathcal{L}_{\mathbf{y}}^2 \to \mathcal{L}_{\mathbf{x}}^2$ (see Eqs. (1) and (2)). Given that the operator is compact (Kostic et al., 2024b), it admits a singular value decomposition. Hence, the kernel function defining the operator Eq. (1) can be expanded in terms of the operator spectral basis:

$$
\kappa(\boldsymbol{x}, \boldsymbol{y}) := \frac{dP_{\mathbf{xy}}(\boldsymbol{x}, \boldsymbol{y})}{d(P_{\mathbf{x}}(\boldsymbol{x}) \times P_{\mathbf{y}}(\boldsymbol{y}))} = \sum_{i=0}^{\infty} \sigma_i u_i(\boldsymbol{x}) v_i(\boldsymbol{y}).
\tag{80}
$$

Where $(\sigma_i)_{i \in \mathbb{N}}$ denotes the operator's singular values, and $(u_i)_{i \in \mathbb{N}}$ and $(v_i)_{i \in \mathbb{N}}$ denote the left and right singular functions, which form complete orthonormal basis sets for $\mathcal{L}_{\mathbf{x}}^2$ and $\mathcal{L}_{\mathbf{y}}^2$, respectively. Given that the operator's first singular value is $\sigma_0 = 1$, associated with the constant functions $u_0 = \mathbb{1}_{\mathcal{X}}$, $v_0 = \mathbb{1}_{\mathcal{Y}}$, the conditional expectation operator can be defined as:

$$
\mathsf{E}_{\mathbf{y}|\mathbf{x}} = \sum_{i=1}^{\infty} \sigma_i u_i \langle v_i, \cdot \rangle_{P_{\mathbf{y}}} = \mathbb{1}_{\mathcal{X}} \langle \mathbb{1}_{\mathcal{Y}}, \cdot \rangle_{P_{\mathbf{y}}} + \underbrace{\sum_{i=1}^{\infty} \sigma_i u_i \langle v_i, \cdot \rangle_{P_{\mathbf{y}}}}_{\mathsf{D}_{\mathbf{y}|\mathbf{x}}}.
\tag{81}
$$

Where $\mathsf{D}_{\mathbf{y}|\mathbf{x}}$ denotes the *deflated* operator, excluding the first eigen triplet $(\sigma_0, u_0, v_0)$. Leveraging the SVD of $\mathsf{E}_{\mathbf{y}|\mathbf{x}}$, we approximate the operator's action for any $h \in \mathcal{L}_{\mathbf{y}}^2$ using a rank-$r$ $(1 < r < \infty)$ operator given by:

$$
\mathbb{E}[h(\mathbf{y})|\mathbf{x} = \boldsymbol{x}] = [\mathsf{E}_{\mathbf{y}|\mathbf{x}} h](\boldsymbol{x}) \approx \mathbb{E}[h(\mathbf{y})] + \sum_{i=1}^{r} \sigma_i u_i^{\boldsymbol{\theta}}(\boldsymbol{x}) \mathbb{E}[v_i^{\boldsymbol{\theta}}(\mathbf{y}) h(\mathbf{y})],
\tag{82}
$$
$$
\text{s.t. } \mathbb{E}[u_i^{\boldsymbol{\theta}}(\mathbf{x})] = \mathbb{E}[v_i^{\boldsymbol{\theta}}(\mathbf{y})] = 0, \forall\, i \geq 1.
$$

Where $(u_i^{\boldsymbol{\theta}})_{i=1}^r$ and $(v_i^{\boldsymbol{\theta}})_{i=1}^r$ denote parametrizations of the top-$r$ left and right singular functions. Given that the operator's kernel Eq. (80) preserves the probability mass, that is $\int_{\mathcal{X} \times \mathcal{Y}} \kappa(\boldsymbol{x}, \boldsymbol{y}) dP_{\mathbf{x}}(\boldsymbol{x}) dP_{\mathbf{y}}(\boldsymbol{y}) = 1$, every non-constant singular function is constrained to be centered, as described in the r.h.s of Eq. (82).

In the context of symmetries, we note that $D_{\mathbf{y}|\mathbf{x}}$ admits a block-diagonal structure w.r.t. to isotypic basis of associated $\mathcal{L}^2$ spaces. Indeed we have the following from Thm. K.4.

$$Q_{\mathbf{x}}^* D_{\mathbf{y}|\mathbf{x}} Q_{\mathbf{y}} = \oplus_{k=1}^{n_{\mathrm{iso}}} Q_{\mathbf{x}}^{(k)*} D_{\mathbf{y}|\mathbf{x}}^{(k)} Q_{\mathbf{y}}^{(k)} = \oplus_{k=1}^{n_{\mathrm{iso}}} \left[ (\mathsf{U}^{(k)} \mathsf{S}^{(k)} \mathsf{V}^{(k)*}) \otimes \boldsymbol{I}_{d_k} \right]. \tag{83}$$

Where the unitary operators $Q_{\mathbf{x}} \colon \mathcal{L}_{\mathbf{x}}^2 \to \mathcal{L}_{\mathbf{x}}^2$ and $Q_{\mathbf{y}} \colon \mathcal{L}_{\mathbf{y}}^2 \to \mathcal{L}_{\mathbf{y}}^2$ change the basis to the isotypic decompositions $\mathbb{I}_{\mathcal{L}_{\mathbf{x}}^2} = \{\phi_{i,j}^{(k)}\}_{k \in [n_{\mathrm{iso}}], i \in [m_k], j \in [d_k]}$ and $\mathbb{I}_{\mathcal{L}_{\mathbf{y}}^2} = \{\psi_{i,j}^{(k)}\}_{k \in [n_{\mathrm{iso}}], i \in [m_k], j \in [d_k]}$, with $i$ indexing each irreducible $\mathbb{G}$-stable subspace and $j$ indexing the dimensions within that subspace (see App. K.2).

Further, by Thm. K.4, the SVD of $D_{\mathbf{y}|\mathbf{x}}$ forces each isotypic subspace to have dimension at least $d_k = \bar{\rho}_k$ for every $k \in [n_{\mathrm{iso}}]$.

$$Q_{\mathbf{x}}^{(k)*} D_{\mathbf{y}|\mathbf{x}}^{(k)} Q_{\mathbf{y}}^{(k)} = \left[ \mathsf{U}^{(k)} \otimes \boldsymbol{I}_{d_k} \right] \left[ \mathsf{S}^{(k)} \otimes \boldsymbol{I}_{d_k} \right] \left[ \mathsf{V}^{(k)} \otimes \boldsymbol{I}_{d_k} \right]^*, \ k \in [n_{\mathrm{iso}}], \tag{84}$$

where $Q_{\mathbf{x}}^{(k)} Q_{\mathbf{x}}^{(k)*}$ and $Q_{\mathbf{y}}^{(k)} Q_{\mathbf{y}}^{(k)*}$ are orthogonal projectors on $k$-th isotypic subspace, and

$$Q_{\mathbf{x}}^* D_{\mathbf{y}|\mathbf{x}} Q_{\mathbf{y}} = \left[ \boldsymbol{I}_{n_{\mathrm{iso}}} \otimes \mathsf{U}^{(k)} \otimes \boldsymbol{I}_{d_k} \right] \left[ \boldsymbol{I}_{n_{\mathrm{iso}}} \otimes \mathsf{S}^{(k)} \otimes \boldsymbol{I}_{d_k} \right] \left[ \boldsymbol{I}_{n_{\mathrm{iso}}} \otimes \mathsf{V}^{(k)} \otimes \boldsymbol{I}_{d_k} \right]^*. \tag{85}$$

Further, observe that the singular values of $D_{\mathbf{y}|\mathbf{x}}$ are elements of positive diagonal operators $\mathsf{S}^{(k)}$, denoted as $(\mathsf{S}^{(k)})_i = \sigma_i^{(k)}$, while the left and right singular functions are $u_i^{(k)} \otimes \boldsymbol{e}_j^{d_k}$ and $v_i^{(k)} \otimes \boldsymbol{e}_j^{d_k}$, respectively, for $i \in \mathbb{N}$, $j \in [d_k]$ and $k \in [n_{\mathrm{iso}}]$, where $\boldsymbol{e}_j^d$ is $j$-th vector of standard basis of $\mathbb{R}^d$.

Given the constraints on the spectral basis of $\mathbb{G}$-equivariant operators (see Cor. K.5), our representation learning procedure approach results in feature maps:

$$\begin{aligned} \boldsymbol{u_\theta}(\cdot) &= \sum_{k \in [n_{\mathrm{iso}}], i \in [m], j \in [d_k]} [\boldsymbol{e}_k^{n_{\mathrm{iso}}} \otimes \boldsymbol{e}_i^m \otimes \boldsymbol{e}_j^{d_k}] u_{i,j}^{\boldsymbol{\theta}(k)}(\cdot) \colon \mathcal{X} \to \mathbb{R}^{r_m} \\ \boldsymbol{v_\theta}(\cdot) &= \sum_{k \in [n_{\mathrm{iso}}], i \in [m], j \in [d_k]} [\boldsymbol{e}_k^{n_{\mathrm{iso}}} \otimes \boldsymbol{e}_i^m \otimes \boldsymbol{e}_j^{d_k}] v_{i,j}^{\boldsymbol{\theta}(k)}(\cdot) \colon \mathcal{X} \to \mathbb{R}^{r_m}, \end{aligned} \tag{86}$$

which can further be separated into $n_{\mathrm{iso}}$ orthogonal blocks $\boldsymbol{u}_{\boldsymbol{\theta}}^{(k)} = \sum_{i \in [m], j \in [d_k]} \phi_{i,j}^{\boldsymbol{\theta}(k)}$ and $\boldsymbol{\psi}_{\boldsymbol{\theta}}^{(k)} = \sum_{i \in [m], j \in [d_k]} \psi_{i,j}^{\boldsymbol{\theta}(k)}$ as

$$\boldsymbol{u}_{\boldsymbol{\theta}}^{(k)} = \sum_{i \in [m], j \in [d_k]} [\boldsymbol{e}_i^m \otimes \boldsymbol{e}_j^{d_k}] u_{i,j}^{\boldsymbol{\theta}(k)}(\cdot) \quad \text{and} \quad \boldsymbol{v}_{\boldsymbol{\theta}}^{(k)} = \sum_{i \in [m], j \in [d_k]} [\boldsymbol{e}_i^m \otimes \boldsymbol{e}_j^{d_k}] v_{i,j}^{\boldsymbol{\theta}(k)}(\cdot). \tag{87}$$

In addition, the singular value matrices have a tensor form $\boldsymbol{S_\theta} = diag(\boldsymbol{S}_{\boldsymbol{\theta}}^{(1)}, \ldots, \boldsymbol{S}_{\boldsymbol{\theta}}^{(n_{\mathrm{iso}})})$, where $\boldsymbol{S}_{\boldsymbol{\theta}}^{(k)} = diag(\sigma_1^{\boldsymbol{\theta}(k)}, \ldots, \sigma_m^{\boldsymbol{\theta}(k)}) \otimes \boldsymbol{I}_{d_k}$ and $\sigma_i^{\boldsymbol{\theta}(k)} \in [0, 1]$, $i \in [m], k \in [n_{\mathrm{iso}}]$. Thus, we obtain the operator $D_{\boldsymbol{\theta}} = E_{\boldsymbol{\theta}} - \mathbb{1}_{P_{\mathbf{x}}} \otimes \mathbb{1}_{P_{\mathbf{y}}}$ in block form, $D_{\boldsymbol{\theta}} = \oplus_{k \in [n_{\mathrm{iso}}]} D_{\boldsymbol{\theta}}^{(k)}$, where each $D_{\boldsymbol{\theta}}^{(k)}$ acts on the $k$-th isotypic subspace as

$$[D_{\boldsymbol{\theta}}^{(k)} f](\boldsymbol{x}) := \boldsymbol{u}_{\boldsymbol{\theta}}^{(k)}(\boldsymbol{x})^\top \boldsymbol{S}_{\boldsymbol{\theta}}^{(k)} \mathbb{E}_{\mathbf{y}}[\boldsymbol{v}_{\boldsymbol{\theta}}^{(k)}(\mathbf{y}) f^{(k)}(\mathbf{y})], \quad f \in \mathcal{L}_{\mathbf{y}}^2, \tag{88}$$

and hence

$$[D_{\boldsymbol{\theta}} f](\boldsymbol{x}) := \boldsymbol{u_\theta}(\boldsymbol{x})^\top \boldsymbol{S_\theta} \mathbb{E}_{\mathbf{y}}[\boldsymbol{v_\theta}(\mathbf{y}) f(\mathbf{y})], \quad f \in \mathcal{L}_{\mathbf{y}}^2. \tag{89}$$

Finally, we extend the definition of $D_{\boldsymbol{\theta}}$ to vector-valued observables $\boldsymbol{h} \colon \mathcal{Y} \to \mathcal{Z}$ via basis expansions.

$$[D_{\boldsymbol{\theta}} \boldsymbol{h}](\boldsymbol{x}) := \sum_\ell \boldsymbol{u_\theta}(\boldsymbol{x})^\top \boldsymbol{S_\theta} \mathbb{E}_{\mathbf{y}}[\boldsymbol{v_\theta}(\mathbf{y})(\langle \boldsymbol{h}(\mathbf{y}), \boldsymbol{z}_\ell \rangle_{\mathcal{Z}} \boldsymbol{z}_\ell)], \quad \boldsymbol{h} \in \mathcal{L}_{\mathbf{y}}^2(\mathcal{Y}, \mathcal{Z}) \tag{90}$$

where $(\boldsymbol{z}_i)_{i \in [n_{\mathcal{Z}}]}$ is the orthonormal basis of $\mathcal{Z}$.

By doing so, we ensure that $D_{\boldsymbol{\theta}}$ and, consequently, $E_{\boldsymbol{\theta}}$ are $\mathbb{G}$-equivariant operators for both the scalar map $\mathcal{L}_{\mathbf{y}}^2 \to \mathcal{L}_{\mathbf{x}}^2$ and the vector-valued map $\mathcal{L}_{\mathbf{y}}^2(\mathcal{Y}, \mathcal{Z}) \to \mathcal{L}_{\mathbf{x}}^2(\mathcal{X}, \mathcal{Z})$. Moreover, a direct consequence of (90) is as follows.

**Proposition M.1.** *Let with $\mathcal{Z}$ being a real Euclidean space endowed with symmetry group $\mathbb{G}$, and let $E_{\boldsymbol{\theta}} \colon \mathcal{L}_{P_{\mathbf{y}}}^2(\mathcal{Y}, \mathcal{Z}) \mapsto \mathcal{L}_{P_{\mathbf{x}}}^2(\mathcal{X}, \mathcal{Z})$ be given by $E_{\boldsymbol{\theta}} \boldsymbol{f} = \mathbb{E}_{\mathbf{y}}[\boldsymbol{f}(\mathbf{y})] + D_{\boldsymbol{\theta}} \boldsymbol{f}$. Then for every $\mathbb{G}$-equivariant $\boldsymbol{f} \in \mathcal{L}_{P_{\mathbf{y}}}^2(\mathcal{Y}, \mathcal{Z})$ and every $\boldsymbol{x} \in \mathcal{X}$*

$$[E_{\boldsymbol{\theta}} \boldsymbol{f}](g \triangleright_{\mathcal{X}} \boldsymbol{x}) = \mathbb{E}_{\mathbf{y}}[\boldsymbol{f}(\mathbf{y})] + [D_{\boldsymbol{\theta}} \boldsymbol{f}](g \triangleright_{\mathcal{X}} \boldsymbol{x}) = \mathbb{E}_{\mathbf{y}}[\boldsymbol{f}(\mathbf{y})] + g \triangleright_{\mathcal{Z}} [D_{\boldsymbol{\theta}} \boldsymbol{f}](\boldsymbol{x}) = g \triangleright_{\mathcal{Z}} [E_{\boldsymbol{\theta}} \boldsymbol{f}](\boldsymbol{x}). \tag{91}$$

*Proof.* Since $\mathsf{D}_{\theta}$ is $\mathbb{G}$-equivaraint, for every $g \in \mathbb{G}$ we have that

$$[\mathsf{D}_{\theta}\boldsymbol{h}](g^{-1} \triangleright_{\mathcal{X}} \boldsymbol{x}) = [\mathsf{D}_{\theta}[\boldsymbol{h}(g^{-1} \triangleright_{\mathcal{Y}} \cdot)]](\boldsymbol{x}) = \sum_i \boldsymbol{u}_{\theta}(\boldsymbol{x})^{\top} \boldsymbol{S}_{\theta} \mathbb{E}_{\mathbf{y}}[\boldsymbol{v}_{\theta}(\mathbf{y})\langle \boldsymbol{h}(g^{-1} \triangleright_{\mathcal{Y}} \mathbf{y}), \boldsymbol{z}_i \rangle_{\mathcal{Z}} \boldsymbol{z}_i],$$

which, using $g$ instead of $g^{-1}$ and the assumption that $f$ is $\mathbb{G}$-equivariant, implies

$$\begin{aligned}
[\mathsf{D}_{\theta}\boldsymbol{h}](g \triangleright_{\mathcal{X}} \boldsymbol{x}) &= \sum_i (\boldsymbol{u}_{\theta}(\boldsymbol{x})^{\top} \boldsymbol{S}_{\theta} \mathbb{E}_{\mathbf{y}}[\boldsymbol{v}_{\theta}(\mathbf{y})\langle g \triangleright_{\mathcal{Z}} \boldsymbol{h}(\mathbf{y}), \boldsymbol{z}_i \rangle_{\mathcal{Z}} \boldsymbol{z}_i] \\
&= \sum_i (\boldsymbol{u}_{\theta}(\mathbf{x})^{\top} \boldsymbol{S}_{\theta} \mathbb{E}_{\mathbf{y}}[\boldsymbol{v}_{\theta}(\mathbf{y})\langle \boldsymbol{h}(\mathbf{y}), g^{-1} \triangleright_{\mathcal{Z}} \boldsymbol{z}_i \rangle_{\mathcal{Z}}) \boldsymbol{z}_i.
\end{aligned}$$

Thus, changing the basis to $(g^{-1} \triangleright_{\mathcal{Z}} \boldsymbol{z}_i)_{i \in [n_{\mathcal{Z}}]}$ we obtain the result when $\mathbb{E}_{\mathbf{y}}[\boldsymbol{h}(\mathbf{y})] = 0$. But since $\mathbb{1}_{\mathcal{X}}(g \triangleright_{\mathcal{X}} \boldsymbol{x}) = 1$ for every $\boldsymbol{x} \in \mathcal{X}$ and $g \in \mathbb{G}$, the same holds for $\mathsf{E}_{\theta}$. $\qquad\square$

Recall that for the effective latent dimension $m$ the true latent dimension is constrained by the dimensionality of the singular spaces, i.e., $r_m = \sum_{k \in [n_{\mathrm{iso}}]} r_k = \sum_{k \in [n_{\mathrm{iso}}]} m[k] d_k$. Further, given a measurable set $\mathbb{A} \subseteq \mathcal{X}$ and collection of group elements $\mathbb{G}' \subseteq \mathbb{G}$, let us define the following symmetry index of a set $\mathbb{A}$ w.r.t. probability distribution of random variable $\mathbf{x}$

$$\gamma_{\mathbb{G}'}(\mathbb{A}) = \frac{1}{|\mathbb{G}'|(|\mathbb{G}'| - 1)} \sum_{\substack{g_1, g_2 \in \mathbb{G}' \\ g_1 \neq g_2}} \frac{\mathbb{P}[\mathbf{x} \in g_1 \triangleright \mathbb{A} \cap g_2 \triangleright \mathbb{A}]}{\mathbb{P}[\mathbf{x} \in \mathbb{A}]}, \tag{92}$$

which in the case when $\mathbb{G}'$ is a subgroup of $\mathbb{G}$ simplifies as

$$\gamma_{\mathbb{G}'}(\mathbb{A}) = \frac{1}{|\mathbb{G}'| - 1} \sum_{\substack{g \in \mathbb{G}' \\ g \neq e}} \frac{\mathbb{P}[\mathbf{x} \in \mathbb{A} \cap g \triangleright \mathbb{A}]}{\mathbb{P}[\mathbf{x} \in \mathbb{A}]}. \tag{93}$$

Observe that always $\gamma_{\mathbb{G}'}(\mathbb{A}) \in [0, 1]$, where extremes correspond to the cases $\gamma_{\mathbb{G}'}(\mathbb{A}) = 1$ when set $\mathbb{A}$ is $\mathbb{G}'$ invariant, and $\gamma_{\mathbb{G}'}(\mathbb{A}) = 0$ when $\mathbb{A}$ equals its coset w.r.t. $\mathbb{G}'$, that is $g \triangleright \mathbb{A} \cap \mathbb{A} = \emptyset$ for all $g \in \mathbb{G}'$, meaning that the set is fully asymmetric w.r.t transformations $g \in \mathbb{G}'$.

We first generalize the approximation error bound in Lemma 1 from (Kostic et al., 2024b) to the case of vector valued functions in the presence of symmetries.

**Theorem M.2** (Approximation error). *Given a group of symmetries $\mathbb{G}$, let $\mathcal{X}$, $\mathcal{Y}$ and $\mathcal{Z}$ be Hilbert spaces endowed with symmetry group $\mathbb{G}$, and let $P_{\mathbf{x}}$, $P_{\mathbf{y}}$ and $P_{\mathbf{xy}}$ be $\mathbb{G}$-invariant probability distributions on $\mathcal{X}$, $\mathcal{Y}$ and $\mathcal{X} \times \mathcal{Y}$. Then, for every $\boldsymbol{h} \in \mathcal{L}_{\mathbf{y}}^2(\mathcal{Y}, \mathcal{Z})$ it holds that*

$$\|\mathbb{E}_{\mathbf{y}}[\boldsymbol{h}(\mathbf{y}) \,|\, \mathbf{x} = \cdot] - \mathsf{E}_{\theta}\boldsymbol{h}\|_{\mathcal{L}_{P_{\mathbf{x}}}^2(\mathcal{X}, \mathcal{Z})} \leq \left(\sigma_{r_m+1}^{\star} + \left\|[\![\mathsf{D}_{\mathbf{y}|\mathbf{x}}]\!]_{r_m} - \mathsf{D}_{\theta}\right\|\right) \|\boldsymbol{h}\|_{\mathcal{L}_{\mathbf{y}}^2(\mathcal{Y}, \mathcal{Z})}. \tag{94}$$

*Moreover, denoting*

$$\mathsf{E}_{\theta}[f(\mathbf{y}) \,|\, \mathbf{x} \in \mathbb{A}] = \mathbb{E}_{\mathbf{y}}[f] + \frac{\mathbb{E}_{\mathbf{x}}[\mathbb{1}_{\mathbb{A}}(\mathbf{x})[\mathsf{D}_{\theta}f](\mathbf{x})]}{\mathbb{P}[\mathbf{x} \in \mathbb{A}]}, \tag{95}$$

*if $\boldsymbol{h}$ is either $\mathbb{G}'$-invariant or $\mathbb{G}'$-equivariant for some $\mathbb{G}' \subseteq \mathbb{G}$, then for every measurable set $\mathbb{A}$*

$$\|\mathbb{E}[\boldsymbol{h}(\mathbf{y}) \,|\, \mathbf{x} \in \mathbb{A}] - \mathsf{E}_{\theta}[\boldsymbol{h}(\mathbf{y}) \,|\, \mathbf{x} \in \mathbb{A}]\|_{\mathcal{Z}} \leq \left(\sigma_{r_m+1}^{\star} + \left\|[\![\mathsf{D}_{\mathbf{y}|\mathbf{x}}]\!]_{r_m} - \mathsf{D}_{\theta}\right\|\right) \frac{\|\boldsymbol{h}\|_{\mathcal{L}_{\mathbf{y}}^2(\mathcal{Y}, \mathcal{Z})}}{\sqrt{\mathbb{P}[\mathbf{x} \in \mathbb{A}]}} \sqrt{\frac{1 + (|\mathbb{G}'| - 1)\gamma_{\mathbb{G}'}(\mathbb{A})}{|\mathbb{G}'|}}. \tag{96}$$

*Proof.* Start by observing that

$$\begin{aligned}
\|\mathbb{E}[\boldsymbol{h}(\mathbf{y}) \,|\, \mathbf{x} = \cdot] - \mathsf{E}_{\theta}\boldsymbol{h}\|_{\mathcal{L}_{P_{\mathbf{x}}}^2(\mathcal{X}, \mathcal{Z})} &\leq \left\|\mathsf{D}_{\mathbf{y}|\mathbf{x}} - \mathsf{D}_{\theta}\right\|_{\mathcal{L}_{\mathbf{y}}^2(\mathcal{Y}, \mathcal{Z}) \to \mathcal{L}_{P_{\mathbf{x}}}^2(\mathcal{X}, \mathcal{Z})} \|\boldsymbol{h}\|_{\mathcal{L}_{\mathbf{y}}^2(\mathcal{Y}, \mathcal{Z})} \\
&= \left\|\mathsf{D}_{\mathbf{y}|\mathbf{x}} - \mathsf{D}_{\theta}\right\|_{\mathcal{L}_{P_{\mathbf{y}}}^2(\mathcal{Y}) \to \mathcal{L}_{P_{\mathbf{x}}}^2(\mathcal{X})} \|\boldsymbol{h}\|_{\mathcal{L}_{\mathbf{y}}^2(\mathcal{Y}, \mathcal{Z})},
\end{aligned}$$

where the equality holds since we extended operators $\mathsf{D}_{\mathbf{y}|\mathbf{x}}$ and $\mathsf{D}_{\theta}$ to vector valued setting as integral operators with the same scalar kernel. Hence, (94) readily follows.

To prove (96), start with noting

$$\mathbb{E}[\boldsymbol{h}(\mathbf{y}) \,|\, \mathbf{x} \in \mathbb{A}] - \mathsf{E}_{\boldsymbol{\theta}}[\boldsymbol{h}(\mathbf{y}) \,|\, \mathbf{x} \in \mathbb{A}] = \frac{\mathbb{E}_{\mathbf{x}}[\mathbb{1}_{\mathbb{A}}(\mathbf{x})[(\mathsf{D}_{\mathbf{y}|\mathbf{x}} - \mathsf{D}_{\boldsymbol{\theta}})\boldsymbol{h}](\mathbf{x})]}{\mathbb{P}[\mathbf{x} \in \mathbb{A}]}.$$

Then, if $\boldsymbol{h}$ is $\mathbb{G}$-equivariant, then, using that invariance of the probability distribution $P_{\mathbf{x}}$, $\mathbb{G}$-equivariance of $\mathsf{D}_{\mathbf{y}|\mathbf{x}}$ and, due to Proposition M.1, of $\mathsf{D}_{\boldsymbol{\theta}}$, we have that for every $g \in \mathbb{G}' \subseteq \mathbb{G}$

$$\begin{aligned}
\mathbb{E}_{\mathbf{x}}[\mathbb{1}_{\mathbb{A}}(\mathbf{x})[(\mathsf{D}_{\mathbf{y}|\mathbf{x}} - \mathsf{D}_{\boldsymbol{\theta}})\boldsymbol{h}](\mathbf{x})] &= \mathbb{E}_{\mathbf{x}}[\mathbb{1}_{\mathbb{A}}(g \triangleright_{\mathcal{X}} \mathbf{x})[(\mathsf{D}_{\mathbf{y}|\mathbf{x}} - \mathsf{D}_{\boldsymbol{\theta}})\boldsymbol{h}](g \triangleright_{\mathcal{X}} \mathbf{x})] \\
&= \mathbb{E}_{\mathbf{x}}[\mathbb{1}_{g^{-1}\triangleright_{\mathcal{X}}\mathbb{A}}(\mathbf{x}) \, g \triangleright_{\mathcal{Z}} [(\mathsf{D}_{\mathbf{y}|\mathbf{x}} - \mathsf{D}_{\boldsymbol{\theta}})\boldsymbol{h}](\mathbf{x})] \\
&= \mathbb{E}_{\mathbf{x}}[\mathbb{1}_{g^{-1}\triangleright_{\mathcal{X}}\mathbb{A}}(\mathbf{x})\bar{\boldsymbol{\rho}}_{\mathcal{Z}}(g) [(\mathsf{D}_{\mathbf{y}|\mathbf{x}} - \mathsf{D}_{\boldsymbol{\theta}})\boldsymbol{h}](\mathbf{x})].
\end{aligned}$$

Hence, averaging over $\mathbb{G}'$ we obtain

$$\mathbb{E}[\boldsymbol{h}(\mathbf{y}) \,|\, \mathbf{x} \in \mathbb{A}] - \mathsf{E}_{\boldsymbol{\theta}}[\boldsymbol{h}(\mathbf{y}) \,|\, \mathbf{x} \in \mathbb{A}] = \mathbb{E}_{\mathbf{x}}[\mathsf{H}(\mathbf{x})\bar{\boldsymbol{z}}(\mathbf{x})],$$

where

$$\mathsf{H}(\boldsymbol{x}) = \frac{1}{|\mathbb{G}'|\mathbb{P}[\mathbf{x} \in \mathbb{A}]} \sum_{g \in \mathbb{G}'} \mathbb{1}_{g^{-1}\triangleright_{\mathcal{X}}\mathbb{A}}(\boldsymbol{x})\bar{\boldsymbol{\rho}}_{\mathcal{Z}}(g) \quad \text{and} \quad \boldsymbol{z}(\boldsymbol{x}) = [(\mathsf{D}_{\mathbf{y}|\mathbf{x}} - \mathsf{D}_{\boldsymbol{\theta}})\boldsymbol{h}](\boldsymbol{x}).$$

Since due to Cauchy-Schwartz inequality we have

$$\|\mathbb{E}_{\mathbf{x}}[\mathsf{H}(\mathbf{x})\boldsymbol{z}(\mathbf{x})]\|_{\mathcal{Z}}^2 \le [\mathbb{E}_{\mathbf{x}} \|\mathsf{H}(\mathbf{x})\|_{\mathcal{Z}\to\mathcal{Z}}^2][\mathbb{E}_{\mathbf{x}} \|\boldsymbol{z}(\mathbf{x})\|_{\mathcal{Z}}^2] = \|\boldsymbol{z}\|_{\mathcal{L}^2_{P_{\mathbf{x}}}(\mathcal{X},\mathcal{Z})}^2 \, [\mathbb{E}_{\mathbf{x}} \|\mathsf{H}(\mathbf{x})\|_{\mathcal{Z}\to\mathcal{Z}}^2]$$

and $\|\boldsymbol{z}\|_{\mathcal{L}^2_{P_{\mathbf{x}}}(\mathcal{X},\mathcal{Z})} \le \|\mathsf{D}_{\mathbf{y}|\mathbf{x}} - \mathsf{D}_{\boldsymbol{\theta}}\|_{\mathcal{L}^2_{\mathbf{y}}(\mathcal{Y},\mathcal{Z})\to\mathcal{L}^2_{P_{\mathbf{x}}}(\mathcal{X},\mathcal{Z})} \|\boldsymbol{h}\|_{\mathcal{L}^2_{\mathbf{y}}(\mathcal{Y},\mathcal{Z})}$, it remains to bound $\mathbb{E}_{\mathbf{x}} \|\mathsf{H}(\mathbf{x})\|_{\mathcal{Z}\to\mathcal{Z}}^2$. But, the group actions in the vector spaces are unitary, so using the $\mathbb{G}$-invariance of the distribution of $\mathbf{x}$ we obtain

$$\begin{aligned}
\mathbb{E}_{\mathbf{x}} \|\mathsf{H}(\mathbf{x})\|_{\mathcal{Z}\to\mathcal{Z}}^2 &\le \mathbb{E}_{\mathbf{x}}\left[\frac{1}{|\mathbb{G}'|\mathbb{P}[\mathbf{x} \in \mathbb{A}]} \sum_{g \in \mathbb{G}'} \mathbb{1}_{g^{-1}\triangleright_{\mathcal{X}}\mathbb{A}}(\boldsymbol{x})\right]^2 \\
&= \frac{1}{|\mathbb{G}'|^2\mathbb{P}[\mathbf{x} \in \mathbb{A}]^2} \sum_{g,g' \in \mathbb{G}'} \mathbb{E}_{\mathbf{x}}[\mathbb{1}_{g^{-1}\triangleright_{\mathcal{X}}\mathbb{A}}(\boldsymbol{x})\mathbb{1}_{g'^{-1}\triangleright_{\mathcal{X}}\mathbb{A}}(\boldsymbol{x})] \\
&= \frac{1}{|\mathbb{G}'|^2\mathbb{P}[\mathbf{x} \in \mathbb{A}]^2} \sum_{g,g' \in \mathbb{G}'} \mathbb{E}_{\mathbf{x}}[\mathbb{1}_{g\triangleright_{\mathcal{X}}\mathbb{A}\cap g'\triangleright_{\mathcal{X}}\mathbb{A}}(\boldsymbol{x})] \\
&= \frac{1}{|\mathbb{G}'|^2\mathbb{P}[\mathbf{x} \in \mathbb{A}]} \sum_{g,g' \in \mathbb{G}'} \frac{\mathbb{P}[\mathbf{x} \in g \triangleright_{\mathcal{X}} \mathbb{A} \cap g' \triangleright_{\mathcal{X}} \mathbb{A}]}{\mathbb{P}[\mathbf{x} \in \mathbb{A}]} \\
&= \frac{1}{\mathbb{P}[\mathbf{x} \in \mathbb{A}]}\frac{1 + (|\mathbb{G}'| - 1)\gamma_{\mathbb{G}'}(\mathbb{A})}{|\mathbb{G}'|},
\end{aligned}$$

which completes the proof of (96) for $\mathbb{G}'$-equivariant functions. Finally, if $f$ is $\mathbb{G}'$-invariant, the proof follows the same lines by replacing group actions ($\triangleright_{\mathcal{Z}}$) by their respective group representation $\boldsymbol{\rho}_{\mathcal{Z}}$ (see Def. I.4) with identity. $\square$

Next we analyze the errors when, instead of applying learned operators $\mathsf{E}_{\boldsymbol{\theta}}$, we apply their empirical counterparts in inference tasks. To that end, we define now estimators of $\mathbb{E}[\boldsymbol{h}(\mathbf{x})]$ and $\mathbb{E}[\boldsymbol{z}(\mathbf{y})]$ exploiting the $\mathbb{G}$-invariance of the distributions of $\mathbf{x}$ and $\mathbf{y}$. First, define the empirical $\mathbb{G}$-invariant distributions

$$\widehat{\mathbb{P}}_{\mathbf{x}} := \frac{1}{|\mathbb{G}|N} \sum_{i=1}^{N}\sum_{g \in g} \delta_{g\triangleright\mathbf{x}_i}(\cdot), \quad \widehat{\mathbb{P}}_{\mathbf{y}} := \frac{1}{|\mathbb{G}|N} \sum_{i=1}^{N}\sum_{g \in g} \delta_{g\triangleright\mathbf{y}_i}(\cdot).$$

Hence we can define the equivariant empirical mean of any function $f \in \mathcal{L}^2_{\mathbf{x}}$, $h \in \mathcal{L}^2_{\mathbf{y}}$ as

$$\widehat{\mathbb{E}}_{\mathbf{x}}[f] = \frac{1}{|\mathbb{G}|N} \sum_{i=1}^{N}\sum_{g \in \mathbb{G}} f(g \triangleright_{\mathcal{X}} \mathbf{x}_i), \quad \widehat{\mathbb{E}}_{\mathbf{y}}[h] = \frac{1}{|\mathbb{G}|N} \sum_{i=1}^{N}\sum_{g \in \mathbb{G}} h(g \triangleright_{\mathcal{Y}} \mathbf{y}_i). \tag{97}$$

This extends naturally to operator on a function space $\mathcal{L}_{\mathbf{y}}^2(\mathcal{Y}, \mathcal{Z})$ where $\mathcal{Z}$ is endowed with an inner product $\langle \cdot, \cdot \rangle = \langle \cdot, \cdot \rangle_{\mathcal{Z}}$. If the distribution of $\mathbf{y}$ is $\mathbb{G}'$-invariant, then for any $h \in \mathcal{L}_{\mathbf{y}}^2(\mathcal{Y}, \mathcal{Z})$, we use the estimator $\widehat{\mathbb{E}}_{\mathbf{y}}[h(\mathbf{y})]$ in (97) as an estimator of $\mathbb{E}[h(\mathbf{y})]$:

$$\widehat{\mathbb{E}}_{\mathbf{y}}[h] = \frac{1}{|\mathbb{G}|N} \sum_{i=1}^{N} \sum_{g \in \mathbb{G}} h(g \triangleright_{\mathcal{Y}} \mathbf{y}_i). \tag{98}$$

In this notation, we define our empirical estimators

$$[\mathsf{E}_{\boldsymbol{\theta}} h](\boldsymbol{x}) \approx [\widehat{\mathsf{E}}_{\boldsymbol{\theta}} h](\boldsymbol{x}) = \widehat{\mathbb{E}}_{\mathbf{y}}[h(\mathbf{y})] + \sum_{k \in [n_{\text{iso}}]} \sum_{i \in [m]} \sum_{j \in [d_k]} \sigma_i^{\boldsymbol{\theta}(k)} u_{i,j}^{\boldsymbol{\theta}(k)}(\boldsymbol{x}) \widehat{\mathbb{E}}_{\mathbf{y}}[v_{i,j}^{\boldsymbol{\theta}(k)} h]$$

and

$$\mathsf{E}_{\boldsymbol{\theta}}[h(\mathbf{y}) \mid \mathbf{x} \in \mathbb{A}] \approx \widehat{\mathsf{E}}_{\boldsymbol{\theta}}[h(\mathbf{y}) \mid \mathbf{x} \in \mathbb{A}] = \widehat{\mathbb{E}}_{\mathbf{y}}[h] + \sum_{k \in [n_{\text{iso}}]} \sum_{i \in [m]} \sum_{j \in [d_k]} \sigma_i^{\boldsymbol{\theta}(k)} \frac{\widehat{\mathbb{E}}_{\mathbf{x}}[u_{i,j}^{\boldsymbol{\theta}(k)} \mathbb{1}_{\mathbb{A}}]}{\widehat{\mathbb{E}}_{\mathbf{x}}[\mathbb{1}_{\mathbb{A}}]} \widehat{\mathbb{E}}_{\mathbf{y}}[v_{i,j}^{\boldsymbol{\theta}(k)} h].$$

and, by choosing $h = \mathbb{1}_{\mathbb{B}}$,

$$P[\mathbf{y} \in \mathbb{B} \mid \mathbf{x} \in \mathbb{A}] \approx \widehat{P}_{\boldsymbol{\theta}}[\mathbf{y} \in \mathbb{B} \mid \mathbf{x} \in \mathbb{A}] = \widehat{\mathbb{E}}_{\mathbf{y}}[\mathbb{1}_{\mathbb{B}}] + \sum_{k \in [n_{\text{iso}}]} \sum_{i \in [m]} \sum_{j \in [d_k]} \sigma_i^{\boldsymbol{\theta}(k)} \frac{\widehat{\mathbb{E}}_{\mathbf{x}}[u_{i,j}^{\boldsymbol{\theta}(k)} \mathbb{1}_{\mathbb{A}}]}{\widehat{\mathbb{E}}_{\mathbf{x}}[\mathbb{1}_{\mathbb{A}}]} \widehat{\mathbb{E}}_{\mathbf{y}}[v_{i,j}^{\boldsymbol{\theta}(k)} \mathbb{1}_{\mathbb{B}}].$$

Direct consequence of the above construction which ensures that $\widehat{P}_{\mathbf{x}}$ and $\widehat{P}_{\mathbf{y}}$ are $\mathbb{G}$-invariant is the following result.

**Proposition M.3.** *Let $P_{\mathbf{x}}$ and $P_{\mathbf{y}}$ are $\mathbb{G}$-invariant, and $\mathsf{D}_{\boldsymbol{\theta}}$ from (89) is $\mathbb{G}$-equivariant model, and let $z \in \mathcal{L}_{\mathbf{x}}^2(\mathcal{X}, \mathbb{R})$ and $h \in \mathcal{L}_{P_{\mathbf{x}}}^2(\mathcal{Y}, \mathcal{Z})$ be arbitrary. If for every $k \in [n_{\text{iso}}]$*

$$\left\{ \left\| \mathsf{D}_{\mathbf{y}|\mathbf{x}}^{(k)} - \mathsf{D}_{\boldsymbol{\theta}}^{(k)} \right\|, \left\| \mathbb{E}_{\mathbf{x}}[\boldsymbol{u}_{\boldsymbol{\theta}}^{(k)}{}_{(1)}(\mathbf{x}) \boldsymbol{u}_{\boldsymbol{\theta}}^{(k)}{}_{(1)}(\mathbf{x})^{\top}] - \boldsymbol{I}_m \right\|, \left\| \mathbb{E}_{\mathbf{y}}[\boldsymbol{v}_{\boldsymbol{\theta}}^{(k)}{}_{(1)}(\mathbf{y}) \boldsymbol{v}_{\boldsymbol{\theta}}^{(k)}{}_{(1)}(\mathbf{y})^{\top}] - \boldsymbol{I}_m \right\| \right\} \leq \mathcal{E}_{\boldsymbol{\theta}}^{(k)}$$

*holds with $\boldsymbol{u}_{\boldsymbol{\theta}}^{(k)}{}_{(1)} = [u_{1,1}^{\boldsymbol{\theta}(k)} | \ldots | u_{m,1}^{\boldsymbol{\theta}(k)}]^{\top} \in \mathbb{R}^m$ and $\boldsymbol{v}_{\boldsymbol{\theta}}^{(k)}{}_{(1)} = [v_{1,1}^{\boldsymbol{\theta}(k)} | \ldots | v_{m,1}^{\boldsymbol{\theta}(k)}]^{\top} \in \mathbb{R}^m$, and if*

$$\begin{aligned}
\frac{\left\| \widehat{\mathbb{E}}_{\mathbf{x}}[\boldsymbol{u}_{\boldsymbol{\theta}}^{(k)}{}_{(1)} z_1^{(k)}] - \mathbb{E}_{\mathbf{x}}[\boldsymbol{u}_{\boldsymbol{\theta}}^{(k)}{}_{(1)}(\mathbf{x}) z_1^{(k)}(\mathbf{x})] \right\|}{\left\| z_1^{(k)} \right\|_{\mathcal{L}_{\mathbf{x}}^2}} &\leq A(\boldsymbol{u}_{\boldsymbol{\theta}}, z), \\
\frac{\left\| \widehat{\mathbb{E}}_{\mathbf{y}}[\boldsymbol{v}_{\boldsymbol{\theta}}^{(k)}{}_{(1)} \otimes \boldsymbol{h}_1^{(k)}] - \mathbb{E}_{\mathbf{y}}[\boldsymbol{v}_{\boldsymbol{\theta}}^{(k)}{}_{(1)}(\mathbf{y}) \otimes \boldsymbol{h}_1^{(k)}(\mathbf{y})] \right\|}{\left\| \boldsymbol{h}_1^{(k)} \right\|_{\mathcal{L}_{\mathbf{y}}^2}} &\leq A(\boldsymbol{v}_{\boldsymbol{\theta}}, \boldsymbol{h}),
\end{aligned} \tag{99}$$

*where $z = \sum_{k \in [n_{\text{iso}}]} \sum_{j \in [d_k]} z_j^{(k)}$ and $\boldsymbol{h} = \sum_{k \in [n_{\text{iso}}]} \sum_{j \in [d_k]} \boldsymbol{h}_j^{(k)}$ are isospectral decompositions, then*

$$\left\| \mathsf{E}_{\boldsymbol{\theta}} \boldsymbol{h} - \widehat{\mathsf{E}}_{\boldsymbol{\theta}} \boldsymbol{h} \right\|_{\mathcal{L}_{P_{\mathbf{x}}}^2(\mathcal{X}, \mathcal{Z})}^2 \leq \left\| \mathbb{E}_{\mathbf{y}}[\boldsymbol{h}(\mathbf{y}) - \widehat{\mathbb{E}}_{\mathbf{y}}[\boldsymbol{h}]] \right\|_{\mathcal{Z}}^2 + \left[ 1 + \max_{k \in [n_{\text{iso}}]} \mathcal{E}_{\boldsymbol{\theta}}^{(k)} \right]^3 \| \boldsymbol{h} \|_{\mathcal{L}_{\mathbf{y}}^2(\mathcal{Y}, \mathcal{Z})}^2 \left[ A(\boldsymbol{v}_{\boldsymbol{\theta}}, \boldsymbol{h}) \right]^2. \tag{100}$$

*Moreover, the empirical estimation error is upper bounded by*

$$\begin{aligned}
&\left\| \mathbb{E}_{\mathbf{x}}[z(\mathbf{x}) [\mathsf{D}_{\boldsymbol{\theta}} \boldsymbol{h}](\mathbf{x})] - \widehat{\mathbb{E}}_{\mathbf{x}}[z[\widehat{\mathsf{D}}_{\boldsymbol{\theta}}^{(k)} \boldsymbol{h}]] \right\|_{\mathcal{Z}}^2 \leq \\
&(1 + \mathcal{E}_{\boldsymbol{\theta}})^3 \left[ A(\boldsymbol{u}_{\boldsymbol{\theta}}, z) + A(\boldsymbol{v}_{\boldsymbol{\theta}}, \boldsymbol{h}) + A(\boldsymbol{u}_{\boldsymbol{\theta}}, z) A(\boldsymbol{v}_{\boldsymbol{\theta}}, \boldsymbol{h}) \right]^2 \| z \|_{\mathcal{L}_{P_{\mathbf{x}}}^2(\mathcal{X})}^2 \| \boldsymbol{h} \|_{\mathcal{L}_{\mathbf{y}}^2(\mathcal{Y}, \mathcal{Z})}^2.
\end{aligned} \tag{101}$$

*Proof.* First, observe that due to $\mathbb{G}$-invariance of distribution $P_{\mathbf{xy}}$ and $\mathbb{G}$-equivaraince of $\mathsf{E}_{\boldsymbol{\theta}}$ and $\mathsf{D}_{\boldsymbol{\theta}}$ we have that

$$\mathsf{E}_{\boldsymbol{\theta}} \boldsymbol{h} = \mathbb{E}_{\mathbf{y}}[\boldsymbol{h}^{(1)}(\mathbf{y})] + \sum_{k \in [n_{\text{iso}}]} \mathsf{D}_{\boldsymbol{\theta}}^{(k)} \boldsymbol{h}^{(k)}, \tag{102}$$

and

$$\mathbb{E}_{\mathbf{x}}[z(\mathbf{x})[\mathsf{E}_{\boldsymbol{\theta}}\boldsymbol{h}](\mathbf{x})] = \mathbb{E}_{\mathbf{x}}[z^{(1)}(\mathbf{x})]\mathbb{E}_{\mathbf{y}}[\boldsymbol{h}^{(1)}(\mathbf{y})] + \sum_{k\in[n_{\mathrm{iso}}]} \mathbb{E}_{\mathbf{x}}[z^{(k)}(\mathbf{x})[\mathsf{D}_{\boldsymbol{\theta}}^{(k)}\boldsymbol{h}^{(k)}](\mathbf{x})]. \tag{103}$$

In the same way, since the empirical distributions $\widehat{P}_{\mathbf{x}}$ and $\widehat{P}_{\mathbf{y}}$ are $\mathbb{G}$-invariant, we have that

$$\widehat{\mathsf{E}}_{\boldsymbol{\theta}}\boldsymbol{h} = \widehat{\mathbb{E}}_{\mathbf{y}}[\boldsymbol{h}^{(1)}] + \sum_{k\in[n_{\mathrm{iso}}]} \widehat{\mathsf{D}}_{\boldsymbol{\theta}}^{(k)}\boldsymbol{h}^{(k)}, \tag{104}$$

and

$$\widehat{\mathbb{E}}_{\mathbf{x}}[z[\widehat{\mathsf{E}}_{\boldsymbol{\theta}}\boldsymbol{h}]] = \widehat{\mathbb{E}}_{\mathbf{x}}[z^{(1)}]\widehat{\mathbb{E}}_{\mathbf{y}}[\boldsymbol{h}^{(1)}] + \sum_{k\in[n_{\mathrm{iso}}]} \widehat{\mathbb{E}}_{\mathbf{x}}[z^{(k)}[\widehat{\mathsf{D}}_{\boldsymbol{\theta}}^{(k)}\boldsymbol{h}^{(k)}]], \tag{105}$$

where

$$[\widehat{\mathsf{D}}_{\boldsymbol{\theta}}^{(k)}\boldsymbol{h}^{(k)}](\boldsymbol{x}) = \boldsymbol{u}_{\boldsymbol{\theta}}^{(k)}(\boldsymbol{x})^{\top}\boldsymbol{S}_{\boldsymbol{\theta}}^{(k)}\widehat{\mathbb{E}}_{\mathbf{y}}[\boldsymbol{v}_{\boldsymbol{\theta}}^{(k)}\otimes\boldsymbol{h}^{(k)}]. \tag{106}$$

Therefore, combining (102) and (104), we obtain that

$$[\mathsf{E}_{\boldsymbol{\theta}}\boldsymbol{h}](\boldsymbol{x}) - [\widehat{\mathsf{E}}_{\boldsymbol{\theta}}\boldsymbol{h}](\boldsymbol{x}) = \Big(\mathbb{E}_{\mathbf{y}}[\boldsymbol{h}^{(1)}(\mathbf{y})] - \widehat{\mathbb{E}}_{\mathbf{y}}[\boldsymbol{h}^{(1)}]\Big)\mathbb{1}_{\mathcal{X}}(\boldsymbol{x})$$
$$+ \sum_{k\in[n_{\mathrm{iso}}]}\Big([\mathsf{D}_{\boldsymbol{\theta}}^{(k)}\boldsymbol{h}^{(k)}](\boldsymbol{x}) - [\widehat{\mathsf{D}}_{\boldsymbol{\theta}}^{(k)}\boldsymbol{h}^{(k)}](\boldsymbol{x})\Big),$$

which after taking the norm in $\mathcal{L}^2_{P_{\mathbf{x}}}(\mathcal{X},\mathcal{Z})$, due to orthonormality of isotypic subspaces gives

$$\left\|\mathsf{E}_{\boldsymbol{\theta}}\boldsymbol{h} - \widehat{\mathsf{E}}_{\boldsymbol{\theta}}[\boldsymbol{h}]\right\|^2_{\mathcal{L}^2_{P_{\mathbf{x}}}(\mathcal{X},\mathcal{Z})} = \left\|\mathbb{E}_{\mathbf{y}}[\boldsymbol{h}^{(1)}(\mathbf{y})] - \widehat{\mathbb{E}}_{\mathbf{y}}[\boldsymbol{h}^{(1)}]\right\|^2_{\mathcal{Z}}$$
$$+ \sum_{k\in[n_{\mathrm{iso}}]}\left\|[\mathsf{D}_{\boldsymbol{\theta}}^{(k)}\boldsymbol{h}^{(k)}] - [\widehat{\mathsf{D}}_{\boldsymbol{\theta}}^{(k)}\boldsymbol{h}^{(k)}]\right\|^2_{\mathcal{L}^2_{P_{\mathbf{x}}}(\mathcal{X},\mathcal{Z})}.$$

Now, observe that, since

$$[\mathsf{D}_{\boldsymbol{\theta}}^{(k)}\boldsymbol{h}^{(k)}](\boldsymbol{x}) - [\widehat{\mathsf{D}}_{\boldsymbol{\theta}}^{(k)}\boldsymbol{h}^{(k)}](\boldsymbol{x}) = \boldsymbol{u}_{\boldsymbol{\theta}}^{(k)}(\boldsymbol{x})^{\top}\boldsymbol{S}_{\boldsymbol{\theta}}^{(k)}\Big(\mathbb{E}_{\mathbf{y}}[\boldsymbol{v}_{\boldsymbol{\theta}}^{(k)}\otimes\boldsymbol{h}^{(k)}] - \widehat{\mathbb{E}}_{\mathbf{y}}[\boldsymbol{v}_{\boldsymbol{\theta}}^{(k)}\otimes\boldsymbol{h}^{(k)}]\Big)$$

applying the norm we have that $\left\|[\mathsf{D}_{\boldsymbol{\theta}}^{(k)}\boldsymbol{h}^{(k)}] - [\widehat{\mathsf{D}}_{\boldsymbol{\theta}}^{(k)}\boldsymbol{h}^{(k)}]\right\|^2_{\mathcal{L}^2_{P_{\mathbf{x}}}(\mathcal{X},\mathcal{Z})}$ equals

$$\Big(\mathbb{E}_{\mathbf{y}}[\boldsymbol{v}_{\boldsymbol{\theta}}^{(k)}\otimes\boldsymbol{h}^{(k)}] - \widehat{\mathbb{E}}_{\mathbf{y}}[\boldsymbol{v}_{\boldsymbol{\theta}}^{(k)}\otimes\boldsymbol{h}^{(k)}]\Big)^{\top}\boldsymbol{S}_{\boldsymbol{\theta}}^{(k)}\Big(\mathbb{E}_{\mathbf{x}}[\boldsymbol{u}_{\boldsymbol{\theta}}^{(k)}(\boldsymbol{x})\boldsymbol{u}_{\boldsymbol{\theta}}^{(k)}(\boldsymbol{x})^{\top}]\Big)\boldsymbol{S}_{\boldsymbol{\theta}}^{(k)}\Big(\mathbb{E}_{\mathbf{y}}[\boldsymbol{v}_{\boldsymbol{\theta}}^{(k)}\otimes\boldsymbol{h}^{(k)}] - \widehat{\mathbb{E}}_{\mathbf{y}}[\boldsymbol{v}_{\boldsymbol{\theta}}^{(k)}\otimes\boldsymbol{h}^{(k)}]\Big)$$

which using constraints within each isotypic block and

$$\mathbb{E}_{\mathbf{x}}[\boldsymbol{u}_{\boldsymbol{\theta}}^{(k)}{}_{(m)}(\boldsymbol{x})\boldsymbol{u}_{\boldsymbol{\theta}}^{(k)}{}_{(m)}(\boldsymbol{x})^{\top}] \preceq \left\|\mathbb{E}_{\mathbf{x}}[\boldsymbol{u}_{\boldsymbol{\theta}}^{(k)}{}_{(m)}(\boldsymbol{x})\boldsymbol{u}_{\boldsymbol{\theta}}^{(k)}{}_{(m)}(\boldsymbol{x})^{\top}]\right\|\boldsymbol{I}_m \leq (1+\mathcal{E}_{\boldsymbol{\theta}}^{(k)})\boldsymbol{I}_m,$$

implies, due to (99), that

$$\left\|\mathsf{D}_{\boldsymbol{\theta}}^{(k)}\boldsymbol{h}^{(k)} - \widehat{\mathsf{D}}_{\boldsymbol{\theta}}^{(k)}\boldsymbol{h}^{(k)}\right\|^2_{\mathcal{L}^2_{P_{\mathbf{x}}}(\mathcal{X},\mathcal{Z})} \leq d_k\,(1+\mathcal{E}_{\boldsymbol{\theta}}^{(k)})\,(\sigma_1^{\boldsymbol{\theta}(k)})^2$$
$$\cdot\left\|\mathbb{E}_{\mathbf{y}}[\boldsymbol{v}_{\boldsymbol{\theta}}^{(k)}{}_{(1)}(\mathbf{y})\otimes\boldsymbol{h}_1^{(k)}(\mathbf{y})] - \widehat{\mathbb{E}}_{\mathbf{y}}[\boldsymbol{v}_{\boldsymbol{\theta}}^{(k)}{}_{(1)}\otimes\boldsymbol{h}_1^{(k)}]\right\|^2_{\mathbb{R}^m\times\mathcal{Z}}$$
$$\leq d_k\,(1+\mathcal{E}_{\boldsymbol{\theta}}^{(k)})\,(\sigma_1^{\boldsymbol{\theta}(k)})^2\,[A(\boldsymbol{v}_{\boldsymbol{\theta}},\boldsymbol{h})]^2\,\left\|\boldsymbol{h}_1^{(k)}\right\|^2_{\mathcal{L}^2_{P_{\mathbf{x}}}(\mathcal{Y},\mathcal{Z})}.$$

Therefore, bounding $\sigma_1^{\boldsymbol{\theta}(k)} \leq \sigma_1^{(k)} + |\sigma_1^{(k)} - \sigma_1^{\boldsymbol{\theta}(k)}| \leq 1 + \left\|\mathsf{D}_{\mathbf{y}|\mathbf{x}}^{(k)} - \mathsf{D}_{\boldsymbol{\theta}}^{(k)}\right\|$ and summing over isotypic components, since $\|\boldsymbol{h}\|^2_{\mathcal{L}^2_{P_{\mathbf{x}}}(\mathcal{Y},\mathcal{Z})} = \sum_{k\in[n_{\mathrm{iso}}],j\in[d_k]}\left\|\boldsymbol{h}_j^{(k)}\right\|^2_{\mathcal{L}^2_{P_{\mathbf{x}}}(\mathcal{Y},\mathcal{Z})} = \sum_{k\in[n_{\mathrm{iso}}]} d_k\left\|\boldsymbol{h}_1^{(k)}\right\|^2_{\mathcal{L}^2_{P_{\mathbf{x}}}(\mathcal{Y},\mathcal{Z})}$, we complete the proof of (100).

To show (101), we combine (103) and (105), and obtain that $\mathbb{E}_{\mathbf{x}}[z(\mathbf{x})[\mathsf{D}_{\boldsymbol{\theta}}\boldsymbol{h}](\mathbf{x})] - \widehat{\mathbb{E}}_{\mathbf{x}}[z[\widehat{\mathsf{D}}_{\boldsymbol{\theta}}\boldsymbol{h}]]$ can be written as

$$\sum_{k \in [n_{\mathrm{iso}}]} d_k \left[ \mathbb{E}_{\mathbf{x}}[\boldsymbol{u}_{\boldsymbol{\theta}}^{(k)}{}_{(1)}(\mathbf{x})z_1^{(k)}(\mathbf{x})]^{\top} \boldsymbol{S}_{\boldsymbol{\theta}}^{(k)} \mathbb{E}_{\mathbf{y}}[\boldsymbol{v}_{\boldsymbol{\theta}}^{(k)}{}_{(1)}(\mathbf{y}) \otimes \boldsymbol{h}_1^{(k)}(\mathbf{y})] - \widehat{\mathbb{E}}_{\mathbf{x}}[\boldsymbol{u}_{\boldsymbol{\theta}}^{(k)} z_1^{(k)}]^{\top} \boldsymbol{S}_{\boldsymbol{\theta}}^{(k)} \widehat{\mathbb{E}}_{\mathbf{y}}[\boldsymbol{v}_{\boldsymbol{\theta}}^{(k)}{}_{(1)} \otimes \boldsymbol{h}_1^{(k)}] \right].$$

Adding and subtracting mixed terms we then obtain for each isotypic component, $\frac{1}{d_k}\mathbb{E}_{\mathbf{x}}[z(\mathbf{x})[\mathsf{D}_{\boldsymbol{\theta}}^{(k)}\boldsymbol{h}](\mathbf{x})] - \widehat{\mathbb{E}}_{\mathbf{x}}[z[\widehat{\mathsf{D}}_{\boldsymbol{\theta}}^{(k)}\boldsymbol{h}]]$ can be expressed as

$$\mathbb{E}_{\mathbf{x}}[\boldsymbol{u}_{\boldsymbol{\theta}}^{(k)}{}_{(1)}(\mathbf{x})z_1^{(k)}(\mathbf{x})]^{\top} \boldsymbol{S}^{(k)} \left( \mathbb{E}_{\mathbf{y}}[\boldsymbol{v}_{\boldsymbol{\theta}}^{(k)}{}_{(1)}(\mathbf{y}) \otimes \boldsymbol{h}_1^{(k)}(\mathbf{y})] - \widehat{\mathbb{E}}_{\mathbf{y}}[\boldsymbol{v}_{\boldsymbol{\theta}}^{(k)}{}_{(1)} \otimes \boldsymbol{h}_1^{(k)}] \right)$$

$$+ \left( \mathbb{E}_{\mathbf{x}}[\boldsymbol{u}_{\boldsymbol{\theta}}^{(k)}{}_{(1)}(\mathbf{x})z_1^{(k)}(\mathbf{x})] - \widehat{\mathbb{E}}_{\mathbf{x}}[\boldsymbol{u}_{\boldsymbol{\theta}}^{(k)}{}_1 z_1^{(k)}] \right)^{\top} \boldsymbol{S}^{(k)} \mathbb{E}_{\mathbf{y}}[\boldsymbol{v}_{\boldsymbol{\theta}}^{(k)}{}_{(1)}(\mathbf{x}) \otimes \boldsymbol{h}_1^{(k)}(\mathbf{y})]$$

$$+ \left( \mathbb{E}_{\mathbf{x}}[\boldsymbol{u}_{\boldsymbol{\theta}}^{(k)}{}_{(1)}(\mathbf{x})z_1^{(k)}(\mathbf{x})] - \widehat{\mathbb{E}}_{\mathbf{x}}[\boldsymbol{u}_{\boldsymbol{\theta}}^{(k)}{}_1 z_1^{(k)}] \right)^{\top} \boldsymbol{S}^{(k)} \left( \mathbb{E}_{\mathbf{y}}[\boldsymbol{v}_{\boldsymbol{\theta}}^{(k)}{}_{(1)}(\mathbf{y}) \otimes \boldsymbol{h}_1^{(k)}(\mathbf{y})] - \widehat{\mathbb{E}}_{\mathbf{y}}[\boldsymbol{v}_{\boldsymbol{\theta}}^{(k)}{}_{(1)} \otimes \boldsymbol{h}_1^{(k)}] \right),$$

and consequently bounded using (99) as

$$\left\| \mathbb{E}_{\mathbf{x}}[z(\mathbf{x})[\mathsf{D}_{\boldsymbol{\theta}}^{(k)}\boldsymbol{h}](\mathbf{x})] - \widehat{\mathbb{E}}_{\mathbf{x}}[z[\widehat{\mathsf{D}}_{\boldsymbol{\theta}}^{(k)}\boldsymbol{h}]] \right\|_{\mathcal{Z}} \leq d_k \sigma_1^{\boldsymbol{\theta}(k)} \Big[ A(\boldsymbol{u}_{\boldsymbol{\theta}}, z) + A(\boldsymbol{v}_{\boldsymbol{\theta}}, \boldsymbol{h})$$
$$+ A(\boldsymbol{u}_{\boldsymbol{\theta}}, z)A(\boldsymbol{v}_{\boldsymbol{\theta}}, \boldsymbol{h}) \Big] \left\| z_1^{(k)} \right\|_{\mathcal{L}_{P_{\mathbf{x}}}^2(\mathcal{X})} \left\| \boldsymbol{h}_1^{(k)} \right\|_{\mathcal{L}_{\mathbf{y}}^2(\mathcal{Y}, \mathcal{Z})}.$$

Summing across isotypic components and bounding $\sigma_1^{\boldsymbol{\theta}(k)}$ as before, we complete the proof. $\qquad\square$

First note that coupling (100) with (94) ensures that we can prove regression bound via concentration result ensuring (99). To obtain similar result for set-wise regression, we set $z = \mathbb{1}_{\mathbb{A}}$ and use (101) to obtain the following.

**Proposition M.4.** *Under the assumptions of Proposition M.3, let $A(\boldsymbol{u}_{\boldsymbol{\theta}}, \mathbb{1}_{\mathbb{A}})A(\boldsymbol{v}_{\boldsymbol{\theta}}, \boldsymbol{h}) \leq A(\boldsymbol{u}_{\boldsymbol{\theta}}, \mathbb{1}_{\mathbb{A}}) + A(\boldsymbol{v}_{\boldsymbol{\theta}}, \boldsymbol{h})$. If*

$$|\mathbb{E}_{\mathbf{x}}[\mathbb{1}_{\mathbb{A}}(\mathbf{x})] - \widehat{\mathbb{E}}_{\mathbf{x}}[\mathbb{1}_{\mathbb{A}}]| / \mathbb{E}_{\mathbf{x}}[\mathbb{1}_{\mathbb{A}}(\mathbf{x})] \leq \eta_{\mathbb{A}} \tag{107}$$

*and $\eta_{\mathbb{A}} < 1/2$, then*

$$\left\| \mathsf{E}_{\boldsymbol{\theta}}[\boldsymbol{h}(\mathbf{y})|\mathbf{x} \in \mathbb{A}] - \widehat{\mathsf{E}}_{\boldsymbol{\theta}}[\boldsymbol{h}(\mathbf{y})|\mathbf{x} \in \mathbb{A}] \right\|_{\mathcal{Z}} \leq \left\| \mathbb{E}_{\mathbf{y}}[\boldsymbol{h}] - \widehat{\mathbb{E}}_{\mathbf{y}}[\boldsymbol{h}] \right\|_{\mathcal{Z}} + \frac{2 \|\boldsymbol{h}\|_{\mathcal{L}_{P_{\mathbf{x}}}^2(\mathcal{Y}, \mathcal{Z})}}{\sqrt{P[\mathbf{x} \in \mathbb{A}]}}$$
$$\times \left[ 2(1 + \mathcal{E}_{\boldsymbol{\theta}}) \Big( A(\boldsymbol{u}_{\boldsymbol{\theta}}, \mathbb{1}_{\mathbb{A}}) + A(\boldsymbol{v}_{\boldsymbol{\theta}}, \boldsymbol{h}) \Big) + \eta_{\mathbb{A}} \right], \tag{108}$$

*and for $\boldsymbol{h} = \mathbb{1}_{\mathbb{B}}$*

$$|P[\mathbf{y} \in \mathbb{B} \mid \mathbf{x} \in \mathbb{A}] - \widehat{P}_{\boldsymbol{\theta}}[\mathbf{y} \in \mathbb{B} \mid \mathbf{x} \in \mathbb{A}]| \leq \left\| \mathbb{E}_{\mathbf{y}}[\boldsymbol{h}] - \widehat{\mathbb{E}}_{\mathbf{y}}[\boldsymbol{h}] \right\|_{\mathcal{Z}}$$
$$+ \frac{2}{\widehat{\mathbb{E}}_{\mathbf{x}}[\boldsymbol{h}]} \sqrt{\frac{P[\mathbf{y} \in \mathbb{B}]}{P[\mathbf{x} \in \mathbb{A}]}} \Big[ 2(1 + \mathcal{E}_{\boldsymbol{\theta}})[A(\boldsymbol{u}_{\boldsymbol{\theta}}, \mathbb{1}_{\mathbb{A}}) + A(\boldsymbol{v}_{\boldsymbol{\theta}}, \mathbb{1}_{\mathbb{B}})] + \eta_{\mathbb{A}} \Big]. \tag{109}$$

*Proof.* Leveraging the representations in (103) and (105) with $z = \mathbb{1}_{\mathbb{A}}$, we get

$$\mathsf{E}_{\boldsymbol{\theta}}[\boldsymbol{h}(\mathbf{y})|\mathbf{x} \in \mathbb{A}] - \widehat{\mathsf{E}}_{\boldsymbol{\theta}}[\boldsymbol{h}(\mathbf{y})|\mathbf{x} \in \mathbb{A}] = \mathbb{E}_{\mathbf{y}}[\boldsymbol{h}] - \widehat{\mathbb{E}}_{\mathbf{y}}[\boldsymbol{h}] + \frac{\mathbb{E}_{\mathbf{x}}[\mathbb{1}_{\mathbb{A}}(\mathbf{x})[\mathsf{D}_{\boldsymbol{\theta}}\boldsymbol{h}](\mathbf{x})]}{\mathbb{E}[\mathbb{1}_{\mathbb{A}}]} - \frac{\widehat{\mathbb{E}}_{\mathbf{x}}[\mathbb{1}_{\mathbb{A}}[\widehat{\mathsf{D}}_{\boldsymbol{\theta}}\boldsymbol{h}]]}{\widehat{\mathbb{E}}_{\mathbf{x}}[\mathbb{1}_{\mathbb{A}}]} =$$
$$\mathbb{E}_{\mathbf{y}}[\boldsymbol{h}] - \widehat{\mathbb{E}}_{\mathbf{y}}[\boldsymbol{h}] + \mathbb{E}_{\mathbf{x}}[\mathbb{1}_{\mathbb{A}}(\mathbf{x})[\mathsf{D}_{\mathbf{y}|\mathbf{x}}\boldsymbol{h}](\mathbf{x})] \left( \frac{1}{\mathbb{E}[\mathbb{1}_{\mathbb{A}}(\mathbf{x})]} - \frac{1}{\widehat{\mathbb{E}}_{\mathbf{x}}[\mathbb{1}_{\mathbb{A}}]} \right) + \frac{\mathbb{E}_{\mathbf{x}}[\mathbb{1}_{\mathbb{A}}(\mathbf{x})[\mathsf{D}_{\boldsymbol{\theta}}\boldsymbol{h}](\mathbf{x})] - \widehat{\mathbb{E}}_{\mathbf{x}}[\mathbb{1}_{\mathbb{A}}[\widehat{\mathsf{D}}_{\boldsymbol{\theta}}\boldsymbol{h}]]}{\widehat{\mathbb{E}}_{\mathbf{x}}[\mathbb{1}_{\mathbb{A}}]}.$$

By triangular inequality applied to the norm in $\mathcal{Z}$, we get

$$\left\|\mathsf{E}_{\boldsymbol{\theta}}[\boldsymbol{h}(\mathbf{y})|\mathbf{x}\in\mathbb{A}]-\widehat{\mathsf{E}}_{\boldsymbol{\theta}}[\boldsymbol{h}(\mathbf{y})|\mathbf{x}\in\mathbb{A}]\right\|_{\mathcal{Z}}$$

$$\leq\left\|\mathbb{E}_{\mathbf{y}}[\boldsymbol{h}]-\widehat{\mathbb{E}}_{\mathbf{y}}[\boldsymbol{h}]\right\|_{\mathcal{Z}}+\|\mathbb{E}[\mathbb{1}_{\mathbb{A}}(\mathbf{x})[\mathsf{D}_{\boldsymbol{\theta}}f(\mathbf{x})]]\|_{\mathcal{Z}}\left|\frac{1}{\mathbb{E}[\mathbb{1}_{\mathbb{A}}(\mathbf{x})]}-\frac{1}{\widehat{\mathbb{E}}_{\mathbf{x}}[\mathbb{1}_{\mathbb{A}}]}\right|+\frac{\left\|\mathbb{E}_{\mathbf{x}}[\mathbb{1}_{\mathbb{A}}(\mathbf{x})[\mathsf{D}_{\boldsymbol{\theta}}h](\mathbf{x})]-\widehat{\mathbb{E}}_{\mathbf{x}}[\mathbb{1}_{\mathbb{A}}[\widehat{\mathsf{D}}_{\boldsymbol{\theta}}h]]\right\|_{\mathcal{Z}}}{\widehat{\mathbb{E}}_{\mathbf{x}}[\mathbb{1}_{\mathbb{A}}]}$$

$$\leq\left\|\mathbb{E}_{\mathbf{y}}[\boldsymbol{h}]-\widehat{\mathbb{E}}_{\mathbf{y}}[\boldsymbol{h}]\right\|_{\mathcal{Z}}+\|\mathbb{E}[\mathbb{1}_{\mathbb{A}}(\mathbf{x})[\mathsf{D}_{\boldsymbol{\theta}}h](\mathbf{x})]\|_{\mathcal{Z}}\frac{2\eta_{\mathbb{A}}}{\mathbb{P}[\mathbf{x}\in\mathbb{A}]}+\frac{\left\|\mathbb{E}_{\mathbf{x}}[\mathbb{1}_{\mathbb{A}}(\mathbf{x})[\mathsf{D}_{\boldsymbol{\theta}}h](\mathbf{x})]-\widehat{\mathbb{E}}_{\mathbf{x}}[\mathbb{1}_{\mathbb{A}}[\widehat{\mathsf{D}}_{\boldsymbol{\theta}}h]]\right\|_{\mathcal{Z}}}{\widehat{\mathbb{E}}_{\mathbf{x}}[\mathbb{1}_{\mathbb{A}}]},$$

where we have used Condition (107) in the last line to get that

$$\left|\frac{1}{\mathbb{P}[\mathbf{x}\in\mathbb{A}]}-\frac{1}{\widehat{\mathbb{E}}_{\mathbf{x}}[\mathbb{1}_{\mathbb{A}}]}\right|\leq\frac{\eta_{\mathbb{A}}}{(1-\eta_{\mathbb{A}})\mathbb{P}[\mathbf{x}\in\mathbb{A}]}\leq\frac{2\eta_{\mathbb{A}}}{\mathbb{P}[\mathbf{x}\in\mathbb{A}]}.$$

From Proposition M.3 and Condition (107) we get that

$$\frac{1}{\widehat{\mathbb{E}}_{\mathbf{x}}[\mathbb{1}_{\mathbb{A}}]}\left\|\mathbb{E}_{\mathbf{x}}[\mathbb{1}_{\mathbb{A}}(\mathbf{x})[\mathsf{D}_{\boldsymbol{\theta}}h](\mathbf{x})]-\widehat{\mathbb{E}}_{\mathbf{x}}[\mathbb{1}_{\mathbb{A}}(\mathbf{x})[\widehat{\mathsf{D}}_{\boldsymbol{\theta}}h]]\right\|_{\mathcal{Z}}\leq\frac{2(1+\mathcal{E}_{\boldsymbol{\theta}})\left[A(\boldsymbol{u}_{\boldsymbol{\theta}},\mathbb{1}_{\mathbb{A}})A(\boldsymbol{v}_{\boldsymbol{\theta}},\boldsymbol{h})\right]}{\mathbb{P}(\mathbf{x}\in\mathbb{A})}\|\mathbb{1}_{\mathbb{A}}\|_{\mathcal{L}^2_{P_{\mathbf{x}}}}\|\boldsymbol{h}\|_{\mathcal{L}^2_{P_{\mathbf{y}}}(\mathcal{Y},\mathcal{Z})}$$

Cauchy's Schwarz's inequality again and $\|\mathsf{D}_{\boldsymbol{\theta}}\|\leq1$ give

$$\|\mathbb{E}[\mathbb{1}_{\mathbb{A}}(\mathbf{x})[\mathsf{D}_{\boldsymbol{\theta}}h](\mathbf{x})]\|_{\mathcal{Z}}\leq\|\mathbb{1}_{\mathbb{A}}\|_{\mathcal{L}^2_{P_{\mathbf{x}}}}\|\mathsf{D}_{\boldsymbol{\theta}}\|\|\boldsymbol{h}\|_{\mathcal{L}^2_{P_{\mathbf{y}}}}\leq\|\mathbb{1}_{\mathbb{A}}\|_{\mathcal{L}^2_{P_{\mathbf{x}}}}\|\boldsymbol{h}\|_{\mathcal{L}^2_{P_{\mathbf{y}}}}=\sqrt{\mathbb{P}[\mathbf{x}\in\mathbb{A}]}\|\boldsymbol{h}\|_{\mathcal{L}^2_{P_{\mathbf{y}}}(\mathcal{Y},\mathcal{Z})}.$$

Combining the last four displays give the first result. The second result follows immediately for $\boldsymbol{h}=\mathbb{1}_{\mathbb{B}}$. $\square$

Consequence of this result is that we can bound the error in probability as we can derive concentration inequalities on the terms in (99) and (107). Then an union bound gives the estimation result for regression conditional on sets.

Next, we recall that $\mathsf{E}_{\mathbf{y}|\mathbf{x}}$ being $(1/\alpha)$-Schatten class operator, implies:

**Assumption M.5.** *Let there exist some constant $c>0$ such that for $\alpha>0$, any $i\geq1$ and any $k\in[n_{\mathrm{iso}}]$, we have $\sigma_i^{(k)}\leq c\,i^{-\alpha}$.*

Further, for any $\boldsymbol{h}\in\mathcal{L}^2_{\mathbf{y}}(\mathcal{Y},\mathcal{Z})$, we define $\overline{\boldsymbol{h}}(\mathbf{y})=\boldsymbol{h}(\mathbf{y})-\mathbb{E}[\boldsymbol{h}(\mathbf{y})]$ and

$$\gamma_{\mathbb{G}'}(\boldsymbol{h}):=\frac{1}{|\mathbb{G}'|-1}\sum_{\substack{g\in\mathbb{G}'\\g\neq e}}\mathbb{E}[\langle\overline{\boldsymbol{h}}(\mathbf{y}),\overline{\boldsymbol{h}}(g\triangleright_{\mathcal{Y}}\mathbf{y})\rangle]. \tag{110}$$

In the following, we consider observables $h$ satisfying the following condition (that is clearly satisfied for an indicator of a set of positive measure)

**Assumption M.6.** *Let there exists an absolute constant $C_0\geq1$ such that $(|\mathbb{G}'|-1)\gamma_{\mathbb{G}'}(h)\leq C_0\,\mathbb{E}[\|h(\mathbf{y})\|^2_{\mathcal{Z}}]$.*

Define

$$\eta_{\mathbb{A}}=\eta_{\mathbb{A}}(\delta):=\left(\frac{1-\mathbb{P}[\mathbf{x}\in\mathbb{G}\triangleright\mathbb{A}]}{\mathbb{P}[\mathbf{x}\in\mathbb{G}\triangleright\mathbb{A}]}\right)\frac{\log2\delta^{-1}}{N}+\sqrt{2\frac{\log2\delta^{-1}}{N}}\sqrt{\frac{1-\mathbb{P}[\mathbf{x}\in\mathbb{G}\triangleright\mathbb{A}]}{\mathbb{P}[\mathbf{x}\in\mathbb{G}\triangleright\mathbb{A}]}}.$$

**Theorem M.7.** *Let Assumptions M.6 and M.5 be satisfied. Let $P_{\mathbf{x}}$ and $P_{\mathbf{y}}$ are $\mathbb{G}$-invariant, and $\mathsf{D}_{\boldsymbol{\theta}}$ from (89) is $\mathbb{G}$-equivariant model, and let $\boldsymbol{h}\in\mathcal{L}^2_{\mathbf{y}}(\mathcal{Y},\mathcal{Z})$ and $\boldsymbol{f}\in\mathcal{L}^2_{\mathbf{x}}(\mathcal{X},\mathcal{Z})$ (with values in $\mathcal{Z}$) be subGaussian random variables. Assume in addition that the event $\mathbb{A}$ is anti-symmetric for $\mathbb{G}$ and that $m_k=m$ for all $k\in[n_{\mathrm{iso}}]$. Assume that $N\geq|\mathbb{G}|$. Then for any $\delta\in(0,1)$, it holds w.p.a.l $1-\delta$*

$$\left\|\mathbb{E}[\boldsymbol{h}(\mathbf{y})\,|\,\mathbf{x}\in\mathbb{A}]-\widehat{\mathsf{E}}_{\boldsymbol{\theta}}[\boldsymbol{h}(\mathbf{y})|\mathbf{x}\in\mathbb{A}]\right\|_{\mathcal{Z}}\lesssim_{C_0}\frac{\|\boldsymbol{h}\|_{\mathcal{L}^2_{\mathbf{y}}(\mathcal{Y},\mathcal{Z})}}{\sqrt{\mathbb{P}[\mathbf{x}\in\mathbb{G}\triangleright_{\mathcal{X}}\mathbb{A}]}}\left(\mathcal{E}_{\boldsymbol{\theta}}+\frac{\log(2n_{\mathrm{iso}}\delta^{-1})}{(d_{\mathrm{iso}}N)^{\frac{\alpha}{1+2\alpha}}}\right),$$

*and*

$$|\mathbb{P}(\mathbf{y}\in\mathbb{B}\,|\,\mathbf{x}\in\mathbb{A})-\widehat{\mathbb{P}}_{\boldsymbol{\theta}}(\mathbf{y}\in\mathbb{B}\,|\,\mathbf{x}\in\mathbb{A})|\lesssim_{C_0}\sqrt{\frac{\mathbb{P}[\mathbf{y}\in\mathbb{B}]}{\mathbb{P}[\mathbf{x}\in\mathbb{G}\triangleright_{\mathcal{X}}\mathbb{A}]}}\left(\mathcal{E}_{\boldsymbol{\theta}}+\frac{\log(2n_{\mathrm{iso}}\delta^{-1})}{(d_{\mathrm{iso}}N)^{\frac{\alpha}{1+2\alpha}}}+\sqrt{|\mathbb{G}|}\eta_{\mathbb{A}}\right).$$

*Proof.* This result follows immediately from Propositions M.3 and M.4 combined with Lemmas M.9 and Lemma M.10. Set

$$A(\boldsymbol{u_\theta}, \boldsymbol{f}) := C \sqrt{\frac{1}{|\mathbb{G}'|N}} \sqrt{C_0 \vee \frac{|\mathbb{G}'|}{N}} \, \log(2n_{\text{iso}}\delta^{-1}),$$

$$A(\boldsymbol{v_\theta}, \boldsymbol{h}) := C \sqrt{\frac{\max_{k\in[n_{\text{iso}}]}\{m_k\}}{|\mathbb{G}'|N}} \sqrt{C_0 \vee \frac{|\mathbb{G}'|}{N}} \, \log(2n_{\text{iso}}\delta^{-1}),$$

for some large enough absolute constant $C > 0$.

Then an union bound based on Lemmas M.9 and M.10 guarantees that (108) is satisfied w.p.a.l. $1 - \delta$ (up to a rescaling of the constant $C$):

$$\left\| \mathsf{E}_\theta[\boldsymbol{h}(\mathbf{y})|\mathbf{x} \in \mathbb{A}] - \widehat{\mathsf{E}}_\theta[\boldsymbol{h}(\mathbf{y})|\mathbf{x} \in \mathbb{A}] \right\|_{\mathcal{Z}} \leq$$
$$C \frac{\|\boldsymbol{h}\|_{\mathcal{L}_\mathbf{x}^2(\mathcal{X},\mathcal{Z})}}{\sqrt{P[\mathbf{x} \in \mathbb{A}]}} \left[ 2(1 + \mathcal{E}_\theta) \left( \sqrt{\frac{\max_{k\in[n_{\text{iso}}]}\{m_k\}}{|\mathbb{G}'|N}} \sqrt{C_0 \vee \frac{|\mathbb{G}'|}{N}} \, \log(2n_{\text{iso}}\delta^{-1}) \right) \right].$$

Next we use our bound on the representation bias in (96)

$$\|\mathbb{E}[\boldsymbol{y}(\mathbf{y}) \mid \mathbf{x} \in \mathbb{A}] - \mathsf{E}_\theta[\boldsymbol{y}(\mathbf{y}) \mid \mathbf{x} \in \mathbb{A}]\|_{\mathcal{Z}} \leq \left(\sigma_{r_m+1}^\star + \mathcal{E}_\theta\right) \frac{\|\boldsymbol{h}\|_{\mathcal{L}_\mathbf{y}^2(\mathcal{Y},\mathcal{Z})}}{\sqrt{\mathbb{P}[\mathbf{x} \in \mathbb{A}]}} \sqrt{\frac{1 + (|\mathbb{G}'|-1)\gamma_\mathbb{G}(\mathbb{A})}{|\mathbb{G}'|}}. \tag{111}$$

Recall that $\mathcal{E}_\theta = \max_{k\in[n_{\text{iso}}]}\{\mathcal{E}_\theta^{(k)}\}$. Under Assumption M.5, we have $\left\|[\mathsf{D}_{\mathbf{y}|\mathbf{x}}]_{r_m} - \mathsf{D}_\theta\right\| \leq \frac{1}{(d_{\text{iso}}m)^\alpha}$. In addition, $(|\mathbb{G}'| - 1)\gamma_\mathbb{G}(\mathbb{A}) \leq C_0$ under Assumption M.6.

Combining the last two display gives w.p.a.l $1 - \delta$

$$\left\| \mathbb{E}[\boldsymbol{h}(\mathbf{y}) \mid \mathbf{x} \in \mathbb{A}] - \widehat{\mathsf{E}}_\theta[\boldsymbol{h}(\mathbf{y})|\mathbf{x} \in \mathbb{A}] \right\|_{\mathcal{Z}} \lesssim_{C_0} \frac{\|\boldsymbol{h}\|_{\mathcal{L}_\mathbf{y}^2(\mathcal{Y},\mathcal{Z})}}{\sqrt{\mathbb{P}[\mathbf{x} \in \mathbb{G} \triangleright_\mathcal{X} \mathbb{A}]}} \left( \mathcal{E}_\theta + \frac{1}{(d_{\text{iso}}m)^\alpha} + \sqrt{\frac{m}{N}} \, \log(2n_{\text{iso}}\delta^{-1}) \right).$$

Balancing the previous display w.r.t. dimension $m$, we get that $m \asymp (d_{\text{iso}}^{-2\alpha}N)^{\frac{1}{1+2\alpha}}$ and the first result follows.

The bound for the conditional probability follows by picking $\boldsymbol{y} = \mathbb{1}_\mathbb{B}$. $\qquad\square$

## M.1. Quadratic Error Regression Bound

Our goal is to estimate the conditional expectation function

$$\boldsymbol{z}(\boldsymbol{x}) = \mathbb{E}[\boldsymbol{h}(\mathbf{y})|\mathbf{x}=\boldsymbol{x}] = \mathbb{E}[\boldsymbol{h}(\mathbf{y})] + [\mathsf{D}_{\mathbf{y}|\mathbf{x}}\boldsymbol{h}](\boldsymbol{x}).$$

Our estimator is

$$\widehat{\boldsymbol{z}}_\theta(\cdot) = \widehat{\mathbb{E}}_\mathbf{y}[\boldsymbol{y}] + [\widehat{\mathsf{D}}_\theta\boldsymbol{h}](\cdot).$$

**Theorem M.8.** *Assume that $Y$ is a sub-Gaussian random vector. Let Assumption M.5 be satisfied. Assume in addition that $\mathcal{E}_\theta \leq 1$, $m_k = m$ for all $k \in [n_{\text{iso}}]$. Then for any $\delta \in (0,1)$ such that $N \geq (c_u \vee c_v)^2 m \log(e\delta^{-1}n_{\text{iso}}) \vee |\mathbb{G}|$, it holds w.p.a.l. $1 - \delta$*

$$\|\boldsymbol{z} - \widehat{\boldsymbol{z}}_\theta\|_{\mathcal{L}_\mathbf{x}^2(\mathcal{X},\mathcal{Z})}^2 \lesssim \text{Tr}(\text{Cov}(Y)) \left( \mathcal{E}_\theta^2 + (d_{\text{iso}}|\mathbb{G}|N)^{\frac{-2\alpha}{1+2\alpha}} \log^2(\delta^{-1}n_{\text{iso}}) \right). \tag{112}$$

**Discussion** When the training of the NN is successful, we expect the statistical rate to dominate the optimization error $\max_{k\in[n_{\text{iso}}]}\{\mathcal{E}_\theta^{(k)}\}$ for large enough sample size $N$. For distribution containing symmetry invariants with large isotopic components ($m$ is large), we observe that exploiting this information in the construction of the NCP operator yields a substantial improvement in the statistical error rate as we go from a rate $N^{-\frac{\alpha}{1+2\alpha}}$ for standard NCP to $(N\,m)^{-\frac{\alpha}{1+2\alpha}}$ for eNCP.

*Proof.* Combining (100) with Lemma M.9 gives w.p.a.l. $1 - \delta$

$$\left\| \mathsf{E}_\theta\, \mathbf{y} - \widehat{\mathsf{E}}_\theta\, \mathbf{y} \right\|_{\mathcal{L}_{P_\mathbf{x}}^2(\mathcal{X})}^2 \lesssim (1 + \mathcal{E}_\theta)^3 \text{Tr}(\text{Cov}(\mathbf{y})) \frac{m}{|\mathbb{G}|N} \log^2(2n_{\text{iso}}\delta^{-1})$$
$$\lesssim \text{Tr}(\text{Cov}(\mathbf{y})) \frac{m}{|\mathbb{G}|N} \log^2(n_{\text{iso}}\delta^{-1}),$$

provide that $\mathcal{E}_{\boldsymbol{\theta}} \leq 1$. We derived in (94) an upper bound on the bias term

$$\left\| \mathbb{E}_{\mathbf{y}|\mathbf{x}}[\mathbf{y} \mid \mathbf{x} = \cdot] - \mathsf{E}_{\boldsymbol{\theta}}\mathbf{y} \right\|_{\mathcal{L}^2_{P_\mathbf{x}}(\mathcal{X},\mathcal{Z})}^2 \leq \mathrm{Tr}(\mathrm{Cov}(\mathbf{y})) \left( \frac{1}{(d_{\mathrm{iso}}m)^{2\alpha}} + \mathcal{E}_{\boldsymbol{\theta}}^2 \right). \tag{113}$$

Balancing the two bounds in the last two displays w.r.t. $m \asymp (|\mathbb{G}|d_{\mathrm{iso}}N)^{\frac{1}{1+2\alpha}}$, we get the result. $\qquad\square$

## M.2. Auxiliary Results.

Consider the function space $\mathcal{L}^2_\mathbf{y}(\mathcal{Y}, \mathcal{Z})$ where $\mathcal{Z}$ is endowed with an inner product $\langle \cdot, \cdot \rangle = \langle \cdot, \cdot \rangle_{\mathcal{Z}}$. If the distribution of $\mathbf{y}$ is $\mathbb{G}'$-invariant, then for any $h \in \mathcal{L}^2_\mathbf{y}(\mathcal{Y}, \mathcal{Z})$, we use the estimator $\widehat{\mathbb{E}}_\mathbf{y}[h]$ in (97) as an estimator of $\mathbb{E}[h(\mathbf{y})]$.

**Lemma M.9.** *Assume that the distribution $P_\mathbf{y}$ of $\mathbf{y}$ is $\mathbb{G}$-invariant and let $\mathbb{G}' \leq \mathbb{G}$. Let there exists a function $\boldsymbol{h} \in \mathcal{L}^2_\mathbf{y}(\mathcal{Y}, \mathcal{Z})$ such that $\boldsymbol{h}(\mathbf{y})$ is subGaussian. Then there exists an absolute constant $C > 0$ such that for any $\delta \in (0, 1)$, it holds w.p.a.l. $1 - \delta$*

$$\left\| \widehat{\mathbb{E}}_\mathbf{y}[\boldsymbol{h}] - \mathbb{E}[\boldsymbol{h}(\mathbf{y})] \right\|_{\mathcal{Z}} \leq C \sqrt{\frac{\log^2 2\delta^{-1}}{|\mathbb{G}'|\,N}} \sqrt{\mathbb{E}[\|\overline{\boldsymbol{h}}(\mathbf{y})\|_{\mathcal{Z}}^2] + (|\mathbb{G}'| - 1)\gamma_{\mathbb{G}'}(\boldsymbol{h}) + \frac{|\mathbb{G}'|\mathbb{E}[\|\overline{\boldsymbol{h}}(\mathbf{y})\|_{\mathcal{Z}}^2]}{N}}.$$

*Assume in addition that there exists an absolute constant $C_0 \geq 1$ such that $(|\mathbb{G}'| - 1)\gamma_{\mathbb{G}'}(\overline{\boldsymbol{h}}) \leq C_0\,\mathbb{E}[\|\boldsymbol{h}(\mathbf{y})\|_{\mathcal{Z}}^2]$. Then for any $\delta \in (0, 1)$, it holds w.p.a.l. $1 - \delta$*

$$\left\| \widehat{\mathbb{E}}_\mathbf{y}[\boldsymbol{h}] - \mathbb{E}[\boldsymbol{h}(\mathbf{y})] \right\|_{\mathcal{Z}} \leq C \sqrt{\frac{\mathbb{E}[\|\overline{\boldsymbol{h}}(\mathbf{y})\|_{\mathcal{Z}}^2]}{|\mathbb{G}'|\,N}} \sqrt{(1 + C_0) + \frac{|\mathbb{G}'|}{N}} \log 2\delta^{-1}.$$

*Note that similar bounds hold valid for the $\mathbb{G}$-invariant distribution $P_\mathbf{x}$ and any function $\boldsymbol{f} \in \mathcal{L}^2_{P_\mathbf{x}}(\mathcal{X}, \mathcal{Z})$ such that $\boldsymbol{f}(\mathbf{x})$ is subGaussian.*

*Proof.* We note that

$$\widehat{\mathbb{E}}_\mathbf{y}[\boldsymbol{h}] - \mathbb{E}[\boldsymbol{h}(\mathbf{y})] = \frac{1}{N} \sum_{i=1}^N Z_i \quad \text{with} \quad Z_i = \frac{1}{|\mathbb{G}'|} \sum_{g \in \mathbb{G}'} \boldsymbol{h}(g \triangleright_\mathcal{Y} \mathbf{y}_i) - \mathbb{E}_{\mathbf{y}_i}[\boldsymbol{h}(g \triangleright_\mathcal{Y} \mathbf{y}_i)], \ \forall i \in [N].$$

Define

$$Z := \frac{1}{|\mathbb{G}'|} \sum_{g \in \mathbb{G}'} \boldsymbol{h}(g \triangleright_\mathcal{Y} \mathbf{y}) - \mathbb{E}_\mathbf{y}[\boldsymbol{h}(g \triangleright_\mathcal{Y} \mathbf{y})], \tag{114}$$

and, for brevity, set $\|\boldsymbol{z}\| = \|\boldsymbol{z}\|_{\mathcal{Z}} = \sqrt{\langle \boldsymbol{z}, \boldsymbol{z} \rangle_{\mathcal{Z}}}$ for any $\boldsymbol{z} \in \mathcal{Z}$. We apply Proposition M.12, to get w.p.a.l. $1 - \delta$

$$\left\| \widehat{\mathbb{E}}_\mathbf{y}[\boldsymbol{h}] - \mathbb{E}_\mathbf{y}[\boldsymbol{h}(\mathbf{y})] \right\| \leq \frac{4\sqrt{2}}{\sqrt{N}} \sqrt{\mathrm{Var}_\mathbf{y}(\|Z\|) + \frac{\|Z\|_{\psi_2}^2}{N} \log \frac{2}{\delta}}.$$

Using the triangular inequality successively on $\|\cdot\|$ and $\|\cdot\|_{\psi_2}$ and the $\mathbb{G}'$-invariance of $P_\mathbf{y}$, $\left\|\overline{\boldsymbol{h}}(g \triangleright_\mathcal{Y} \mathbf{y})\right\|_{\psi_2} = \left\|\overline{\boldsymbol{h}}(\mathbf{y})\right\|_{\psi_2}$ for any $g \in \mathbb{G}'$, we get that

$$\left\| \|Z\| \right\|_{\psi_2} \lesssim \left\| \|\overline{\boldsymbol{h}}(\mathbf{y})\| \right\|_{\psi_2}.$$

We note next that $\left\|\overline{\boldsymbol{h}}(\mathbf{y})\right\|$ is subGaussian. Consequently the well-known property of equivalence of moments for subGaussian distributions gives $\|Z\|_{\psi_2} \lesssim \left\| \|\overline{\boldsymbol{h}}(\mathbf{y})\| \right\|_{\psi_2} \lesssim \mathbb{E}[\|\overline{\boldsymbol{h}}(\mathbf{y})\|^2]$. We derive now a control on $\mathrm{Var}_\mathbf{y}(\|Z\|) \leq \mathbb{E}[\|Z\|^2]$. Using the $\mathbb{G}'$-invariance of $P_\mathbf{y}$, we get

$$\begin{aligned} \mathrm{Var}(\|Z\|) &\leq \frac{\mathbb{E}[\|\overline{\boldsymbol{h}}(\mathbf{y})\|^2]}{|\mathbb{G}'|} + \frac{1}{|\mathbb{G}'|} \sum_{\substack{g \in \mathbb{G}' \\ g \neq e}} \mathbb{E}[\langle \boldsymbol{h}(\mathbf{y}) - \mathbb{E}[\boldsymbol{h}(\mathbf{y})], \boldsymbol{h}(g \triangleright_\mathcal{Y} \mathbf{y}) - \mathbb{E}[\boldsymbol{h}(\mathbf{y})] \rangle] \\ &= \frac{\mathbb{E}[\|\overline{\boldsymbol{h}}(\mathbf{y})\|^2]}{|\mathbb{G}'|} + \frac{(|\mathbb{G}'| - 1)\gamma_{\mathbb{G}'}(\boldsymbol{h})}{|\mathbb{G}'|} \leq (1 + C_0)\frac{\mathbb{E}[\|\overline{\boldsymbol{h}}(\mathbf{y})\|^2]}{|\mathbb{G}'|}. \end{aligned} \tag{115}$$

Hence we get the result.

$\square$

We focus now on a concentration bound for indicator functions $z = \mathbb{1}_\mathbb{A}$ for any event $\mathbb{A} \in \Sigma_\mathcal{X}$. We define

$$Z_A := \widehat{\mathbb{E}}_\mathbf{x}[\mathbb{1}_\mathbb{A}] - \mathbb{P}[\mathbf{x} \in \mathbb{A}] = \frac{1}{|\mathbb{G}'|} \sum_{g \in \mathbb{G}'} (\mathbb{1}_{g^{-1} \triangleright_\mathcal{X} \mathbb{A}}(\mathbf{x}) - \mathbb{E}[\mathbb{1}_{g^{-1} \triangleright_\mathcal{X} \mathbb{A}}(\mathbf{x})])$$

$$= \frac{1}{|\mathbb{G}'|} \sum_{g \in \mathbb{G}'} \left( \mathbb{1}_{g^{-1} \triangleright_\mathcal{X} \mathbb{A}}(\mathbf{x}) - \mathbb{P}[\mathbf{x} \in \mathbb{A}] \right) = \left( \frac{1}{|\mathbb{G}'|} \sum_{g \in \mathbb{G}'} \mathbb{1}_{g^{-1} \triangleright_\mathcal{X} \mathbb{A}}(\mathbf{x}) \right) - \mathbb{P}[\mathbf{x} \in \mathbb{A}]. \tag{116}$$

Note that we always have $|Z_\mathbb{A}| \leq 1$ but this bound can be quite conservative as we could get a much sharper bound for some events $\mathbb{A}$. We denote by $\gamma_{\mathbb{G}',\infty}(\mathbb{A})$ the smallest deterministic upper-bound on $\frac{1}{|\mathbb{G}'|} \sum_{g \in \mathbb{G}'} \mathbb{1}_{g^{-1} \triangleright_\mathcal{X} \mathbb{A}}(\mathbf{x})$ (For instance when $\mathbb{A}$ is an antisymmetric event, then we have $\gamma_{\mathbb{G}',\infty}(\mathbb{A}) = 1/|\mathbb{G}'|$). Then we have

$$-\mathbb{P}[\mathbf{x} \in \mathbb{A}] \leq Z_\mathbb{A} \leq \gamma_{\mathbb{G}',\infty}(\mathbb{A}) - \mathbb{P}[\mathbf{x} \in \mathbb{A}]. \tag{117}$$

Define also

$$\Upsilon_{\mathbb{G}',X}(\mathbb{A}) := \mathbb{P}(\mathbf{x} \in \mathbb{A})(1 - \mathbb{P}(\mathbf{x} \in \mathbb{A})) + (|\mathbb{G}'| - 1)\left(\gamma_{\mathbb{G}'}(\mathbb{A}) - \mathbb{P}[\mathbf{x} \in \mathbb{A}]\right)\mathbb{P}[\mathbf{x} \in \mathbb{A}]. \tag{118}$$

**Lemma M.10.** *Let the distribution of* $\mathbf{x}$ *be* $\mathbb{G}'$-*invariant. Then for any* $\mathbb{A} \in \Sigma_\mathcal{X}$ *and any* $\delta \in (0,1)$, *it holds w.p.a.l.* $1 - \delta$

$$|\widehat{\mathbb{E}}_\mathbf{x}[\mathbb{1}_\mathbb{A}] - \mathbb{E}[\mathbb{1}_\mathbb{A}(\mathbf{x})]| \leq |\gamma_{\mathbb{G}',\infty}(\mathbb{A}) - \mathbb{P}[\mathbf{x} \in \mathbb{A}]| \frac{\log 2\delta^{-1}}{N} + \sqrt{\frac{\Upsilon_{\mathbb{G}',\mathbf{x}}(\mathbb{A})}{|\mathbb{G}'|}} \sqrt{2\frac{\log 2\delta^{-1}}{N}}.$$

*Assume in addition that* $g \triangleright \mathbb{A} \cap \mathbb{A} = \emptyset$ *for all* $g \in \mathbb{G}' \setminus \{e\}$. *Then it holds w.p.a.l.* $1 - \delta$

$$\frac{|\widehat{\mathbb{E}}_\mathbf{x}[\mathbb{1}_\mathbb{A}] - \mathbb{E}[\mathbb{1}_\mathbb{A}(\mathbf{x})]|}{\mathbb{E}[\mathbb{1}_\mathbb{A}(\mathbf{x})]} \leq \left( \frac{1 - \mathbb{P}[\mathbf{x} \in \mathbb{G}' \triangleright \mathbb{A}]}{\mathbb{P}[\mathbf{x} \in \mathbb{G}' \triangleright \mathbb{A}]} \right) \frac{\log 2\delta^{-1}}{N} + \sqrt{2\frac{\log 2\delta^{-1}}{N}} \sqrt{\frac{1 - \mathbb{P}[\mathbf{x} \in \mathbb{G}' \triangleright \mathbb{A}]}{\mathbb{P}[\mathbf{x} \in \mathbb{G}' \triangleright \mathbb{A}]}}.$$

*If the distribution of* $\mathbf{y}$ *is* $\mathbb{G}'$-*invariant, then an identical result is immediately available for* $\mathbf{y}$ *by the same proof argument.*

**Remark M.11.** Using the standard empirical mean estimator that does not take advantage of $\mathbb{G}$-invariance, we obtain a concentration bound with a slower rate. For example, for an antisymmetric event $A$, we would achieve, w.p.a.l. $1 - \delta$, the following result:

$$\frac{|\widehat{\mathbb{E}}_\mathbf{x}[\mathbb{1}_\mathbb{A}] - \mathbb{E}[\mathbb{1}_\mathbb{A}(\mathbf{x})]|}{\mathbb{E}[\mathbb{1}_\mathbb{A}(\mathbf{x})]} \leq \left( \frac{1 - \mathbb{P}[\mathbf{x} \in \mathbb{A}]}{\mathbb{P}[\mathbf{x} \in \mathbb{A}]} \right) \frac{\log 2\delta^{-1}}{N} + \sqrt{2\frac{\log 2\delta^{-1}}{N}} \sqrt{\frac{1 - \mathbb{P}[\mathbf{x} \in \mathbb{A}]}{\mathbb{P}[\mathbf{x} \in \mathbb{A}]}}.$$

Specifically, leveraging $\mathbb{G}'$-invariance allows us to replace $\mathbb{P}[\mathbf{x} \in \mathbb{A}]$ with $\mathbb{P}[\mathbf{x} \in \mathbb{G}' \triangleright_\mathcal{X} \mathbb{A}]$, which represents the probability of the entire orbit of $A$ under the action of $\mathbb{G}'$. This becomes particularly interesting when $\mathbb{P}[\mathbf{x} \in \mathbb{A}] \ll \mathbb{P}[\mathbf{x} \in \mathbb{G}' \triangleright_\mathcal{X} \mathbb{A}]$, especially in the case of rare events where $\mathbb{P}[\mathbf{x} \in \mathbb{A}] \approx 0$.

*Proof.* Since $P_\mathbf{x}$ is $\mathbb{G}'$-invariant, we have $\mathbb{E}[\mathbb{1}_{g^{-1} \triangleright \mathbb{A}}(\mathbf{x})] = \mathbb{P}[\mathbf{x} \in \mathbb{A}]$ and $\text{Var}(\mathbb{1}_{g^{-1} \triangleright \mathbb{A}}(\mathbf{x})) = \text{Var}(\mathbb{1}_\mathbb{A}(\mathbf{x})) = \mathbb{P}[\mathbf{x} \in \mathbb{A}](1 - \mathbb{P}[\mathbf{x} \in \mathbb{A}])$, for any $g \in \mathbb{G}'$. Hence

$$\widehat{\mathbb{E}}_\mathbf{x}[\mathbb{1}_\mathbb{A}] - \mathbb{E}[\mathbb{1}_\mathbb{A}(\mathbf{x})] = \frac{1}{N} \sum_{i=1}^N Z_i \text{ with } Z_i = \frac{1}{|\mathbb{G}'|} \sum_{g \in \mathbb{G}'} \mathbb{1}_{g^{-1} \triangleright \mathbb{A}}(\mathbf{x}_i) - \mathbb{E}[\mathbb{1}_{g^{-1} \triangleright \mathbb{A}}(\mathbf{x}_i)], \ \forall i \in [N].$$

The $Z_i$'s are i.i.d. copies of $Z = Z_\mathbb{A}$. In view of (117), we can apply Hoeffding's inequality Bercu et al. (2015, Theorem 2.16). We get for any $\delta \in (0,1)$ w.p.a.l $1 - \delta$

$$|\widehat{\mathbb{E}}_\mathbf{x}[\mathbb{1}_\mathbb{A}] - \mathbb{E}[\mathbb{1}_\mathbb{A}(\mathbf{x})]| \leq \gamma_{\mathbb{G}',\infty}(\mathbb{A}) \sqrt{\frac{\log 2\delta^{-1}}{2N}}. \tag{119}$$

We propose to prove another bound based on application of Bernstein's inequality. We first prove an improved bound on $\mathrm{Var}(Z)$ as compared to the standard empirical mean estimator which does not exploit $\mathbb{G}$-invariance. Indeed we have

$$\mathrm{Var}(Z) = \frac{1}{|\mathbb{G}'|^2}\left(\sum_{g\in\mathbb{G}'}\mathrm{Var}(\mathbb{1}_{g^{-1}\triangleright\mathbb{A}}(\mathbf{x})) + \sum_{g\neq g'}\mathrm{Cov}\left(\mathbb{1}_{g^{-1}\triangleright\mathbb{A}}(\mathbf{x}),\mathbb{1}_{(g')^{-1}\triangleright\mathbb{A}}(\mathbf{x})\right)\right)$$

$$= \frac{\mathbb{P}(\mathbf{x}\in\mathbb{A})(1-\mathbb{P}(\mathbf{x}\in\mathbb{A}))}{|\mathbb{G}'|} + \frac{1}{|\mathbb{G}'|^2}\sum_{g\neq g'}\mathrm{Cov}\left(\mathbb{1}_{g^{-1}\triangleright\mathbb{A}}(\mathbf{x}),\mathbb{1}_{(g')^{-1}\triangleright\mathbb{A}}(\mathbf{x})\right).$$

Next, using again that $\mathbb{P}_X$ is $\mathbb{G}$-invariant, we get for any $g,g'\in\mathbb{G}'$

$$\mathrm{Cov}\left(\mathbb{1}_{g^{-1}\triangleright\mathbb{A}}(\mathbf{x}),\mathbb{1}_{(g')^{-1}\triangleright\mathbb{A}}(\mathbf{x})\right) = \tag{120}$$

$$\mathbb{P}[\mathbf{x}\in g^{-1}\triangleright\mathbb{A}\cap(g')^{-1}\triangleright\mathbb{A}] - \mathbb{P}[\mathbf{x}\in g^{-1}\triangleright\mathbb{A}]\,\mathbb{P}[\mathbf{x}\in(g')^{-1}\triangleright\mathbb{A}]$$
$$= \mathbb{P}[\mathbf{x}\in g^{-1}\triangleright\mathbb{A}\cap(g')^{-1}\triangleright\mathbb{A}] - \mathbb{P}[\mathbf{x}\in\mathbb{A}]^2. \tag{121}$$

Using again the invariance assumption, we note that

$$\sum_{g\neq g'}\mathrm{Cov}\big(\mathbb{1}_{g^{-1}\triangleright\mathbb{A}}(\mathbf{x}),\mathbb{1}_{(g')^{-1}\triangleright\mathbb{A}}(\mathbf{x})\big) = |\mathbb{G}'|\Big(\sum_{g\in\mathbb{G}',g\neq e}\mathbb{P}[\mathbf{x}\in\mathbb{A}\cap g\triangleright\mathbb{A}]\Big) - |\mathbb{G}'|(|\mathbb{G}'|-1)\mathbb{P}[\mathbf{x}\in\mathbb{A}]^2$$

Consequently by definition of $\gamma_{\mathbb{G}'}(A)$ in (92) and (93), we get

$$\sum_{g\in\mathbb{G}',g\neq e}\mathbb{P}[\mathbf{x}\in\mathbb{A}\cap g\triangleright A] = (|\mathbb{G}'|-1)\,\gamma_{\mathbb{G}'}(A)\,\mathbb{P}(\mathbf{x}\in\mathbb{A}).$$

Combining the last four displays, we get

$$\mathrm{Var}(Z) = \frac{\mathbb{P}(\mathbf{x}\in\mathbb{A})(1-\mathbb{P}(\mathbf{x}\in\mathbb{A}))+(|\mathbb{G}'|-1)\,(\gamma_{\mathbb{G}'}(A)-\mathbb{P}[\mathbf{x}\in\mathbb{A}])\mathbb{P}[\mathbf{x}\in\mathbb{A}]}{|\mathbb{G}'|} = \frac{\Upsilon_{\mathbb{G}',\mathbf{x}}(A)}{|\mathbb{G}'|}. \tag{122}$$

We note that for any $p\geq 3$

$$\sum_{i=1}^N \mathbb{E}[\big(\max(0,Z_i)\big)^p] \leq \frac{p!}{2}\max\big(0,\gamma_{\mathbb{G}',\infty}(\mathbb{A})-\mathbb{P}[\mathbf{x}\in\mathbb{A}]\big)^{p-2}N\,\mathrm{Var}(Z).$$

Then Bercu et al. (2015, Theorem 2.1) gives w.p.a.l. $1-\delta$

$$\widehat{\mathbb{E}}_{\mathbf{x}}[\mathbb{1}_\mathbb{A}] - \mathbb{E}[\mathbb{1}_\mathbb{A}(\mathbf{x})] \leq \max\big(0,\gamma_{\mathbb{G}',\infty}(\mathbb{A})-\mathbb{P}[\mathbf{x}\in\mathbb{A}]\big)\frac{\log\delta^{-1}}{N} + \sqrt{\mathrm{Var}(Z)}\sqrt{2\frac{\log\delta^{-1}}{N}}.$$

Applying the same reasoning to variables $-Z_1,\ldots,-Z_N$ and an union bound gives gives w.p.a.l. $1-2\delta$

$$|\widehat{\mathbb{E}}_{\mathbf{x}}[\mathbb{1}_\mathbb{A}] - \mathbb{E}[\mathbb{1}_\mathbb{A}(\mathbf{x})]| \leq \big|\gamma_{\mathbb{G}',\infty}(\mathbb{A})-\mathbb{P}[\mathbf{x}\in\mathbb{A}]\big|\frac{\log\delta^{-1}}{N} + \sqrt{\mathrm{Var}(Z)}\sqrt{2\frac{\log\delta^{-1}}{N}}. \tag{123}$$

Next, we note that when $g\triangleright\mathbb{A}\cap\mathbb{A}=\emptyset$ for all $g\in\mathbb{G}'\setminus\{e\}$, then $\gamma_{\mathbb{G}'}(\mathbb{A})=0$ and $\mathbb{P}[\mathbf{x}\in\mathbb{A}]=\mathbb{P}[\mathbf{x}\in\mathbb{G}'\triangleright\mathbb{A}]/|\mathbb{G}'|$. Consequently we get

$$\Upsilon_{\mathbb{G}',\mathbf{x}}(\mathbb{A}) = \frac{\mathbb{P}[\mathbf{x}\in\mathbb{G}'\triangleright\mathbb{A}](1-\mathbb{P}[\mathbf{x}\in\mathbb{G}'\triangleright\mathbb{A}])}{|\mathbb{G}'|} \quad\text{and}\quad \frac{\gamma_{\mathbb{G}',\infty}(\mathbb{A})-\mathbb{P}[\mathbf{x}\in\mathbb{A}]}{\mathbb{P}[\mathbf{x}\in\mathbb{A}]} = \frac{1}{\mathbb{P}[\mathbf{x}\in\mathbb{G}'\triangleright\mathbb{A}]} - 1.$$

Hence under the additional assumptions, dividing by $\mathbb{E}[\mathbb{1}_\mathbb{A}(\mathbf{x})]=\mathbb{P}[\mathbf{x}\in\mathbb{A}]$ gives w.p.a.l. $1-2\delta$

$$\frac{|\widehat{\mathbb{E}}_{\mathbf{x}}[\mathbb{1}_\mathbb{A}] - \mathbb{E}[\mathbb{1}_\mathbb{A}(\mathbf{x})]|}{\mathbb{E}[\mathbb{1}_\mathbb{A}(\mathbf{x})]} \leq \left(\frac{1}{\mathbb{P}[\mathbf{x}\in\mathbb{G}'\triangleright\mathbb{A}]}-1\right)\frac{\log\delta^{-1}}{N} + \sqrt{2\frac{\log\delta^{-1}}{N}}\sqrt{\frac{1-\mathbb{P}[\mathbf{x}\in\mathbb{G}'\triangleright\mathbb{A}]}{\mathbb{P}[\mathbf{x}\in\mathbb{G}'\triangleright\mathbb{A}]}}.$$

Replacing $\delta$ by $\delta/2$ gives w.p.a.l. $1-\delta$

$$\frac{|\widehat{\mathbb{E}}_{\mathbf{x}}[\mathbb{1}_\mathbb{A}] - \mathbb{E}[\mathbb{1}_\mathbb{A}(\mathbf{x})]|}{\mathbb{E}[\mathbb{1}_\mathbb{A}(\mathbf{x})]} \leq \left(\frac{1}{\mathbb{P}[\mathbf{x}\in\mathbb{G}'\triangleright\mathbb{A}]}-1\right)\frac{\log 2\delta^{-1}}{N} + \sqrt{2\frac{\log 2\delta^{-1}}{N}}\sqrt{\frac{1-\mathbb{P}[\mathbf{x}\in\mathbb{G}'\triangleright\mathbb{A}]}{\mathbb{P}[\mathbf{x}\in\mathbb{G}'\triangleright\mathbb{A}]}}. \tag{124}$$

$\square$

**Proposition M.12.** *Let $A_i$, $i \in [N]$ be i.i.d copies of a random variable $A$ in a separable Hilbert space with norm $\|\cdot\|$. If there exist constants $L > 0$ and $\sigma > 0$ such that for every $m \geq 2$, $\mathbb{E}\|A\|^m \leq \frac{1}{2}m!L^{m-2}\sigma^2$, then with probability at least $1 - \delta$*

$$\left\| \frac{1}{N} \sum_{i \in [N]} A_i - \mathbb{E}A \right\| \leq \frac{4\sqrt{2}}{\sqrt{N}} \sqrt{\sigma^2 + \frac{L^2}{N}} \log \frac{2}{\delta}. \tag{125}$$

**Lemma M.13** ((Sub-Gaussian random variable) Lemma 5.5. in (Vershynin, 2012))**.** *Let $Z$ be a random variable. Then, the following assertions are equivalent with parameters $K_i > 0$ differing from each other by at most an absolute constant factor.*

1. *Tails: $\mathbb{P}\{|Z| > t\} \leq \exp(1 - t^2/K_1^2)$ for all $t \geq 0$;*

2. *Moments: $(\mathbb{E}|Z|^p)^{1/p} \leq K_2\sqrt{p}$ for all $p \geq 1$;*

3. *Super-exponential moment: $\mathbb{E}\exp(Z^2/K_3^2) \leq 2$.*

*A random variable $Z$ satisfying any of the above assertions is called a sub-Gaussian random variable. We will denote by $K_3$ the sub-Gaussian norm.*

Consequently, a sub-Gaussian random variable satisfies the following equivalence of moments property. There exists an absolute constant $c > 0$ such that for any $m \geq 2$,

$$\left(\mathbb{E}|Z|^m\right)^{1/m} \leq cK_3\sqrt{m}\left(\mathbb{E}|Z|^2\right)^{1/2}.$$

**Lemma M.14.** *Assume that $Y$ is sub-Gaussian with sub-Gaussian norm $K$. We set $\sigma_\theta^2(Y) := \mathrm{Var}(\|Y - \mathbb{E}[\mathbf{y}]\|)$. Then there exists an absolute constant $C > 0$ such that for any $\delta \in (0, 1)$, it holds w.p.a.l. $1 - \delta$*

$$\left\| \widehat{\mathbb{E}}_{\mathbf{y}}[\mathbf{y}] - \mathbb{E}[\mathbf{y}] \right\| \leq \frac{C}{\sqrt{N}} \sqrt{\sigma^2(\mathbf{y}) + \frac{K^2}{N}} \log(2\delta^{-1}).$$

*Proof.* Set $Z := \|\mathbf{y} - \mathbb{E}\mathbf{y}\|$ and we recall that $\sigma^2(\mathbf{y}) := \mathrm{Var}(\|\mathbf{y} - \mathbb{E}[\mathbf{y}]\|)$. We check that the moment condition,

$$\mathbb{E}Z^m \leq \frac{1}{2}m!L^{m-2}\sigma^2(\mathbf{y})^2, \quad \forall m \geq 2,$$

for some constant $L > 0$ to be specified.

The condition is obviously satisfied for $m = 2$. Next for any $m \geq 3$, the Cauchy-Schwarz inequality and the equivalence of moment property give

$$\mathbb{E}Z^m \leq \left(\mathbb{E}Z^{2(m-2)}\right)^{1/2}\left(\mathbb{E}Z^4\right)^{1/2} \leq 4K_3^2\sigma_\theta^2(Y)^2\left(\mathbb{E}Z^{2(m-2)}\right)^{1/2}.$$

Next, by homogeneity, rescaling $Z$ to $Z/K_1$ we can assume that $K_1 = 1$ in Lemma M.13. We recall that if $Z$ is in addition non-negative random variable, then for every integer $p \geq 1$, we have

$$\mathbb{E}Z^p = \int_0^\infty \mathbb{P}\{Z \geq t\} pt^{p-1}\, dt \leq \int_0^\infty e^{1-t^2} pt^{p-1}\, dt = \left(\frac{ep}{2}\right)\Gamma\left(\frac{p}{2}\right).$$

With $p = 2(m - 2)$, we get that $\mathbb{E}Z^p \leq e(m-2)\Gamma(m-2) = e(m-2)! = em!/2$. Using again Lemma M.13, we can take $L = cK$ for some large enough absolute constant $c > 0$. Then Proposition M.12 gives the result.

$\square$

