# OpenReview forum: "Representation Learning for Equivariant Inference with Guarantees"
_ICML.cc/2026/Conference — ICML 2026 regular_

### Official Review · Reviewer_f5XK · 2026-02-26

**Soundness:** 3
**Presentation:** 3
**Significance:** 3
**Originality:** 3
**Overall Recommendation:** 5
**Confidence:** 3

**Summary:**

This paper presents a framework for symmetry-aware uncertainty quantification -- equivariant Neural Conditional Probability (eNCP).
It aims to answer the question of how symmetry-aware representations that best capture conditional structure in the data can be learned systematically.
To this end, the authors bridge the fields of spectral contrastive learning and geometric deep learning (GDL).
The paper contributes with 1) eNCP methodological framework, 2) task-agnostic representation learning, and 3) non-asymptotic statistical learning guarantees, supported by empirical results on synthetic and real-world tasks.

**Compliance With Llm Reviewing Policy:**

Affirmed.

**Final Justification:**

I strongly encourage the authors to incorporate their rebuttal into the manuscript.
I recommend this paper for acceptance with Accept, and my concerns have been fully resolved.

**Key Questions For Authors:**

1. For the baselines, what would be the impact of (random, on-the-fly) data augmentation with symmetry in question? (For example, as it is commonly done for SO(3)-equivariant benchmarks (e.g. 3D point cloud classification), without increasing the number of samples.)
2. Could the authors elaborate on how the equivariant models are built for the experiments?
3. What is the computational overhead introduced by the symmetry-awareness addition to the NCP in the proposed method?
4. To continue the line of Q3 and with reference to Appendix E.1, how practical would it be to apply eNCP to a (discretised) SO(3)?

Addressing these questions will help me finalise the score.

**Limitations:**

yes

**Strengths And Weaknesses:**

Strengths:

1. The combination of the GDL framework with spectral contrastive learning appears to be original and is well formulated in this paper.

2. The paper is well organised and written. Especially appreciated are the acronyms and the notation list.

3. The demonstration supports the presented theory across multiple tasks.


____
Weaknesses:

1. Model architectures: I recommend that the main paper should include a more detailed architecture description, which I currently find superficial.
Ideally, it should be demonstrated to be indeed equivariant, e.g. with some numerical experiment.

2. It would serve the paper well if it included baselines trained with (random) augmentation (not just more samples), especially in the G-equivariant regression case.

3. While there is sample efficiency/complexity discussion, computational complexity analysis is missing.

4. Not so major: As pointed out by the authors in the limitations, the user needs to fully specify the symmetry prior. I recommend that this be stated upfront, at the beginning of the paper.

---

> ### Author Rebuttal · Authors · 2026-03-31
>
> We thank the reviewer for the thoughtful and constructive feedback, and for recognizing both the originality of the work and the quality of its presentation. We agree that the main paper should include additional architectural detail, and that augmentation baselines are informative. We address the comments and questions jointly by topic below.
> ## Architecture details and equivariance/invariance numerical verification
> In the revision, we will include architectural details from App. E and G in the main body, making the construction of the equivariant models explicit. Specifically, we will emphasize that the **NNs are built** using the `symm_learning` and `escnn` GDL libraries, which readily enable us to **parameterize the representations in the disentangled basis** (see Eq. 21 and `symm_learning.nn.Change2DisentangledBasis`). Furthermore, to support reproducibility and adoption, we provide [a code repository](https://anonymous.4open.science/r/symm_rep_learn-2B3A) containing the full experimental code and tutorial notebooks.
>
> Regarding numerical verification of equivariance, we refer the reviewer to Fig. 5, where we quantify the violation of the G-invariance of the learned PMD for all models. Notably, the **PMD approximations of eNCP and iDRF's are invariant up to numerical tolerance**, confirming that equivariance is enforced structurally by the parameterization, and not approximately learned. For eNCP, this implies **the G-equivariance of the representation functions and truncated operator**.
> ## Data augmentation baselines
> Symmetry priors can be leveraged either through data augmentation or through constraint enforcement, e.g., via equivariant NNs or penalty-based methods. Since we focused on the latter, additional data augmentation for fully equivariant models is typically not expected to provide any additional benefit. However, adding data augmentation to the symmetry-agnostic NCP baseline should improve performance. So, to address the reviewer’s point directly, we have added an NCP-with-augmentation (NCPaug) baseline to the synthetic experiment in App. G4. The results, summarized in [Fig. A1](https://anonymous.4open.science/r/symm_rep_learn-2B3A/plots/rebutal_results.pdf), show that augmentation improves over vanilla NCP, but still underperforms relative to eNCP, indicating that **explicit symmetry constraints are more effective than augmentation alone**. We thank the reviewer for this suggestion. We commit to adding augmentation baselines for completeness in the revision and discussing these results.
> ## Computational overhead
> We agree that computational overhead should be discussed more explicitly. In practice, the main overhead from equivariance arises during training, since the forward/backward passes of equivariant layers involve additional operations. However, inference-time cost remains essentially unchanged, since after training an equivariant NN can be run as a standard NN (Cesa et al., 2022). For the experiment in App. G4, we report that the average forward time for predicting the full cCDF over the discretized range is statistically equivalent for the two models (results over 30 forward passes):
> - NCP: `0.628 ± 0.422` ms vs. eNCP: `0.648 ± 0.613` ms
>
> This is consistent with the fact that equivariance primarily increases training time, while the learned model remains a standard NN at inference. We will add a short computational-complexity discussion to make this clearer.
> ## Practicality for discretized SO(3)
> In practice, eNCP can be applied to SO(3)/O(3) symmetries through a finite subgroup or discretization, and this is exactly the route we take in our 3D experiment, presented in Fig. 4-right, which uses the icosahedral group, a standard discretization of O(3). Such discretizations are common in the GDL literature and in practical implementations of steerable CNNs (Weiler et al., 2024), in part because the computational cost of the forward/backward training passes is not dramatically affected by the order of the symmetry group (Cesa et al., 2022). Consequently, our disentangled contrastive loss can be used to train G-steerable CNNs with essentially the same computational complexity as when training the same architectures with standard supervised losses.
> ## Symmetry prior assumption
> We fully agree with this point. Indeed, **our framework follows the standard GDL assumption** that symmetries are known a priori (as is typical in physics- and geometry-driven applications). That said, the reviewer is absolutely right that this should be stated more explicitly earlier in the paper. We will revise the introduction accordingly, provide a sensitivity analysis under misspecification (see reply to i3Ar), and note that extending the theory to partially unknown symmetries is an interesting future direction.
> ___
> We thank the reviewer again for the constructive feedback, and *we would be grateful if the reviewer would consider updating their score if these clarifications address their concerns*.

---

> > ### Author Rebuttal · Reviewer_f5XK · 2026-03-31
> >
> > I thank the authors for the detailed rebuttal addressing my questions.
> > The authors have done a good job.
> >
> > - Regarding the computational speed comparison, what explains the large standard deviation in both cases?
> > - Could the authors also provide the training speed comparison, e.g. per epoch or batch?
> >
> > I strongly encourage the authors to incorporate their rebuttal into the manuscript.
> >
> > *Conditioned on that* and given the quality of the rebuttal, I retain my original positive assessment of the work and increase the score.

---

> > > ### Author Response · Authors · 2026-04-07
> > >
> > > We thank the reviewer for the score increase and continued engagement.
> > >
> > > ## Large standard deviations in the runtime comparison
> > >
> > > The previously reported variance was an artifact of insufficient measurement passes and suboptimal GPU profiling; specifically, the earlier measurements did not account for warmup effects, CUDA kernel initialization, and thermal fluctuations. We apologize for this.
> > >
> > > ## Training and inference computational cost
> > > Below we report the computational cost of the eNCP and NCP models across different groups, averaged over 200 passes with proper warmup, using batch size $2048$, data dimensionality $|\mathcal{X}| = |\mathcal{Y}| = 128$, and encoder networks $φ_{θ}$ and $ψ_{θ}$ parameterized by 3-hidden-layer MLP/eMLP neural networks:
> > >
> > > | Group | Iso. Subspaces | Group Order | Model | No. trainable params (k) | Train Forward (ms) | Train Backward (ms) | Val Forward (ms) |
> > > |---|---|---|---|---:|---:|---:|---:|
> > > | ICO | 5 | 60 | eNCP | 9.2 *(60x fewer)* | 3.067 ± 0.105 *(3.7x)* | 2.237 ± 0.084 *(2.5x)* | 0.741 ± 0.091 *(1.1x)* |
> > > |  |  |  | NCP | 550.8 | 0.825 ± 0.032 | 0.896 ± 0.051 | 0.653 ± 0.080 |
> > > | D10 | 8 | 20 | eNCP | 16.7 *(20x fewer)* | 3.761 ± 0.128 *(5.4x)* | 3.185 ± 0.109 *(4.1x)* | 0.487 ± 0.030 *(1.2x)* |
> > > |  |  |  | NCP | 333.2 | 0.692 ± 0.063 | 0.773 ± 0.046 | 0.415 ± 0.025 |
> > > | C10 | 6 | 10 | eNCP | 28.7 *(10x fewer)* | 4.250 ± 0.492 *(5.6x)* | 2.707 ± 0.269 *(3.4x)* | 0.450 ± 0.033 *(1.2x)* |
> > > |  |  |  | NCP | 287.3 | 0.760 ± 0.298 | 0.797 ± 0.161 | 0.382 ± 0.028 |
> > > | C2 | 2 | 2 | eNCP | 139.3 *(2x fewer)* | 2.315 ± 0.705 *(3.2x)* | 1.540 ± 0.552 *(2.2x)* | 0.353 ± 0.014 *(1.2x)* |
> > > |  |  |  | NCP | 278.5 | 0.721 ± 0.167 | 0.710 ± 0.114 | 0.301 ± 0.011 |
> > >
> > > eNCP incurs a 3–6× per-batch training overhead, arising from basis expansions in equivariant layers and symmetry-aware estimation of means, covariances, and cross-covariances in the loss (Eq. 14). This overhead scales with the number of isotypic subspaces rather than the order of the group. Due to the same measurement issues described above, we also previously understated inference cost: eNCP incurs a mild ~1.2× increase over NCP due to the additional change-of-basis step in the encoder (Eq. 21).
> > >
> > > The per-batch overhead is offset by substantially faster convergence: as shown in [Fig. A1(d)](https://anonymous.4open.science/r/symm_rep_learn-2B3A/plots/rebutal_results.pdf), eNCP reaches a lower validation loss before epoch 500 than NCP achieves after 2500 epochs, a roughly 5× reduction in training batches required to reach a better solution. Combined with 2–60× fewer parameters depending on the group, the overall computational picture is favorable.
> > >
> > > We appreciate the reviewer’s suggestion to examine this more carefully, which led us to substantially optimize the symmetry-aware covariance computation, resulting in the runtime figures reported above. We will incorporate the rebuttal experiments and discussions into the final manuscript.

---

### Official Review · Reviewer_Wg3E · 2026-03-04

**Soundness:** 3
**Presentation:** 2
**Significance:** 3
**Originality:** 3
**Overall Recommendation:** 4
**Confidence:** 2

**Summary:**

This paper presents a new method called Equivariant Neural Conditional Probability (eNCP) for learning data representations that respect physical or geometric symmetries. The authors combine geometric deep learning with operator theory to approximate the conditional expectation operator, which allows the model to perform tasks like regression and uncertainty quantification with fewer data samples.

**Compliance With Llm Reviewing Policy:**

Affirmed.

**Final Justification:**

The authors have addressed my concerns to some extent. However, I still have reservations regarding the experimental evaluation. Despite these remaining concerns, the clarifications provided are sufficient for me to increase my rating to 4.

**Key Questions For Authors:**

1) The robotics experiments represent a very specific type of physical symmetry. How does the eNCP framework perform on other common benchmarks where symmetries are well-defined but the data structure is different, such as in 3D point cloud classification?

2) The theoretical guarantees rely on the assumption that the symmetry groups are known and correctly specified. In real-world applications where our physical priors might be slightly "off" or incomplete, how robust is the eNCP framework? Have you conducted any sensitivity analyses regarding imperfect symmetry priors?

**Limitations:**

yes

**Strengths And Weaknesses:**

*Disclaimer*: I do not feel confident in reviewing this paper because it uses very advanced concepts from group representation theory and operator mathematics that are outside my main area of expertise.

That said, the paper's effort addressing equivariant representation learning in a more principled manner seems very relevant. However, most of the paper focuses on dense mathematical notation, while the experimental validation is very limited. In my view, the paper would benefit from adding high-level explanations or intuitive summaries. Additional experiments on synthetic datasets with well-defined symmetries demonstrating the utility of the proposed framework would help as well.

---

> ### Author Rebuttal · Authors · 2026-03-31
>
> We thank the reviewer for the thoughtful and constructive feedback, and especially for acknowledging the technical background required to assess the paper. We agree that accessibility, intuition, and scope should be further improved. Importantly, many of the requested elements (synthetic experiments, implementation details, and tutorials) are **already present in the appendix and/or code repository**, and we will make them more prominent in the main paper.
>
> ## On notation and intuition
> At a high level, the method can be understood simply as learning a **symmetry-constrained low-rank operator** that captures **conditional structure** in the data, and we agree that this perspective should be emphasized more clearly early in the main text.
> On the other hand, our contribution sits at the intersection of geometric deep learning, operator learning, and spectral representation learning, which necessitates a unified mathematical language. Furthermore, **since these three areas of ML are rarely treated together in a singler framework**, we made a considerable effort to provide a **self-contained pedagogical supplementary material**, adding illustrative diagrams (see Figs. 2, 3, 6, 7, 8, 9, 10, 11, 12, 15, 16, and 17) with the sole purpose of **providing graphical intuition for complex theoretical concepts**.
> Furthermore, to improve accessibility, **we already include a notation table**, and in the revision, we will further:
> - introduce a short high-level overview of the method in algorithmic terms before formal development,
> - improve signposting and separation between intuition and technical results.
>
> If the reviewer has any specific suggestions on portions of the text, notation, or explanations that could be improved, we are happy to follow them.
>
> ##  Synthetic demonstrations
> We agree that intuitive synthetic examples should be more prominent. The paper **already includes two low-dimensional synthetic experiments** (App. G), where symmetry actions, constraints, and learned representations are directly visualized and easily interpreted.
>
> These experiments explicitly show how symmetry constraints shape both the learned representations and the operator structure (Fig. 9). In the revision, we propose to move the most illustrative example (currently Fig. 12) into the main paper, so that readers encounter an interpretable setting before the more complex experiments. We are also providing [tutorial-style notebooks](https://anonymous.4open.science/r/symm_rep_learn-2B3A), **supporting both accessibility and reproducibility**.
>
> ## Applicability beyond robotics (3D point clouds)
> In the submitted version, we prioritized the cGMM and robotics experiments because they provided the strongest evidence of practical relevance. While we do not study 3D point-cloud classification in this paper, our framework is directly compatible with that setting. **eNCP is not tied to a specific data modality or architecture**; rather, it is an equivariant representation-learning framework that **can be built on top of any GDL backbone**, including the E(3)-equivariant architectures commonly used for 3D data. As discussed in line 224 and App. E.1, our framework handles continuous compact groups through the same discretization strategy routinely used in G-steerable networks, and non-compact groups through an appropriate selection of NN backbone.
> Importantly, we chose our practical setup to avoid introducing theoretical overhead that is orthogonal to the paper’s main contribution. That said, our experiments already partially cover this regime. In Fig. 4-right, we study a 3D environment with icosahedral symmetry. The icosahedral group is a standard finite subgroup used to discretize O(3), so we view these results as evidence that the framework is compatible with 3D symmetry-aware settings beyond the robotics examples shown in the paper.
>
> ## Robustness to imperfect symmetry priors
> We agree that robustness is important. While **our framework follows the standard GDL assumption** that symmetries are known a priori (as is typical in physics and geometry-driven applications), **we have prepared a sensitivity analysis under symmetry misspecification** (discussed in detail in the reply to *i3Ar*, and summarized in [Figs. A2-4](https://anonymous.4open.science/r/symm_rep_learn-2B3A/plots/rebutal_results.pdf))
>
> We find that performance degrades **gracefully** as misspecification increases, with degradation remaining localized to regions where the symmetry assumption is violated. This indicates that **the method remains reliable in practice even when symmetry assumptions are only approximately satisfied**.
> ___
> We thank the reviewer again for the constructive feedback and hope these clarifications improve the accessibility and clarify the scope and applicability of the work. We would be grateful if the reviewer would consider updating their score if these clarifications address their concerns.

---

> > ### Author Rebuttal · Reviewer_Wg3E · 2026-04-02
> >
> > I thank the authors for the detailed rebuttal addressing my questions. I'm updating my score. Nevertheless, I still believe that the paper would benefit from an experiment that adds a clear demonstration of their core contribution.

---

> > > ### Author Response · Authors · 2026-04-08
> > >
> > > We thank the reviewer sincerely for the careful follow-up, for increasing the score, and for the more favorable assessment of our work. We are especially grateful that the rebuttal helped address part of the reviewer’s concerns.
> > >
> > > Regarding the remaining reservation about the experimental evaluation, we would like to emphasize that we substantially strengthened the empirical case during the rebuttal, both through additional experiments and the present appendix material. In particular, we added misspecification experiments and further synthetic analyses aimed at making the core contribution more intuitive and easier to assess.
> > >
> > > At the same time, we agree with the reviewer that the core contributions can be showcased in a single core experiment: the synthetic cGMM experiment already serves as the intended core demonstration of the method, but we did not exploit it fully as a visual and intuitive illustration in the manuscript. This experiment allows us to vary symmetry groups and data dimensionality, evaluate PMD approximation error directly, and obtain a graphical understanding of both the symmetry priors (cf. Fig. 3) and their effect on approximation quality. While the paper reports this experiment quantitatively, we agree that its qualitative and visual side should have been brought forward more prominently.
> > >
> > > To make this especially clear, we prepared a dedicated tutorial notebook for the 2D cGMM setting:  [2D cGMM tutorial notebook](https://anonymous.4open.science/r/symm_rep_learn-2B3A/paper/examples/cGMM_2D/2D_cGMM.ipynb)
> > >
> > > This notebook was designed to extract as much intuition as possible from the synthetic setup. In particular, it includes training-dynamics comparisons across models, sample-efficiency plots, visualizations of the conditional expectations predicted by eNCP and NCP, 2D views of PMD approximation error as a function of dataset size, and direct image-level PMD predictions for all models. Together, these plots make it much easier to visually assess the gain brought by symmetry constraints.
> > >
> > > We value this experiment especially because it provides a clear graphical interpretation of the symmetry priors, their consequences for the learned representations, and their impact on approximation quality. More broadly, throughout both the manuscript and the rebuttal, we made a considerable effort to provide visual intuition for otherwise abstract theoretical ideas, and we thank the reviewer for pushing us to make this core synthetic demonstration sharper and more explicit.
> > >
> > > In the final version, we plan to include part of these additional visualizations in the main paper and/or appendix, so that the improvement is depicted more directly in the manuscript itself rather than only quantitatively.

---

### Official Review · Reviewer_L4iB · 2026-03-12

**Soundness:** 3
**Presentation:** 2
**Significance:** 3
**Originality:** 2
**Overall Recommendation:** 4
**Confidence:** 1

**Summary:**

The authors propose to extend NCP [Kostic 2024a] to group-equivariant distributions, using equivariant neural networks. NCP is a recent framework that proposes to solve the task of conditional probability estimation by learning the conditional expectation operator - an approach that allows to solve a variety of tasks (such as conditional density estimation, regression, and uncertainty quantification) using a single training objective. The proposed approach adapts NCP by decomposing the learned latent space into several blocks based on the isotopic decomposition of the G-symmetric operator Hilbert space. They demonstrate that this decomposition improves provably the expectation operator approximation of NCP in the case of group-equivariant distributions. Moreover, they evaluate their approach on the task of conditional expectation operator learning (using gaussian mixtures data), regression and uncertainty quantification (using a regression task from robotics [Ordonez-Apraez 2024]).

**Compliance With Llm Reviewing Policy:**

Affirmed.

**Final Justification:**

The current paper is well-motivated and a solid extension of NCP. I therefore propose to increase my current score to 4.

**Key Questions For Authors:**

1) How is the decomposition of $E_\theta$, $\phi_\theta$ and $\psi_\theta$ obtained in practice ?

2) How can the proposed framework be adapted in the case where the variables X and Y are not equivariant to the same groups ?

**Limitations:**

Yes.

**Strengths And Weaknesses:**

**Strengths**

The idea of extending NCP to group equivariance is novel, to the best of my knowledge. The authors provide theoretical guarantees for their proposed approach, which demonstrate the advantage of encoding equivariance into NCP. Moreover, the approach is evaluated on synthetic and real-world tasks. Experimental results demonstrate a clear advantage of the proposed approach in the settings of regression and uncertainty quantification, and a mild advantage in the context of conditional expectation operator learning.

**Weaknesses**

**Clarity.** The paper is generally hard to follow. This is mainly due to the use of heavy notations that tend to obscure the main argument. It seems that the paper would benefit from using simplified notations, at least in the main body, that would focus on communicating only the main concepts and intuitions about the introduced framework.

**Limited experimental validation.** Although the work is mainly theoretical, the experimental validation is limited, both in datasets and baseline, especially if we compare to the reference work [Kostic 2024b] that their approach extends.

**Reproducibility.** Important parts of the framework, such as the decomposition of the encoders $\phi_\theta$ and $\psi_\theta$ in practical cases, do not seem to be described with enough detail for the reproducibility of their results.

---

> ### Author Rebuttal · Authors · 2026-03-31
>
> We thank the reviewer for the thoughtful feedback, and for recognizing the novelty of extending NCP to the equivariant setting, the theoretical guarantees, and the empirical gains on regression and uncertainty quantification tasks. We also agree with the concerns regarding clarity, experimental breadth, and reproducibility. Importantly, many of the practical details and additional supporting experiments are already contained in Apps. E and G and in the accompanying code repository, but we agree that these elements should have been made more prominent in the main paper. Below, we address the main concerns and questions by topic.
>
> ## Clarity
> We agree that the presentation can be made substantially more accessible. At a high level, the method can be understood simply as learning a symmetry-constrained low-rank operator that captures conditional structure in the data, and we agree that this perspective should be emphasized much earlier in the main text. At the same time, the work sits at the intersection of geometric deep learning, operator learning, and spectral representation learning, which requires a non-trivial unified mathematical language. To improve accessibility, we made a considerable effort to provide a notation table and self-contained pedagogical supplementary material, adding illustrative diagrams (see Figs. 2, 3, 6, 7, 8, 9, 10, 11, 12, 15, 16, and 17) with the sole purpose of providing graphical intuition for complex theoretical concepts. In the revision, we will further add a short algorithmic overview and improve the separation between intuition and technical development.
>
> ## Practical implementation and reproducibility
> We agree that the architectural implementation details currently presented in Apps. E and G should also be brought into the main body to make the practical construction of the disentangled representations clearer. In practice, once the symmetry group and its actions on $X$ and $Y$ have been identified, the encoders $ψ_θ$ and $φ_θ$ can be parameterized using **any** $G$-equivariant NN backbone from standard GDL libraries such as `symm_learning` or `escnn`, which readily enable parameterization in the disentangled/isotypic basis (see Eq. 21 and `symm_learning.nn.Change2DisentangledBasis`, or `escnn.group.Representation.disentangle`). In this basis, the truncated operator $E_θ$ decomposes block-diagonally, with each block parameterized as an equivariant linear map (e.g., `symm_learning.nn.eLinear` or `escnn.nn.Linear`).
>
> Regarding reproducibility, App. E provides the technical details for the parameterization of the disentangled representations and truncated operator, while App. G gives a detailed description of the experimental setup. In addition, to further support accessibility and reproducibility, we have prepared a [code repository](https://anonymous.4open.science/r/symm_rep_learn-2B3A) that enables reproduction of all experiments and also contains several tutorial notebooks aimed at facilitating adoption. We would also be grateful for any suggestions on missing information that should be included either in the main body or in the appendix.
>
> ## Experimental validation
> In contrast to NCP focused on UQ tasks, our goal was to validate **how symmetries impact three different aspects**: (i) operator approximation (synthetic data with known ground truth), (ii) regression, and (iii) uncertainty quantification (robotics). While, together with additional ones in App. G, these experiments isolate the effect of symmetry independently of architecture design, we acknowledge that the empirical contribution can be further expanded. So, in addition, we address robustness concerns with additional experiments under symmetry misspecification (see response to *i3Ar* and [Figs. A1-4](https://anonymous.4open.science/r/symm_rep_learn-2B3A/plots/rebutal_results.pdf)), showing graceful performance degradation localized only on subsets of the data that violate symmetry priors.
>
> ## Questions
> - Q1: See above.
> - Q2:  In practice, the method, like any other GDL one, works whenever the spaces $X$ and $Y$ share symmetries, that is,  when $G_X$ and $G_Y$ are subgroups of a common ambient group $G$. In that setting, the conditional expectation operator acts as an *equivariant operator* between the corresponding representation spaces sharing symmetry components (i.e., isotypic spaces of same type). If there is no common ambient space, symmetry cannot be exploited and one should revert to NCP. We will clarify this point in the final version, and are happy to provide further details if requested.
> ___
> We thank the reviewer again for the constructive feedback and would be grateful if the reviewer would consider updating their score if these clarifications address their concerns.

---

> > ### Author Rebuttal · Reviewer_L4iB · 2026-04-03
> >
> > I thank the authors for their detailed answer, including additional experiments. I briefly comment on their answer below.
> >
> > - **Clarity.** Although the presentation of the framework is pedagogical, I still believe that the notations could be drastically simplified in the main part of the paper, without sacrificing the generality of the description, and with great gains in clarity.
> > - **Reproducibility.** I acknowledge that the availability of a code repository is a great advantage for the reproducibility of the results and the adoption of the eNCP framework.
> > - **Latent space decomposition.** I was not referring to the encoder / decoder implementation as neural networks, but rather to the decomposition of the latent space into its isotypic subspaces in practice.
> > - **Theoretical results.** I am not familiar enough with the NCP framework to assess the validity of the numerous theoretical results that are presented in the paper, nor their strength.
> >
> > However, the current paper is well-motivated and proposes a solid extension of NCP. I therefore increase my current score to 4.

---

> > > ### Author Response · Authors · 2026-04-07
> > >
> > > We thank the reviewer for the careful follow-up, for increasing the score, and for the generous assessment of the paper as a well-motivated and solid extension of NCP.
> > >
> > > **Clarity.** We agree with the reviewer that the notation overhead in the main body can be reduced substantially without sacrificing generality. In the final version, we will work on simplifying the notation and leverage part of the additional page allowance to provide more intuition and a clearer high-level presentation of the framework before introducing the full formalism.
> > >
> > > **Reproducibility.** We also thank the reviewer for acknowledging that providing reproducible code and tutorial notebooks strengthens the paper and facilitates adoption of the eNCP framework. This was an important goal for us, and we will continue to make the practical implementation as transparent as possible in the final manuscript.
> > >
> > > **Latent-space decomposition.** We appreciate this clarification. The key point is that the parameterization of the encoders in the isotypic/disentangled basis already entails the latent-space decomposition. Indeed, decomposing the group representation acting on the approximated latent space is equivalent to decomposing that latent space into orthogonal isotypic subspaces. This is precisely the core structural result underlying the method—the isotypic decomposition—and it is described in detail in App. I.2, where we make the connection between the decomposition of the space and the decomposition of the representation more explicit. We agree that this point should be brought out much more clearly in the main paper, and we will do so in the final version.
> > >
> > > **Theoretical results.** Finally, we thank the reviewer for acknowledging the level of background required to fully assess the theoretical contributions, and for recognizing the work as well motivated and technically grounded. We will also use the revision to better separate what is inherited from NCP from what is genuinely new in eNCP, so that the theoretical contribution is easier to parse.
> > >
> > > We thank the reviewer again for the constructive engagement and for the improved assessment.

---

### Official Review · Reviewer_i3Ar · 2026-03-13

**Soundness:** 3
**Presentation:** 4
**Significance:** 4
**Originality:** 4
**Overall Recommendation:** 5
**Confidence:** 2

**Summary:**

This paper proposes eNCP, an equivariant extension of Neural Conditional Probability that learns symmetry-aware representations by approximating the conditional expectation operator under symmetry priors. The core idea is to combine operator-theoretic conditional probability estimation with equivariant representation theory: the learned feature spaces are decomposed into isotypic components, the operator approximation is block-diagonal across these components, and the resulting model can be used for equivariant regression, conditional probability estimation, and uncertainty quantification. The paper also derives non-asymptotic bounds in which the estimation term improves with symmetry structure, and presents experiments on synthetic conditional-density tasks and robotics regression / uncertainty-estimation problems.

**Compliance With Llm Reviewing Policy:**

Affirmed.

**Final Justification:**

The clarifications made during the rebuttal solved my remaining concerns and hence I increase my score.

**Key Questions For Authors:**

**Questions**

- The bound in Equation 19 improves with larger irrep dimension  $d_{\mathrm{iso}}$  and worsens with larger  $n_{\mathrm{iso}}$. More intuition for this dependence would be helpful. Why do these two forms of increased representational structure have opposite effects in the bound?
-   An experiment with deliberately misspecified or approximate symmetries would substantially strengthen the practical case for the method.

**Limitations:**

Yes.

**Strengths And Weaknesses:**

**Strengths**
-  **Clear methodological through-line.**  The progression from symmetry assumptions on  $P(x)$  and  $P(y\mid x)$, to equivariance of the conditional expectation operator, to invariance of the PMD kernel is coherent. The parametrisation of an invariant kernel by equivariant encoders parameterised in irreps is well-motivated and follows established design principles.
- **Presentation is mostly strong for a difficult paper.** The paper does a good job of conveying a mathematically dense set of ideas in an accessible way. In particular, Figure 3 is helpful in illustrating how the marginal and joint densities, conditional distributions and expectations, and the PMD transform under the symmetry assumptions.

**Weaknesses**
- **Baseline set is limited relative to the paper’s scope.**  The comparisons to NCP, MLP/eMLP, DRF/iDRF, and CQR/eCQR are reasonable, but for a method framed as a general equivariant representation-learning framework, I would have expected stronger comparisons to more modern equivariant probabilistic modeling baselines.
- **The method depends on correctly specified symmetry priors.**  The authors acknowledge this limitation themselves. In practice, partial symmetry, approximate symmetry, or misspecified symmetry is common, and the paper does not test robustness in those cases.
- **Theoretical contribution could be exposed more clearly in the main text.** The paper’s value rests heavily on the appendix theory, especially the symmetry-constrained spectral structure and the learning bounds. In the main text, the proof burden is too high relative to what is exposed. For a paper making “first guarantees” claims, the core assumptions and what is genuinely new versus inherited from NCP are not separated sharply enough. In particular, the bound depends on the representation error  $E_\theta^r,$ but the paper does not give much operational guidance on when optimising the proposed loss will reliably control that term in realistic neural settings.

---

> ### Author Rebuttal · Authors · 2026-03-31
>
> We thank the reviewer for the careful reading and positive assessment, and for highlighting the operator-theoretic formulation and symmetry-aware structure. We address the main concerns below and would be grateful if the reviewer would consider increasing the score if these points are satisfactorily addressed
> ## Q1: Impact on the statistical bound
> We thank the reviewer for pointing out the lack of clarity. The key point is that the ratio between the number of isotypic components $n_{iso}$ and the total dimension $d_{iso}$ is upper bounded by one. Thus, a better presentation of the estimation term would indicate that it scales as $\log(d_{iso}) / d_{iso}$, reflecting an effective dimension reduction induced by the isotypic decomposition. Here, larger irreducible representations (higher-dim. blocks -> larger $d_{iso}$) correspond to quotient structures of larger order and yield stronger gains, explaining the dependence in Eq. 19.
> Importantly, **this makes explicit that symmetry acts as a dimension-reduction mechanism at the operator level**, yielding strictly improved statistical rates compared to the non-equivariant setting. We will revise the presentation accordingly.
>
> ## Q2: Robustness to misspecification
> We thank the reviewer for pointing to this important aspect. We performed a sensitivity analysis following Wang et al. A general theory of correct, incorrect, and extrinsic equivariance. NeurIPS 2023. We consider: (i) **Extrinsic misspecification**: training data violates G-invariance of $P_x$ (restricted support), (ii) **Incorrect misspecification**: localized violations of $P_{y|x}$ G-invariance. Findings (see [Figs. A2-4](https://anonymous.4open.science/r/symm_rep_learn-2B3A/plots/rebutal_results.pdf)) show that:
> - Under **extrinsic misspecification**, enforcing G-invariance acts as a regularizer and improves out-of-distribution generalization without harming in-support performance (Fig. A2),
> - under **incorrect misspecification**, performance degrades continuously with the degree of violation, and the degradation remains *localized* to regions where symmetry is violated (Fig. A3-4).
> - in all cases, the method does not collapse and retains advantages in regions consistent with the symmetry prior.
>
> Crucially, this demonstrates that **the method is robust beyond the idealized GDL setting** and can potentially be used to detect local symmetry violations. We will include this additional experiment in the revised manuscript.
> ## Clarification of theoretical contribution
> The learning bound decomposes into two terms:  an **estimation error** , and a **representation error** $E^r_\theta$ inherited from NCP. Our main contribution is in the former: by exploiting symmetry, the conditional expectation operator decomposes into isotypic components, reducing the effective dimension from $d$ to $d_{iso}$ and improving the rates by $\log(d_{iso})/d_{iso}$. This explicitly shows that **any nontrivial symmetry improves sample complexity**. Importantly, combining (i) symmetry-aware operator learning, (ii) finite sample guarantees, and (iii) applicability to regression and uncertainty quantification yields, to our knowledge, a formulation not achieved by any prior work. This positions eNCP as a principled extension of NCP that both improves statistical efficiency and broadens applicability in symmetry-structured domains.
> ## Representation error term
> We agree that $E^r_θ$ is not fully characterized *statistically*. This was deliberate, as a full treatment would require additional technical assumptions without changing the main qualitative conclusions. Instead, we briefly reported in the caption of Table 1 its connection to the excess loss (noting a typo due to omitting the constant $\sqrt{σ_r/(σ_r-σ_{r+1})}$), and focused on providing *empirical insights* via synthetic experiments where the true operator is known: Fig. 4 quantifies operator approximation error, and Fig. 5 isolates the gain from enforcing equivariance. These results show that symmetry constraints significantly reduce this term in practice. A sharper theoretical analysis necessarily depends on specific architectures and regularity assumptions, and is an important direction for future work. We will clarify this in the revision
> ## Baselines
> Our experiments and baselines are driven by the object we learn: the **conditional expectation operator** (given by the PMD kernel), which yields optimal spectral representations and induces regression, event probabilities, and uncertainty quantification. Our aim is to see how the equivariance of architectures impacts representation quality **across these tasks**. Accordingly, we compare to the strongest available methods aligned with these tasks: NCP (operator learning), eMLP/MLP (equivariant/non-equivariant regression), CQR/eCQR (quantile-based UQ), and DRF/iDRF (direct PMD estimation). Yet, if the reviewer thinks we should go beyond this we would appreciate clarification on which tasks and baselines should be considered

---

> > ### Author Rebuttal · Reviewer_i3Ar · 2026-04-04
> >
> > I thank the authors for the exhaustive rebuttal. I continue to recommend the paper for acceptance and increase my score.

---

> > > ### Author Response · Authors · 2026-04-07
> > >
> > > We thank the reviewer for the careful reading, constructive engagement throughout the rebuttal process, and for increasing the score.
> > >
> > > We are especially glad that the clarifications on the statistical bound, the additional misspecification experiments, and the separation between the new theoretical contribution of eNCP and the parts inherited from NCP helped resolve the remaining concerns. We also appreciate the reviewer’s recognition of the paper’s methodological coherence, theoretical contribution, and practical relevance.

---

### Decision · Program_Chairs · 2026-04-30

**Decision:**

Accept (regular)

**Comment:**

After discussion, the reviewers reached a positive consensus to accept the paper. They found the contribution interesting and technically strong. Even though 3/4 reviewers had low confidence because the paper is quite technical, Reviewer f5XK, an expert in geometric deep learning, gave a positive assessment of the paper’s soundness with reasonably high confidence, which provides additional reassurance.

The main concern was clarity, and some reviewers also wanted additional experimental evaluation. The rebuttal helped a lot by adding clearer explanations, extra experiments, and reproducibility details (repository).

Overall, I recommend accepting the paper. However, I believe the final version should improve clarity and include the added material from the rebuttal.